# Acute and post-acute respiratory complications of SARS-CoV-2 infection: population-based cohort study in South Korea and Japan

Yujin Choi [1,2,14], Hyeon Jin Kim[1,3,14], Jaeyu Park[1,3,14], Myeongcheol Lee[1,3], Sunyoung Kim[4], Ai Koyanagi[5], Lee Smith [6], Min Seo Kim [7], Masoud Rahmati [8,9,10], Hayeon Lee [1,15] ✉, Jiseung Kang [11,12,15] ✉ & Dong Keon Yon [1,3,13,15] ✉

Considering the significant burden of post-acute COVID-19 conditions among patients infected with SARS-CoV-2, we aimed to identify the risk of acute respiratory complications or post-acute respiratory sequelae. A binational population-based cohort study was conducted to analyze the risk of acute respiratory complications or post-acute respiratory sequelae after SARS-CoV-2 infection. We used a Korean nationwide claim-based cohort (K-COV-N; $n = 2,312,748$; main cohort) and a Japanese claim-based cohort (JMDC; $n = 3,115,606$; replication cohort) after multi-to-one propensity score matching. Among 2,312,748 Korean participants (mean age, 47.2 years [SD, 15.6]; 1,109,708 [48.0%] female), 17.1% (394,598/2,312,748) were infected with SARS-CoV-2. The risk of acute respiratory complications or post-acute respiratory sequelae is significantly increased in people with SARS-CoV-2 infection compared to the general population (acute respiratory complications: HR, 8.06 [95% CI, 6.92-9.38]; post-acute respiratory sequelae: 1.68 [1.62-1.75]), and the risk increased with increasing COVID-19 severity. We identified COVID-19 vaccination as an attenuating factor, showing a protective association against acute or post-acute respiratory conditions. Furthermore, while the excess post-acute risk diminished with time following SARS-CoV-2 infection, it persisted beyond 6 months post-infection. The replication cohort showed a similar pattern in the association. Our study comprehensively evaluates respiratory complications in post-COVID-19 conditions, considering attenuating factors such as vaccination status, post-infection duration, COVID-19 severity, and specific respiratory conditions.

Severe acute respiratory syndrome coronavirus 2 (SARS-CoV-2), responsible for the COVID-19 pandemic, has caused global attention not only due to its immediate symptoms derived from the virus itself but also owing to subsequent physical and mental health sequelae[1].

Since 2020, there have been over 761 million reported cases of SARS-CoV-2 infections, with ~6.8 million deaths among infected individuals[2]. Despite its relatively low fatality rate of ~1.3%[3], ~10% of patients with SARS-CoV-2 infection report persistent, long-lasting comorbidities after

infection termed post-acute COVID-19 condition[4]. Post-acute COVID-19 condition refers to persistent or new-onset health outcomes that last more than a month since SARS-CoV-2 infection, including both short-term (4–12 weeks) and long-term (>12 weeks) symptoms and sequelae[5].

Given the nature of the SARS-CoV-2, the infection can trigger adverse effects on the respiratory system and post-acute COVID-19 respiratory sequelae. One previous study highlighted the association between acute respiratory complication or post-acute respiratory sequelae and COVID-19[6]. Acute respiratory complication is an umbrella term that describes illnesses that affect the respiratory system in a sudden onset[7,8]. Post-acute respiratory sequelae refers to a broad category of long-term non-infectious respiratory diseases that affect the lungs and airways[7,8]. For perspective, the influenza virus is also a well-known viral inducer of respiratory failure[9]. However, insufficient attention has been given to the impact of SARS-CoV-2 infection on acute respiratory complications or post-acute respiratory sequelae in comparison with the influenza infection as a common respiratory viral infection.

Therefore, by using a binational, large-scale, long-term, population-based database with more than 22 million participants in South Korea and Japan, we aimed to investigate the impact of SARS-CoV-2 infection on pathological developments of acute respiratory complication or post-acute respiratory sequelae. We also examined whether COVID-19 vaccinations offer protection against COVID-19-related respiratory outcomes. Furthermore, we analyze the comparison of complications following SARS-CoV-2 infection versus following influenza infection.

## Results

In the main cohort, there were a total of 10,027,506 participants with a mean age of 48.4 (standard deviation [SD], 13.4) years, of which 49.9% (5,000,621/10,027,506) were female (Table S1). The replication cohort includes 4,909,861 participants with a mean age of 46.8 (SD, 11.9) years and 38.3% (1,882,174/4,909,861) females (Table S2). Table 1 shows the baseline characteristics of the 1:5 propensity score matched cohort of South Korea. After 1:5 propensity score matching based on SARS-CoV-2 infection, we identified 82.9% (1,918,150/2,312,748) of participants without SARS-CoV-2 infection and 17.1% (394,598/2,312,748) of participants with SARS-CoV-2 infection, respectively.

In the 1:3 propensity score-matched replication cohort, 74.4% (2,318,505/3,115,606) of participants without SARS-CoV-2 infection and 25.6% (797,101/3,115,606) of participants with SARS-CoV-2 infection were included in our final analyses (Table S3). The standardized mean differences (SMD) of all matching covariates in both multi-to-one propensity score-matched main and replication cohorts were smaller than 0.1 (Table 1).

In the main and replication cohorts, individuals with SARS-CoV-2 infection had a higher adjusted hazard ratio (HR) for post-acute respiratory sequelae compared to the general population (main: HR, 1.68 [95% confidence interval (CI), 1.62–1.75]; replication: HR, 3.32 [95% CI, 3.27–3.37]) in Table 2. Furthermore, patients with SARS-CoV-2 infection had an increased risk for acute respiratory complication compared to non-infected controls (main: HR, 8.06 [95% CI, 6.92–9.38]; replication: HR, 4.17 [95% CI, 3.90–4.45]). When directly comparing the risk for acute respiratory complication between SARS-CoV-2 and influenza infections, SARS-CoV-2 infection was significantly associated with an increased risk (main: HR, 4.32 [95% CI, 2.73–6.83]; replication: HR, 6.51 [95% CI, 5.38–7.87]) in Tables S4–S6.

Relative to the general population, patients with SARS-CoV-2 infection had significantly increased risk for several subtypes of post-acute respiratory sequelae, including chronic respiratory failure (main: HR, 8.92 [95% CI, 4.92-16.17]; replication: HR, 7.55 [95% CI, 6.35-8.97]), chronic obstructive pulmonary disease (COPD), emphysema, asthma, pulmonary sarcoidosis, and interstitial lung disease (main: HR, 10.38 [95% CI, 8.75-12.31]; replication: HR, 4.75 [95% CI, 4.54-4.97]) in Table 3.

Notably, the risk for acute respiratory complication, including aspergillosis pneumonia (main: HR, 6.85 [95% CI, 3.48-13.50]; replication: HR, 4.97 [95% CI, 4.26-5.79]), pneumothorax, acute respiratory failure (main: HR, 112.04 [95% CI, 64.00-196.16]; replication: HR, 6.49 [95% CI, 6.32-6.65]) showed an increase in patients with SARS-CoV-2 infection compared to the general population. This tendency of increased risk for several subtypes of respiratory diseases was also shown when compared to patients with influenza infection and the overlap-weighted cohort. (Tables S7–S10). Estimates of marginal prevalence showed that patients with COVID-19 had a higher prevalence compared to the general population (Tables S11 and S12).

The risk of acute respiratory complication showed decreasing trends according to the number of SARS-CoV-2 vaccinations from individuals after once receiving vaccination (HR, 0.51 [95% CI, 0.38-0.68]) to those with two or more vaccinations (HR, 0.24 [95% CI, 0.19-0.30]). Interestingly, mixed types of vaccination showed the lowest risk of developing post-acute respiratory sequelae of all SARS-CoV-2 vaccination methods (HR, 0.18 [95% CI, 0.08-0.38]). The risks of acute respiratory complications were higher in patients with moderate to severe COVID-19 symptoms (HR, 39.54 [95% CI, 33.54-46.62]). Both the original strain and the delta variant of SARS-CoV-2 were shown to have a higher risk of acute respiratory complications (original strain: HR, 9.21 [95% CI, 7.19-11.80]; delta strain: HR, 7.44 [95% CI, 6.13-9.03]). In addition, the risk for post-acute respiratory sequelae also exhibited a similar pattern (Table 4 and S13).

Table 5 shows the risk of developing acute respiratory complications or post-acute respiratory sequelae based on how long it has been since the participant was infected with SARS-CoV-2 compared to the general population. The analysis of post-acute respiratory sequelae did not include the data of the first month of SARS-CoV-2 infection. The first 3 months after infection with SARS-CoV-2 had the highest risk of developing post-acute respiratory sequelae (main: HR, 2.51 [95% CI, 2.38-2.64]; replication: HR, 4.40 [95% CI, 4.30-4.51]). With increasing duration post-SARS-CoV-2 infection, the risk of post-acute respiratory sequelae significantly decreased, but the risk remained even after 6 months (main: HR, 1.10 [95% CI, 1.01-1.19]; replication: HR, 2.67 [95% CI, 2.61-2.73]). HR of time attenuation effect after SARS-CoV-2 infection showed significance compared to influenza infection likewise (Table S14). Similar associations were observed in the stratification analysis according to sex, age, household income, Charlson comorbidity index (CCI), body mass index (BMI), alcohol consumption, physical activity, region of residence, and income level and polymerase chain reaction (PCR) test in the propensity score-matched cohorts (Tables S15–S24).

## Discussion
### Findings of this study
This is the first study to use population-based, binational large-scale cohort study databases from South Korean and Japanese nationwide cohorts that expresses the association of SARS-CoV-2 infection with acute respiratory complication or post-acute respiratory sequelae. First, the risk of acute respiratory complication or post-acute respiratory sequelae is significantly increased in participants with SARS-CoV-2 infection, compared to the general population. Second, SARS-CoV-2 infection induced a significantly increased risk for several specific post-acute respiratory sequelae, including chronic respiratory failure, COPD, emphysema, asthma, and interstitial lung disease, compared to the general population. In addition, several acute respiratory complications, including aspergillosis pneumonia, pneumothorax, acute respiratory failure, and pulmonary embolism, also depicted a notable increase in risk after SARS-CoV-2 infection compared to the general population. Third, people who were vaccinated, especially multiple vaccinations and mixed vaccinations, had a lower risk of developing post-acute respiratory sequelae than infected patients of SARS-CoV-2 without vaccination. Fourth, the risk of post-acute respiratory sequelae

**Table 1 | Baseline characteristics for 1:5 propensity score–matched cohort (COVID-19 vs. general population) in South Korea (main)**

| Characteristic | COVID-19 vs. general population (n = 2,312,748) | | |
|---|---|---|---|
| | COVID-19 (n = 394,598) | General population (n = 1,918,150) | SMD* |
| **Mean age (SD), y** | 47.5 (16.8) | 46.8 (14.3) | 0.046 |
| **Age, n (%)** | | | 0.026 |
| 20–39 y | 143,273 (36.3) | 702,322 (36.6) | |
| 40–59 y | 145,169 (36.8) | 709,343 (37.0) | |
| ≥60 y | 106,156 (26.9) | 506,485 (26.4) | |
| **Sex, n (%)** | | | 0.006 |
| Male | 205,058 (52.0) | 997,982 (52.0) | |
| Female | 189,540 (48.0) | 920,168 (48.0) | |
| **Region of residence, n (%)** | | | <0.001 |
| Urban | 213,052 (54.0) | 1,035,261 (54.0) | |
| Rural | 181,546 (46.0) | 882,889 (46.0) | |
| **Medical history, n (%)** | | | |
| Cardiovascular disease | 59,947 (15.2) | 286,623 (14.9) | 0.009 |
| Chronic kidney disease | 18,963 (4.8) | 88,687 (4.6) | 0.005 |
| Medication use for diabetes | 71,625 (18.2) | 342,656 (17.9) | 0.008 |
| Medication use for hyperlipidemia | 59,947 (15.2) | 286,623 (14.9) | 0.007 |
| Medication use for hypertension | 32,402 (8.2) | 154,789 (8.1) | 0.005 |
| **Unmatching covariates, n (%)†** | | | |
| Charlson Comorbidity Index score | | | 0.230 |
| 0 | 346,579 (87.8) | 1,806,906 (94.2) | |
| 1 | 30,584 (7.8) | 60,556 (3.2) | |
| ≥2 | 17,435 (4.4) | 50,688 (2.6) | |
| Household income | | | <0.001 |
| Low (0th–39th percentile) | 182,632 (46.3) | 887,593 (46.3) | |
| Middle (40th –79th percentile) | 140,084 (35.5) | 681,180 (35.5) | |
| High (80th–100th percentile) | 71,882 (18.2) | 349,377 (18.2) | |
| Body mass index | | | 1.212 |
| Underweight (< 18.5 kg/m²) | 6875 (1.7) | 67,970 (3.5) | |
| Normal (18.5-22.9 kg/m²) | 77,779 (19.7) | 685,687 (35.8) | |
| Overweight (23.0-24.9 kg/m²) | 54,427 (13.8) | 440,197 (23.0) | |
| Obese (≥ 25.0 kg/m²) | 95,082 (24.1) | 724,074 (37.8) | |
| Unknown | 160,435 (40.7) | 222 (0.012) | |
| Blood pressure | | | 1.157 |
| SBP < 140 mmHg and DBP < 90 mmHg | 199,342 (50.5) | 1,673,777 (87.3) | |
| SBP ≥ 140 mmHg or DBP ≥ 90 mmHg | 33,711 (8.5) | 240,991 (12.6) | |
| Unknown | 161,545 (40.9) | 3382 (0.2) | |
| Fasting blood glucose | | | 1.179 |
| <100 mg/dL | 140,656 (35.7) | 1,189,021 (62.0) | |
| ≥100 mg/dL | 92,374 (23.4) | 725,682 (37.8) | |
| Unknown | 161,568 (40.9) | 3447 (0.2) | |
| Serum total cholesterol | | | 0.416 |
| <200 mg/dL | 67,886 (17.2) | 525,684 (27.4) | |

**Table 1 (continued) | Baseline characteristics for 1:5 propensity score–matched cohort (COVID-19 vs. general population) in South Korea (main)**

| Characteristic | COVID-19 vs. general population (n = 2,312,748) | | |
|---|---|---|---|
| | COVID-19 (n = 394,598) | General population (n = 1,918,150) | SMD* |
| 200 to 239 mg/dL | 39,508 (10.0) | 316,959 (16.5) | |
| ≥240 mg/dL | 16,860 (4.3) | 134,405 (7.0) | |
| Unknown | 270,344 (68.5) | 941,102 (49.1) | |
| Glomerular filtration rate | | | 1.180 |
| <60 mL/min/1.73 m² | 8332 (2.1) | 52,421 (2.7) | |
| 60 to 89 mL/min/1.73 m² | 102,604 (26.0) | 812,214 (42.3) | |
| ≥90 mL/min/1.73 m² | 121,914 (30.9) | 1,048,318 (54.7) | |
| Unknown | 161,748 (41.0) | 5197 (0.3) | |
| Smoking status | | | 1.191 |
| Never | 154,105 (39.1) | 1,214,221 (63.3) | |
| Former | 42,814 (10.9) | 299,027 (15.6) | |
| Current | 37,263 (9.4) | 404,323 (21.1) | |
| Unknown | 160,416 (40.7) | 579 (0.030) | |
| Alcohol consumption | | | 1.156 |
| <1 day/week | 136,870 (34.7) | 1,143,488 (59.6) | |
| 1 to 2 days/week | 66,311 (16.8) | 545,090 (28.4) | |
| 3 to 4 days/week | 23,113 (5.9) | 173,245 (9.0) | |
| ≥ 5 days/week | 7893 (2.0) | 55,687 (2.9) | |
| Unknown | 160,411 (40.7) | 640 (0.034) | |
| Aerobic physical activity | | | 1.179 |
| Insufficient | 118,792 (30.1) | 959,088 (50.0) | |
| Sufficient | 115,321 (29.2) | 958,194 (50.0) | |
| Unknown | 160,485 (40.7) | 868 (0.1) | |
| Strain of SARS-CoV-2 | | | 0.004 |
| Original | 121,521 (30.8) | 594,134 (31.0) | |
| Delta | 273,077 (69.2) | 1,324,016 (69.0) | |

DBP, diastolic blood pressure; SARS-CoV-2, severe acute respiratory syndrome coronavirus 2; SBP, systolic blood pressure; SD, standard deviation; SMD, standardized mean difference.
* An SMD < 0.1 indicates no significant imbalance. All SMDs were <0.1 in the propensity score–matched cohorts.
† Unmatched covariates were included as adjustment factors in statistical analyses.

and acute respiratory complications increased with the severity of COVID-19. Fifth, infection of SARS-CoV-2 was associated with an increase of post-acute respiratory sequelae and acute respiratory complications regardless of the strain type. Lastly, the risk of post-acute respiratory sequelae diminished with time following SARS-CoV-2 infection yet persisted beyond 6 months post-infection.

### Comparisons with previous studies

Some previous studies have examined the relationship between COVID-19 and respiratory complications. However, previous research encompassed countries such as Brazil (n = 88)[10], Palestine (n = 705)[11], and Netherlands (n = 257)[12], with small cohorts and without a general population group as controls or influenza cohort as an external comparator. In addition, previous studies did not consider the subtypes of specific respiratory diseases, vaccinations, and severity of COVID-19 on the association between respiratory outcomes and SARS-CoV-2 infections[13]. Therefore, our study is distinct from other studies in that we compared the association of COVID-19 and respiratory diseases with that of influenza by using population-based binational cohorts with a generalizable scale (main cohort, total N = 10,027,506; replication cohort, total N = 4,909,861).

**Table 2 | HR (95% CI) for the post-acute respiratory sequelae or acute respiratory complications after SARS-CoV-2 infection in the propensity score-matched cohorts of South Korea (main) and Japan (replication)**

| Cohort | South Korea | | | Japan | | |
|---|---|---|---|---|---|---|
| | COVID-19 *vs.* general population (*n* = 2,312,748) | | | COVID-19 *vs.* general population (*n* = 3,115,606) | | |
| | | HR (95% CI) | | | HR (95% CI) | |
| | Events, *n* (%) | Model 1[*] | Model 2[†] | Events, *n* (%) | Model 3[*] | Model 4[‖] |
| **Post-acute respiratory sequelae** | | | | | | |
| Comparators (general population or patients with influenza) | 16,122 (0.84) | 1.0 (reference) | 1.0 (reference) | 35,300 (1.52) | 1.0 (reference) | 1.0 (reference) |
| Patients with COVID-19 | 5292 (1.34) | **1.64 (1.59-1.69)** | **1.68 (1.62-1.75)** | 41,074 (5.15) | **3.50 (3.45-3.55)** | **3.32 (3.27-3.37)** |
| **Acute respiratory complications** | | | | | | |
| Comparators | 331 (0.017) | 1.0 (reference) | 1.0 (reference) | 1468 (0.06) | 1.0 (reference) | 1.0 (reference) |
| Patients with COVID-19 | 618 (0.16) | **9.70 (8.46-11.11)** | **8.06 (6.92-9.38)** | 2304 (0.29) | **4.60 (4.31-4.91)** | **4.17 (3.90-4.45)** |

CCI, Charlson comorbidity index; CI, confidence interval; HR, hazard ratio.
Bold indicates that hazard ratio is statistically significant ($P < 0.05$).
*Models 1 and 3: Adjusted for age (20–39, 40–59, and ≥60 years) and sex.
†Model 2: Adjusted for age (20–39, 40–59, and ≥60 years); sex, household income (low income, middle income, and high income); region of residence (urban and rural); CCI score (0, 1, and ≥2); obesity (underweight [<18.5 kg/m²], normal [18.5–22.9 kg/m²]; overweight [23.0–24.9 kg/m²], obese [≥25.0 kg/m²], and unknown); blood pressure (systolic blood pressure <140 mmHg and diastolic blood pressure < 90 mmHg, systolic blood pressure ≥140 mmHg or diastolic blood pressure ≥ 90 mmHg, and unknown); fasting blood glucose (<100, ≥ 100 mg/dL, and unknown); serum total cholesterol (<200, 200–239, ≥ 240 mg/dL, and unknown); glomerular filtration rate (<60, 60–89, ≥ 90 mL/min/1.73 m², and unknown); smoking status (never, former, current smoker, and unknown); alcoholic drinks (<1, 1–2, 3–4, ≥ 5 days per week, and unknown); aerobic physical activity (sufficient, insufficient, and unknown); previous history of cardiovascular disease, and chronic kidney disease; history of medication use for diabetes mellitus, dyslipidemia, and hypertension; and strain of SARS-CoV-2 (original and delta).
‖Model 4: Adjusted for age (20–39, 40–59, and ≥ 60 years); sex; insurance status (insured and dependent); CCI score (0, 1, and ≥2); body mass index (underweight [< 18.5 kg/m²], normal [18.5–22.9 kg/m²], overweight [23.0–24.9 kg/m²], obese [≥25.0 kg/m²], and unknown); blood pressure (systolic blood pressure <140 mmHg and diastolic blood pressure < 90 mmHg, systolic blood pressure ≥140 mmHg or diastolic blood pressure ≥90 mmHg, and unknown); fasting blood glucose (<100, ≥100 mg/dL, and unknown); serum total cholesterol (<200, 200–239, ≥240 mg/dL, and unknown); glomerular filtration rate (<60, 60–89, ≥90 mL/min/1.73 m², and unknown); smoking status (non- and current smoker, and unknown); alcoholic drinks (rarely, sometimes, everyday, and unknown); aerobic physical activity (sufficient, insufficient, and unknown); previous history of cardiovascular disease, and chronic kidney disease; history of medication use for diabetes mellitus, dyslipidemia, and hypertension; and strain of SARS-CoV-2 (original and delta).

## Possible explanations

Influenza and COVID-19 have similar symptoms such as fever, cough, shortness of breath, and sore throat[14]. However, it is known that people who are infected with SARS-CoV-2 may have more severe symptoms and take longer to recover than those infected with influenza[15]. The longer time and severity of COVID-19 may have had a greater influence on the patients and the overall immune system[16,17]. Severe COVID-19 can lead to lasting changes in hematopoietic stem and progenitor cells and immune cell phenotypes in individuals while they are recovering from COVID-19[16]. Also, T cells can be impaired in severe disease and can be associated with intense activation and lymphopenia[18]. Therefore, SARS-CoV-2 infection can make the patients more vulnerable to developing other respiratory diseases. This aligns with our findings that indicate that infection of SARS-CoV-2 has a greater influence on developing acute respiratory complications or post-acute respiratory sequelae compared to influenza.

Unlike influenza, SARS-CoV-2 infection induces a fibrosis-associated transcriptional profile in pulmonary macrophages, characterized by elevated levels of transforming growth factor beta 1 and transforming growth factor beta induced, as well as other proteins like macrophage mannose receptor 1 and cluster of differentiation 163[19,20]. This gene expression pattern enhances the profibrotic functions of macrophages[20], potentially leading to acute respiratory distress syndrome. Aspergillosis pneumonia, an infection caused by inhaling spores of the fungus Aspergillus, prevalent in the natural environment[21], does not usually develop. However, there are many countries that use antibiotics for COVID-19 treatment[22–24], and overuse of antibiotics can heighten the risk of aspergillosis pneumonia development since the antibiotics may cause a disturbance to the immune system[25,26]. This explains our finding that specific diseases of acute respiratory complication or post-acute respiratory sequelae depicted a notable increase of risk after SARS-CoV-2 infection compared to influenza virus infection.

Many studies have confirmed that vaccination significantly reduces the infection rate of SARS-CoV-2 and the severity of COVID-19 symptoms[27,28]. The efficacy of vaccination is more profound in preventing severe cases and deaths[29]. The reduced severity due to vaccination may positively affect immune resilience[30], thereby decreasing the incidence of acute respiratory complication or post-acute respiratory sequelae. Given that the efficacy of the vaccination drops in the first 6 months, booster vaccination might be essential to sustain protective effects[29]. This is consistent with the result of this study in that multiple vaccinations decreases the development of acute respiratory complications or post-acute respiratory sequelae. Furthermore, many studies showed that mixing types of vaccination for COVID-19 may lower the risk of SARS-CoV-2 infection[31]. This approach could also mitigate the development of future respiratory complications.

Post-recovery from COVID-19, the immune system undergoes reconstruction[32]. However, the elevated interferon responsive genes in monocytes can still be found after 4 months since the infection[33], which implies that the immune system is not fully recovered after 4 months, and constant attention must be paid to the patients. Our findings show that as time passes after initial infection with SARS-CoV-2, the risk of developing acute respiratory complications or post-acute respiratory sequelae gradually decreased. However, the risk for respiratory sequelae in post-acute COVID-19 condition persisted beyond 6 months post-infection.

## Limitations and strengths

This is the first study to utilize binational, large-scale, population-based databases to examine risk for respiratory sequelae in acute or post-acute COVID-19 conditions in patients with SARS-CoV-2 infection. However, some limitations must be taken into consideration. First, although the database used is a highly credible database that covers 98% of the Korean population and 40% of the Japanese population, individuals who could be vulnerable to influenza and COVID-19, such as immigrants and undocumented immigrants, are left out of the database[34–36]. Likewise, the JMDC database does not include the entire Japanese population and may have potential bias. Second, our data is limited to the East Asian population, specifically South Korea and Japan. Therefore, our study is difficult to generalize to other ethnic

**Table 3 | HR (95% CI) for the post-acute respiratory sequelae or acute respiratory complications subtypes after SARS-CoV-2 infection in the propensity score-matched cohorts in South Korea (main) and Japan (replication)**

| Cohort | South Korea | | | Japan | | |
|---|---|---|---|---|---|---|
| | COVID-19 vs. general population (n = 2,312,748) | | | COVID-19 vs. general population (n = 3,115,606) | | |
| | | HR (95% CI) | | | HR (95% CI) | |
| | Events, n (%) | Model 1[*] | Model 2[*] | Events, n (%) | Model 3[*] | Model 4[‖] |
| **Post-acute respiratory sequelae** | | | | | | |
| **Chronic respiratory failure** | | | | | | |
| Comparators (general population or patients with influenza) | 18 (0.00094) | 1.0 (reference) | 1.0 (reference) | 170 (0.0073) | 1.0 (reference) | 1.0 (reference) |
| Patients with COVID-19 | 46 (0.012) | **12.80 (7.42-22.07)** | **8.92 (4.92-16.17)** | 688 (0.086) | **11.85 (10.02-14.02)** | **7.55 (6.35-8.97)** |
| **Pulmonary hypertension** | | | | | | |
| Comparators | 15 (0.00078) | 1.0 (reference) | 1.0 (reference) | 156 (0.0067) | 1.0 (reference) | 1.0 (reference) |
| Patients with COVID-19 | 3 (0.00076) | 1.00 (0.29-3.45) | 0.60 (0.11-3.39) | 217 (0.027) | **4.07 (3.31-5.00)** | **3.11 (2.51-3.85)** |
| **Sleep apnea** | | | | | | |
| Comparators | 1143 (0.060) | 1.0 (reference) | 1.0 (reference) | 6643 (0.29) | 1.0 (reference) | 1.0 (reference) |
| Patients with COVID-19 | 235 (0.060) | 1.02 (0.89-1.17) | 1.13 (0.95-1.33) | 5198 (0.65) | **2.30 (2.22-2.39)** | **2.21 (2.13-2.29)** |
| **COPD** | | | | | | |
| Comparators | 10846 (0.57) | 1.0 (reference) | 1.0 (reference) | 11003 (0.47) | 1.0 (reference) | 1.0 (reference) |
| Patients with COVID-19 | 3359 (0.85) | **1.54 (1.49-1.61)** | **1.57 (1.50-1.65)** | 15520 (1.95) | **4.17 (4.07-4.27)** | **3.93 (3.83-4.03)** |
| **Emphysema** | | | | | | |
| Comparators | 386 (0.020) | 1.0 (reference) | 1.0 (reference) | 1550 (0.067) | 1.0 (reference) | 1.0 (reference) |
| Patients with COVID-19 | 133 (0.034) | **1.73 (1.42-2.10)** | **1.60 (1.27-2.01)** | 2085 (0.26) | **3.95 (3.70-4.22)** | **3.44 (3.22-3.68)** |
| **Asthma** | | | | | | |
| Comparators | 4197 (0.22) | 1.0 (reference) | 1.0 (reference) | 21314 (0.92) | 1.0 (reference) | 1.0 (reference) |
| Patients with COVID-19 | 1431 (0.36) | **1.70 (1.60-1.80)** | **1.74 (1.62-1.87)** | 25311 (3.18) | **3.53 (3.46-3.59)** | **3.44 (3.38-3.50)** |
| **Pulmonary sarcoidosis** | | | | | | |
| Comparators | 29 (0.0015) | 1.0 (reference) | 1.0 (reference) | 972 (0.042) | 1.0 (reference) | 1.0 (reference) |
| Patients with COVID-19 | 4 (0.0010) | 0.69 (0.24-1.95) | 0.96 (0.34-2.75) | 1255 (0.16) | **3.78 (3.48-4.11)** | **3.44 (3.16-3.75)** |
| **Interstitial lung disease** | | | | | | |
| Comparators | 223 (0.012) | 1.0 (reference) | 1.0 (reference) | 2942 (0.13) | 1.0 (reference) | 1.0 (reference) |
| Patients with COVID-19 | 453 (0.11) | **10.13 (8.63-11.90)** | **10.38 (8.75-12.31)** | 5996 (0.75) | **6.00 (5.74-6.27)** | **4.75 (4.54-4.97)** |
| **Acute respiratory complications** | | | | | | |
| **Pneumocystis pneumonia** | | | | | | |
| Comparators | 5 (0.00026) | 1.0 (reference) | 1.0 (reference) | 934 (0.040) | 1.0 (reference) | 1.0 (reference) |
| Patients with COVID-19 | 2 (0.00051) | 1.96 (0.38-10.10) | 0.03 (0.00-8550.49) | 1426 (0.18) | **4.46 (4.10-4.84)** | **3.28 (3.01-3.58)** |
| **Aspergillosis pneumonia** | | | | | | |
| Comparators | 16 (0.00083) | 1.0 (reference) | 1.0 (reference) | 249 (0.011) | 1.0 (reference) | 1.0 (reference) |
| Patients with COVID-19 | 32 (0.0081) | **9.73 (5.34-17.72)** | **6.85 (3.48-13.50)** | 601 (0.075) | **7.05 (6.08-8.17)** | **4.97 (4.26-5.79)** |
| **Pleural empyema** | | | | | | |
| Comparators | 10 (0.00052) | 1.0 (reference) | 1.0 (reference) | 24 (0.0010) | 1.0 (reference) | 1.0 (reference) |
| Patients with COVID-19 | 6 (0.0015) | **2.93 (1.06-8.05)** | 1.45 (0.32-6.63) | 226 (0.028) | **27.44 (18.02-41.80)** | **22.00 (14.38-33.65)** |
| **Lung abscess** | | | | | | |
| Comparators | 17 (0.00089) | 1.0 (reference) | 1.0 (reference) | 57 (0.0025) | 1.0 (reference) | 1.0 (reference) |
| Patients with COVID-19 | 7 (0.0018) | 2.01 (0.83-4.85) | 2.20 (0.81-6.00) | 301 (0.038) | **15.39 (11.60-20.43)** | **13.57 (10.19-18.07)** |
| **Pneumothorax** | | | | | | |
| Comparators | 43 (0.0022) | 1.0 (reference) | 1.0 (reference) | 3818 (0.16) | 1.0 (reference) | 1.0 (reference) |
| Patients with COVID-19 | 50 (0.013) | **5.69 (3.78-8.55)** | **5.29 (3.32-8.42)** | 3234 (0.41) | **2.49 (2.37-2.60)** | **2.41 (2.30-2.53)** |
| **Acute respiratory failure** | | | | | | |
| Comparators | 13 (0.00068) | 1.0 (reference) | 1.0 (reference) | 8767 (0.38) | 1.0 (reference) | 1.0 (reference) |
| Patients with COVID-19 | 363 (0.092) | **135.7 (78.05-235.91)** | **112.04 (64.00-196.16)** | 20983 (2.63) | **7.10 (6.92-7.28)** | **6.49 (6.32-6.65)** |

**Table 3 (continued) | HR (95% CI) for the post-acute respiratory sequelae or acute respiratory complications subtypes after SARS-CoV-2 infection in the propensity score-matched cohorts in South Korea (main) and Japan (replication)**

| Cohort | South Korea | | | Japan | | |
|---|---|---|---|---|---|---|
| | COVID-19 *vs.* general population (*n* = 2,312,748) | | | COVID-19 *vs.* general population (*n* = 3,115,606) | | |
| | | HR (95% CI) | | | HR (95% CI) | |
| | Events, *n* (%) | Model 1[*] | Model 2[†] | Events, *n* (%) | Model 3[*] | Model 4[‖] |
| Pulmonary embolism | | | | | | |
| Comparators | 209 (0.011) | 1.0 (reference) | 1.0 (reference) | 2212 (0.10) | 1.0 (reference) | 1.0 (reference) |
| Patients with COVID-19 | 162 (0.041) | **3.79 (3.08-4.65)** | **2.98 (2.32-3.82)** | 3972 (0.50) | **5.26 (5.00-5.55)** | **4.58 (4.34-4.83)** |

CCI, Charlson comorbidity index; CI, confidence interval; COPD, chronic obstructive pulmonary disease; HR, hazard ratio; NA, not available.
Bold indicates that hazard ratio is statistically significant (*P* < 0.05).
*Models 1 and 3: Adjusted for age (20–39, 40–59, and ≥60 years) and sex.
†Model 2: Adjusted for age (20–39, 40–59, and ≥ 60 years); sex, household income (low income, middle income, and high income); region of residence (urban and rural); CCI score (0, 1, and ≥2); obesity (underweight [<18.5 kg/m²], normal [18.5–22.9 kg/m²]; overweight [23.0–24.9 kg/m²], obese [≥25.0 kg/m²], and unknown); blood pressure (systolic blood pressure <140 mmHg and diastolic blood pressure <90 mmHg, systolic blood pressure ≥140 mmHg or diastolic blood pressure ≥ 90 mmHg, and unknown); fasting blood glucose (<100, ≥ 100 mg/dL, and unknown); serum total cholesterol (<200, 200–239, ≥ 240 mg/dL, and unknown); glomerular filtration rate (<60, 60–89, ≥90 mL/min/1.73 m², and unknown); smoking status (never, former, current smoker, and unknown); alcoholic drinks (<1, 1–2, 3–4, ≥5 days per week, and unknown); aerobic physical activity (sufficient, insufficient, and unknown); previous history of cardiovascular disease, and chronic kidney disease; history of medication use for diabetes mellitus, dyslipidemia, and hypertension; and strain of SARS-CoV-2 (original and delta).
‖Model 4: Adjusted for age (20–39, 40–59, and ≥60 years); sex; insurance status (insured and dependent); CCI score (0, 1, and ≥2); body mass index (underweight [<18.5 kg/m², normal [18.5–22.9 kg/m², overweight [23.0–24.9 kg/m², obese [≥25.0 kg/m²], and unknown); blood pressure (systolic blood pressure <140 mmHg and diastolic blood pressure <90 mmHg, systolic blood pressure ≥140 mmHg or diastolic blood pressure ≥90 mmHg, and unknown); fasting blood glucose (<100, ≥100 mg/dL, and unknown); serum total cholesterol (<200, 200–239, ≥240 mg/dL, and unknown); glomerular filtration rate (<60, 60–89, ≥90 mL/min/1.73 m², and unknown); smoking status (non- and current smoker, and unknown); alcoholic drinks (rarely, sometimes, everyday, and unknown); aerobic physical activity (sufficient, insufficient, and unknown); previous history of cardiovascular disease, and chronic kidney disease; history of medication use for diabetes mellitus, dyslipidemia, and hypertension; and strain of SARS-CoV-2 (original and delta).

groups. Third, the K-COV-N and JMDC datasets are heterogeneous. Therefore, we opted against merging the datasets, using the K-COV-N data for the main cohort and the JMDC data for the replication cohort. In addition, we used different lists of the covariates for each main and replication cohort due to the difference in data structure. Fourth, the dataset we utilized has a risk of underdiagnoses of SARS-CoV-2 and influenza infection. There is a possibility of overlooking patients who were infected with SARS-CoV-2 or influenza but did not take the PCR test or visit a hospital to receive treatment. However, to assess the potential underdiagnoses, we analyzed the HR of post-acute respiratory sequelae and short-term acute respiratory complications with participants after PCR tests. Fifth, the HR and its 95% CI for the risk of asthma may differ from previous research due to the difference in experimental designs, including study population or definition of exposure. Sixth, the propensity score-matched cohort had differential missingness between those infected with SARS-CoV-2 and the general population for national health examination information variables (BMI, blood pressure, fasting blood glucose, glomerular filtration rate, smoking status, alcohol consumption, and aerobic physical activity; >40% versus <1%), due to their exclusion from matching criteria.

### Policy implications
This binational, large-scale, population-based cohort study further emphasizes risks in relation to SARS-CoV-2 infection, the importance of vaccination, efficient vaccination methods, and post-acute COVID-19 conditions with an emphasis on acute respiratory complications or post-acute respiratory sequelae. These findings depict a need for different health policies to manage social health. To minimize adverse respiratory outcomes after being infected with SARS-CoV-2, the government should make policies to mix and match the vaccine types to individuals. Individuals should be investigated even after full recovery from COVID-19 to resolve post-acute COVID-19 conditions.

In conclusion, this study emphasizes that the risk of developing acute respiratory complications or post-acute respiratory sequelae in post-COVID-19 condition is associated with infection of SARS-CoV-2, and the risk was more pronounced with increasing COVID-19 severity. People who were vaccinated had a lower risk of developing acute respiratory complications or post-acute respiratory sequelae than those without vaccination. While the risk of acute respiratory complications or post-acute respiratory sequelae decreases with

time post-SARS-CoV-2 infection, it remains evident beyond 6 months. Therefore, our findings suggest that the potential risk of respiratory sequelae in acute or post-acute COVID-19 conditions accentuates the imperative for continued vigilance and response to SARS-CoV-2.

## Methods
### Data source
Utilizing large-scale, population-based binational cohorts, this study incorporated a South Korean nationwide claim-based cohort (K-COV-N cohort; total *N* = 10,027,506) for the main cohort and a Japanese claim-based cohort (JMDC cohort; total *N* = 4,909,861) for the replication cohort (Fig. 1)[37]. Both the K-COV-N and the JMDC cohorts were constructed through data derived from a universal health insurance claims system. This study received approvals from the Korea Disease Control and Prevention Agency (KDCA), National Health Insurance Service (NHIS; KDCA-NHIS-2022-1-632), JMDC (PHP-00002201-04), and the Institutional Review Board of Kyung Hee University (KHSIRB-23-241). Under the terms of the approval, patient consent was not required to use routine health records for our study.

### K-COV-N cohort for main cohort
We utilized the NHIS database, which is a large-scale, nationwide, general, population-based cohort in South Korea, covering 98% of the population for the main cohort[34]. The NHIS and the KDCA provided data for the cohort constructed for study purposes, which includes participants ≥20 years old with a record of medical examination from January 1, 2018, to December 31, 2021 (total *N* = 10,027,506). The dataset consists of national health examination information, death records, health insurance data including individual demographic information, outpatient/inpatient records, and pharmaceutical data from the NHIS and COVID-19 vaccination data, SARS-CoV-2 test results, and COVID-19-related outcomes from the KDCA. The constructed K-COV-N database embodies the following characteristics, thereby affirming its significance: (1) the Korean government has established an extensive healthcare system to provide coverage for individuals infected with SARS-CoV-2; (2) all patient-related data was anonymized by the Korean government[34,36]; and (3) according to the prior study, the diagnostic records from the NHIS had a predictive accuracy of 82%[38].

**Table 4 | Subgroup analysis (COVID-19 *vs.* general population) of HR (95% CI) of the post-acute respiratory sequelae or acute respiratory complications after SARS-CoV-2 infection stratified by vaccination, COVID-19 severity, and SARS-CoV-2 strain in the cohort of South Korea (main)**

| Variable | Events/total, *n/N* (%) | HR (95% CI) | |
|---|---|---|---|
| | | Model 1[*] | Model 2[†] |
| **Post-acute respiratory sequelae** | | | |
| **Number of SARS-CoV-2 vaccinations** | | | |
| Non-infected control[‖] | 16,122/1,918,150 (0.84) | **0.62 (0.60-0.65)** | **0.60 (0.58-0.62)** |
| COVID-19 without SARS-CoV-2 vaccination | 4331/200,539 (2.16) | 1.0 (reference) | 1.0 (reference) |
| COVID-19 after SARS-CoV-2 vaccination received once | 493/38,852 (1.27) | **0.90 (0.82-0.99)** | **0.85 (0.77-0.94)** |
| COVID-19 after SARS-CoV-2 vaccination received twice or more | 468/155,207 (0.30) | **0.69 (0.62-0.76)** | **0.64 (0.57-0.71)** |
| **Type of SARS-CoV-2 vaccinations** | | | |
| Non-infected control[‖] | 16,122/1,918,150 (0.84) | **0.62 (0.60-0.65)** | **0.60 (0.58-0.62)** |
| COVID-19 without SARS-CoV-2 vaccination | 4331/200,539 (2.16) | 1.0 (reference) | 1.0 (reference) |
| COVID-19 with viral vector SARS-CoV-2 vaccination | 465/109,066 (0.43) | **1.12 (1.01-1.23)** | 0.99 (0.90-1.10) |
| COVID-19 with mRNA SARS-CoV-2 vaccination | 477/66,891 (0.71) | **1.20 (1.09-1.33)** | 1.11 (0.99-1.22) |
| COVID-19 with both types of SARS-CoV-2 vaccination | 19/18,102 (0.10) | 0.74 (0.47-1.16) | 0.66 (0.42-1.03) |
| **COVID-19 severity** | | | |
| Non-infected control[‖] | 16,122/1,918,150 (0.84) | 1.0 (reference) | 1.0 (reference) |
| Mild COVID-19 | 3492/340,813 (1.02) | **1.28 (1.24-1.33)** | **1.37 (1.32-1.43)** |
| Moderate to severe COVID-19 | 1800/53,785 (3.35) | **3.60 (3.43-3.78)** | **3.20 (3.03-3.38)** |
| **Original strain of SARS-CoV-2 (overall population)** | | | |
| Non-infected control before the delta-dominant phase[§] | 11,667/594,134 (1.96) | 1.0 (reference) | 1.0 (reference) |
| Infection with original strain | 3649/121,521 (3.00) | **1.58 (1.52-1.64)** | **1.59 (1.52-1.66)** |
| **Delta variant of SARS-CoV-2 (overall population)** | | | |
| Non-infected control during the delta-dominant phase[§] | 4455/1,324,016 (0.34) | 1.0 (reference) | 1.0 (reference) |
| Infection with delta variant | 1643/273,077 (0.60) | **1.81 (1.71-1.92)** | **1.94 (1.81-2.08)** |
| **Acute respiratory complications** | | | |
| **Number of SARS-CoV-2 vaccinations** | | | |
| Non-infected control[‖] | 311/1,918,150 (0.016) | **0.07 (0.06-0.08)** | **0.08 (0.06-0.09)** |
| COVID-19 without SARS-CoV-2 vaccination | 415/200,539 (0.21) | 1.0 (reference) | 1.0 (reference) |
| COVID-19 after SARS-CoV-2 vaccination received once | 56/38,852 (0.14) | **0.62 (0.47-0.83)** | **0.51 (0.38-0.68)** |
| COVID-19 after SARS-CoV-2 vaccination received twice or more | 147/155,207 (0.09) | **0.32 (0.26-0.39)** | **0.24 (0.19-0.30)** |
| **Type of SARS-CoV-2 vaccinations** | | | |
| Non-infected control[‖] | 311/1,918,150 (0.016) | **0.07 (0.06-0.08)** | **0.08 (0.06-0.09)** |
| COVID-19 without SARS-CoV-2 vaccination | 415/200,539 (0.21) | 1.0 (reference) | 1.0 (reference) |
| COVID-19 with viral vector SARS-CoV-2 vaccination | 79/109,066 (0.072) | **0.44 (0.34-0.56)** | **0.36 (0.28-0.47)** |
| COVID-19 with mRNA SARS-CoV-2 vaccination | 117/66,891 (0.17) | **0.40 (0.32-0.50)** | **0.42 (0.33-0.53)** |
| COVID-19 with both types of SARS-CoV-2 vaccination | 7/18,102 (0.039) | **0.19 (0.09-0.41)** | **0.18 (0.08-0.38)** |
| **COVID-19 severity** | | | |
| Non-infected control[‖] | 311/1,918,150 (0.016) | 1.0 (reference) | 1.0 (reference) |
| Mild COVID-19 | 50/340,813 (0.015) | 0.95 (0.71-1.28) | 0.99 (0.73-1.34) |
| Moderate to severe COVID-19 | 568/53,783 (1.06) | **51.18 (44.38-59.02)** | **39.54 (33.54-46.62)** |
| **Original strain of SARS-CoV-2 (overall population)** | | | |
| Non-infected control before the delta-dominant phase[§] | 110/594,134 (0.019) | 1.0 (reference) | 1.0 (reference) |
| Infection with original strain | 230/121,521 (0.19) | **10.27 (8.18-12.88)** | **9.21 (7.19-11.80)** |
| **Delta variant of SARS-CoV-2 (overall population)** | | | |
| Non-infected control during the delta-dominant phase[§] | 201/1,324,016 (0.015) | 1.0 (reference) | 1.0 (reference) |
| Infection with delta variant | 388/273,077 (0.14) | **9.39 (7.92-11.14)** | **7.44 (6.13-9.03)** |

CCI, Charlson comorbidity index; CI, confidence interval; HR, hazard ratio; SARS-CoV-2, severe acute respiratory syndrome coronavirus 2.
Bold indicates that hazard ratio is statistically significant (P < 0.05).
‖HR of the non-infected control represents the risk of respiratory diseases, and HRs of patients with COVID-19 indicate the risk of post-acute respiratory complications following SARS-CoV-2 infection.
§ Only 1:5–matched comparators in each patient group at the same index date were included to reduce immortal time bias.
*Model 1: Adjusted for age (20–39, 40–59, and ≥60 years) and sex.
†Model 2: Adjusted for age (20–39, 40–59, and ≥60 years); sex, household income (low income, middle income, and high income); region of residence (urban and rural); CCI score (0, 1, and ≥2); obesity (underweight [<18.5 kg/m$^2$], normal [18.5–22.9 kg/m$^2$]; overweight [23.0–24.9 kg/m$^2$], obese [≥25.0 kg/m$^2$], and unknown); blood pressure (systolic blood pressure <140 mmHg and diastolic blood pressure < 90 mmHg, systolic blood pressure ≥ 140 mmHg or diastolic blood pressure ≥90 mmHg, and unknown); fasting blood glucose (<100, ≥100 mg/dL, and unknown); serum total cholesterol (<200, 200–239, ≥240 mg/dL, and unknown); glomerular filtration rate (<60, 60–89, ≥90 mL/min/1.73 m$^2$, and unknown); smoking status (never, former, current smoker, and unknown); alcoholic drinks (<1, 1–2, 3–4, ≥5 days per week, and unknown); aerobic physical activity (sufficient, insufficient, and unknown); previous history of cardiovascular disease, and chronic kidney disease; history of medication use for diabetes mellitus, dyslipidemia, and hypertension; and strain of SARS-CoV-2 (original and delta).

**Table 5 | Time attenuation effect analysis of HR (95% CI) for the risk of post-acute respiratory sequelae after SARS-CoV-2 infection in South Korea (main cohort) and Japan (replication cohort)**

| Time | COVID-19 *vs.* general population | |
|---|---|---|
| | Main cohort[†] | Replication cohort[‖] |
| **Post-acute respiratory sequelae** | | |
| <3 months | **2.51 (2.38-2.64)** | **4.40 (4.30-4.51)** |
| 3–6 months | **1.24 (1.15-1.33)** | **2.66 (2.57-2.75)** |
| ≥6 months | **1.10 (1.01-1.19)** | **2.67 (2.61-2.73)** |

CCI, Charlson comorbidity index; CI, confidence interval; HR, hazard ratio; SARS-CoV-2, severe acute respiratory syndrome coronavirus 2.

Bold indicates that hazard ratio is statistically significant (*P* < 0.05).

[†] **Adjusted HR (main):** Adjusted for age (20–39, 40–59, and ≥60 years); sex, household income (low income, middle income, and high income); region of residence (urban and rural); CCI score (0, 1, and ≥2); obesity (underweight [<18.5 kg/m²], normal [18.5–22.9 kg/m²]; overweight [23.0–24.9 kg/m²], obese [≥25.0 kg/m²], and unknown); blood pressure (systolic blood pressure < 140 mmHg and diastolic blood pressure < 90 mmHg, systolic blood pressure ≥ 140 mmHg or diastolic blood pressure ≥ 90 mmHg, and unknown); fasting blood glucose (<100, ≥100 mg/dL, and unknown); serum total cholesterol (<200, 200–239, ≥240 mg/dL, and unknown); glomerular filtration rate (<60, 60–89, ≥90 mL/min/1.73 m², and unknown); smoking status (never, former, current smoker, and unknown); alcoholic drinks (<1, 1–2, 3–4, ≥5 days per week, and unknown); aerobic physical activity (sufficient, insufficient, and unknown); previous history of cardiovascular disease, and chronic kidney disease; history of medication use for diabetes mellitus, dyslipidemia, and hypertension; and strain of SARS-CoV-2 (original and delta).

[‖] **Adjusted HR (replication):** Adjusted for age (20–39, 40–59, and ≥60 years); sex; insurance status (insured and dependent); CCI score (0, 1, and ≥2); body mass index (underweight [<18.5 kg/m²], normal [18.5–22.9 kg/m²], overweight [23.0–24.9 kg/m²], obese [≥25.0 kg/m²], and unknown); blood pressure (systolic blood pressure < 140 mmHg and diastolic blood pressure < 90 mmHg, systolic blood pressure ≥ 140 mmHg or diastolic blood pressure ≥ 90 mmHg, and unknown); fasting blood glucose (<100, ≥ 100 mg/dL, and unknown); serum total cholesterol (<200, 200–239, ≥240 mg/dL, and unknown); glomerular filtration rate (<60, 60–89, ≥90 mL/min/1.73 m², and unknown); smoking status (non- and current smoker, and unknown); alcoholic drinks (rarely, sometimes, everyday, and unknown); aerobic physical activity (sufficient, insufficient, and unknown); previous history of cardiovascular disease, and chronic kidney disease; history of medication use for diabetes mellitus, dyslipidemia, and hypertension; and strain of SARS-CoV-2 (original and delta).

The previous diagnostic history was assessed during the pre-observation period from 2018 to 2019, the follow-up observation period was between 2020 and 2021. The follow-up date ended on December 31, 2021, at death, or at development of primary outcomes (Fig. S1). We excluded participants with the following criteria: (1) insufficient demographic information and those who died before (excluded *n* = 3,967,482); and (2) previous history of chronic respiratory disease in the pre-observation period (excluded *n* = 710,468).

## Exposures

Exposure to SARS-CoV-2 is defined as an infection validated using a real-time reverse transcriptase polymerase chain reaction (RT-PCR) assay or antigen test on nasal and pharyngeal swabs, as approved by the KDCA[39]. Patients necessitating intensive care, oxygen therapy, extracorporeal membrane oxygenation, renal replacement, or cardio resuscitation were classified as having moderate to severe COVID-19[40]. All other cases were categorized as having mild COVID-19[41]. COVID-19 vaccination was classified according to the number of vaccinations (unvaccinated, 1, and ≥2 times) and vaccine type (unvaccinated, mRNA vaccinated [Pfizer-BioNTech and Moderna], viral vector vaccinated [Oxford-AstraZeneca and Johnson & Johnson/Janssen], and vaccinated with both types)[36]. Only vaccination status before SARS-CoV-2 infection was considered in our analysis. In South Korea, SARS-CoV-2 infection from January 2020 to July 31, 2021, was defined as the original stain, and the SARS-CoV-2 infection from August 1, 2021, to December 31, 2021, was defined as the delta[36,37,42]. In Japan, diagnosis of SARS-CoV-2 infection was categorized as infection with the original strain until May 31, 2021, and delta variants from June 1, 2021, to December 31, 2021[36,43]. To examine the relative severity of COVID-19 in

comparison with another contagious viral respiratory disease, additional exposure to influenza infection was defined. It refers to cases diagnosed through an RT-PCR assay or antigen test on nasal and pharyngeal swabs during the observation period. For individuals infected with both SARS-CoV-2 and influenza, it includes instances of influenza infection developing after the SARS-CoV-2 infection.

## Outcomes

To investigate the impact of SARS-CoV-2 infection on acute respiratory complication and post-acute respiratory sequelae, respectively, our experimental designs included two distinct 'primary outcomes'. First, we used the incidence of various post-acute respiratory sequelae after 30 days of SARS-CoV-2 as the 'primary outcome' for respiratory sequelae in post-acute COVID-19 conditions. Second, the 'primary outcome' for acute respiratory complication is the incidence of various respiratory diseases within 1 month following a diagnosis of SARS-CoV-2[44–47]. Outcomes were defined based on appropriate International Classification of Diseases 10th (ICD-10) codes for the new-onset of the specific diagnosis with at least one claim within one year. Post-acute respiratory sequelae were defined as chronic respiratory failure, pulmonary hypertension, sleep apnea, COPD, emphysema, asthma, pulmonary sarcoidosis, and interstitial lung disease[48]. In addition, acute respiratory complication was defined as pneumocystis pneumonia, aspergillosis pneumonia, pleural empyema, lung abscess, pneumothorax, acute respiratory failure, and pulmonary embolism (Table S25)[48].

## Covariates

Participant demographic data was sourced from the insurance database, which included age (20–39, 40–59, and ≥60 years), sex (male and female), household income percentiles (low [0–39], middle [40–79], high [80–100]), and region of residence (urban and rural)[34]. CCI (0, 1, and ≥2), histories of cardiovascular disease and chronic kidney disease, and previous use of medication for diabetes, hyperlipidemia, and hypertension were identified using the ICD-10 codes, combined with results of general health examinations and personal medical interview[34,37,41]. From the health examination, BMI (underweight [<18.5 kg/m²], normal [18.5–22.9 kg/m²], overweight [23.0–24.9 kg/m²], obese [≥25.0 kg/m²], and unknown), blood pressure (systolic blood pressure < 140 mmHg and diastolic blood pressure < 90 mmHg, systolic blood pressure ≥ 140 mmHg or diastolic blood pressure ≥ 90 mmHg, and unknown), fasting blood glucose (<100, ≥100 mg/dL, and unknown), serum total cholesterol (<200, 200–239, ≥240 mg/dL, and unknown), glomerular filtration rate (<60, 60–89, ≥90 mL/min/1.73 m², and unknown), smoking status (never, former, current smoker, and unknown), alcoholic drinks (<1, 1–2, 3–4, ≥5 days per week, and unknown), aerobic physical activity (sufficient [≥150 min/week of moderate-intensity activity or ≥75 min/week of vigorous-intensity activity or greater than an equivalent combination], insufficient, and unknown), and type of SARS-CoV-2 (original and delta) were obtained[39].

## Propensity score matching

To enhance the robustness and generalizability of our primary findings and balance baseline covariates, we employed exposure-driven propensity score matching. This approach compared individuals with SARS-CoV-2 infection to those without infection as a general population[49]. The propensity score was calculated by using a logistic regression model, adjusted for age (20–39, 40–59, and ≥60 years), sex (male and female), region of residence (urban and rural), history of cardiovascular and chronic disease, and medication use for diabetes, hyperlipidemia, and hypertension. Individuals were paired in a 1:5 ratio between the exposure group (SARS-CoV-2) and the non-exposure group. Through the prior procedures, we generated multi-to-one matched cohorts utilizing a 'greedy nearest-neighbour' algorithm, maintaining a caliper width of 0.001 standard deviations. The quality

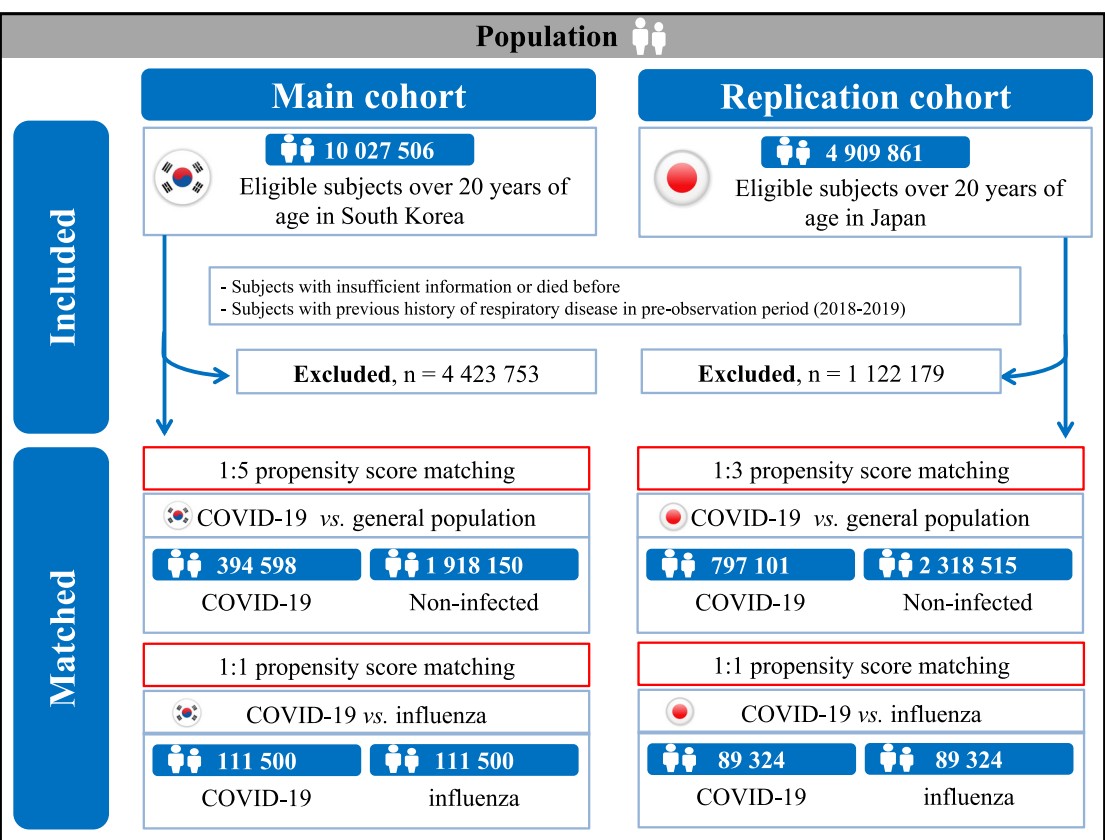

**Fig. 1 | Study population in the main cohort (South Korea) and replication cohort (Japan).**

of the match was evaluated through the SMD, with an SMD <0.1, indicating minimal imbalances between the groups[49]. In addition, to investigate the relative severity of COVID-19 compared to other infectious viral respiratory diseases, an influenza group within the general population was utilized as another control group, directly matching SARS-CoV-2 infections at a 1:1 ratio.

### Replication cohort in Japan
We employed the same definition of ICD-10 codes, exposure, and outcome assessments, general health examination, follow-up duration, and propensity score matching for the JMDC cohort (total $N = 4,909,861$) as we did with the main cohort. However, due to the lack of COVID-19 vaccination data in the JMDC cohort, we utilized this cohort primarily to validate the findings from the main cohort. Supplementary material was provided to address more detailed information about the validation cohort, which was caused by the different components and structures of the main cohort (Supplementary Material).

### Statistical analysis
To estimate the HRs with 95% CIs, we applied the Cox proportional hazards model. Therefore, we assigned the 'individual index date' by following the criterion. For the exposed group, it is the date of the first diagnosis of SARS-CoV-2. For the individuals in the non-exposed group, the index date was allocated to match the index date of the corresponding exposed case. This approach was implemented to mitigate immortal time bias, ensuring an equitable comparison between groups.

In the matched COVID-19 cohort, we conducted various statistical analyses, detailed in Table S26. These analyses involved respiratory disease (post-acute respiratory sequelae and acute respiratory complication) and its subtypes, number of vaccinations (without, 1 time, and ≥2 times), type of previous vaccination (mRNA, viral vector, and

both)[36], and strain of SARS-CoV-2 (original and delta). We further assessed the time attenuation effect of respiratory diseases after SARS-CoV-2 (<3, 3–6, and ≥6 months) to reduce reverse causation. To minimize the impact of potential confounders, the following variables were used for the adjusted model: age, sex, household income, region of residence, CCI score, obesity, blood pressure, fasting blood glucose, serum total cholesterol, glomerular filtration rate, smoking status, alcoholic drinks, aerobic physical activity, previous history of cardiovascular and chronic kidney diseases, history of medication use for diabetes mellitus, dyslipidemia, and hypertension, and strain of SARS-CoV-2 (original and delta). All statistical analyses were conducted using SAS (version 9.4; SAS Institute Inc., Cary, NC, USA)[50,51]. A two-sided $P$-value < 0.05 was considered statistically significant.

### Sensitivity analysis
Several sensitivity analyses were conducted to enhance the credibility of the manuscript and our primary analyses. First, to identify detection bias and validate the results of our cohort, we performed a negative control analysis by exploring the association between tympanic membrane perforation disease and SARS-CoV-2 infection (Table S27). Second, we conducted an exposure-driven propensity score matching analysis based on the claim record of individuals who tested positive for SARS-CoV-2 within 2 weeks following an RT-PCR test and those who did not, for a more sophisticated analysis of the association between SARS-CoV-2 infection and respiratory symptoms (Table S15). Third, stratification analyses were further employed in two matched cohorts stratified by factors, including sex, age group, household income, CCI, BMI status, smoking status, alcoholic drinks, region of residence, and aerobic physical activity. Fourth, to thoroughly evaluate the marginal risks and prevalence of acute and post-acute respiratory conditions post-COVID-19, we utilized the average treatment effect in the overlap method, calculating overlap-weighted hazard ratios (Tables S7–S9)[52–54].

## Patient and public involvement

The Korean government and JMDC anonymized patient data by excluding patient-related data such as personal identification numbers or names for confidentiality. While direct identification of individuals was rendered impossible due to the removal of names, all other pertinent data remained intact and accessible for our analyses. Research questions and outcome measures were autonomously determined without the intervention of individuals. The research design and implementation proceeded without external consultation. However, it can be extended to include contributions from other qualified public participants who are able to offer valuable insights into the research design, analysis, and interpretation. Upon request, the researchers intend to disseminate the results of this research to all research participants and relevant communities.

## Reporting summary

Further information on research design is available in the Nature Portfolio Reporting Summary linked to this article.

## Data availability

The datasets analyzed during the current study are available in the National Health Insurance Service in South Korea (https://nhiss.nhis.or.kr/bd/ab/bdaba000eng.do) and the JMDC in Japan (https://www.jmdc.co.jp/en/jmdc-claims-database/). This protects the confidentiality of the data and ensures that Information Governance is robust. Applications to access health data in South Korea are submitted to the National Health Insurance Service in South Korea. Information can be found at https://nhiss.nhis.or.kr/bd/ab/bdaba000eng.do. Applications to access health data in Japan are submitted to the JMDC, Japan. Information can be found at https://www.jmdc.co.jp/en/jmdc-claims-database/.

## Code availability

This study did not generate new or customized code/algorithm. Statistical analyses were performed using SAS (version 9.4; SAS Institute Inc., Cary, NC, USA) for analysis of big data. The codes utilized in the analysis are available from the corresponding author.

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

## Acknowledgements

This study used the database of the Korea Disease Control and Prevention Agency (KDCA) and the National Health Insurance Service (NHIS) for policy and academic research. The research number of this study is KDCA-NHIS-2022-1-632. This research was supported by grants from National Research Foundation of Korea (NRF) funded by the Korea government (MSIT; RS-2023-00248157; D.K.Y.), Ministry of Food and Drug Safety in 2024 (21153MFDS601; D.K.Y.), and the Korea Health Technology R&D Project through the Korea Health Industry Development Institute (KHIDI), funded by the Ministry of Health & Welfare, Republic of Korea (HV22C0233; D.K.Y.) and Cosmax BTI. We also thank Dong-Geol Lee (Cosmax BTI) and Seunghyun Kang (Cosmax BTI) for helpful advice and comments. The funders had no role in study design, data collection, data analysis, data interpretation, or writing of the report.

## Author contributions

D.K.Y. had full access to all of the data in the study and took responsibility for the integrity of the data and the accuracy of the data analysis. Y.C., H.J.K., J.P., M.L., S.K., A.K., L.S., M.S.K., M.R., H.L., J.K., and D.K.Y. approved the final version before submission. Study concept and design: Y.C., H.J.K., J.P., H.L., J.K., and D.K.Y.; Acquisition, analysis, or interpretation of data: Y.C., H.J.K., J.P., H.L., J.K., and D.K.Y.; Drafting of the manuscript: Y.C., H.J.K., J.P., H.L., J.K., and D.K.Y.; Critical revision of the manuscript for important intellectual content: Y.C., H.J.K., J.P., M.L., S.K., A.K., L.S., M.S.K., M.R., H.L., J.K., and D.K.Y.; Statistical analysis: Y.C., H.J.K., J.P., H.L., J.K., and D.K.Y.; Study supervision: D.K.Y. D.K.Y. is a guarantor for this study. The corresponding author attests that all listed authors meet authorship criteria and that no others meeting the criteria have been omitted.

## Competing interests

The authors declare no competing interests.

## Additional information

[1]Center for Digital Health, Medical Science research Institute, Kyung Hee University College of Medicine, Seoul, South Korea. [2]Department of Korean Medicine, Kyung Hee University College of Korean Medicine, Seoul, South Korea. [3]Department of Regulatory Science, Kyung Hee University, Seoul, South Korea. [4]Department of Family Medicine, Kyung Hee University Medical Center, Kyung Hee University College of Medicine, Seoul, South Korea. [5]Research and Development Unit, Parc Sanitari Sant Joan de Deu, Barcelona, Spain. [6]Centre for Health, Performance and Wellbeing, Anglia Ruskin University, Cambridge, UK. [7]Cardiovascular Disease Initiative, Broad Institute of MIT and Harvard, Cambridge, MA, USA. [8]CEReSS Health Service Research and Quality of Life Center, Assistance Publique-Hôpitaux de Marseille, Aix-Marseille University, Marseille, France. [9]Department of Physical Education and Sport Sciences, Faculty of Literature and Humanities, Vali-E-Asr University of Rafsanjan, Rafsanjan, Iran. [10]Department of Physical Education and Sport Sciences, Faculty of Literature and Human Sciences, Lorestan University, Khoramabad, Iran. [11]Division of Sleep Medicine, Harvard Medical School, Boston, MA, USA. [12]Department of Anesthesia, Critical Care and Pain Medicine, Massachusetts General Hospital, Boston, MA, USA. [13]Department of Pediatrics, Kyung Hee University Medical Center, Kyung Hee University College of Medicine, Seoul, South Korea. [14]These authors contributed equally: Yujin Choi, Hyeon Jin Kim, Jaeyu Park. [15]These authors jointly supervised this work: Hayeon Lee, Jiseung Kang, Dong Keon Yon. ✉e-mail: wwhy28@khu.ac.kr; wltmd1006@gmail.com; yonkkang@gmail.com

