## [Peer Review File · Nature Communications]

Acute and post-acute respiratory complications of SARS-CoV-2 infection: population-based cohort study in South Korea and JapanREVIEWER COMMENTS

Reviewer #1 (Remarks to the Author):

Lee and colleagues' manuscript investigates whether COVID-19 infection is associated with long-term post-acute respiratory sequelae compared to individuals without infection. They also compare this to influenza infection using data from a population-based database in South Korea and Japan. The authors further examine whether COVID-19 vaccinations provide protection against related respiratory outcomes. A strength of the manuscript is its use of a large-scale, population-based, binational database, encompassing over 10 million participants in the discovery cohort and over 12 million in the validation cohort. This sample appears to be nationally representative of both countries. The efforts and initiatives of the investigative team in collating claim data from two countries are commendable. However, several comments below may help to further strengthen the manuscript.

Major comments:

It is unclear at what time point the authors selected the non-infected group when they performed the propensity score matching in the COVID and influenza cohorts. The timing for COVID or influenza infection is clear, immediately following the infections. However, for individuals without infection (the matching set), the chosen timing is unclear. The authors need to provide more details for clarification. The timing of the uninfected group is crucial to ensure comparability between the infected and uninfected groups.

The authors compare the hazards of acute and long-term respiratory sequelae between their COVID-19 and flu patients and those without infection (Table 2). These analyses offer minimal scientific insight because patients with respiratory infections certainly have a higher risk of short- or long-term respiratory symptoms post-infection.

The authors suggest that COVID-19 patients may have a higher risk of post-acute respiratory complications compared to those post-flu infection. However, this conclusion is not fully supported by the results in Table 2. The risk estimates for long-term sequelae in the COVID-19 and influenza cohorts are somewhat similar in the Discovery cohort, with some differences in the validation cohorts. Overall, there is no direct comparison in the SAME models to evaluate the risk of post-acute sequelae by comparing COVID infection to flu.

Similarly, the authors use data from Korea as a discovery cohort and from Japan as a validation cohort. However, the estimates differ significantly between these two countries, making it difficult to quantify the exact risk associated with long-term sequelae. Both countries have major differences in their healthcare systems, treatment guidelines for COVID-19, and data acquisition methods, which could significantly impact risk estimates. The authors might consider presenting both datasets on equal terms, rather than as discovery versus validation cohorts, which are terms more commonly used in machine learning approaches.

One significant contribution of this paper is the results reported in Table 3, where the author details individual conditions of long-term sequelae, potentially contributing to the definition of long-COVID conditions. This table could be strengthened by providing the marginal predicted prevalence for each condition, aiding in understanding the risk and prevalence of these conditions post COVID.

Table 4 attempts to evaluate the risk of acute and long-term sequelae by comparing vaccination status and COVID severity. The reference group seems to be the non-infected control. Similar to previous comments, it is unclear how the non-infected control could have acute and long-term sequelae respiratory complications and at what time point the authors are referring to.

Additionally, vaccination is initially protective against COVID-19 infection, and those who had COVID-19 post-vaccination represent breakthrough infections. The authors seem to conflate multiple groups (i.e., vaccine status among people without COVID-19). To properly assess the protective effect of vaccination against long COVID, the authors might consider focusing on people with breakthrough infections, comparing their vaccination status and acute/long-term COVID outcomes.

The authors report a large number of non-COVID or non-flu comparison groups in their analyses. Given that this is claims data, can the author address how they dealt with the issue of underdiagnosis of COVID-19 and flu in their control (non-infected) group? Misclassification of COVID/flu cases to non-infected cases is highly likely given the nature of claims data.

Another limitation is the timing of the data. The study period appears to be pre-Omicron, which is less relevant to current times. Updating the dataset to a more recent period would substantially strengthen the paper.

Minor:

There are minor editorial errors throughout the paper and in the tables. For example, Table 2 incorrectly reports "Ratio of HR," which is redundant as hazard ratio already includes the term ratio. Thorough editing and proofreading of the manuscript may enhance its quality.

Reviewer #2 (Remarks to the Author):

I enjoyed reviewing this manuscript. It is well written and the analysis appears to be robust. I have some concerns that I would like to be addressed. Most important is the potential for sample bias.

Comments for authors:

- Can you be sure that participants did not have the other infection (i.e. that people with SARS-CoV-2 infection never had influenza and vice versa)?
- The statement "the majority of patients with SARS-CoV-2 infection report persistent, long-lasting comorbidities after infection termed long COVID" is untrue. Please correct it. The balance of evidence suggests that long-COVID prevalence following symptomatic SARS-CoV-2 infection is between 5 and 10%.
- Please elaborate on the meaning of claim-based cohort, and any potential bias that may be introduced. How representative are these cohorts of the populations of South Korea and Japan?
- Be more specific about the time period of infection and follow-up. Did you consider including dominant variant as a covariate in analysis?
- Figure S1 requires clarification. Are the number of exclusions the total number of participants excluded in each group (A-D) for all reasons (insufficient information, death, previous history of respiratory disease)? Please provide numbers excluded for each reason. The proportions excluded from the Japanese cohort are far higher than those excluded from the South Korean cohort. Please comment on possible reasons for this. Could there be bias due to under or over-recording of disease?
- Table S1 is unnecessary and out of place. The content should be moved to the text in methods, results and discussion where relevant.
- Table S3 is referred to out of context at the bottom of page 10.
- Table S4 is missing urban/rural residence for validation cohort.
- Table 1 reports substantial missingness. How was this dealt with?
- Tables 2 and 3 report very large differences in effect size between the discovery and validation cohorts. Please comment on this and its implications.
- How can the risk of acute respiratory complication remain after 6 months (page 16)? By definition that is not acute.
- It needs to be made more clear that people with SARS-CoV-2 infection and influenza were not directly compared. Rather, each group was compared separately with uninfected individuals. The language used in interpretation should reflect this.
- The population of South Korea is >50 million people. Therefore, how can you claim that the discovery cohort of ~10 million covers 98% of the population?
- The methods section implies that vaccination status including type of vaccine is recorded at an individual level. However, on page 20 is the sentence "Third, this study illustrated the method of mixed vaccination in general, not showing the specific vaccines used on each person." Please clarify whether vaccination is recorded for each individual or not.
- I do not expect you to act on this advice posthoc. However, public and patient involvement is beneficial in the secondary analysis and interpretation of data. Their insights can be very useful and there are many people with long-COVID who are interested and engaged in research.

Response Letter

The following table summarizes the information about the revised manuscript versus the previous version.

	Original version	Revised version
Main manuscript page	45p	48p
Supplement page	23p	87p
Supplement Figure	1	2
Supplement Table	7	25

Table of Contents

1. Reviewer #1	3
2. Reviewer #2	119
End	265

Reviewer #1**Comment 1.**

Lee and colleagues' manuscript investigates whether COVID-19 infection is associated with long-term post-acute respiratory sequelae compared to individuals without infection. They also compare this to influenza infection using data from a population-based database in South Korea and Japan. The authors further examine whether COVID-19 vaccinations provide protection against related respiratory outcomes. A strength of the manuscript is its use of a large-scale, population-based, binational database, encompassing over 10 million participants in the discovery cohort and over 12 million in the validation cohort. This sample appears to be nationally representative of both countries. The efforts and initiatives of the investigative team in collating claim data from two countries are commendable. However, several comments below may help to further strengthen the manuscript.

Response:

Thank you for your thoughtful review of our paper. As you mentioned, our study can contribute to investigating whether COVID-19 infection is associated with post-acute respiratory sequelae compared to uninfected groups by using data from population-based binational cohorts from South Korea and Japan. We are grateful for the recognition of our research efforts. We sincerely appreciate your invaluable comments and the opportunity to refine our manuscript. Your feedback has been instrumental in guiding our revisions. We have carefully considered your suggestions and have enhanced our manuscript as below. We hope these refinements align with your expectations and look forward to a favorable response.

Comment 2.

It is unclear at what time point the authors selected the non-infected group when they performed the propensity score matching in the COVID and influenza cohorts. The timing for COVID or influenza infection is clear, immediately following the infections. However, for individuals without infection (the matching set), the chosen timing is unclear. The authors need to provide more details for clarification. The timing of the uninfected group is crucial to ensure comparability between the infected and uninfected groups.

Response:

Thank you for mentioning such an important point regarding the index time for individuals without infection. The criteria of the exposure of this research are participants who were diagnosed with COVID-19 from Jan 1, 2020, to Dec 31, 2021, and individual index dates are established as the first diagnosis of SARS-CoV-2 or influenza infection for the infected group, as you mentioned. The index date of the individuals in the uninfected group was assigned using the propensity score matching methodology procedure. The propensity score was calculated through a multivariate logistic regression model, which was the criteria of multi-to-one matching (1:5, South Korea; 1:3, Japan) between the infected group and the uninfected group. The index dates of the individuals in the uninfected group were assigned according to the index date of the infected individual they were matched to. We apologize for the lack of information on the index date for uninfected individuals. We added explanations about the index date of the uninfected patients in the Methods of our manuscript. We also added Figure S2 in the Supplementary Materials to clarify the determination of index dates. The changes in texts are shown below. Thank you for giving us an opportunity to clarify the details and enhance the credibility of our manuscript.

Changes in text:**Methods / Statistical analysis**

Therefore, we assigned the ‘individual index date’ by following the criterion. For the exposed group, it is the date of the first diagnosis of SARS-CoV-2. For the individuals in the non-exposed group, the index date was allocated to match the index date of the corresponding exposed case. This approach was implemented to mitigate immortal time bias, ensuring an equitable comparison between groups.

Figure S2. Study flow of cohorts

Comment 3.

The authors compare the hazards of acute and long-term respiratory sequelae between their COVID-19 and flu patients and those without infection (Table 2). These analyses offer minimal scientific insight because patients with respiratory infections certainly have a higher risk of short- or long-term respiratory symptoms post-infection.

The authors suggest that COVID-19 patients may have a higher risk of post-acute respiratory complications compared to those post-flu infection. However, this conclusion is not fully supported by the results in Table 2. The risk estimates for long-term sequelae in the COVID-19 and influenza cohorts are somewhat similar in the Discovery cohort, with some differences in the validation cohorts. Overall, there is no direct comparison in the SAME models to evaluate the risk of post-acute sequelae by comparing COVID infection to flu.

Response:

Thank you for your insightful feedback. We agree that clear scientific insight into the differential risk of post-acute respiratory sequelae following SARS-CoV-2 and influenza infection is important. We deeply acknowledge your comment that our initial analysis provided a limited perspective by not directly comparing risks of the post-acute respiratory sequelae between participants with SARS-CoV-2 infection and participants with influenza. Therefore, we have conducted a new matching of the entire dataset and re-analysis. To aid your understanding, we present our revised study design as shown in the Venn diagram below.

(only for reviewer)

PS matching cohort

- A: participants infected with only SARS-CoV-2
- B: participants only infected with influenza from January 2020 to December 2021
- C: participants infected with SARS-CoV-2 and were infected with influenza after being diagnosed with COVID-19
- D: participants without SARS-CoV-2 or influenza infection

COVID-19 vs. general population: A, C vs. B, D

COVID-19 vs. influenza: A, C vs. B

In response to your other comments and those from Reviewer #2 regarding control groups, we divided the re-analysis into two comparisons. First, we investigated the risk of respiratory complications following SARS-CoV-2 infection compared to the general population, which is a key finding of this study, presented in the main table. We defined the general population as individuals not infected with SARS-CoV-2, which may include patients with influenza. We conducted analyses of the hazard ratio (HR) of COVID-19, using the general population as a reference, as shown in Tables 1-5. Second, to incorporate your suggestion, instead of an indirect comparison, we conducted a direct comparison through 1:1 exposure-driven propensity score matching to compare COVID-19 with influenza. To examine the relative severity of COVID-19 in comparison with another contagious viral respiratory disease, the patient group with influenza infection excluded patients with both SARS-CoV-2 and influenza infections. We conducted analyses of the HR of COVID-19, using patients with influenza as a reference. The results of these analyses have been added to supplementary Tables S11-S14 and S16. Furthermore, we have also revised the Methods and Results sections accordingly. For your information, the overall findings of our study through re-analysis are

consistent with the findings of the original version. Once again, we sincerely appreciate the opportunity to improve our manuscript.

Changes in text:

Abstract

- We aimed to identify the risk of acute respiratory complications or long-term post-acute respiratory sequelae in **long COVID**.
- After exposure-driven 1:5 propensity score matching, we found that the risk of acute respiratory complication or long-term respiratory sequelae is significantly increased in people with SARS-CoV-2 infection **compared to the general population (acute respiratory complication: HR, 8.06 [95% CI, 6.92-9.38]; long-term respiratory sequelae: HR, 1.68 [95% CI, 1.62-1.75])**, and the risk increased with increasing COVID-19 severity. The SARS-CoV-2 infection induced a significantly increased risk for several acute respiratory complications **(aspergillosis pneumonia, pneumothorax, acute respiratory failure, and pulmonary embolism)** and long-term respiratory sequelae **(chronic respiratory failure, COPD, emphysema, asthma, and interstitial lung disease)**.
- Through this large-scale, binational, and population-based cohort study, acute or long-term respiratory sequelae in long COVID were observed **compared to the general population**.

Methods/Exposures

- **To examine the relative severity of COVID-19 in comparison with another contagious viral respiratory disease, additional exposure to influenza infection was defined. It refers to cases diagnosed through an RT-PCR assay or antigen test on nasal and pharyngeal swabs during the observation period. For individuals infected with both SARS-CoV-2 and influenza, it includes instances of influenza infection developing after the SARS-CoV-2 infection.**

Methods/Propensity score matching

- To enhance the robustness and generalizability of our primary findings and balance baseline covariates, we employed exposure-driven propensity score matching. **This approach compared individuals with SARS-CoV-2 infection to those without infection as a general population.²¹ The propensity score was calculated by using a logistic regression model, adjusted for age (20–39, 40–59, and ≥60 years), sex (male and female), region of residence (urban and rural), history of cardiovascular and chronic disease, and medication use for diabetes, hyperlipidemia, and**

hypertension. Individuals were paired in 1:5 ratio between the exposure group (SARS-CoV-2) and the non-exposure group. Through the prior procedures, we generated multi-to-one matched cohorts utilized a ‘greedy nearest-neighbor’ algorithm, maintaining a caliper width of 0.001 standard deviations. The quality of the match was evaluated through the standardized mean differences (SMD), with an SMD less than 0.1 signifying minimal imbalances between the groups.²² In addition, to investigate the relative severity of COVID-19 compared to other infectious viral respiratory diseases, an influenza group within the general population was utilized as another control group, directly matching SARS-CoV-2 infections at a 1:1 ratio.

Results

- **Table 1** shows the baseline characteristics of 1:5 propensity score matched cohort of South Korea. After 1:5 propensity score matching based on SARS-CoV-2 infection, we identified 82.9% (1,918,150/2,312,748) of participants without SARS-CoV-2 infection and 17.1% (394,598/2,312,748) of participants with SARS-CoV-2 infection, respectively.

- In the 1:3 propensity score matched replication cohort, 74.4% (2,318,505/3,115,606) of participants without SARS-CoV-2 infection and 25.6% (797,101/3,115,606) of participants with SARS-CoV-2 infection were included in our final analyses (**Table S10**).

- In the main and replication cohorts, individuals with SARS-CoV-2 infection had a higher adjusted HR for long-term post-acute respiratory sequelae compared to the general population (main: HR, 1.68 [95% CI, 1.62-1.75]; replication: HR, 3.32 [95% CI, 3.27-3.37]) in **Table 2**. Furthermore, patients with SARS-CoV-2 infection had an increased risk for acute respiratory complication compared to non-infected controls (main: HR, 8.06 [95% CI, 6.92-9.38]; replication: HR, 4.17 [95% CI, 3.90-4.45]). When directly comparing the risk for acute respiratory complication between SARS-CoV-2 and influenza infections, SARS-CoV-2 infection was significantly associated with an increased risk (main: HR, 4.32 [95% CI, 2.73-6.83]; replication: HR, 6.51 [95% CI, 5.38-7.87]) in **Tables S11-S13**.

- Relative to the general population, patients with SARS-CoV-2 infection had significantly increased risk for several subtypes of long-term respiratory sequelae, including chronic respiratory failure (main: HR, 8.92 [95% CI, 4.92-16.17]; replication: HR, 7.55 [95% CI, 6.35-8.97]), COPD, emphysema, asthma, pulmonary sarcoidosis, and interstitial lung disease (main: HR, 10.38 [95% CI, 8.75-12.31]; replication: HR, 4.75 [95% CI, 4.54-4.97]) in **Table 3**. Notably, the risk for acute respiratory complication, including aspergillosis pneumonia (main: HR, 6.85 [95% CI, 3.48-13.50]; replication: HR, 4.97 [95% CI, 4.26-5.79]), pneumothorax,

acute respiratory failure (main: HR, 112.04 [95% CI, 64.00-196.16]; replication: HR, 6.49 [95% CI, 6.32-6.65]) showed an increase in patients with SARS-CoV-2 infection compared to the general population. This tendency of increased risk for several subtypes of respiratory diseases was also shown when compared to patients with influenza infection (Table S14).

- The risk of acute respiratory complication showed decreasing trends according to the number of SARS-CoV-2 vaccinations from individuals after once receiving vaccination (HR, 0.51 [95% CI, 0.38-0.68]) to those with two or more vaccinations (HR, 0.24 [95% CI, 0.19-0.30]). Interestingly, mixed types of vaccination showed the lowest risk of developing long-term respiratory sequelae of all SARS-CoV-2 vaccination methods (HR, 0.18 [95% CI, 0.08-0.38]). The risks of acute respiratory complication were higher in patients with moderate to severe COVID-19 symptoms (HR, 39.54 [95% CI, 33.54-46.62]). Both the original strain and the delta variant of SARS-CoV-2 were shown to have a higher risk of acute respiratory complications (original strain: HR, 9.21 [95% CI, 7.19-11.80]; delta strain: HR, 7.44 [95% CI, 6.13-9.03]). In addition, the risk for long-term post-acute respiratory sequelae also exhibited a similar pattern (Tables 4 and S15).

- Table 5 shows the risk of developing acute respiratory complication or long-term post-acute respiratory sequelae based on how long it has been since the participant was infected with SARS-CoV-2 compared to the general population.

- The first three months after infection with SARS-CoV-2 had the highest risk of developing long-term respiratory sequelae (main: HR, 2.51 [95% CI, 2.38-2.64]; replication: HR, 4.40 [95% CI, 4.30-4.51]). With increasing duration post-SARS-CoV-2 infection, the risk of long-term post-acute respiratory sequelae significantly decreased, but the risk remained even after 6 months (main: HR, 1.10 [95% CI, 1.01-1.19]; replication: HR, 2.67 [95% CI, 2.61-2.73]). HR of time attenuation effect after SARS-CoV-2 infection showed significance compared to influenza infection likewise (Table S16).

Discussion/Findings of this study

- First, the risk of acute respiratory complication or long-term post-acute respiratory sequelae is significantly increased in participants with SARS-CoV-2 infection, compared to the general population. Second, SARS-CoV-2 infection induced a significantly increased risk for several specific long-term respiratory sequelae, including chronic respiratory failure, COPD, emphysema, asthma, and interstitial lung disease, compared to the general. In addition, several acute respiratory complications, including aspergillosis pneumonia, pneumothorax, acute

respiratory failure, and pulmonary embolism, also depicted a notable increase in risk after SARS-CoV-2 infection, compared to the general population.

Table S2. Statistical analyses and justification

No. Statistical analysis (1 to 9)	Cohort	Justification
1. Incident respiratory diseases in COVID-19 versus general population	1. Main cohort after 1:5 propensity score matching 2. Replication cohort 1:3 propensity score matching	- Main results - To investigate the association of acute respiratory complication and long-term respiratory sequelae and long COVID-19. - Matching covariates: age (20–39, 40–59, and ≥60 years); sex; household income (low income, middle income, and high income); region of residence (urban and rural); previous history of cardiovascular disease, chronic kidney disease; and history of medication use for diabetes mellitus, dyslipidemia, and hypertension. - Model (main cohort): adjusting covariates for age (20–39, 40–59, and ≥60 years); sex, household income (low income, middle income, and high income); region of residence (urban and rural); CCI score (0, 1, and ≥2); obesity (underweight [<18.5 kg/m ²], normal [18.5–22.9 kg/m ²], overweight [23.0–24.9 kg/m ²], obese [≥25.0 kg/m ²], and unknown); blood pressure (systolic blood pressure <140 mmHg and diastolic blood pressure <90 mmHg, systolic blood pressure ≥140 mmHg or diastolic blood pressure ≥90 mmHg, and unknown); fasting blood glucose (<100 , ≥100 mg/dL, and unknown); serum total cholesterol (<200 , 200–239, ≥240 mg/dL, and unknown); glomerular filtration rate (<60 , 60–89, ≥90 mL/min/1.73 m ² , and unknown); smoking status (never, former, current smoker, and unknown); alcoholic drinks (<1 , 1–2, 3–4, ≥5 days per week, and unknown); aerobic physical activity (sufficient, insufficient, and unknown); previous history of cardiovascular disease, and chronic kidney disease; history of medication use for diabetes mellitus, dyslipidemia, and hypertension; and strain of SARS-CoV-2 (original and delta). - Model (replication cohort): adjusting covariates for age (20–39, 40–59, and ≥60 years); sex; insurance status (insured and dependent); CCI score (0, 1, and ≥2); body mass index (underweight [<18.5 kg/m ²], normal [18.5–22.9 kg/m ²], overweight [23.0–25.0 kg/m ²], obese [≥25.0 kg/m ²], and unknown); blood pressure (systolic blood pressure <140 mmHg and diastolic blood pressure <90 mmHg, systolic blood pressure

		≥140 mmHg or diastolic blood pressure ≥90 mmHg, and unknown); fasting blood glucose (<100, ≥100 mg/dL, and unknown); serum total cholesterol (<200, 200–239, ≥240 mg/dL, and unknown); glomerular filtration rate (<60, 60–89, ≥90 mL/min/1.73 m², and unknown); smoking status (non- and current smoker, and unknown); alcoholic drinks (rarely, sometimes, everyday, and unknown); aerobic physical activity (sufficient, insufficient, and unknown); previous history of cardiovascular disease, and chronic kidney disease; history of medication use for diabetes mellitus, dyslipidemia, and hypertension; and strain of SARS-CoV-2 (original and delta).
2. Subgroup analysis of the risk of several respiratory diseases after SARS-CoV-2 infection	1. Main cohort after 1:5 propensity score matching 2. Replication cohort 1:3 propensity score matching	- Main results - To investigate the likelihood of respiratory diseases (long-term post-acute respiratory sequelae and acute respiratory complication) after SARS-CoV-2 infection, compared with general population. - Long-term post-acute respiratory sequelae: chronic respiratory failure, pulmonary hypertension, sleep apnea, chronic obstructive pulmonary disease (COPD), emphysema, asthma, pulmonary sarcoidosis, and interstitial lung disease. - Acute respiratory complication: pneumocystis pneumonia, aspergillosis pneumonia, pleural empyema, lung abscess, pneumothorax, acute respiratory failure, and pulmonary embolism.
3. Subgroup analysis of the risk of incident respiratory diseases after SARS-CoV-2 according to the SARS-CoV-2 vaccinations, severity of COVID-19, and strain type	Main cohort after 1:5 propensity score matching	- Main results - To investigate the association of subsequent all-cause respiratory diseases following COVID-19, stratified by vaccination dose (once and twice or more), and type of vaccinations (vector, mRNA, and mix), severity of COVID-19 (mild and moderate-to-severe), and strain of SARS-CoV-2 (original and delta).
4. Attenuation time effect of long-term post-acute respiratory sequelae development after COVID-19	1. Main cohort after 1:5 propensity score matching 2. Replication cohort 1:3 propensity score matching	- Main results - To investigate the time attenuation effect of long-term post-acute respiratory sequelae development after SARS-CoV-2 infection (<3, 3 to 6, and ≥6 months).
5. Incident respiratory diseases in COVID-19 versus contemporary control (influenza)	1. Main cohort after 1:1 propensity score matching 2. Replication cohort 1:1 propensity score matching	- Contemporary control results, consistent with the methodology employed in statistical analysis no. 1, 2, and 4.

6. Negative control analysis of the risk of incident tympanic membrane perforation disease.	1. Main cohort after 1:5 propensity score matching 2. Replication cohort 1:3 propensity score matching	- To verify the validity of our findings and identify potential misclassification bias, we conducted a negative control analysis of tympanic membrane perforation disease following the COVID-19 diagnosis.
7. Incident respiratory diseases in positive for COVID-19 versus negative for COVID-19 after the PCR test	Positive for COVID-19 versus negative for COVID-19 after the PCR test after 1:5 propensity score matching in main cohort	- To conduct more sophisticated analysis of the association between SARS-CoV-2 infection and respiratory disease - The cohort was based on the claim record of individuals who tested positive for SARS-CoV-2 within two weeks following an RT-PCR test and those who did not.
8. Subgroup analysis of the risk of respiratory diseases after SARS-CoV-2 infection in overlap-weighted cohort	1. Main cohort after overlap-weighted cohort 2. Replication cohort after overlap-weighted cohort	- To assess the risks and prevalence of respiratory conditions following COVID-19, the average treatment effect in the overlap (ATO) method was used, with overlap-weighted hazard ratios calculated for precision. - Matching covariates: age (20–39, 40–59, and ≥60 years); sex; household income (low income, middle income, and high income); region of residence (urban and rural); previous history of cardiovascular disease, chronic kidney disease; and history of medication use for diabetes mellitus, dyslipidemia, and hypertension. - Model: adjusting covariates for age (20–39, 40–59, and ≥60 years); sex, household income (low income, middle income, and high income); region of residence (urban and rural); CCI score (0, 1, and ≥2); obesity (underweight [$<18.5 \text{ kg/m}^2$], normal [$18.5\text{--}22.9 \text{ kg/m}^2$]; overweight [$23.0\text{--}24.9 \text{ kg/m}^2$], obese [$\geq 25.0 \text{ kg/m}^2$], and unknown); blood pressure (systolic blood pressure $<140 \text{ mmHg}$ and diastolic blood pressure $<90 \text{ mmHg}$, systolic blood pressure $\geq 140 \text{ mmHg}$ or diastolic blood pressure $\geq 90 \text{ mmHg}$, and unknown); fasting blood glucose (<100 , $\geq 100 \text{ mg/dL}$, and unknown); serum total cholesterol (<200 , $200\text{--}239$, $\geq 240 \text{ mg/dL}$, and unknown); glomerular filtration rate (<60 , $60\text{--}89$, $\geq 90 \text{ mL/min/1.73 m}^2$, and unknown); smoking status (never, former, current smoker, and unknown); alcoholic drinks (<1 , $1\text{--}2$, $3\text{--}4$, ≥ 5 days per week, and unknown); aerobic physical activity (sufficient, insufficient, and unknown); previous history of cardiovascular disease, and chronic kidney disease; history of medication use for diabetes mellitus, dyslipidemia, and hypertension; and strain of SARS-CoV-2 (original and delta).

9. Stratification analysis of the risk of respiratory diseases and its subtypes development after COVID-19	1. Main cohort after 1:5 propensity score matching 2. Replication cohort 1:3 propensity score matching 3. Main cohort after 1:1 propensity score matching 4. Replication cohort 1:1 propensity score matching	- To investigate unexpected mediated effects by sex, age, region of residence, household income, body mass index, Charlson comorbidity index, smoking status, alcohol use, aerobic physical activity, and strain of SARS-CoV-2.
--	--	---

CCI, Charlson comorbidity index; PCR, polymerase chain reaction; SARS-CoV-2, severe acute respiratory syndrome coronavirus 2.

Table 1. Baseline characteristics for 1:5 propensity score–matched cohort (COVID-19 vs. general population) in South Korea (main)

Characteristic	COVID-19 vs. general population (n=2,312,748)		SMD*
	COVID-19 (n=394,598)	General population (n=1,918,150)	
Mean age (SD), y	47.5 (16.8)	46.8 (14.3)	0.046
Age, n (%)			0.026
20–39 y	143,273 (36.3)	702,322 (36.6)	
40–59 y	145,169 (36.8)	709,343 (37.0)	
≥60 y	106,156 (26.9)	506,485 (26.4)	
Sex, n (%)			0.006
Male	205,058 (52.0)	997,982 (52.0)	
Female	189,540 (48.0)	920,168 (48.0)	
Region of residence, n (%)			<0.001
Urban	213,052 (54.0)	1,035,261 (54.0)	
Rural	181,546 (46.0)	882,889 (46.0)	
Medical history, n (%)			
Cardiovascular disease	59,947 (15.2)	286,623 (14.9)	0.009
Chronic kidney disease	18,963 (4.8)	88,687 (4.6)	0.005
Medication use for diabetes	71,625 (18.2)	342,656 (17.9)	0.008
Medication use for hyperlipidemia	59,947 (15.2)	286,623 (14.9)	0.007
Medication use for hypertension	32,402 (8.2)	154,789 (8.1)	0.005
Unmatching covariates, n (%)[†]			
Charlson Comorbidity Index score			0.230
0	346,579 (87.8)	1,806,906 (94.2)	
1	30,584 (7.8)	60,556 (3.2)	
≥2	17,435 (4.4)	50,688 (2.6)	

Household income			<0.001
Low (0th–39th percentile)	182,632 (46.3)	887,593 (46.3)	
Middle (40th –79th percentile)	140,084 (35.5)	681,180 (35.5)	
High (80th–100th percentile)	71,882 (18.2)	349,377 (18.2)	
Body mass index			1.212
Underweight (<18.5 kg/m ²)	6875 (1.7)	67,970 (3.5)	
Normal (18.5-22.9 kg/m ²)	77,779 (19.7)	685,687 (35.8)	
Overweight (23.0-24.9 kg/m ²)	54,427 (13.8)	440,197 (23.0)	
Obese (≥25.0 kg/m ²)	95,082 (24.1)	724,074 (37.8)	
Unknown	160,435 (40.7)	222 (0.012)	
Blood pressure			1.157
SBP <140 mmHg and DBP <90 mmHg	199,342 (50.5)	1,673,777 (87.3)	
SBP ≥140mmHg or DBP ≥90 mmHg	33,711 (8.5)	240,991 (12.6)	
Unknown	161,545 (40.9)	3382 (0.2)	
Fasting blood glucose			1.179
<100 mg/dL	140,656 (35.7)	1,189,021 (62.0)	
≥100 mg/dL	92,374 (23.4)	725,682 (37.8)	
Unknown	161568 (40.9)	3447 (0.2)	
Serum total cholesterol			0.416
<200 mg/dL	67,886 (17.2)	525,684 (27.4)	
200 to 239 mg/dL	39,508 (10.0)	316,959 (16.5)	
≥240 mg/dL	16,860 (4.3)	134,405 (7.0)	
Unknown	270,344 (68.5)	941,102 (49.1)	
Glomerular filtration rate			1.180
<60 mL/min/1.73 m ²	8332 (2.1)	52,421 (2.7)	
60 to 89 mL/min/1.73 m ²	102,604 (26.0)	812,214 (42.3)	
≥90 mL/min/1.73 m ²	121,914 (30.9)	1,048,318 (54.7)	

Unknown	161,748 (41.0)	5197 (0.3)	
Smoking status			1.191
Never	154,105 (39.1)	1,214,221 (63.3)	
Former	42,814 (10.9)	299,027 (15.6)	
Current	37,263 (9.4)	404,323 (21.1)	
Unknown	160,416 (40.7)	579 (0.030)	
Alcohol consumption			1.156
<1 day/week	136,870 (34.7)	1,143,488 (59.6)	
1 to 2 days/week	66,311 (16.8)	545,090 (28.4)	
3 to 4 days/week	23,113 (5.9)	173,245 (9.0)	
≥5 days/week	7893 (2.0)	55,687 (2.9)	
Unknown	160,411 (40.7)	640 (0.034)	
Aerobic physical activity			1.179
Insufficient	118,792 (30.1)	959,088 (50.0)	
Sufficient	115,321 (29.2)	958,194 (50.0)	
Unknown	160,485 (40.7)	868 (0.1)	
Strain of SARS-CoV-2			0.004
Original	121,521 (30.8)	594,134 (31.0)	
Delta	273,077 (69.2)	1,324,016 (69.0)	

DBP, diastolic blood pressure; SARS-CoV-2, severe acute respiratory syndrome coronavirus 2; SBP, systolic blood pressure; SD, standard deviation; SMD, standardized mean difference.

* An SMD <0.1 indicates no significant imbalance. All SMDs were <0.100 in the propensity score–matched cohorts.

† Unmatched covariates were included as adjustment factors in statistical analyses.

Table 2. Hazard ratio (95% CI) for the **long-term post-acute respiratory sequelae** or **short-term acute respiratory complication** after SARS-CoV-2 infection in the propensity score-matched cohorts of South Korea (main) and Japan (replication)

Cohort	South Korea			Japan		
	COVID-19 vs. general population (n=2,312,748)			COVID-19 vs. general population (n=3,115,606)		
	Events, n (%)	HR (95% CI)		Events, n (%)	HR (95% CI)	
Model 1*		Model 2†	Model 3*		Model 4‡	
Long-term post-acute respiratory sequelae						
Comparators (general population or patients with influenza)	16,122 (0.84)	1.0 (reference)	1.0 (reference)	35300 (1.52)	1.0 (reference)	1.0 (reference)
Patients with COVID-19	5292 (1.34)	1.64 (1.59-1.69)	1.68 (1.62-1.75)	41074 (5.15)	3.50 (3.45-3.55)	3.32 (3.27-3.37)
Acute respiratory complication						
Comparators	331 (0.017)	1.0 (reference)	1.0 (reference)	1468 (0.06)	1.0 (reference)	1.0 (reference)
Patients with COVID-19	618 (0.16)	9.70 (8.46-11.11)	8.06 (6.92-9.38)	2304 (0.29)	4.60 (4.31-4.91)	4.17 (3.90-4.45)

CCI, Charlson comorbidity index; CI, confidence interval; HR, hazard ratio.

The data in bold indicate significant differences ($P < 0.05$).

***Model 1 and 3:** Adjusted for age (20–39, 40–59, and ≥ 60 years) and sex.

†**Model 2:** Adjusted for age (20–39, 40–59, and ≥ 60 years); sex, household income (low income, middle income, and high income); region of residence (urban and rural); CCI score (0, 1, and ≥ 2); obesity (underweight [$< 18.5 \text{ kg/m}^2$], normal [$18.5\text{--}22.9 \text{ kg/m}^2$]; overweight [$23.0\text{--}24.9 \text{ kg/m}^2$], obese [$\geq 25.0 \text{ kg/m}^2$], and unknown); blood pressure (systolic blood pressure $< 140 \text{ mmHg}$ and diastolic blood pressure $< 90 \text{ mmHg}$, systolic blood pressure $\geq 140 \text{ mmHg}$ or diastolic blood pressure $\geq 90 \text{ mmHg}$, and unknown); fasting blood glucose (< 100 , $\geq 100 \text{ mg/dL}$, and unknown); serum total cholesterol (< 200 , $200\text{--}239$, $\geq 240 \text{ mg/dL}$, and unknown); glomerular filtration rate (< 60 , $60\text{--}89$, $\geq 90 \text{ mL/min/1.73 m}^2$, and unknown); smoking status (never, former, current smoker, and unknown); alcoholic drinks (< 1 , $1\text{--}2$, $3\text{--}4$, ≥ 5 days per week, and unknown); aerobic physical activity (sufficient, insufficient, and unknown); previous history of cardiovascular disease, and chronic kidney disease; history of medication use for diabetes mellitus, dyslipidemia, and hypertension; and strain of SARS-CoV-2 (original and delta).

‡**Model 4:** Adjusted for age (20–39, 40–59, and ≥ 60 years); sex; insurance status (insured and dependent); CCI score (0, 1, and ≥ 2); body mass index (underweight [$< 18.5 \text{ kg/m}^2$], normal [$18.5\text{--}22.9 \text{ kg/m}^2$], overweight [$23.0\text{--}25.0 \text{ kg/m}^2$], obese [$\geq 25.0 \text{ kg/m}^2$], and unknown); blood pressure (systolic blood pressure $< 140 \text{ mmHg}$ and diastolic blood pressure $< 90 \text{ mmHg}$, systolic blood pressure $\geq 140 \text{ mmHg}$ or diastolic blood

pressure ≥ 90 mmHg, and unknown); fasting blood glucose (< 100 , ≥ 100 mg/dL, and unknown); serum total cholesterol (< 200 , 200–239, ≥ 240 mg/dL, and unknown); glomerular filtration rate (< 60 , 60–89, ≥ 90 mL/min/1.73 m², and unknown); smoking status (non- and current smoker, and unknown); alcoholic drinks (rarely, sometimes, everyday, and unknown); aerobic physical activity (sufficient, insufficient, and unknown); previous history of cardiovascular disease, and chronic kidney disease; history of medication use for diabetes mellitus, dyslipidemia, and hypertension; and strain of SARS-CoV-2 (original and delta).

Table 3. HR (95% CI) for the long-term post-acute respiratory sequelae or short-term acute respiratory complication subtypes after SARS-CoV-2 infection in the propensity score-matched cohorts in South Korea (main) and Japan (replication)

Cohort	South Korea			Japan		
	COVID-19 vs. general population (n=2,312,748)			COVID-19 vs. general population (n=3,115,606)		
	Events, n (%)	HR (95% CI)		Events, n (%)	HR (95% CI)	
Model 1*		Model 2†	Model 3*		Model 4	
Long-term post-acute respiratory sequelae						
Chronic respiratory failure						
Comparators (general population or patients with influenza)	18 (0.00094)	1.0 (reference)	1.0 (reference)	170 (0.0073)	1.0 (reference)	1.0 (reference)
Patients with COVID-19	46 (0.012)	12.80 (7.42-22.07)	8.92 (4.92-16.17)	688 (0.086)	11.85 (10.02-14.02)	7.55 (6.35-8.97)
Pulmonary hypertension						
Comparators	15 (0.00078)	1.0 (reference)	1.0 (reference)	156 (0.0067)	1.0 (reference)	1.0 (reference)
Patients with COVID-19	3 (0.00076)	1.00 (0.29-3.45)	0.60 (0.11-3.39)	217 (0.027)	4.07 (3.31-5.00)	3.11 (2.51-3.85)
Sleep apnea						
Comparators	1143 (0.060)	1.0 (reference)	1.0 (reference)	6643 (0.29)	1.0 (reference)	1.0 (reference)
Patients with COVID-19	235 (0.060)	1.02 (0.89-1.17)	1.13 (0.95-1.33)	5198 (0.65)	2.30 (2.22-2.39)	2.21 (2.13-2.29)
COPD						
Comparators	10846 (0.57)	1.0 (reference)	1.0 (reference)	11003 (0.47)	1.0 (reference)	1.0 (reference)
Patients with COVID-19	3359 (0.85)	1.54 (1.49-1.61)	1.57 (1.50-1.65)	15520 (1.95)	4.17 (4.07-4.27)	3.93 (3.83-4.03)
Emphysema						
Comparators	386 (0.020)	1.0 (reference)	1.0 (reference)	1550 (0.067)	1.0 (reference)	1.0 (reference)
Patients with COVID-19	133 (0.034)	1.73 (1.42-2.10)	1.60 (1.27-2.01)	2085 (0.26)	3.95 (3.70-4.22)	3.44 (3.22-3.68)
Asthma						
Comparators	4197 (0.22)	1.0 (reference)	1.0 (reference)	21314 (0.92)	1.0 (reference)	1.0 (reference)
Patients with COVID-19	1431 (0.36)	1.70 (1.60-1.80)	1.74 (1.62-1.87)	25311 (3.18)	3.53 (3.46-3.59)	3.44 (3.38-3.50)

Pulmonary sarcoidosis

Comparators	29 (0.0015)	1.0 (reference)	1.0 (reference)	972 (0.042)	1.0 (reference)	1.0 (reference)
Patients with COVID-19	4 (0.0010)	0.69 (0.24-1.95)	0.96 (0.34-2.75)	1255 (0.16)	3.78 (3.48-4.11)	3.44 (3.16-3.75)

Interstitial lung disease

Comparators	223 (0.012)	1.0 (reference)	1.0 (reference)	2942 (0.13)	1.0 (reference)	1.0 (reference)
Patients with COVID-19	453 (0.11)	10.13 (8.63-11.90)	10.38 (8.75-12.31)	5996 (0.75)	6.00 (5.74-6.27)	4.75 (4.54-4.97)

Acute respiratory complication**Pneumocystis pneumonia**

Comparators	5 (0.00026)	1.0 (reference)	1.0 (reference)	934 (0.040)	1.0 (reference)	1.0 (reference)
Patients with COVID-19	2 (0.00051)	1.96 (0.38-10.10)	0.03 (0.00-8550.49)	1426 (0.18)	4.46 (4.10-4.84)	3.28 (3.01-3.58)

Aspergillosis pneumonia

Comparators	16 (0.00083)	1.0 (reference)	1.0 (reference)	249 (0.011)	1.0 (reference)	1.0 (reference)
Patients with COVID-19	32 (0.0081)	9.73 (5.34-17.72)	6.85 (3.48-13.50)	601 (0.075)	7.05 (6.08-8.17)	4.97 (4.26-5.79)

Pleural empyema

Comparators	10 (0.00052)	1.0 (reference)	1.0 (reference)	24 (0.0010)	1.0 (reference)	1.0 (reference)
Patients with COVID-19	6 (0.0015)	2.93 (1.06-8.05)	1.45 (0.32-6.63)	226 (0.028)	27.44 (18.02-41.80)	22.00 (14.38-33.65)

Lung abscess

Comparators	17 (0.00089)	1.0 (reference)	1.0 (reference)	57 (0.0025)	1.0 (reference)	1.0 (reference)
Patients with COVID-19	7 (0.0018)	2.01 (0.83-4.85)	2.20 (0.81-6.00)	301 (0.038)	15.39 (11.60-20.43)	13.57 (10.19-18.07)

Pneumothorax

Comparators	43 (0.0022)	1.0 (reference)	1.0 (reference)	3818 (0.16)	1.0 (reference)	1.0 (reference)
Patients with COVID-19	50 (0.013)	5.69 (3.78-8.55)	5.29 (3.32-8.42)	3234 (0.41)	2.49 (2.37-2.60)	2.41 (2.30-2.53)

Acute respiratory failure

Comparators	13 (0.00068)	1.0 (reference)	1.0 (reference)	8767 (0.38)	1.0 (reference)	1.0 (reference)
Patients with COVID-19	363 (0.092)	135.7 (78.05-235.91)	112.04 (64.00-196.16)	20983 (2.63)	7.10 (6.92-7.28)	6.49 (6.32-6.65)

Pulmonary embolism

Comparators	209 (0.011)	1.0 (reference)	1.0 (reference)	2212 (0.10)	1.0 (reference)	1.0 (reference)
Patients with COVID-19	162 (0.041)	3.79 (3.08-4.65)	2.98 (2.32-3.82)	3972 (0.50)	5.26 (5.00-5.55)	4.58 (4.34-4.83)

CCI, Charlson comorbidity index; CI, confidence interval; COPD, chronic obstructive pulmonary disease; HR, hazard ratio; NA, not available.

The data in bold indicate significant differences ($P < 0.05$).

***Model 1 and 3:** Adjusted for age (20–39, 40–59, and ≥ 60 years) and sex.

†**Model 2:** Adjusted for age (20–39, 40–59, and ≥ 60 years); sex, household income (low income, middle income, and high income); region of residence (urban and rural); CCI score (0, 1, and ≥ 2); obesity (underweight [$< 18.5 \text{ kg/m}^2$], normal [$18.5\text{--}22.9 \text{ kg/m}^2$]; overweight [$23.0\text{--}24.9 \text{ kg/m}^2$], obese [$\geq 25.0 \text{ kg/m}^2$], and unknown); blood pressure (systolic blood pressure $< 140 \text{ mmHg}$ and diastolic blood pressure $< 90 \text{ mmHg}$, systolic blood pressure $\geq 140 \text{ mmHg}$ or diastolic blood pressure $\geq 90 \text{ mmHg}$, and unknown); fasting blood glucose (< 100 , $\geq 100 \text{ mg/dL}$, and unknown); serum total cholesterol (< 200 , $200\text{--}239$, $\geq 240 \text{ mg/dL}$, and unknown); glomerular filtration rate (< 60 , $60\text{--}89$, $\geq 90 \text{ mL/min/1.73 m}^2$, and unknown); smoking status (never, former, current smoker, and unknown); alcoholic drinks (< 1 , $1\text{--}2$, $3\text{--}4$, ≥ 5 days per week, and unknown); aerobic physical activity (sufficient, insufficient, and unknown); previous history of cardiovascular disease, and chronic kidney disease; history of medication use for diabetes mellitus, dyslipidemia, and hypertension; and strain of SARS-CoV-2 (original and delta).

‡**Model 4:** Adjusted for age (20–39, 40–59, and ≥ 60 years); sex; insurance status (insured and dependent); CCI score (0, 1, and ≥ 2); body mass index (underweight [$< 18.5 \text{ kg/m}^2$], normal [$18.5\text{--}22.9 \text{ kg/m}^2$], overweight [$23.0\text{--}25.0 \text{ kg/m}^2$], obese [$\geq 25.0 \text{ kg/m}^2$], and unknown); blood pressure (systolic blood pressure $< 140 \text{ mmHg}$ and diastolic blood pressure $< 90 \text{ mmHg}$, systolic blood pressure $\geq 140 \text{ mmHg}$ or diastolic blood pressure $\geq 90 \text{ mmHg}$, and unknown); fasting blood glucose (< 100 , $\geq 100 \text{ mg/dL}$, and unknown); serum total cholesterol (< 200 , $200\text{--}239$, $\geq 240 \text{ mg/dL}$, and unknown); glomerular filtration rate (< 60 , $60\text{--}89$, $\geq 90 \text{ mL/min/1.73 m}^2$, and unknown); smoking status (non- and current smoker, and unknown); alcoholic drinks (rarely, sometimes, everyday, and unknown); aerobic physical activity (sufficient, insufficient, and unknown); previous history of cardiovascular disease, and chronic kidney disease; history of medication use for diabetes mellitus, dyslipidemia, and hypertension; and strain of SARS-CoV-2 (original and delta).

Table 4. Subgroup analysis (COVID-19 vs. general population) of HR (95% CI) of the **long-term post-acute respiratory sequelae** or **short-term acute respiratory complication** after SARS-CoV-2 infection stratified by vaccination, COVID-19 severity, and SARS-CoV-2 strain in the cohort of South Korea (main)

Variable	Events/total, n/N (%)	HR (95% CI)	
		Model 1*	Model 2†
Long-term post-acute respiratory sequelae			
Number of SARS-CoV-2 vaccinations			
Non-infected control	16,122/1,918,150 (0.84)	0.62 (0.60-0.65)	0.60 (0.58-0.62)
COVID-19 without SARS-CoV-2 vaccination	4331/200,539 (2.16)	1.0 (reference)	1.0 (reference)
COVID-19 after SARS-CoV-2 vaccination received once	493/38,852 (1.27)	0.90 (0.82-0.99)	0.85 (0.77-0.94)
COVID-19 after SARS-CoV-2 vaccination received twice or more	468/155,207 (0.30)	0.69 (0.62-0.76)	0.64 (0.57-0.71)
Type of SARS-CoV-2 vaccinations			
Non-infected control	16,122/1,918,150 (0.84)	0.62 (0.60-0.65)	0.60 (0.58-0.62)
COVID-19 without SARS-CoV-2 vaccination	4331/200,539 (2.16)	1.0 (reference)	1.0 (reference)
COVID-19 with viral vector SARS-CoV-2 vaccination	465/109,066 (0.43)	1.12 (1.01-1.23)	0.99 (0.90-1.10)
COVID-19 with mRNA SARS-CoV-2 vaccination	477/66,891 (0.71)	1.20 (1.09-1.33)	1.11 (0.99-1.22)
COVID-19 with both types of SARS-CoV-2 vaccination	19/18,102 (0.10)	0.74 (0.47-1.16)	0.66 (0.42-1.03)
COVID-19 severity			
Non-infected control	16,122/1,918,150 (0.84)	1.0 (reference)	1.0 (reference)
Mild COVID-19	3492/340,813 (1.02)	1.28 (1.24-1.33)	1.37 (1.32-1.43)
Moderate to severe COVID-19	1800/53785 (3.35)	3.60 (3.43-3.78)	3.20 (3.03-3.38)
Original strain of SARS-CoV-2 (overall population)			
Non-infected control before the delta-dominant phase [§]	11,667/594,134 (1.96)	1.0 (reference)	1.0 (reference)

Infection with original strain	3649/121,521 (3.00)	1.58 (1.52-1.64)	1.59 (1.52-1.66)
Delta variant of SARS-CoV-2 (overall population)			
Non-infected control during the delta-dominant phase [§]	4455/1,324,016 (0.34)	1.0 (reference)	1.0 (reference)
Infection with Delta variant	1643/273,077 (0.60)	1.81 (1.71-1.92)	1.94 (1.81-2.08)
Acute respiratory complication			
Number of SARS-CoV-2 vaccinations			
Non-infected control	311/1,918,150 (0.016)	0.07 (0.06-0.08)	0.08 (0.06-0.09)
COVID-19 without SARS-CoV-2 vaccination	415/200,539 (0.21)	1.0 (reference)	1.0 (reference)
COVID-19 after SARS-CoV-2 vaccination received once	56/38,852 (0.14)	0.62 (0.47-0.83)	0.51 (0.38-0.68)
COVID-19 after SARS-CoV-2 vaccination received twice or more	147/155,207 (0.09)	0.32 (0.26-0.39)	0.24 (0.19-0.30)
Type of SARS-CoV-2 vaccinations			
Non-infected control	311/1,918,150 (0.016)	0.07 (0.06-0.08)	0.08 (0.06-0.09)
COVID-19 without SARS-CoV-2 vaccination	415/200,539 (0.21)	1.0 (reference)	1.0 (reference)
COVID-19 with viral vector SARS-CoV-2 vaccination	79/109,066 (0.072)	0.44 (0.34-0.56)	0.36 (0.28-0.47)
COVID-19 with mRNA SARS-CoV-2 vaccination	117/66,891 (0.17)	0.40 (0.32-0.50)	0.42 (0.33-0.53)
COVID-19 with both types of SARS-CoV-2 vaccination	7/18,102 (0.039)	0.19 (0.09-0.41)	0.18 (0.08-0.38)
COVID-19 severity			
Non-infected control	311/1,918,150 (0.016)	1.0 (reference)	1.0 (reference)
Mild COVID-19	50/340,813 (0.015)	0.95 (0.71-1.28)	0.99 (0.73-1.34)
Moderate to severe COVID-19	568/53,783 (1.06)	51.18 (44.38-59.02)	39.54 (33.54-46.62)
Original strain of SARS-CoV-2 (overall population)			
Non-infected control before the delta-dominant phase [§]	110/594,134 (0.019)	1.0 (reference)	1.0 (reference)
Infection with original strain	230/121,521 (0.19)	10.27 (8.18-12.88)	9.21 (7.19-11.80)

Delta variant of SARS-CoV-2 (overall population)

Non-infected control during the delta-dominant phase §	201/1,324,016 (0.015)	1.0 (reference)	1.0 (reference)
Infection with Delta variant	388/273,077 (0.14)	9.39 (7.92-11.14)	7.44 (6.13-9.03)

CCI, Charlson comorbidity index; CI, confidence interval; HR, hazard ratio; SARS-CoV-2, severe acute respiratory syndrome coronavirus 2.

The data in bold indicate significant differences ($P < 0.05$).

|| HR of the non-infected control represents the risk of respiratory diseases, and HRs of patients with COVID-19 indicate the risk of post-acute respiratory complications following SARS-CoV-2 infection.

§ Only 1:5-matched comparators in each patient group at the same index date were included to reduce immortal time bias.

***Model 1:** Adjusted for age (20–39, 40–59, and ≥ 60 years) and sex.

†**Model 2:** Adjusted for age (20–39, 40–59, and ≥ 60 years); sex, household income (low income, middle income, and high income); region of residence (urban and rural); CCI score (0, 1, and ≥ 2); obesity (underweight [$< 18.5 \text{ kg/m}^2$], normal [$18.5\text{--}22.9 \text{ kg/m}^2$]; overweight [$23.0\text{--}24.9 \text{ kg/m}^2$], obese [$\geq 25.0 \text{ kg/m}^2$], and unknown); blood pressure (systolic blood pressure $< 140 \text{ mmHg}$ and diastolic blood pressure $< 90 \text{ mmHg}$, systolic blood pressure $\geq 140 \text{ mmHg}$ or diastolic blood pressure $\geq 90 \text{ mmHg}$, and unknown); fasting blood glucose (< 100 , $\geq 100 \text{ mg/dL}$, and unknown); serum total cholesterol (< 200 , $200\text{--}239$, $\geq 240 \text{ mg/dL}$, and unknown); glomerular filtration rate (< 60 , $60\text{--}89$, $\geq 90 \text{ mL/min/1.73 m}^2$, and unknown); smoking status (never, former, current smoker, and unknown); alcoholic drinks (< 1 , $1\text{--}2$, $3\text{--}4$, ≥ 5 days per week, and unknown); aerobic physical activity (sufficient, insufficient, and unknown); previous history of cardiovascular disease, and chronic kidney disease; history of medication use for diabetes mellitus, dyslipidemia, and hypertension; and strain of SARS-CoV-2 (original and delta).

Table 5. Time attenuation effect analysis of HR (95% CI) for the risk of **long-term post-acute respiratory sequelae** after SARS-CoV-2 infection in South Korea (main cohort) and Japan (replication cohort)

Time	COVID-19 vs. general population	
	Main cohort [†]	Replication cohort
long-term post-acute respiratory sequelae		
<3 months	2.51 (2.38-2.64)	4.40 (4.30-4.51)
3–6 months	1.24 (1.15-1.33)	2.66 (2.57-2.75)
≥6 months	1.10 (1.01-1.19)	2.67 (2.61-2.73)

CCI, Charlson comorbidity index; CI, confidence interval; HR, hazard ratio; SARS-CoV-2, severe acute respiratory syndrome coronavirus 2. The data in bold indicate significant differences ($P < 0.05$).

[†] **Adjusted HR (main):** Adjusted for age (20–39, 40–59, and ≥60 years); sex, household income (low income, middle income, and high income); region of residence (urban and rural); CCI score (0, 1, and ≥2); obesity (underweight [$<18.5 \text{ kg/m}^2$], normal [$18.5\text{--}22.9 \text{ kg/m}^2$]; overweight [$23.0\text{--}24.9 \text{ kg/m}^2$], obese [$\geq 25.0 \text{ kg/m}^2$], and unknown); blood pressure (systolic blood pressure $<140 \text{ mmHg}$ and diastolic blood pressure $<90 \text{ mmHg}$, systolic blood pressure $\geq 140 \text{ mmHg}$ or diastolic blood pressure $\geq 90 \text{ mmHg}$, and unknown); fasting blood glucose (<100 , $\geq 100 \text{ mg/dL}$, and unknown); serum total cholesterol (<200 , $200\text{--}239$, $\geq 240 \text{ mg/dL}$, and unknown); glomerular filtration rate (<60 , $60\text{--}89$, $\geq 90 \text{ mL/min/1.73 m}^2$, and unknown); smoking status (never, former, current smoker, and unknown); alcoholic drinks (<1 , $1\text{--}2$, $3\text{--}4$, ≥ 5 days per week, and unknown); aerobic physical activity (sufficient, insufficient, and unknown); previous history of cardiovascular disease, and chronic kidney disease; history of medication use for diabetes mellitus, dyslipidemia, and hypertension; and strain of SARS-CoV-2 (original and delta).

^{||} **Adjusted HR (replication):** Adjusted for age (20–39, 40–59, and ≥60 years); sex; insurance status (insured and dependent); CCI score (0, 1, and ≥ 2); body mass index (underweight [$<18.5 \text{ kg/m}^2$], normal [$18.5\text{--}22.9 \text{ kg/m}^2$], overweight [$23.0\text{--}25.0 \text{ kg/m}^2$], obese [$\geq 25.0 \text{ kg/m}^2$], and unknown); blood pressure (systolic blood pressure $<140 \text{ mmHg}$ and diastolic blood pressure $<90 \text{ mmHg}$, systolic blood pressure $\geq 140 \text{ mmHg}$ or diastolic blood pressure $\geq 90 \text{ mmHg}$, and unknown); fasting blood glucose (<100 , $\geq 100 \text{ mg/dL}$, and unknown); serum total cholesterol (<200 , $200\text{--}239$, $\geq 240 \text{ mg/dL}$, and unknown); glomerular filtration rate (<60 , $60\text{--}89$, $\geq 90 \text{ mL/min/1.73 m}^2$, and unknown); smoking status (non- and current smoker, and unknown); alcoholic drinks (rarely, sometimes, everyday, and unknown); aerobic physical activity (sufficient, insufficient, and unknown); previous history of cardiovascular disease, and chronic kidney disease; history of medication use for diabetes mellitus, dyslipidemia, and hypertension; and strain of SARS-CoV-2 (original and delta).

Table S11. Baseline characteristics for 1:1 propensity score-matched cohort (COVID-19 vs. influenza) in South Korea (main)

Characteristic	COVID-19 vs. influenza (n=223,000)		SMD*
	COVID-19 (n=111,500)	Influenza (n=111,500)	
Mean age (SD), y	45.3 (15.3)	45.0 (12.7)	0.017
Age, n (%)			0.060
20–39 y	42,625 (38.2)	40,131 (36.0)	
40–59 y	50,713 (45.5)	53,295 (47.8)	
≥60 y	18,162 (16.3)	18,074 (16.2)	
Sex, n (%)			0.046
Male	46,400 (41.6)	49,587 (44.5)	
Female	65,100 (58.4)	61,913 (55.5)	
Region of residence, n (%)			0.005
Urban	49,431 (44.3)	49,160 (44.1)	
Rural	62,069 (55.7)	62,340 (55.9)	
Medical history, n (%)			
Cardiovascular disease	4166 (3.7)	4258 (3.8)	0.004
Chronic kidney disease	2011 (1.8)	1991 (1.8)	0.001
Medication use for diabetes	17,218 (15.4)	16,665 (15.0)	0.014
Medication use for hyperlipidemia	17,656 (15.8)	17,167 (15.4)	0.012
Medication use for hypertension	7408 (6.6)	6640 (6.0)	0.028
Unmatching covariates, n (%)[†]			
Charlson Comorbidity Index score			0.103
0	99,673 (89.4)	103,133 (92.5)	
1	7867 (7.1)	5296 (4.8)	
≥2	3960 (3.6)	3071 (2.8)	

Household income			0.078
Low (0th–39th percentile)	48,878 (43.8)	46,012 (41.3)	
Middle (40th –79th percentile)	45,339 (40.7)	45,119 (40.5)	
High (80th–100th percentile)	17,283 (15.5)	20,369 (18.3)	
Body mass index			1.180
Underweight (<18.5 kg/m ²)	2120 (1.9)	3997 (3.6)	
Normal (18.5-22.9 kg/m ²)	22,894 (20.5)	39,561 (35.5)	
Overweight (23.0-24.9 kg/m ²)	14,256 (12.8)	24,459 (21.9)	
Obese (≥25.0 kg/m ²)	25,159 (22.6)	43,474 (39.0)	
Unknown	47,071 (42.2)	9 (0.0081)	
Blood pressure			1.207
SBP <140 mmHg and DBP <90 mmHg	55,801 (50.1)	99,099 (88.9)	
SBP ≥140mmHg or DBP ≥90 mmHg	8400 (7.5)	12,237 (11.0)	
Unknown	47299 (42.4)	164 (0.15)	
Fasting blood glucose			1.229
<100 mg/dL	40,399 (36.2)	72,206 (64.8)	
≥100 mg/dL	23,797 (21.3)	39,131 (35.1)	
Unknown	47,304 (42.4)	163 (0.15)	
Serum total cholesterol			0.462
<200 mg/dL	17,459 (15.7)	29,991 (26.9)	
200 to 239 mg/dL	10485 (9.4)	18,760 (16.8)	
≥240 mg/dL	4859 (4.4)	7923 (7.1)	
Unknown	78,697 (70.6)	54,826 (49.2)	
Glomerular filtration rate			1.208
<60 mL/min/1.73 m ²	1689 (1.5)	2223 (2.0)	
60 to 89 mL/min/1.73 m ²	26,592 (23.9)	45,211 (40.6)	
≥90 mL/min/1.73 m ²	35,867 (32.2)	63,785 (57.2)	

Unknown	47,352 (42.5)	281 (0.25)	
Smoking status			1.155
Never	45,487 (40.8)	75,040 (67.3)	
Former	9450 (8.5)	15,936 (14.3)	
Current	9506 (8.5)	20,488 (18.4)	
Unknown	47,057 (42.2)	36 (0.032)	
Alcohol consumption			1.208
<1 day/week	38,511 (34.5)	68078 (61.1)	
1 to 2 days/week	17,887 (16.0)	31025 (27.8)	
3 to 4 days/week	6125 (5.5)	9620 (8.6)	
≥5 days/week	1922 (1.7)	2741 (2.5)	
Unknown	47,055 (42.2)	36 (0.032)	
Aerobic physical activity			1.228
Insufficient	33,898 (30.4)	58,183 (52.2)	
Sufficient	30,539 (27.4)	53,276 (47.8)	
Unknown	47,063 (42.2)	41 (0.037)	
Strain of SARS-CoV-2			<0.001
Original	34685 (31.1)	34,718 (31.1)	
Delta	76,815 (68.9)	76,782 (68.9)	

DBP, diastolic blood pressure; SARS-CoV-2, severe acute respiratory syndrome coronavirus 2; SBP, systolic blood pressure; SD, standard deviation; SMD, standardized mean difference.

* An SMD <0.1 indicates no significant imbalance. All SMDs were <0.100 in the propensity score-matched cohorts.

† Unmatched covariates were included as adjustment factors in statistical analyses.

Table S10. Baseline characteristics for 1:3 propensity score–matched cohort (COVID-19 vs. general population) in Japan (replication)

Characteristic	COVID-19 vs. general population (n=3,115,606)		SMD*
	COVID-19 (n=797,101)	General population (n=2,318,505)	
Mean age (SD), y	44 (11.88)	44 (12.03)	0.034
Age, n (%)			<0.001
20–39 y	302,404 (37.94)	878,119 (37.87)	
40–59 y	411,351 (51.61)	1,199,717 (51.75)	
≥60 y	83,346 (10.46)	240,669 (10.38)	
Sex, n (%)			0.005
Male	495,460 (62.16)	1,447,190 (62.42)	
Female	301,641 (37.84)	871,315 (37.58)	
Insurance status, n (%)			0.001
Insured	718,811 (90.18)	2,091,308 (90.20)	
Dependent	78,290 (9.82)	227,197 (9.80)	
Medical history, n (%)			
Cardiovascular disease	60,450 (7.58)	170,601 (7.36)	0.009
Chronic kidney disease	32,292 (4.05)	90,680 (3.91)	0.007
Medication use for diabetes	19,777 (2.48)	56,919 (2.45)	0.002
Medication use for hyperlipidemia	49,255 (6.18)	141,631 (6.11)	0.003
Medication use for hypertension	67,167 (8.43)	191,951 (8.28)	0.005
Unmatching covariates, n (%)[†]			
Charlson Comorbidity Index score			0.203
0	764,052 (95.85)	2,293,774 (98.93)	
1	10,954 (1.37)	8,640 (0.37)	
≥2	22,095 (2.77)	16,091 (0.69)	

Body mass index			<0.001
Underweight (<18.5 kg/m ²)	427,244 (53.60)	1,263,598 (54.50)	
Normal (18.5-22.9 kg/m ²)	151,460 (19.00)	439,941 (18.98)	
Overweight (23.0-24.9 kg/m ²)	169,962 (21.32)	484,142 (20.88)	
Obese (≥25.0 kg/m ²)	47,210 (5.92)	127,534 (5.50)	
Unknown	1225 (0.15)	3290 (0.14)	
Blood pressure			0.038
SBP <140 mmHg and DBP <90 mmHg	691,676 (86.77)	1,986,000 (85.66)	
SBP ≥140mmHg or DBP ≥90 mmHg	46,474 (5.83)	149,954 (6.47)	
Unknown	58,951 (7.40)	182,551 (7.87)	
Fasting blood glucose			<0.001
<100 mg/dL	511,681 (64.19)	1,474,020 (63.58)	
≥100 mg/dL	160,434 (20.13)	474,726 (20.48)	
Unknown	124,986 (15.68)	369,759 (15.95)	
Serum total cholesterol			0.054
<200 mg/dL	373,377 (46.84)	1,040,983 (44.90)	
200 to 239 mg/dL	285,475 (35.81)	843,032 (36.36)	
≥240 mg/dL	123,823 (15.53)	379,819 (16.38)	
Unknown	14,426 (1.81)	54,671 (2.36)	
Glomerular filtration rate			0.037
<60 mL/min/1.73 m ²	4383 (0.55)	8174 (0.35)	
60 to 89 mL/min/1.73 m ²	72,932 (9.15)	215,112 (9.28)	
≥90 mL/min/1.73 m ²	433,528 (54.39)	1,237,792 (53.39)	
Unknown	286,258 (35.91)	857,427 (36.98)	
Smoking status			0.062
Non-smoker	181,033 (22.71)	554,190 (23.90)	
Smoker	586,552 (73.59)	1,677,800 (72.37)	

Unknown	29,516 (3.70)	86,515 (3.73)	
Alcohol consumption			0.040
Everyday	161,809 (20.30)	466,183 (20.11)	
Sometimes	264,583 (33.19)	762,501 (32.89)	
Rarely	311,632 (39.10)	919,772 (39.67)	
Unknown	59,077 (7.41)	170,049 (7.33)	
Aerobic physical activity			0.046
Insufficient	161,205 (20.22)	497,575 (21.46)	
Sufficient	562,300 (70.54)	1,605,294 (69.24)	
Unknown	73,596 (9.23)	215,636 (9.30)	
Strain of SARS-CoV-2			0.005
Original	335,571 (42.10)	981,871 (42.35)	
Delta	461,530 (57.90)	1,336,634 (57.65)	

DBP, diastolic blood pressure; SARS-CoV-2, severe acute respiratory syndrome coronavirus 2; SBP, systolic blood pressure; SD, standard deviation; SMD, standardized mean difference.

* An SMD <0.1 indicates no significant imbalance. All SMDs were <0.100 in the propensity score-matched cohorts.

† Unmatched covariates were included as adjustment factors in statistical analyses.

Table S12. Baseline characteristics for 1:1 propensity score-matched cohort (COVID-19 vs. influenza) in Japan (replication)

Characteristic	COVID-19 vs. influenza (n=178,648)		SMD*
	COVID-19 (n=89,324)	Influenza (n=89,324)	
Mean age (SD), y	44 (11.76)	44 (11.57)	0.018
Age, n (%)			<0.001
20–39 y	31,256 (34.99)	31,210 (34.94)	
40–59 y	48,959 (54.81)	49,241 (55.13)	
≥60 y	9109 (10.20)	8873 (9.93)	
Sex, n (%)			0.005
Male	55,617 (62.26)	55,396 (62.02)	
Female	33,707 (37.74)	33,928 (37.98)	
Insurance status, n (%)			<0.001
Insured	80,141 (89.72)	80,141 (89.72)	
Dependent	9183 (10.28)	9183 (10.28)	
Medical history, n (%)			
Cardiovascular disease	6235 (6.98)	6214 (6.96)	0.001
Chronic kidney disease	3031 (3.39)	3252 (3.64)	0.013
Medication use for diabetes	2422 (2.71)	2200 (2.46)	0.016
Medication use for hyperlipidemia	6159 (6.90)	5877 (6.58)	0.013
Medication use for hypertension	8016 (8.97)	7716 (8.64)	0.012
Unmatching covariates, n (%)[†]			
Charlson Comorbidity Index score			0.153
0	85,599 (95.83)	87,875 (98.38)	
1	1150 (1.29)	501 (0.56)	
≥2	2575 (2.88)	948 (1.06)	

Body mass index			0.007
Underweight (<18.5 kg/m ²)	47,532 (53.21)	47,767 (53.48)	
Normal (18.5-22.9 kg/m ²)	17,018 (19.05)	16,935 (18.96)	
Overweight (23.0-24.9 kg/m ²)	19,260 (21.56)	19,131 (21.42)	
Obese (≥25.0 kg/m ²)	5373 (6.02)	5366 (6.01)	
Unknown	125 (0.14)	141 (0.16)	
Blood pressure			<0.001
SBP <140 mmHg and DBP <90 mmHg	77,118 (86.34)	77,019 (86.22)	
SBP ≥140mmHg or DBP ≥90 mmHg	5456 (6.11)	5383 (6.03)	
Unknown	6750 (7.56)	6922 (7.75)	
Fasting blood glucose			0.028
<100 mg/dL	57,035 (63.85)	57,644 (64.53)	
≥100 mg/dL	18,237 (20.42)	18,284 (20.47)	
Unknown	14,052 (15.73)	13,396 (15.00)	
Serum total cholesterol			0.022
<200 mg/dL	41,317 (46.26)	39,896 (44.66)	
200 to 239 mg/dL	32,270 (36.13)	33,092 (37.05)	
≥240 mg/dL	14,211 (15.91)	14,672 (16.43)	
Unknown	1526 (1.71)	1664 (1.86)	
Glomerular filtration rate			0.073
<60 mL/min/1.73 m ²	501 (0.56)	296 (0.33)	
60 to 89 mL/min/1.73 m ²	8344 (9.34)	7987 (8.94)	
≥90 mL/min/1.73 m ²	48,818 (54.65)	46,549 (52.11)	
Unknown	31,661 (35.45)	34,492 (38.61)	
Smoking status			0.062
Non-smoker	20,354 (22.79)	21,458 (24.02)	

Smoker	65,734 (73.59)	64,706 (72.44)	
Unknown	3236 (3.62)	3160 (3.54)	
Alcohol consumption			0.049
Everyday	18,486 (20.70)	18,284 (20.47)	
Sometimes	29,295 (32.80)	28,687 (32.12)	
Rarely	35,096 (39.29)	35,716 (39.98)	
Unknown	6447 (7.22)	6637 (7.43)	
Aerobic physical activity			0.035
Insufficient	18,249 (20.43)	18,303 (20.49)	
Sufficient	62,979 (70.51)	62,858 (70.37)	
Unknown	8096 (9.06)	8163 (9.14)	
Strain of SARS-CoV-2			1.652
Original	36,886 (41.29)	88,835 (99.45)	
Delta	52,438 (58.71)	489 (0.55)	

DBP, diastolic blood pressure; SARS-CoV-2, severe acute respiratory syndrome coronavirus 2; SBP, systolic blood pressure; SD, standard deviation; SMD, standardized mean difference.

* An SMD <0.1 indicates no significant imbalance. All SMDs were <0.100 in the propensity score–matched cohorts.

† Unmatched covariates were included as adjustment factors in statistical analyses.

Table S13. Hazard ratio (95% CI) for the **long-term post-acute respiratory sequelae** or **short-term acute respiratory complication** after SARS-CoV-2 infection in the propensity score-matched cohorts (COVID-19 vs. influenza) of South Korea (main) and Japan (replication)

Cohort	South Korea			Japan		
	COVID-19 vs. influenza (n=223,000)			COVID-19 vs. influenza (n=169,924)		
	Events, n (%)	HR (95% CI)		Events, n (%)	HR (95% CI)	
Model 1*		Model 2†	Model 3*		Model 4‡	
Long-term post-acute respiratory sequelae						
Comparators (general population or patients with influenza)	1081 (0.97)	1.0 (reference)	1.0 (reference)	905 (1.07)	1.0 (reference)	1.0 (reference)
Patients with COVID-19	1500 (1.35)	1.55 (1.43-1.67)	1.66 (1.52-1.82)	4757 (5.60)	5.44 (5.07-5.84)	5.17 (4.82-5.55)
Acute respiratory complication						
Comparators	45 (0.040)	1.0 (reference)	1.0 (reference)	122 (0.14)	1.0 (reference)	1.0 (reference)
Patients with COVID-19	115 (0.10)	4.75 (3.08-7.33)	4.32 (2.73-6.83)	924 (1.09)	7.57 (6.27-9.14)	6.51 (5.38-7.87)

CCI, Charlson comorbidity index; CI, confidence interval; HR, hazard ratio.

The data in bold indicate significant differences ($P < 0.05$).

***Model 1 and 3:** Adjusted for age (20–39, 40–59, and ≥ 60 years) and sex.

†**Model 2:** Adjusted for age (20–39, 40–59, and ≥ 60 years); sex, household income (low income, middle income, and high income); region of residence (urban and rural); CCI score (0, 1, and ≥ 2); obesity (underweight [$< 18.5 \text{ kg/m}^2$], normal [$18.5\text{--}22.9 \text{ kg/m}^2$]; overweight [$23.0\text{--}24.9 \text{ kg/m}^2$], obese [$\geq 25.0 \text{ kg/m}^2$], and unknown); blood pressure (systolic blood pressure $< 140 \text{ mmHg}$ and diastolic blood pressure $< 90 \text{ mmHg}$, systolic blood pressure $\geq 140 \text{ mmHg}$ or diastolic blood pressure $\geq 90 \text{ mmHg}$, and unknown); fasting blood glucose (< 100 , $\geq 100 \text{ mg/dL}$, and unknown); serum total cholesterol (< 200 , $200\text{--}239$, $\geq 240 \text{ mg/dL}$, and unknown); glomerular filtration rate (< 60 , $60\text{--}89$, $\geq 90 \text{ mL/min/1.73 m}^2$, and unknown); smoking status (never, former, current smoker, and unknown); alcoholic drinks (< 1 , $1\text{--}2$, $3\text{--}4$, ≥ 5 days per week, and unknown); aerobic physical activity (sufficient, insufficient, and unknown); previous history of cardiovascular disease, and chronic kidney disease; history of medication use for diabetes mellitus, dyslipidemia, and hypertension; and strain of SARS-CoV-2 (original and delta).

‡**Model 4:** Adjusted for age (20–39, 40–59, and ≥ 60 years); sex; insurance status (insured and dependent); CCI score (0, 1, and ≥ 2); body mass index (underweight [$< 18.5 \text{ kg/m}^2$], normal [$18.5\text{--}22.9 \text{ kg/m}^2$], overweight [$23.0\text{--}25.0 \text{ kg/m}^2$], obese [$\geq 25.0 \text{ kg/m}^2$], and unknown); blood

pressure (systolic blood pressure <140 mmHg and diastolic blood pressure <90 mmHg, systolic blood pressure \geq 140 mmHg or diastolic blood pressure \geq 90 mmHg, and unknown); fasting blood glucose (<100, \geq 100 mg/dL, and unknown); serum total cholesterol (<200, 200–239, \geq 240 mg/dL, and unknown); glomerular filtration rate (<60, 60–89, \geq 90 mL/min/1.73 m², and unknown); smoking status (non- and current smoker, and unknown); alcoholic drinks (rarely, sometimes, everyday, and unknown); aerobic physical activity (sufficient, insufficient, and unknown); previous history of cardiovascular disease, and chronic kidney disease; history of medication use for diabetes mellitus, dyslipidemia, and hypertension; and strain of SARS-CoV-2 (original and delta).

Table S14. HR (95% CI) for the long-term post-acute respiratory sequelae or short-term acute respiratory complication subtypes after SARS-CoV-2 infection in the propensity score-matched cohorts (COVID-19 vs. influenza) in South Korea (main) and Japan (replication)

Cohort	South Korea			Japan		
	COVID-19 vs. influenza (n=223,000)			COVID-19 vs. influenza (n=169,924)		
	Events, n (%)	HR (95% CI)		Events, n (%)	HR (95% CI)	
Model 1*		Model 2†	Model 3*		Model 4	
Long-term post-acute respiratory sequelae						
Chronic respiratory failure						
Comparators (general population or patients with influenza)	3 (0.0027)	1.0 (reference)	1.0 (reference)	3 (0.0035)	1.0 (reference)	1.0 (reference)
Patients with COVID-19	12 (0.011)	4.19 (1.18-14.86)	4.82 (1.28-18.12)	77 (0.091)	25.55 (8.06-80.95)	16.15 (5.04-51.80)
Pulmonary hypertension						
Comparators	0 (0.00)	1.0 (reference)	1.0 (reference)	4 (0.0047)	1.0 (reference)	1.0 (reference)
Patients with COVID-19	0 (0.00)	NA	NA	28 (0.033)	6.97 (2.44-19.87)	5.59 (1.93-16.14)
Sleep apnea						
Comparators	23 (0.021)	1.0 (reference)	1.0 (reference)	88 (0.10)	1.0 (reference)	1.0 (reference)
Patients with COVID-19	49 (0.044)	4.26 (2.26-8.01)	4.12 (2.09-8.12)	549 (0.65)	6.26 (5.00-7.84)	5.95 (4.74-7.46)
COPD						
Comparators	734 (0.66)	1.0 (reference)	1.0 (reference)	271 (0.32)	1.0 (reference)	1.0 (reference)
Patients with COVID-19	994 (0.89)	1.48 (1.34-1.63)	1.60 (1.44-1.79)	1703 (2.00)	0.99 (0.91-1.09)	1.01 (0.91-1.12)
Emphysema						
Comparators	19 (0.017)	1.0 (reference)	1.0 (reference)	31 (0.036)	1.0 (reference)	1.0 (reference)
Patients with COVID-19	33 (0.030)	1.96 (1.10-3.47)	1.99 (1.06-3.74)	232 (0.27)	7.47 (5.14-10.87)	6.46 (4.43-9.43)
Asthma						
Comparators	337 (0.30)	1.0 (reference)	1.0 (reference)	477 (0.56)	1.0 (reference)	1.0 (reference)
Patients with COVID-19	421 (0.38)	1.37 (1.18-1.58)	1.45 (1.23-1.71)	2672 (3.14)	5.72 (5.19-6.30)	5.59 (5.07-6.16)
Pulmonary sarcoidosis						

Comparators	2 (0.0018)	1.0 (reference)	1.0 (reference)	25 (0.029)	1.0 (reference)	1.0 (reference)
Patients with COVID-19	2 (0.0018)	2.07 (0.19-22.84)	3.24 (0.29-36.47)	127 (0.15)	5.08 (3.31-7.81)	4.52 (2.93-6.96)
Interstitial lung disease						
Comparators	32 (0.029)	1.0 (reference)	1.0 (reference)	67 (0.079)	1.0 (reference)	1.0 (reference)
Patients with COVID-19	94 (0.084)	3.25 (2.15-4.90)	3.37 (2.19-5.19)	679 (0.80)	10.19 (7.93-13.1)	8.26 (6.42-10.64)
Acute respiratory complication						
Pneumocystis pneumonia						
Comparators	0 (0.00)	1.0 (reference)	1.0 (reference)	18 (0.021)	1.0 (reference)	1.0 (reference)
Patients with COVID-19	0 (0.00)	NA	NA	66 (0.078)	3.65 (2.17-6.14)	2.85 (1.68-4.85)
Aspergillosis pneumonia						
Comparators	0 (0.00)	1.0 (reference)	1.0 (reference)	4 (0.0047)	1.0 (reference)	1.0 (reference)
Patients with COVID-19	9 (0.0081)	NA	NA	17 (0.020)	4.21 (1.42-12.52)	3.49 (1.16-10.54)
Pleural empyema						
Comparators	2 (0.0018)	1.0 (reference)	1.0 (reference)	0 (0.00)	1.0 (reference)	1.0 (reference)
Patients with COVID-19	0 (0.00)	NA	NA	18 (0.021)	NA	NA
Lung abscess						
Comparators	6 (0.0054)	1.0 (reference)	1.0 (reference)	6 (0.0071)	1.0 (reference)	1.0 (reference)
Patients with COVID-19	0 (0.00)	NA	NA	18 (0.021)	2.98 (1.18-7.50)	2.25 (0.88-5.80)
Pneumothorax						
Comparators	18 (0.016)	1.0 (reference)	1.0 (reference)	22 (0.026)	1.0 (reference)	1.0 (reference)
Patients with COVID-19	10 (0.0090)	1.05 (0.44-2.53)	1.02 (0.37-2.81)	114 (0.13)	5.17 (3.28-8.16)	4.92 (3.11-7.77)
Acute respiratory failure						
Comparators	2 (0.0018)	1.0 (reference)	1.0 (reference)	58 (0.068)	1.0 (reference)	1.0 (reference)
Patients with COVID-19	67 (0.060)	NA	NA	583 (0.69)	10.04 (7.67-13.15)	8.44 (6.43-11.07)
Pulmonary embolism						
Comparators	18 (0.06)	1.0 (reference)	1.0 (reference)	18 (0.021)	1.0 (reference)	1.0 (reference)
Patients with COVID-19	29 (0.026)	2.94 (1.43-6.03)	3.27 (1.53-7.02)	175 (0.21)	9.68 (5.96-15.73)	7.96 (4.88-12.98)

CCI, Charlson comorbidity index; CI, confidence interval; COPD, chronic obstructive pulmonary disease; HR, hazard ratio; NA, not available.

The data in bold indicate significant differences ($P < 0.05$).

***Model 1 and 3:** Adjusted for age (20–39, 40–59, and ≥ 60 years) and sex.

†**Model 2:** Adjusted for age (20–39, 40–59, and ≥ 60 years); sex, household income (low income, middle income, and high income); region of residence (urban and rural); CCI score (0, 1, and ≥ 2); obesity (underweight [$< 18.5 \text{ kg/m}^2$], normal [$18.5\text{--}22.9 \text{ kg/m}^2$]; overweight [$23.0\text{--}24.9 \text{ kg/m}^2$], obese [$\geq 25.0 \text{ kg/m}^2$], and unknown); blood pressure (systolic blood pressure $< 140 \text{ mmHg}$ and diastolic blood pressure $< 90 \text{ mmHg}$, systolic blood pressure $\geq 140 \text{ mmHg}$ or diastolic blood pressure $\geq 90 \text{ mmHg}$, and unknown); fasting blood glucose (< 100 , $\geq 100 \text{ mg/dL}$, and unknown); serum total cholesterol (< 200 , $200\text{--}239$, $\geq 240 \text{ mg/dL}$, and unknown); glomerular filtration rate (< 60 , $60\text{--}89$, $\geq 90 \text{ mL/min/1.73 m}^2$, and unknown); smoking status (never, former, current smoker, and unknown); alcoholic drinks (< 1 , $1\text{--}2$, $3\text{--}4$, ≥ 5 days per week, and unknown); aerobic physical activity (sufficient, insufficient, and unknown); previous history of cardiovascular disease, and chronic kidney disease; history of medication use for diabetes mellitus, dyslipidemia, and hypertension; and strain of SARS-CoV-2 (original and delta).

‡ **Model 4:** Adjusted for age (20–39, 40–59, and ≥ 60 years); sex; insurance status (insured and dependent); CCI score (0, 1, and ≥ 2); body mass index (underweight [$< 18.5 \text{ kg/m}^2$], normal [$18.5\text{--}22.9 \text{ kg/m}^2$], overweight [$23.0\text{--}25.0 \text{ kg/m}^2$], obese [$\geq 25.0 \text{ kg/m}^2$], and unknown); blood pressure (systolic blood pressure $< 140 \text{ mmHg}$ and diastolic blood pressure $< 90 \text{ mmHg}$, systolic blood pressure $\geq 140 \text{ mmHg}$ or diastolic blood pressure $\geq 90 \text{ mmHg}$, and unknown); fasting blood glucose (< 100 , $\geq 100 \text{ mg/dL}$, and unknown); serum total cholesterol (< 200 , $200\text{--}239$, $\geq 240 \text{ mg/dL}$, and unknown); glomerular filtration rate (< 60 , $60\text{--}89$, $\geq 90 \text{ mL/min/1.73 m}^2$, and unknown); smoking status (non- and current smoker, and unknown); alcoholic drinks (rarely, sometimes, everyday, and unknown); aerobic physical activity (sufficient, insufficient, and unknown); previous history of cardiovascular disease, and chronic kidney disease; history of medication use for diabetes mellitus, dyslipidemia, and hypertension; and strain of SARS-CoV-2 (original and delta).

Table S16. Time attenuation effect analysis (COVID-19 vs. influenza) of HR (95% CI) for the risk of the **long-term post-acute respiratory sequelae** after SARS-CoV-2 infection in South Korea (main cohort) and Japan (replication cohort)

Time	COVID-19 vs. influenza	
	Main cohort [†]	Replication cohort
long-term post-acute respiratory sequelae		
<3 months	2.28 (2.01-2.59)	8.88 (7.76-10.16)
3–6 months	1.24 (1.04-1.47)	3.81 (3.27-4.44)
≥6 months	1.01 (0.83-1.23)	3.65 (3.30-4.05)

CCI, Charlson comorbidity index; CI, confidence interval; HR, hazard ratio; SARS-CoV-2, severe acute respiratory syndrome coronavirus 2. The data in bold indicate significant differences ($P < 0.05$).

[†] **Adjusted HR (main):** Adjusted for age (20–39, 40–59, and ≥60 years); sex, household income (low income, middle income, and high income); region of residence (urban and rural); CCI score (0, 1, and ≥2); obesity (underweight [$<18.5 \text{ kg/m}^2$], normal [$18.5\text{--}22.9 \text{ kg/m}^2$]; overweight [$23.0\text{--}24.9 \text{ kg/m}^2$], obese [$\geq 25.0 \text{ kg/m}^2$], and unknown); blood pressure (systolic blood pressure $<140 \text{ mmHg}$ and diastolic blood pressure $<90 \text{ mmHg}$, systolic blood pressure $\geq 140 \text{ mmHg}$ or diastolic blood pressure $\geq 90 \text{ mmHg}$, and unknown); fasting blood glucose (<100 , $\geq 100 \text{ mg/dL}$, and unknown); serum total cholesterol (<200 , $200\text{--}239$, $\geq 240 \text{ mg/dL}$, and unknown); glomerular filtration rate (<60 , $60\text{--}89$, $\geq 90 \text{ mL/min/1.73 m}^2$, and unknown); smoking status (never, former, current smoker, and unknown); alcoholic drinks (<1 , $1\text{--}2$, $3\text{--}4$, ≥ 5 days per week, and unknown); aerobic physical activity (sufficient, insufficient, and unknown); previous history of cardiovascular disease, and chronic kidney disease; history of medication use for diabetes mellitus, dyslipidemia, and hypertension; and strain of SARS-CoV-2 (original and delta).

^{||} **Adjusted HR (replication):** Adjusted for age (20–39, 40–59, and ≥60 years); sex; insurance status (insured and dependent); CCI score (0, 1, and ≥ 2); body mass index (underweight [$<18.5 \text{ kg/m}^2$], normal [$18.5\text{--}22.9 \text{ kg/m}^2$], overweight [$23.0\text{--}25.0 \text{ kg/m}^2$], obese [$\geq 25.0 \text{ kg/m}^2$], and unknown); blood pressure (systolic blood pressure $<140 \text{ mmHg}$ and diastolic blood pressure $<90 \text{ mmHg}$, systolic blood pressure $\geq 140 \text{ mmHg}$ or diastolic blood pressure $\geq 90 \text{ mmHg}$, and unknown); fasting blood glucose (<100 , $\geq 100 \text{ mg/dL}$, and unknown); serum total cholesterol (<200 , $200\text{--}239$, $\geq 240 \text{ mg/dL}$, and unknown); glomerular filtration rate (<60 , $60\text{--}89$, $\geq 90 \text{ mL/min/1.73 m}^2$, and unknown); smoking status (non- and current smoker, and unknown); alcoholic drinks (rarely, sometimes, everyday, and unknown); aerobic physical activity (sufficient, insufficient, and unknown); previous history of cardiovascular disease, and chronic kidney disease; history of medication use for diabetes mellitus, dyslipidemia, and hypertension; and strain of SARS-CoV-2 (original and delta).

Table S17. Stratification analysis for the **long-term** risk of **post-acute respiratory sequelae** following COVID-19 in the propensity score matching cohorts (COVID-19 vs. general population) of South Korea (main)

	Events, n (%)	COVID-19 vs. general population (n =2,312,748)	
		HR (95% CI)	
		Model 1*	Model 2†
Sex			
Male			
Comparators (general population or patients with influenza)	7849 (0.79)	1.0 (reference)	1.0 (reference)
Patients with COVID-19	75 (1.28)	1.67 (1.60-1.75)	1.64 (1.55-1.73)
Female			
Comparators	8273 (0.90)	1.0 (reference)	1.0 (reference)
Patients with COVID-19	2666 (1.41)	1.61 (1.54-1.68)	1.67 (1.59-1.76)
Age			
20–39 y			
Comparators	5809 (0.83)	1.0 (reference)	1.0 (reference)
Patients with COVID-19	1501 (1.05)	1.27 (1.20-1.35)	1.33 (1.23-1.44)
40–59 years			
Comparators	5243 (0.74)	1.0 (reference)	1.0 (reference)
Patients with COVID-19	1861 (1.28)	1.76 (1.66-1.85)	1.79 (1.68-1.89)
≥60 y			
Comparators	5070 (1.0010)	1.0 (reference)	1.0 (reference)
Patients with COVID-19	1930 (1.82)	1.96 (1.86-2.07)	1.85 (1.74-1.96)
Region of residence			
Rural			
Comparators	8382 (0.81)	1.0 (reference)	1.0 (reference)
Patients with COVID-19	2821 (1.32)	1.68 (1.61-1.75)	1.71 (1.63-1.80)
Urban			
Comparators	7740 (0.88)	1.0 (reference)	1.0 (reference)

Patients with COVID-19	2471 (1.36)	1.60 (1.53-1.67)	1.60 (1.51-1.68)
Income level			
Low (0th–39th percentile)			
Comparators	7444 (0.84)	1.0 (reference)	1.0 (reference)
Patients with COVID-19	2370 (1.30)	1.59 (1.52-1.67)	1.60 (1.51-1.69)
Middle (40th –79th percentile)			
Comparators	5786 (0.85)	1.0 (reference)	1.0 (reference)
Patients with COVID-19	1861 (1.33)	1.60 (1.52-1.69)	1.61 (1.51-1.71)
High (80th–100th percentile)			
Comparators	2892 (0.83)	1.0 (reference)	1.0 (reference)
Patients with COVID-19	1061 (1.48)	1.86 (1.73-1.99)	1.86 (1.72-2.02)
CCI score			
0 score			
Comparators	14,263 (0.79)	1.0 (reference)	1.0 (reference)
Patients with COVID-19	4171 (1.20)	1.59 (1.54-1.65)	1.72 (1.66-1.80)
≥1 score			
Comparators	1859 (1.67)	1.0 (reference)	1.0 (reference)
Patients with COVID-19	1121 (2.33)	1.31 (1.22-1.42)	1.39 (1.27-1.51)
BMI			
<18.5 kg/m²			
Comparators	549 (0.81)	1.0 (reference)	1.0 (reference)
Patients with COVID-19	89 (1.29)	1.66 (1.32-2.07)	1.59 (1.27-2.00)
18.5-23.0 kg/m²			
Comparators	5575 (0.81)	1.0 (reference)	1.0 (reference)
Patients with COVID-19	989 (1.27)	1.58 (1.48-1.69)	1.49 (1.39-1.60)
23.0-25.0 kg/m²			
Comparators	6362 (0.88)	1.0 (reference)	1.0 (reference)
Patients with COVID-19	1548 (1.63)	1.83 (1.74-1.94)	1.72 (1.63-1.82)
≥25.0 kg/m²			
Comparators	2 (0.90)	1.0 (reference)	1.0 (reference)

Patients with COVID-19	1822 (1.14)	1.10 (0.27-4.40)	5.28 (0.41-67.90)
Unknown			
Comparators	3634 (0.83)	1.0 (reference)	1.0 (reference)
Patients with COVID-19	844 (1.55)	1.87 (1.73-2.01)	1.75 (1.62-1.89)
Smoking status			
Non-smoker			
Comparators	10,215 (0.84)	1.0 (reference)	1.0 (reference)
Patients with COVID-19	2333 (1.51)	1.79 (1.71-1.88)	1.70 (1.62-1.78)
Smoker			
Comparators	5905 (0.84)	1.0 (reference)	1.0 (reference)
Patients with COVID-19	1138 (1.42)	1.71 (1.60-1.82)	1.58 (1.48-1.69)
Unknown			
Comparators	2 (0.35)	1.0 (reference)	1.0 (reference)
Patients with COVID-19	1821 (1.14)	3.88 (0.97-15.49)	0.23 (0.00-60.08)
Alcohol consumption			
Non-drinker			
Comparators	10,042 (0.88)	1.0 (reference)	1.0 (reference)
Patients with COVID-19	2140 (1.56)	1.79 (1.70-1.87)	1.67 (1.60-1.75)
Drinker			
Comparators	6077 (0.79)	1.0 (reference)	1.0 (reference)
Patients with COVID-19	1331 (1.37)	1.73 (1.63-1.83)	1.63 (1.53-1.73)
Unknown			
Comparators	3 (0.47)	1.0 (reference)	1.0 (reference)
Patients with COVID-19	1821 (1.14)	2.62 (0.84-8.12)	0.18 (0.00-24.24)
Physical activity			
Insufficient physical activity			
Comparators	8479 (0.88)	1.0 (reference)	1.0 (reference)
Patients with COVID-19	1773 (1.49)	1.69 (1.60-1.77)	1.57 (1.49-1.65)
Sufficient physical activity			
Comparators	7635 (0.80)	1.0 (reference)	1.0 (reference)
Patients with COVID-19	1698 (1.47)	1.85 (1.75-1.95)	1.76 (1.67-1.85)

Unknown

Comparators	8 (0.92)	1.0 (reference)	1.0 (reference)
Patients with COVID-19	1821 (1.13)	1.48 (0.74-2.96)	0.09 (0.00-3.16)

BMI, body mass index; CCI, Charlson comorbidity index; CI, confidence interval; HR, hazard ratio; NA, not available; SARS-CoV-2, severe acute respiratory syndrome coronavirus 2.

The data in bold indicate significant differences ($P < 0.05$).

***Model 1:** Adjusted for age (20–39, 40–59, and ≥ 60 years) and sex.

†Model 2 (main): Adjusted for age (20–39, 40–59, and ≥ 60 years); sex, household income (low income, middle income, and high income); region of residence (urban and rural); CCI score (0, 1, and ≥ 2); obesity (underweight [$< 18.5 \text{ kg/m}^2$], normal [$18.5\text{--}22.9 \text{ kg/m}^2$]; overweight [$23.0\text{--}24.9 \text{ kg/m}^2$], obese [$\geq 25.0 \text{ kg/m}^2$], and unknown); blood pressure (systolic blood pressure $< 140 \text{ mmHg}$ and diastolic blood pressure $< 90 \text{ mmHg}$, systolic blood pressure $\geq 140 \text{ mmHg}$ or diastolic blood pressure $\geq 90 \text{ mmHg}$, and unknown); fasting blood glucose (< 100 , $\geq 100 \text{ mg/dL}$, and unknown); serum total cholesterol (< 200 , $200\text{--}239$, $\geq 240 \text{ mg/dL}$, and unknown); glomerular filtration rate (< 60 , $60\text{--}89$, $\geq 90 \text{ mL/min/1.73 m}^2$, and unknown); smoking status (never, former, current smoker, and unknown); alcoholic drinks (< 1 , $1\text{--}2$, $3\text{--}4$, ≥ 5 days per week, and unknown); aerobic physical activity (sufficient, insufficient, and unknown); previous history of cardiovascular disease, and chronic kidney disease; history of medication use for diabetes mellitus, dyslipidemia, and hypertension; and strain of SARS-CoV-2 (original and delta).

Table S18. Stratification analysis for the **long-term** risk of **post-acute respiratory sequelae** following COVID-19 in the propensity score matching cohorts of (COVID-19 vs. influenza) South Korea (main)

	Events, n (%)	COVID-19 vs. influenza (n =223,000)	
		Model 1*	HR (95% CI) Model 2†
Sex			
Male			
Comparators (general population or patients with influenza)	515 (1.04)	1.0 (reference)	1.0 (reference)
Patients with COVID-19	581 (1.25)	1.34 (1.19-1.52)	1.40 (1.21-1.61)
Female			
Comparators	566 (0.91)	1.0 (reference)	1.0 (reference)
Patients with COVID-19	919 (1.41)	1.69 (1.52-1.88)	1.82 (1.62-2.05)
Age			
20–39 y			
Comparators	481 (1.20)	1.0 (reference)	1.0 (reference)
Patients with COVID-19	465 (1.09)	0.97 (0.85-1.11)	1.05 (0.89-1.23)
40–59 years			
Comparators	436 (0.82)	1.0 (reference)	1.0 (reference)
Patients with COVID-19	651 (1.28)	1.68 (1.49-1.90)	1.75 (1.53-2.00)
≥60 y			
Comparators	164 (0.91)	1.0 (reference)	1.0 (reference)
Patients with COVID-19	384 (2.11)	2.91 (2.40-3.53)	3.15 (2.56-3.88)
Region of residence			
Rural			
Comparators	448 (0.91)	1.0 (reference)	1.0 (reference)
Patients with COVID-19	624 (1.26)	1.56 (1.38-1.77)	1.73 (1.51-1.99)
Urban			
Comparators	633 (1.02)	1.0 (reference)	1.0 (reference)

Patients with COVID-19	876 (1.41)	1.51 (1.36-1.68)	1.57 (1.39-1.76)
Income level			
Low (0th–39th percentile)			
Comparators	454 (0.99)	1.0 (reference)	1.0 (reference)
Patients with COVID-19	671 (1.37)	1.54 (1.36-1.74)	1.69 (1.47-1.94)
Middle (40th –79th percentile)			
Comparators	432 (0.96)	1.0 (reference)	1.0 (reference)
Patients with COVID-19	596 (1.31)	1.50 (1.32-1.70)	1.59 (1.38-1.83)
High (80th–100th percentile)			
Comparators	195 (0.96)	1.0 (reference)	1.0 (reference)
Patients with COVID-19	233 (1.35)	1.59 (1.31-1.94)	1.62 (1.30-2.02)
CCI score			
0 score			
Comparators	988 (0.96)	1.0 (reference)	1.0 (reference)
Patients with COVID-19	1195 (1.20)	1.39 (1.27-1.51)	1.52 (1.38-1.67)
≥1 score			
Comparators	93 (1.11)	1.0 (reference)	1.0 (reference)
Patients with COVID-19	305 (2.58)	2.34 (1.84-2.98)	2.75 (2.12-3.56)
BMI			
<18.5 kg/m²			
Comparators	35 (0.88)	1.0 (reference)	1.0 (reference)
Patients with COVID-19	26 (1.23)	1.62 (0.96-2.72)	1.72 (1.01-2.92)
18.5-23.0 kg/m²			
Comparators	357 (0.90)	1.0 (reference)	1.0 (reference)
Patients with COVID-19	311 (1.36)	1.67 (1.43-1.96)	1.68 (1.44-1.96)
23.0-25.0 kg/m²			
Comparators	235 (0.96)	1.0 (reference)	1.0 (reference)
Patients with COVID-19	233 (1.63)	1.78 (1.48-2.13)	1.80 (1.50-2.17)
≥25.0 kg/m²			
Comparators	454 (1.04)	1.0 (reference)	1.0 (reference)

Patients with COVID-19	384 (1.53)	1.56 (1.35-1.79)	1.52 (1.32-1.75)
Unknown			
Comparators	0 (0.00)	1.0 (reference)	1.0 (reference)
Patients with COVID-19	546 (1.16)	N/A	N/A
Smoking status			
Non-smoker			
Comparators	717 (0.96)	1.0 (reference)	1.0 (reference)
Patients with COVID-19	681 (1.50)	1.67 (1.50-1.86)	1.66 (1.49-1.85)
Smoker			
Comparators	364 (1.00)	1.0 (reference)	1.0 (reference)
Patients with COVID-19	274 (1.45)	1.59 (1.35-1.86)	1.59 (1.35-1.87)
Unknown			
Comparators	0 (0.00)	1.0 (reference)	1.0 (reference)
Patients with COVID-19	545 (1.16)	N/A	N/A
Alcohol consumption			
Non-drinker			
Comparators	631 (0.93)	1.0 (reference)	1.0 (reference)
Patients with COVID-19	609 (1.58)	1.87 (1.66-2.09)	1.84 (1.64-2.07)
Drinker			
Comparators	450 (1.04)	1.0 (reference)	1.0 (reference)
Patients with COVID-19	346 (1.33)	1.36 (1.18-1.57)	1.37 (1.19-1.58)
Unknown			
Comparators	0 (0.00)	1.0 (reference)	1.0 (reference)
Patients with COVID-19	545 (1.16)	N/A	N/A
Physical activity			
Insufficient physical activity			
Comparators	554 (0.95)	1.0 (reference)	1.0 (reference)
Patients with COVID-19	507 (1.50)	1.69 (1.49-1.91)	1.67 (1.48-1.89)
Sufficient physical activity			
Comparators	526 (0.99)	1.0 (reference)	1.0 (reference)
Patients with COVID-19	448 (1.47)	1.61 (1.41-1.83)	1.60 (1.41-1.82)

Unknown

Comparators	1 (2.44)	1.0 (reference)	1.0 (reference)
Patients with COVID-19	545 (1.16)	0.47 (0.07-3.37)	N/A

BMI, body mass index; CCI, Charlson comorbidity index; CI, confidence interval; HR, hazard ratio; NA, not available; SARS-CoV-2, severe acute respiratory syndrome coronavirus 2.

The data in bold indicate significant differences ($P < 0.05$).

***Model 1:** Adjusted for age (20–39, 40–59, and ≥ 60 years) and sex.

†Model 2 (main): Adjusted for age (20–39, 40–59, and ≥ 60 years); sex, household income (low income, middle income, and high income); region of residence (urban and rural); CCI score (0, 1, and ≥ 2); obesity (underweight [$< 18.5 \text{ kg/m}^2$], normal [$18.5\text{--}22.9 \text{ kg/m}^2$]; overweight [$23.0\text{--}24.9 \text{ kg/m}^2$], obese [$\geq 25.0 \text{ kg/m}^2$], and unknown); blood pressure (systolic blood pressure $< 140 \text{ mmHg}$ and diastolic blood pressure $< 90 \text{ mmHg}$, systolic blood pressure $\geq 140 \text{ mmHg}$ or diastolic blood pressure $\geq 90 \text{ mmHg}$, and unknown); fasting blood glucose (< 100 , $\geq 100 \text{ mg/dL}$, and unknown); serum total cholesterol (< 200 , $200\text{--}239$, $\geq 240 \text{ mg/dL}$, and unknown); glomerular filtration rate (< 60 , $60\text{--}89$, $\geq 90 \text{ mL/min/1.73 m}^2$, and unknown); smoking status (never, former, current smoker, and unknown); alcoholic drinks (< 1 , $1\text{--}2$, $3\text{--}4$, ≥ 5 days per week, and unknown); aerobic physical activity (sufficient, insufficient, and unknown); previous history of cardiovascular disease, and chronic kidney disease; history of medication use for diabetes mellitus, dyslipidemia, and hypertension; and strain of SARS-CoV-2 (original and delta).

Table S19. Stratification analysis for the **short-term risk of acute respiratory complication** following COVID-19 in the propensity score matching cohorts (COVID-19 vs. general population) of South Korea (main)

	Events, n (%)	COVID-19 vs. general population (n =2,312,748)	
		HR (95% CI)	
		Model 1*	Model 2†
Sex			
Male			
Comparators (general population or patients with influenza)	167 (0.017)	1.0 (reference)	1.0 (reference)
Patients with COVID-19	345 (0.17)	10.08 (8.38-12.13)	8.72 (7.11-10.68)
Female			
Comparators	144 (0.016)	1.0 (reference)	1.0 (reference)
Patients with COVID-19	273 (0.14)	9.26 (7.56-11.33)	6.67 (5.28-8.43)
Age			
20–39 y			
Comparators	61 (0.0087)	1.0 (reference)	1.0 (reference)
Patients with COVID-19	65 (0.045)	5.23 (3.69-7.41)	5.85 (3.85-8.91)
40–59 years			
Comparators	83 (0.012)	1.0 (reference)	1.0 (reference)
Patients with COVID-19	157 (0.11)	9.26 (7.10-12.09)	7.14 (5.31-9.60)
≥60 y			
Comparators	167 (0.033)	1.0 (reference)	1.0 (reference)
Patients with COVID-19	396 (0.37)	11.55 (9.64-13.84)	8.69 (7.10-10.62)
Region of residence			
Rural			
Comparators	185 (0.018)	1.0 (reference)	1.0 (reference)
Patients with COVID-19	271 (0.13)	7.15 (5.93-8.62)	5.61 (4.52-6.96)
Urban			
Comparators	126 (0.014)	1.0 (reference)	1.0 (reference)

Patients with COVID-19	347 (0.19)	13.46 (10.97-16.50)	10.81 (8.64-13.51)
Income level			
Low (0th–39th percentile)			
Comparators	142 (0.016)	1.0 (reference)	1.0 (reference)
Patients with COVID-19	241 (0.13)	8.28 (6.73-10.19)	7.06 (5.58-8.94)
Middle (40th –79th percentile)			
Comparators	99 (0.015)	1.0 (reference)	1.0 (reference)
Patients with COVID-19	229 (0.16)	11.26 (8.90-14.26)	8.64 (6.65-11.22)
High (80th–100th percentile)			
Comparators	70 (0.020)	1.0 (reference)	1.0 (reference)
Patients with COVID-19	148 (0.21)	10.36 (7.80-13.77)	7.72 (5.61-10.61)
CCI score			
0 score			
Comparators	197 (0.011)	1.0 (reference)	1.0 (reference)
Patients with COVID-19	338 (0.10)	9.36 (7.85-11.16)	8.83 (7.25-10.76)
≥1 scores			
Comparators	114 (0.10)	1.0 (reference)	1.0 (reference)
Patients with COVID-19	280 (0.58)	5.60 (4.50-6.96)	5.85 (4.60-7.44)
BMI			
<18.5 kg/m²			
Comparators	13 (0.019)	1.0 (reference)	1.0 (reference)
Patients with COVID-19	11 (0.16)	8.37 (3.73-18.78)	8.08 (3.57-18.32)
18.5-23.0 kg/m²			
Comparators	81 (0.012)	1.0 (reference)	1.0 (reference)
Patients with COVID-19	70 (0.090)	7.22 (5.24-9.95)	6.15 (4.43-8.53)
23.0-25.0 kg/m²			
Comparators	71 (0.016)	1.0 (reference)	1.0 (reference)
Patients with COVID-19	85 (0.16)	8.94 (6.52-12.25)	7.92 (5.75-10.91)
≥25.0 kg/m²			
Comparators	146 (0.020)	1.0 (reference)	1.0 (reference)
Patients with COVID-19	201 (0.21)	9.51 (7.68-11.78)	8.25 (6.64-10.26)

Unknown			
Comparators	0 (0.00)	1.0 (reference)	1.0 (reference)
Patients with COVID-19	251 (0.16)	N/A	N/A
			
Smoking status			
Non-smoker			
Comparators	189 (0.016)	1.0 (reference)	1.0 (reference)
Patients with COVID-19	239 (0.16)	8.95 (7.40-10.84)	7.72 (6.35-9.38)
Smoker			
Comparators	122 (0.017)	1.0 (reference)	1.0 (reference)
Patients with COVID-19	128 (0.16)	8.50 (6.63-10.90)	7.56 (5.88-9.73)
Unknown			
Comparators	0 (0.00)	1.0 (reference)	1.0 (reference)
Patients with COVID-19	251 (0.16)	N/A	N/A
			
Alcohol consumption			
Non-drinker			
Comparators	208 (0.018)	1.0 (reference)	1.0 (reference)
Patients with COVID-19	243 (0.18)	8.88 (7.38-10.69)	7.75 (6.41-9.36)
Drinker			
Comparators	103 (0.013)	1.0 (reference)	1.0 (reference)
Patients with COVID-19	124 (0.13)	8.90 (6.85-11.56)	7.53 (5.77-9.83)
Unknown			
Comparators	0 (0.00)	1.0 (reference)	1.0 (reference)
Patients with COVID-19	251 (0.16)	N/A	N/A
			
Physical activity			
Insufficient physical activity			
Comparators	154 (0.016)	1.0 (reference)	1.0 (reference)
Patients with COVID-19	212 (0.18)	9.84 (7.99-12.12)	8.46 (6.84-10.46)
Sufficient physical activity			
Comparators	157 (0.016)	1.0 (reference)	1.0 (reference)
Patients with COVID-19	155 (0.13)	7.68 (6.15-9.59)	6.82 (5.45-8.55)
Unknown			
Comparators	0 (0.00)	1.0 (reference)	1.0 (reference)

Patients with COVID-19

251 (0.16)

N/A

N/A

BMI, body mass index; CCI, Charlson comorbidity index; CI, confidence interval; HR, hazard ratio; NA, not available; SARS-CoV-2, severe acute respiratory syndrome coronavirus 2.

The data in bold indicate significant differences ($P < 0.05$).

***Model 1:** Adjusted for age (20–39, 40–59, and ≥ 60 years) and sex.

†Model 2 (main): Adjusted for age (20–39, 40–59, and ≥ 60 years); sex, household income (low income, middle income, and high income); region of residence (urban and rural); CCI score (0, 1, and ≥ 2); obesity (underweight [$< 18.5 \text{ kg/m}^2$], normal [$18.5\text{--}22.9 \text{ kg/m}^2$]; overweight [$23.0\text{--}24.9 \text{ kg/m}^2$], obese [$\geq 25.0 \text{ kg/m}^2$], and unknown); blood pressure (systolic blood pressure $< 140 \text{ mmHg}$ and diastolic blood pressure $< 90 \text{ mmHg}$, systolic blood pressure $\geq 140 \text{ mmHg}$ or diastolic blood pressure $\geq 90 \text{ mmHg}$, and unknown); fasting blood glucose (< 100 , $\geq 100 \text{ mg/dL}$, and unknown); serum total cholesterol (< 200 , $200\text{--}239$, $\geq 240 \text{ mg/dL}$, and unknown); glomerular filtration rate (< 60 , $60\text{--}89$, $\geq 90 \text{ mL/min/1.73 m}^2$, and unknown); smoking status (never, former, current smoker, and unknown); alcoholic drinks (< 1 , $1\text{--}2$, $3\text{--}4$, ≥ 5 days per week, and unknown); aerobic physical activity (sufficient, insufficient, and unknown); previous history of cardiovascular disease, and chronic kidney disease; history of medication use for diabetes mellitus, dyslipidemia, and hypertension; and strain of SARS-CoV-2 (original and delta).

Table S20. Stratification analysis for the **short-term** risk of **acute respiratory complication** following COVID-19 in the propensity score matching cohorts (COVID-19 vs. influenza) of South Korea (main)

	Events, n (%)	COVID-19 vs. influenza (n =223,000)	
		Model 1*	Model 2†
Sex			
Male			
Comparators (general population or patients with influenza)	23 (0.046)	1.0 (reference)	1.0 (reference)
Patients with COVID-19	51 (0.11)	27.75 (6.76-114.00)	25.32 (6.04-106.13)
Female			
Comparators	22 (0.036)	1.0 (reference)	1.0 (reference)
Patients with COVID-19	64 (0.098)	10.38 (4.50-23.97)	8.23 (3.46-19.58)
Age			
20–39 y			
Comparators	18 (0.045)	1.0 (reference)	1.0 (reference)
Patients with COVID-19	15 (0.035)	6.94 (1.59-30.34)	10.78 (2.35-49.52)
40–59 years			
Comparators	12 (0.023)	1.0 (reference)	1.0 (reference)
Patients with COVID-19	43 (0.085)	46.16 (6.36-335.06)	42.17 (5.74-309.82)
≥60 y			
Comparators	15 (0.083)	1.0 (reference)	1.0 (reference)
Patients with COVID-19	57 (0.31)	11.53 (4.62-28.76)	7.80 (3.00-20.28)
Region of residence			
Rural			
Comparators	29 (0.059)	1.0 (reference)	1.0 (reference)
Patients with COVID-19	39 (0.079)	5.57 (2.49-12.46)	5.05 (2.18-11.71)
Urban			
Comparators	16 (0.026)	1.0 (reference)	1.0 (reference)

Patients with COVID-19	76 (0.12)	79.28 (11.03-570.15)	65.32 (8.99-474.56)
Income level			
Low (0th–39th percentile)			
Comparators	16 (0.035)	1.0 (reference)	1.0 (reference)
Patients with COVID-19	36 (0.074)	17.62 (4.25-73.15)	17.31 (4.07-73.64)
Middle (40th –79th percentile)			
Comparators	21 (0.047)	1.0 (reference)	1.0 (reference)
Patients with COVID-19	55 (0.12)	9.33 (4.02-21.67)	8.26 (3.46-19.72)
High (80th–100th percentile)			
Comparators	8 (0.039)	1.0 (reference)	1.0 (reference)
Patients with COVID-19	24 (0.14)	N/A	N/A
CCI score			
0 score			
Comparators	37 (0.036)	1.0 (reference)	1.0 (reference)
Patients with COVID-19	61 (0.061)	8.25 (3.95-17.25)	6.79 (3.12-14.75)
≥1 scores			
Comparators	8 (0.10)	1.0 (reference)	1.0 (reference)
Patients with COVID-19	54 (0.46)	N/A	N/A
BMI			
<18.5 kg/m²			
Comparators	6 (0.15)	1.0 (reference)	1.0 (reference)
Patients with COVID-19	3 (0.14)	2.84 (0.47-17.04)	2.70 (0.38-19.31)
18.5-23.0 kg/m²			
Comparators	15 (0.038)	1.0 (reference)	1.0 (reference)
Patients with COVID-19	16 (0.070)	26.54 (3.52-200.29)	21.48 (2.83-163.25)
23.0-25.0 kg/m²			
Comparators	12 (0.049)	1.0 (reference)	1.0 (reference)
Patients with COVID-19	13 (0.091)	11.02 (2.48-48.95)	11.51 (2.57-51.54)
≥25.0 kg/m²			
Comparators	12 (0.028)	1.0 (reference)	1.0 (reference)

Patients with COVID-19	37 (0.147)	19.63 (6.05-63.68)	16.15 (4.95-52.73)
Unknown			
Comparators	0 (0.00)	1.0 (reference)	1.0 (reference)
Patients with COVID-19	46 (0.10)	N/A	N/A
			
Smoking status			
Non-smoker			
Comparators	25 (0.033)	1.0 (reference)	1.0 (reference)
Patients with COVID-19	50 (0.11)	13.10 (5.61-30.56)	11.09 (4.73-25.99)
Smoker			
Comparators	20 (0.055)	1.0 (reference)	1.0 (reference)
Patients with COVID-19	19 (0.10)	16.91 (3.94-72.67)	16.11 (3.73-69.58)
Unknown			
Comparators	0 (0.00)	1.0 (reference)	1.0 (reference)
Patients with COVID-19	46 (0.10)	N/A	N/A
			
Alcohol consumption			
Non-drinker			
Comparators	28 (0.041)	1.0 (reference)	1.0 (reference)
Patients with COVID-19	46 (0.12)	15.83 (6.28-39.88)	13.73 (5.43-34.71)
Drinker			
Comparators	17 (0.039)	1.0 (reference)	1.0 (reference)
Patients with COVID-19	23 (0.089)	11.81 (3.54-39.37)	10.20 (3.03-34.35)
Unknown			
Comparators	0 (0.00)	1.0 (reference)	1.0 (reference)
Patients with COVID-19	46 (0.10)	N/A	N/A
			
Physical activity			
Insufficient physical activity			
Comparators	23 (0.040)	1.0 (reference)	1.0 (reference)
Patients with COVID-19	36 (0.11)	18.51 (5.70-60.16)	15.87 (4.87-51.79)
Sufficient physical activity			
Comparators	22 (0.041)	1.0 (reference)	1.0 (reference)
Patients with COVID-19	33 (0.11)	11.29 (4.41-28.94)	9.81 (3.80-25.28)

Unknown

Comparators	0 (0.00)	1.0 (reference)	1.0 (reference)
Patients with COVID-19	46 (0.10)	N/A	N/A

BMI, body mass index; CCI, Charlson comorbidity index; CI, confidence interval; HR, hazard ratio; NA, not available; SARS-CoV-2, severe acute respiratory syndrome coronavirus 2.

The data in bold indicate significant differences ($P < 0.05$).

***Model 1:** Adjusted for age (20–39, 40–59, and ≥ 60 years) and sex.

†Model 2 (main): Adjusted for age (20–39, 40–59, and ≥ 60 years); sex, household income (low income, middle income, and high income); region of residence (urban and rural); CCI score (0, 1, and ≥ 2); obesity (underweight [$< 18.5 \text{ kg/m}^2$], normal [$18.5\text{--}22.9 \text{ kg/m}^2$]; overweight [$23.0\text{--}24.9 \text{ kg/m}^2$], obese [$\geq 25.0 \text{ kg/m}^2$], and unknown); blood pressure (systolic blood pressure $< 140 \text{ mmHg}$ and diastolic blood pressure $< 90 \text{ mmHg}$, systolic blood pressure $\geq 140 \text{ mmHg}$ or diastolic blood pressure $\geq 90 \text{ mmHg}$, and unknown); fasting blood glucose (< 100 , $\geq 100 \text{ mg/dL}$, and unknown); serum total cholesterol (< 200 , $200\text{--}239$, $\geq 240 \text{ mg/dL}$, and unknown); glomerular filtration rate (< 60 , $60\text{--}89$, $\geq 90 \text{ mL/min/1.73 m}^2$, and unknown); smoking status (never, former, current smoker, and unknown); alcoholic drinks (< 1 , $1\text{--}2$, $3\text{--}4$, ≥ 5 days per week, and unknown); aerobic physical activity (sufficient, insufficient, and unknown); previous history of cardiovascular disease, and chronic kidney disease; history of medication use for diabetes mellitus, dyslipidemia, and hypertension; and strain of SARS-CoV-2 (original and delta).

Table S21. Stratification analysis for the **long-term** risk of **post-acute respiratory sequelae** following COVID-19 in the propensity score matching cohorts (COVID-19 vs. general population) of Japan (replication)

	COVID-19 vs. general population (n=3,115,606)		
	Events, n (%)	HR (95% CI)	
		Model 3*	Model 4
Sex			
Male			
Comparators (general population or patients with influenza)	21,197 (1.46)	1.0 (reference)	1.0 (reference)
Patients with COVID-19	24,947 (5.035)	3.56 (3.49-3.62)	3.34 (3.28-3.40)
Female			
Comparators	14,103 (1.62)	1.0 (reference)	1.0 (reference)
Patients with COVID-19	16,127 (5.35)	3.41 (3.34-3.49)	3.29 (3.21-3.36)
Age			
20–39 y			
Comparators	12,546 (1.43)	1.0 (reference)	1.0 (reference)
Patients with COVID-19	14,595 (4.83)	3.47 (3.39-3.56)	3.38 (3.30-3.46)
40–59 years			
Comparators	18,329 (1.53)	1.0 (reference)	1.0 (reference)
Patients with COVID-19	20,739 (5.042)	3.41 (3.35-3.48)	3.22 (3.15-3.28)
≥60 y			
Comparators	4425 (1.84)	1.0 (reference)	1.0 (reference)
Patients with COVID-19	5740 (6.89)	3.93 (3.77-4.08)	3.56 (3.42-3.70)
CCI score			
0 score			
Comparators	34,452 (1.50)	1.0 (reference)	1.0 (reference)

Patients with COVID-19	36,256 (4.75)	3.28 (3.23-3.33)	3.28 (3.23-3.33)
≥1 scores			
Comparators	848 (3.43)	1.0 (reference)	1.0 (reference)
Patients with COVID-19	4818 (14.58)	4.15 (3.86-4.47)	4.09 (3.80-4.40)
BMI			
<18.5 kg/m²			
Comparators	24,324 (1.43)	1.0 (reference)	1.0 (reference)
Patients with COVID-19	28,166 (4.87)	3.53 (3.47-3.59)	3.36 (3.31-3.42)
18.5-23.0 kg/m²			
Comparators	8232 (1.70)	1.0 (reference)	1.0 (reference)
Patients with COVID-19	9483 (5.58)	3.39 (3.29-3.49)	3.19 (3.10-3.29)
23.0-25.0 kg/m²			
Comparators	2681 (2.10)	1.0 (reference)	1.0 (reference)
Patients with COVID-19	3368 (7.13)	3.45 (3.28-3.63)	3.21 (3.05-3.38)
≥25.0 kg/m²			
Comparators	63 (1.91)	1.0 (reference)	1.0 (reference)
Patients with COVID-19	57 (4.65)	2.57 (1.80-3.68)	2.50 (1.73-3.59)
Smoking status			
Non-smoker			
Comparators	34,154 (1.53)	1.0 (reference)	1.0 (reference)
Patients with COVID-19	39,793 (5.18)	3.50 (3.45-3.55)	3.31 (3.26-3.36)
Smoker			
Comparators	1146 (1.32)	1.0 (reference)	1.0 (reference)
Patients with COVID-19	1281 (4.34)	3.50 (3.23-3.79)	3.37 (3.11-3.66)
Alcohol consumption			
Non-drinker			
Comparators	18,088 (1.47)	1.0 (reference)	1.0 (reference)

Patients with COVID-19	20,797 (4.88)	3.44 (3.37-3.51)	3.27 (3.20-3.34)
Drinker			
Comparators	14,746 (1.60)	1.0 (reference)	1.0 (reference)
Patients with COVID-19	17,402 (5.58)	3.58 (3.50-3.66)	3.38 (3.31-3.46)
Unknown			
Comparators	2466 (1.45)	1.0 (reference)	1.0 (reference)
Patients with COVID-19	2875 (4.87)	3.49 (3.31-3.69)	3.36 (3.18-3.55)
Physical activity			
Insufficient physical activity			
Comparators	7205 (1.45)	1.0 (reference)	1.0 (reference)
Patients with COVID-19	7543 (4.68)	3.36 (3.25-3.47)	3.20 (3.09-3.30)
Sufficient physical activity			
Comparators	25,121 (1.56)	1.0 (reference)	1.0 (reference)
Patients with COVID-19	30,100 (5.35)	3.52 (3.47-3.58)	3.34 (3.28-3.40)
Unknown			
Comparators	2974 (1.38)	1.0 (reference)	1.0 (reference)
Patients with COVID-19	3431 (4.66)	3.55 (3.38-3.73)	3.39 (3.23-3.56)
Strain of SARS-CoV-2			
Original			
Comparators	27,023 (2.75)	1.0 (reference)	1.0 (reference)
Patients with COVID-19	29,447 (8.78)	3.31 (3.26-3.36)	3.13 (3.08-3.18)
Delta			
Comparators	8277 (0.62)	1.0 (reference)	1.0 (reference)
Patients with COVID-19	11,627 (2.52)	4.13 (4.01-4.24)	3.94 (3.83-4.05)

BMI, body mass index; CCI, Charlson comorbidity index; CI, confidence interval; HR, hazard ratio; NA, not available; SARS-CoV-2, severe acute respiratory syndrome coronavirus 2.

The data in bold indicate significant differences ($P < 0.05$).

***Model 3:** Adjusted for age (20–39, 40–59, and ≥ 60 years) and sex.

|| **Model 4 (replication):** Adjusted for age (20–39, 40–59, and ≥ 60 years); sex; insurance status (insured and dependent); CCI score (0, 1, and ≥ 2); body mass index (underweight [$< 18.5 \text{ kg/m}^2$], normal [$18.5\text{--}22.9 \text{ kg/m}^2$], overweight [$23.0\text{--}25.0 \text{ kg/m}^2$], obese [$\geq 25.0 \text{ kg/m}^2$], and unknown); blood pressure (systolic blood pressure $< 140 \text{ mmHg}$ and diastolic blood pressure $< 90 \text{ mmHg}$, systolic blood pressure $\geq 140 \text{ mmHg}$ or diastolic blood pressure $\geq 90 \text{ mmHg}$, and unknown); fasting blood glucose (< 100 , $\geq 100 \text{ mg/dL}$, and unknown); serum total cholesterol (< 200 , $200\text{--}239$, $\geq 240 \text{ mg/dL}$, and unknown); glomerular filtration rate (< 60 , $60\text{--}89$, $\geq 90 \text{ mL/min/1.73 m}^2$, and unknown); smoking status (non- and current smoker, and unknown); alcoholic drinks (rarely, sometimes, everyday, and unknown); aerobic physical activity (sufficient, insufficient, and unknown); previous history of cardiovascular disease, and chronic kidney disease; history of medication use for diabetes mellitus, dyslipidemia, and hypertension; and strain of SARS-CoV-2 (original and delta).

Table S22. Stratification analysis for the **long-term** risk of **post-acute respiratory sequelae** following COVID-19 in the propensity score matching cohorts (COVID-19 vs. influenza) of Japan (replication)

	Events, n (%)	COVID-19 vs. influenza (n =169,924)	
		Model 3*	Model 4 ^{II}
Sex			
Male			
Comparators (general population or patients with influenza)	526 (1.00)	1.0 (reference)	1.0 (reference)
Patients with COVID-19	2938 (5.54)	5.76 (5.25-6.32)	5.41 (4.93-5.94)
Female			
Comparators	379 (1.18)	1.0 (reference)	1.0 (reference)
Patients with COVID-19	1819 (5.69)	5.00 (4.48-5.59)	4.83 (4.33-5.40)
Age			
20–39 y			
Comparators	373 (1.25)	1.0 (reference)	1.0 (reference)
Patients with COVID-19	1556 (5.19)	4.29 (3.83-4.80)	4.19 (3.75-4.70)
40–59 years			
Comparators	429 (0.92)	1.0 (reference)	1.0 (reference)
Patients with COVID-19	2567 (5.51)	6.23 (5.63-6.90)	5.92 (5.34-6.56)
≥60 y			
Comparators	103 (1.25)	1.0 (reference)	1.0 (reference)
Patients with COVID-19	634 (7.51)	6.30 (5.11-7.75)	5.64 (4.57-6.96)
CCI score			
0 score			
Comparators	877 (1.05)	1.0 (reference)	1.0 (reference)
Patients with COVID-19	4196 (5.15)	5.11 (4.75-5.49)	5.11 (4.75-5.50)
≥1 scores			

Comparators	28 (2.43)	1.0 (reference)	1.0 (reference)
Patients with COVID-19	561 (16.01)	6.61 (4.52-9.66)	6.63 (4.53-9.71)
BMI			
<18.5 kg/m²			
Comparators	636 (1.03)	1.0 (reference)	1.0 (reference)
Patients with COVID-19	3290 (5.35)	5.39 (4.95-5.86)	5.16 (4.74-5.62)
18.5-23.0 kg/m²			
Comparators	195 (1.08)	1.0 (reference)	1.0 (reference)
Patients with COVID-19	1075 (5.88)	5.67 (4.86-6.60)	5.36 (4.59-6.24)
23.0-25.0 kg/m²			
Comparators	73 (1.47)	1.0 (reference)	1.0 (reference)
Patients with COVID-19	387 (7.58)	5.26 (4.10-6.76)	4.84 (3.76-6.23)
≥25.0 kg/m²			
Comparators	1 (0.85)	1.0 (reference)	1.0 (reference)
Patients with COVID-19	5 (3.88)	4.87 (0.56-42.31)	3.58 (0.32-40.57)
Smoking status			
Non-smoker			
Comparators	868 (1.06)	1.0 (reference)	1.0 (reference)
Patients with COVID-19	4604 (5.62)	5.49 (5.11-5.90)	5.20 (4.83-5.59)
Smoker			
Comparators	37 (1.23)	1.0 (reference)	1.0 (reference)
Patients with COVID-19	153 (4.96)	4.36 (3.04-6.25)	4.29 (2.98-6.16)
Alcohol consumption			
Non-drinker			
Comparators	443 (0.99)	1.0 (reference)	1.0 (reference)
Patients with COVID-19	2411 (5.30)	5.58 (5.04-6.17)	5.28 (4.77-5.84)
Drinker			
Comparators	397 (1.18)	1.0 (reference)	1.0 (reference)
Patients with COVID-19	2012 (6.03)	5.28 (4.74-5.88)	5.01 (4.50-5.59)
Unknown			

Comparators	65 (1.03)	1.0 (reference)	1.0 (reference)
Patients with COVID-19	334 (5.44)	5.54 (4.25-7.23)	5.53 (4.23-7.22)
Physical activity			
Insufficient physical activity			
Comparators	169 (0.97)	1.0 (reference)	1.0 (reference)
Patients with COVID-19	883 (5.10)	5.50 (4.67-6.49)	5.22 (4.43-6.16)
Sufficient physical activity			
Comparators	651 (1.09)	1.0 (reference)	1.0 (reference)
Patients with COVID-19	3477 (5.80)	5.48 (5.04-5.96)	5.20 (4.78-5.65)
Unknown			
Comparators	85 (1.10)	1.0 (reference)	1.0 (reference)
Patients with COVID-19	397 (5.15)	5.01 (3.96-6.33)	4.88 (3.86-6.18)
Strain of SARS-CoV-2			
Original			
Comparators	641 (1.80)	1.0 (reference)	1.0 (reference)
Patients with COVID-19	3269 (9.19)	5.33 (4.89-5.80)	5.05 (4.64-5.50)
Delta			
Comparators	264 (0.53)	1.0 (reference)	1.0 (reference)
Patients with COVID-19	1488 (3.01)	5.73 (5.02-6.53)	5.47 (4.79-6.23)

BMI, body mass index; CCI, Charlson comorbidity index; CI, confidence interval; HR, hazard ratio; NA, not available; SARS-CoV-2, severe acute respiratory syndrome coronavirus 2.

The data in bold indicate significant differences ($P < 0.05$).

***Model 3:** Adjusted for age (20–39, 40–59, and ≥ 60 years) and sex.

|| **Model 4 (replication):** Adjusted for age (20–39, 40–59, and ≥ 60 years); sex; insurance status (insured and dependent); CCI score (0, 1, and ≥ 2); body mass index (underweight [$< 18.5 \text{ kg/m}^2$], normal [$18.5\text{--}22.9 \text{ kg/m}^2$], overweight [$23.0\text{--}25.0 \text{ kg/m}^2$], obese [$\geq 25.0 \text{ kg/m}^2$], and unknown); blood pressure (systolic blood pressure $< 140 \text{ mmHg}$ and diastolic blood pressure $< 90 \text{ mmHg}$, systolic blood pressure $\geq 140 \text{ mmHg}$ or diastolic blood pressure $\geq 90 \text{ mmHg}$, and unknown); fasting blood glucose (< 100 , $\geq 100 \text{ mg/dL}$, and unknown); serum total cholesterol (< 200 , $200\text{--}239$, $\geq 240 \text{ mg/dL}$, and unknown); glomerular filtration rate (< 60 , $60\text{--}89$, $\geq 90 \text{ mL/min/1.73 m}^2$, and unknown); smoking status (non- and current smoker, and unknown); alcoholic drinks (rarely, sometimes, everyday, and unknown); aerobic physical activity (sufficient, insufficient, and unknown); previous history of cardiovascular disease, and chronic kidney disease; history of medication use for diabetes mellitus, dyslipidemia, and hypertension; and strain of SARS-CoV-2 (original and delta).

Table S23. Stratification analysis for the **short-term risk of acute respiratory complication** following COVID-19 in the propensity score matching cohorts (COVID-19 vs. general population) of Japan (replication)

	Events, n (%)	COVID-19 vs. general population (n =3,115,606)	
		HR (95% CI)	
		Model 3*	Model 4 ^{II}
Sex			
Male			
Comparators (general population or patients with influenza)	954 (0.066)	1.0 (reference)	1.0 (reference)
Patients with COVID-19	1569 (0.32)	4.84 (4.46-5.24)	4.36 (4.01-4.73)
Female			
Comparators	514 (0.059)	1.0 (reference)	1.0 (reference)
Patients with COVID-19	735 (0.24)	4.16 (3.72-4.66)	3.82 (3.40-4.28)
Age			
20–39 y			
Comparators	466 (0.053)	1.0 (reference)	1.0 (reference)
Patients with COVID-19	721 (0.24)	4.51 (4.01-5.07)	4.39 (3.91-4.94)
40–59 years			
Comparators	817 (0.068)	1.0 (reference)	1.0 (reference)
Patients with COVID-19	1176 (0.29)	4.23 (3.87-4.63)	3.77 (3.44-4.13)
≥60 y			
Comparators	185 (0.077)	1.0 (reference)	1.0 (reference)
Patients with COVID-19	407 (0.49)	6.43 (5.41-7.65)	5.33 (4.45-6.37)
CCI score			
0 score			
Comparators	1409 (0.061)	1.0 (reference)	1.0 (reference)
Patients with COVID-19	1923 (0.25)	4.16 (3.88-4.46)	4.14 (3.86-4.43)

≥1 scores			
Comparators	59 (0.24)	1.0 (reference)	1.0 (reference)
Patients with COVID-19	381 (1.15)	4.32 (3.28-5.68)	4.40 (3.34-5.79)
BMI			
<18.5 kg/m²			
Comparators	1097 (0.064)	1.0 (reference)	1.0 (reference)
Patients with COVID-19	1636 (0.28)	4.44 (4.12-4.80)	4.05 (3.75-4.38)
18.5-23.0 kg/m²			
Comparators	288 (0.059)	1.0 (reference)	1.0 (reference)
Patients with COVID-19	508 (0.30)	5.04 (4.37-5.83)	4.51 (3.89-5.22)
23.0-25.0 kg/m²			
Comparators	82 (0.064)	1.0 (reference)	1.0 (reference)
Patients with COVID-19	154 (0.33)	4.90 (3.75-6.41)	4.37 (3.32-5.74)
≥25.0 kg/m²			
Comparators	1 (0.030)	1.0 (reference)	1.0 (reference)
Patients with COVID-19	6 (0.49)	16.98 (2.04-141.11)	18.83 (2.12-167.43)
Smoking status			
Non-smoker			
Comparators	1396 (0.063)	1.0 (reference)	1.0 (reference)
Patients with COVID-19	2211 (0.29)	4.63 (4.33-4.95)	4.19 (3.91-4.49)
Smoker			
Comparators	72 (0.083)	1.0 (reference)	1.0 (reference)
Patients with COVID-19	93 (0.32)	4.05 (2.97-5.51)	3.84 (2.82-5.24)
Alcohol consumption			
Non-drinker			
Comparators	752 (0.061)	1.0 (reference)	1.0 (reference)
Patients with COVID-19	1193 (0.28)	4.64 (4.24-5.09)	4.23 (3.86-4.65)
Drinker			
Comparators	590 (0.064)	1.0 (reference)	1.0 (reference)

Patients with COVID-19	935 (0.30)	4.65 (4.19-5.15)	4.16 (3.75-4.62)
Unknown			
Comparators	126 (0.074)	1.0 (reference)	1.0 (reference)
Patients with COVID-19	176 (0.30)	4.11 (3.27-5.17)	3.91 (3.10-4.92)
Physical activity			
Insufficient physical activity			
Comparators	339 (0.068)	1.0 (reference)	1.0 (reference)
Patients with COVID-19	471 (0.29)	4.36 (3.79-5.02)	3.91 (3.39-4.50)
Sufficient physical activity			
Comparators	967 (0.060)	1.0 (reference)	1.0 (reference)
Patients with COVID-19	1610 (0.29)	4.76 (4.39-5.15)	4.30 (3.97-4.66)
Unknown			
Comparators	162 (0.075)	1.0 (reference)	1.0 (reference)
Patients with COVID-19	223 (0.30)	4.18 (3.42-5.12)	3.97 (3.23-4.87)
Strain of SARS-CoV-2			
Original			
Comparators	612 (0.062)	1.0 (reference)	1.0 (reference)
Patients with COVID-19	951 (0.28)	4.58 (4.14-5.07)	4.01 (3.61-4.45)
Delta			
Comparators	856 (0.064)	1.0 (reference)	1.0 (reference)
Patients with COVID-19	1353 (0.29)	4.59 (4.21-5.00)	4.28 (3.92-4.66)

BMI, body mass index; CCI, Charlson comorbidity index; CI, confidence interval; HR, hazard ratio; NA, not available; SARS-CoV-2, severe acute respiratory syndrome coronavirus 2.

The data in bold indicate significant differences ($P < 0.05$).

***Model 3:** Adjusted for age (20–39, 40–59, and ≥ 60 years) and sex.

|| **Model 4 (replication):** Adjusted for age (20–39, 40–59, and ≥ 60 years); sex; insurance status (insured and dependent); CCI score (0, 1, and ≥ 2); body mass index (underweight [$< 18.5 \text{ kg/m}^2$], normal [$18.5\text{--}22.9 \text{ kg/m}^2$], overweight [$23.0\text{--}25.0 \text{ kg/m}^2$], obese [$\geq 25.0 \text{ kg/m}^2$], and unknown); blood pressure (systolic blood pressure $< 140 \text{ mmHg}$ and diastolic blood pressure $< 90 \text{ mmHg}$, systolic blood pressure $\geq 140 \text{ mmHg}$ or diastolic blood pressure $\geq 90 \text{ mmHg}$, and unknown); fasting blood glucose (< 100 , $\geq 100 \text{ mg/dL}$, and unknown); serum total cholesterol (< 200 , $200\text{--}239$, $\geq 240 \text{ mg/dL}$, and unknown); glomerular filtration rate (< 60 , $60\text{--}89$, $\geq 90 \text{ mL/min/1.73 m}^2$, and unknown); smoking status

(non- and current smoker, and unknown); alcoholic drinks (rarely, sometimes, everyday, and unknown); aerobic physical activity (sufficient, insufficient, and unknown); previous history of cardiovascular disease, and chronic kidney disease; history of medication use for diabetes mellitus, dyslipidemia, and hypertension; and strain of SARS-CoV-2 (original and delta).

Table S24. Stratification analysis for the **short-term** risk of **acute respiratory complication** following COVID-19 in the propensity score matching cohorts (COVID-19 vs. influenza) of Japan (replication)

	Events, n (%)	COVID-19 vs. influenza (n =169,924)	
		HR (95% CI)	
		Model 3*	Model 4 ^{II}
Sex			
Male			
Comparators (general population or patients with influenza)	80 (0.15)	1.0 (reference)	1.0 (reference)
Patients with COVID-19	641 (1.21)	7.97 (6.32-10.05)	6.76 (5.35-8.54)
Female			
Comparators	42 (0.13)	1.0 (reference)	1.0 (reference)
Patients with COVID-19	283 (0.89)	6.80 (4.92-9.40)	6.04 (4.36-8.37)
Age			
20–39 y			
Comparators	31 (0.10)	1.0 (reference)	1.0 (reference)
Patients with COVID-19	205 (0.68)	6.61 (4.53-9.64)	6.21 (4.25-9.06)
40–59 years			
Comparators	64 (0.14)	1.0 (reference)	1.0 (reference)
Patients with COVID-19	535 (1.15)	8.42 (6.50-10.91)	7.05 (5.43-9.16)
≥60 y			
Comparators	27 (0.33)	1.0 (reference)	1.0 (reference)
Patients with COVID-19	184 (2.18)	6.67 (4.45-9.99)	5.61 (3.73-8.44)
CCI score			
0 score			
Comparators	107 (0.13)	1.0 (reference)	1.0 (reference)

Patients with COVID-19	694 (0.85)	6.71 (5.47-8.22)	6.71 (5.47-8.22)
≥1 scores			
Comparators	15 (1.30)	1.0 (reference)	1.0 (reference)
Patients with COVID-19	230 (6.56)	4.88 (2.89-8.23)	4.82 (2.85-8.16)
BMI			
<18.5 kg/m²			
Comparators	81 (0.13)	1.0 (reference)	1.0 (reference)
Patients with COVID-19	629 (1.02)	7.84 (6.22-9.88)	6.94 (5.50-8.76)
18.5-23.0 kg/m²			
Comparators	29 (0.16)	1.0 (reference)	1.0 (reference)
Patients with COVID-19	214 (1.17)	7.24 (4.91-10.67)	5.90 (3.99-8.74)
23.0-25.0 kg/m²			
Comparators	12 (0.24)	1.0 (reference)	1.0 (reference)
Patients with COVID-19	81 (1.59)	6.52 (3.55-11.95)	5.29 (2.86-9.79)
≥25.0 kg/m²			
Comparators	0 (0.00)	1.0 (reference)	1.0 (reference)
Patients with COVID-19	0 (0.00)	1.56 (1.54 to 1.59)	1.54 (1.51 to 1.57)
Smoking status			
Non-smoker			
Comparators	118 (0.14)	1.0 (reference)	1.0 (reference)
Patients with COVID-19	892 (1.09)	7.57 (6.25-9.17)	6.50 (5.36-7.89)
Smoker			
Comparators	4 (0.13)	1.0 (reference)	1.0 (reference)
Patients with COVID-19	32 (1.04)	8.00 (2.83-22.64)	6.43 (2.24-18.45)
Alcohol consumption			
Non-drinker			
Comparators	54 (0.12)	1.0 (reference)	1.0 (reference)

Patients with COVID-19	471 (1.04)	8.64 (6.52-11.45)	7.28 (5.48-9.66)
Drinker			
Comparators	60 (0.18)	1.0 (reference)	1.0 (reference)
Patients with COVID-19	398 (1.19)	6.72 (5.12-8.81)	5.87 (4.47-7.72)
Unknown			
Comparators	8 (0.13)	1.0 (reference)	1.0 (reference)
Patients with COVID-19	55 (0.90)	6.92 (3.30-14.53)	6.22 (2.95-13.14)
			
Physical activity			
Insufficient physical activity			
Comparators	24 (0.14)	1.0 (reference)	1.0 (reference)
Patients with COVID-19	184 (1.06)	7.76 (5.07-11.87)	6.78 (4.42-10.41)
Sufficient physical activity			
Comparators	89 (0.15)	1.0 (reference)	1.0 (reference)
Patients with COVID-19	662 (1.10)	7.42 (5.95-9.26)	6.32 (5.06-7.90)
Unknown			
Comparators	9 (0.12)	1.0 (reference)	1.0 (reference)
Patients with COVID-19	78 (1.01)	8.56 (4.30-17.07)	7.55 (3.77-15.11)
			
Strain of SARS-CoV-2			
Original			
Comparators	59 (0.17)	1.0 (reference)	1.0 (reference)
Patients with COVID-19	496 (1.39)	8.41 (6.42-11.02)	6.81 (5.18-8.94)
Delta			
Comparators	63 (0.13)	1.0 (reference)	1.0 (reference)
Patients with COVID-19	428 (0.87)	6.80 (5.22-8.85)	6.21 (4.76-8.10)

BMI, body mass index; CCI, Charlson comorbidity index; CI, confidence interval; HR, hazard ratio; NA, not available; SARS-CoV-2, severe acute respiratory syndrome coronavirus 2.

The data in bold indicate significant differences ($P < 0.05$).

***Model 3:** Adjusted for age (20–39, 40–59, and ≥60 years) and sex.

|| **Model 4 (replication):** Adjusted for age (20–39, 40–59, and ≥60 years); sex; insurance status (insured and dependent); CCI score (0, 1, and ≥ 2); body mass index (underweight [$<18.5 \text{ kg/m}^2$], normal [$18.5\text{--}22.9 \text{ kg/m}^2$], overweight [$23.0\text{--}25.0 \text{ kg/m}^2$], obese [$\geq 25.0 \text{ kg/m}^2$], and unknown); blood pressure (systolic blood pressure $<140 \text{ mmHg}$ and diastolic blood pressure $<90 \text{ mmHg}$, systolic blood pressure $\geq 140 \text{ mmHg}$ or diastolic blood pressure $\geq 90 \text{ mmHg}$, and unknown); fasting blood glucose (<100 , $\geq 100 \text{ mg/dL}$, and unknown); serum total cholesterol (<200 , $200\text{--}239$, $\geq 240 \text{ mg/dL}$, and unknown); glomerular filtration rate (<60 , $60\text{--}89$, $\geq 90 \text{ mL/min/1.73 m}^2$, and unknown); smoking status (non- and current smoker, and unknown); alcoholic drinks (rarely, sometimes, everyday, and unknown); aerobic physical activity (sufficient, insufficient, and unknown); previous history of cardiovascular disease, and chronic kidney disease; history of medication use for diabetes mellitus, dyslipidemia, and hypertension; and strain of SARS-CoV-2 (original and delta).

Comment 4.

Similarly, the authors use data from Korea as a discovery cohort and from Japan as a validation cohort. However, the estimates differ significantly between these two countries, making it difficult to quantify the exact risk associated with long-term sequelae. Both countries have major differences in their healthcare systems, treatment guidelines for COVID-19, and data acquisition methods, which could significantly impact risk estimates. The authors might consider presenting both datasets on equal terms, rather than as discovery versus validation cohorts, which are terms more commonly used in machine learning approaches.

Response:

Thank you for your valuable comment about the difference in the estimates and, therefore, the necessity of changing the terms. We originally intended to show the analyses of South Korea to evaluate our findings and use the Japanese dataset for validation, hence using the terms ‘discovery’ and ‘validation’ cohorts. However, we understood that such terms may cause misunderstanding for the readers of our research since, as you mentioned, the countries have major differences in their healthcare systems, treatment guidelines for COVID-19, and data acquisition methods. Therefore, we changed the terms to ‘main’ and ‘replication’ cohort. The changes in the terms throughout the manuscript are shown below. Thank you for guiding us in using appropriate terms for our research that would enhance its accuracy.

Changes in text:**Title**

- Acute respiratory complication and long-term respiratory sequelae in long COVID in South Korea and Japan: binational population-based **main** and **replication** cohort study

Abstract

- We conducted a binational population-based **main and replication cohort** study to analyze the risk of acute respiratory complication or long-term post-acute respiratory sequelae after SARS-CoV-2 infection. We used a Korean nationwide claim-based cohort (K-CoV-2 cohort; total N=10,027,506) for the **main cohort** and a Japanese claim-based cohort (JMDC cohort; total N=4,909,861) for the **replication cohort**.

Methods/Data source

- Utilizing large-scale, population-based binational cohorts, this study incorporated a South Korean nationwide claim-based cohort (K-CoV-N cohort; total N=10,027,506) for the **main cohort** and a Japanese claim-based cohort (JMDC cohort; total N=4,909,861) for the **replication cohort** (Figure S1).

Methods/K-CoV-N cohort for **main cohort**

- We utilized the NHIS database, which is a large-scale, nationwide, general, population-based cohort in South Korea, covering 98% of the population for the **main cohort**.

Methods/Replication cohort** in Japan**

- We employed the same definition of ICD-10 codes, exposure, and outcome assessments, general health examination, follow-up duration, and propensity score matching for the JMDC cohort (total N=4,909,861) as we did with the **main cohort**. However, due to the lack of COVID-19 vaccination data in the JMDC cohort, we utilized this cohort primarily to validate the findings from the **main cohort**. Supplementary material was provided to address more detailed information about the validation cohort, which was caused by the different components and structures of the **main cohort** (**Supplementary material**).

Results

- In the **main cohort**, there were a total of 10,027,506 participants with a mean age of 48.4 (standard deviation [SD], 13.4) years, of which 49.9% (5,000,621/10,027,506) were female (**Table S8**). The **replication cohort** includes 4,909,861 participants with a mean age of 46.8 (SD, 11.9) years and 38.3% (1,882,174/4,909,861) females (**Table S9**).

- SMDs of all matching covariates in both multi-to-one propensity score-matched **main and replication cohorts** were smaller than 0.1 (**Table 1**).

- In the **main and replication cohorts**, individuals with SARS-CoV-2 infection had a higher adjusted HR for long-term post-acute respiratory sequelae compared to the general population (**main**: HR, 1.68 [95% CI, 1.62-1.75]; **replication**: HR, 3.32 [95% CI, 3.27-3.37]) in **Table 2**. Furthermore, patients with SARS-CoV-2 infection had an increased risk for acute respiratory complication compared to non-infected controls (**main**: HR, 8.06 [95% CI, 6.92-9.38]; **replication**: HR, 4.17 [95% CI, 3.90-4.45]). When directly comparing the risk for acute respiratory complication between SARS-CoV-2 and influenza infections, SARS-CoV-2

infection was significantly associated with an increased risk (**main**: HR, 4.32 [95% CI, 2.73-6.83]; **replication**: HR, 6.51 [95% CI, 5.38-7.87]) in **Tables S11-S13**.

- Relative to the general population, patients with SARS-CoV-2 infection had significantly increased risk for several subtypes of long-term respiratory sequelae, including chronic respiratory failure (**main**: HR, 8.92 [95% CI, 4.92-16.17]; **replication**: HR, 7.55 [95% CI, 6.35-8.97]), COPD, emphysema, asthma, pulmonary sarcoidosis, and interstitial lung disease (**main**: HR, 10.38 [95% CI, 8.75-12.31]; **replication**: HR, 4.75 [95% CI, 4.54-4.97]) in **Table 3**. Notably, the risk for acute respiratory complication, including aspergillosis pneumonia (**main**: HR, 6.85 [95% CI, 3.48-13.50]; **replication**: HR, 4.97 [95% CI, 4.26-5.79]), pneumothorax, acute respiratory failure (**main**: HR, 112.04 [95% CI, 64.00-196.16]; **replication**: HR, 6.49 [95% CI, 6.32-6.65]) showed an increase in patients with SARS-CoV-2 infection compared to the general population.

- The first three months after infection with SARS-CoV-2 had the highest risk of developing long-term respiratory sequelae (**main**: HR, 2.51 [95% CI, 2.38-2.64]; **replication**: HR, 4.40 [95% CI, 4.30-4.51]). With increasing duration post-SARS-CoV-2 infection, the risk of long-term post-acute respiratory sequelae significantly decreased, but the risk remained even after 6 months (**main**: HR, 1.10 [95% CI, 1.01-1.19]; **replication**: HR, 2.67 [95% CI, 2.61-2.73]).

Discussion/Comparisons with previous studies

- Therefore, our study is distinct from other studies in that we compared the association of COVID-19 and respiratory diseases with that of influenza by using population-based binational cohorts with a generalizable scale (**main cohort**, N=10,027,506; **replication cohort**, N=4,909,861).

Discussion/Limitations and strengths

- Therefore, we opted against merging the datasets, using the K-CoV-N data for the **main cohort** and the JMDC data for the **replication cohort**. In addition, we used different lists of the covariates for each **main and replication cohort** due to the difference in data structure.

Tables 1-5 and Tables S2-S25

We have changed all related terms in the table. Please refer to the attached tables for Comment 4.

Comment 5.

One significant contribution of this paper is the results reported in Table 3, where the author details individual conditions of long-term sequelae, potentially contributing to the definition of long-COVID conditions. This table could be strengthened by providing the marginal predicted prevalence for each condition, aiding in understanding the risk and prevalence of these conditions post COVID.

Response:

Thank you for the constructive comment. We deeply appreciate your acknowledgment of our contribution regarding the results reported in Table 3, where the author details individual conditions of long-term sequelae, potentially contributing to the definition of long-COVID conditions. Also, we appreciate your suggestion to enhance Table 3 by providing the marginal predicted prevalence for each condition.

In response to the suggestion of providing marginal predicted prevalence for each condition, we have expanded our analysis by employing the average treatment effect in the overlap (ATO) method. Specifically, we calculated the overlap-weighted hazard ratios to assess the risk and prevalence of acute respiratory complications and long-term respiratory sequelae following a COVID-19 diagnosis. This methodological enhancement enables us to provide a detailed understanding of each condition risk, adjusted for the common support between exposed and non-exposed populations, thereby improving the robustness of our findings. Based on your recommendation, additional analyses on the marginal predicted prevalence have been incorporated into Tables S5-S7 below. We hope these revisions can address your concerns and appreciate the opportunity to refine our manuscript based on your valuable insights.

Changes in text:

Methods/Sensitivity analysis

- Fourth, to thoroughly evaluate the risks and prevalence of acute and long-term respiratory conditions post-COVID-19, we utilized the average treatment effect in the overlap (ATO) method, calculating overlap-weighted hazard ratios (**Tables S5-S7**).²⁴

Added references

24 Thomas, L. E., Li, F. & Pencina, M. J. Overlap Weighting: A Propensity Score Method That Mimics Attributes of a Randomized Clinical Trial. *JAMA* **323**, 2417-2418 (2020). <https://doi.org/10.1001/jama.2020.7819>

Table S5. Baseline characteristics for the overlap-weighted cohort in South Korea (main)

Characteristic	Main cohort (n=754,742)		SMD
	COVID-19 (n=377,371)	Non-COVID-19 (n=377,371)	
Mean age (SD), y	47.6 (16.5)	46.9 (2.9)	0.053
Age, n (%)			<0.001
20–39 y	136,554 (36.2)	136,554 (36.2)	
40–59 y	139,057 (36.9)	139,056 (36.9)	
≥60 y	101,760 (27.0)	101,761 (27.0)	
Sex, n (%)			<0.001
Male	196,487 (52.1)	196,487 (52.1)	
Female	180,884 (47.9)	180,884 (47.9)	
Region of residence, n (%)			<0.001
Urban	203,682 (54.0)	203,682 (54.0)	
Rural	173,689 (46.0)	173,689 (46.0)	
Medical history, n (%)			
Cardiovascular disease	17,644 (4.7)	17,644 (4.7)	<0.001
Chronic kidney disease	8406 (2.2)	8405 (2.2)	<0.001
Medication use for diabetes	68,892 (18.3)	68,892 (18.3)	<0.001
Medication use for hyperlipidemia	57,510 (15.2)	57,510 (15.2)	<0.001
Medication use for hypertension	31,163 (8.3)	31,163 (8.3)	<0.001
Unmatching covariates, n (%)[†]			
Charlson Comorbidity Index score			0.230
0	331,387 (87.8)	355,625 (94.2)	
1	29,236 (7.8)	11,726 (3.1)	
≥2	16,748 (4.4)	10,020 (2.7)	
Household income			<0.001

Low (0th–39th percentile)	174,656 (46.3)	174,656 (46.3)	
Middle (40th –79th percentile)	134,003 (35.5)	134,003 (35.5)	
High (80th–100th percentile)	68,712 (18.2)	68,712 (18.2)	
Body mass index			1.212
Underweight (<18.5 kg/m ²)	6581 (1.7)	13,600 (3.6)	
Normal (18.5-22.9 kg/m ²)	74,312 (19.7)	135,565 (35.9)	
Overweight (23.0-24.9 kg/m ²)	52,016 (13.8)	85,947 (22.8)	
Obese (≥25.0 kg/m ²)	91,218 (24.2)	142,215 (37.7)	
Unknown	153,244 (40.6)	45 (0.01)	
Blood pressure			1.181
SBP <140 mmHg and DBP <90 mmHg	190,480 (50.5)	329,221 (87.2)	
SBP ≥140mmHg or DBP ≥90 mmHg	32,542 (8.6)	47,369 (12.6)	
Unknown	154,349 (40.9)	781 (0.2)	
Fasting blood glucose			1.181
<100 mg/dL	133,935 (35.5)	234,987 (62.3)	
≥100 mg/dL	89,068 (23.6)	141,591 (37.5)	
Unknown	154,368 (40.9)	793 (0.2)	
Serum total cholesterol			0.395
<200 mg/dL	64,465 (17.1)	103,189 (27.3)	
200 to 239 mg/dL	37,829 (10.0)	61,589 (16.3)	
≥240 mg/dL	16,051 (4.3)	25,767 (6.8)	
Unknown	259,026 (68.6)	186,826 (49.5)	
Glomerular filtration rate			1.180
<60 mL/min/1.73 m ²	8176 (2.2)	10,228 (2.7)	
60 to 89 mL/min/1.73 m ²	98,133 (26.0)	159,110 (42.2)	
≥90 mL/min/1.73 m ²	116,544 (30.9)	206,880 (54.8)	
Unknown	154,518 (41.0)	1153 (0.3)	
Smoking status			1.191

Never	147,844 (39.2)	239,173 (63.4)	
Former	40,867 (10.8)	59,401 (15.7)	
Current	35,423 (9.4)	78,677 (20.9)	
Unknown	153,237 (40.6)	120 (0.03)	
Alcohol consumption			1.156
<1 day/week	131,356 (34.8)	224,829 (59.6)	
1 to 2 days/week	62,664 (16.6)	107,501 (28.5)	
3 to 4 days/week	22,326 (5.9)	33,901 (9.0)	
≥5 days/week	7773 (2.1)	11,015 (2.9)	
Unknown	153,252 (40.6)	125 (0.03)	
Aerobic physical activity			1.179
Insufficient	114,084 (30.2)	187,416 (49.7)	
Sufficient	109,951 (29.1)	189,781 (50.3)	
Unknown	153,336 (40.6)	174 (0.1)	
Strain of SARS-CoV-2			0.003
Original	115,269 (30.6)	115,831 (30.7)	
Delta	262,102 (69.5)	261,540 (69.3)	

DBP, diastolic blood pressure; SARS-CoV-2, severe acute respiratory syndrome coronavirus 2; SBP, systolic blood pressure; SD, standard deviation; SMD, standardized mean difference.

* An SMD <0.1 indicates no significant imbalance. All SMDs were <0.100 in the propensity score–matched cohorts.

† Unmatched covariates were included as adjustment factors in statistical analyses.

Table S6. Baseline characteristics for the overlap-weighted cohort in Japan (replication)

Characteristic	Replication cohort (n=1,276,694)		SMD*
	COVID-19 (n=638,347)	General population (n=638,347)	
Mean age (SD), y	44.4 (10.6)	44.8 (5.2)	0.050
Age, n (%)			<0.001
20–39 y	227,172 (35.6)	227,172 (35.6)	
40–59 y	339,533 (53.2)	339,533 (53.2)	
≥60 y	71,642 (11.2)	71,642 (11.2)	
Sex, n (%)			<0.001
Male	397,791 (62.3)	397,791 (62.3)	
Female	240,556 (37.7)	240,556 (37.7)	
Insurance status, n (%)			<0.001
Insured	572,607 (89.7)	572,607 (89.7)	
Dependent	65,740 (10.3)	65,740 (10.3)	
Medical history, n (%)			
Cardiovascular disease	47,750 (7.5)	47,750 (7.5)	<0.001
Chronic kidney disease	25,451 (4.0)	25,451 (4.0)	<0.001
Medication use for diabetes	16,160 (2.5)	16,160 (2.5)	<0.001
Medication use for hyperlipidemia	40,639 (6.4)	40,639 (6.4)	<0.001
Medication use for hypertension	54,908 (8.6)	54,908 (8.6)	<0.001
Unmatching covariates, n (%)[†]			
Charlson Comorbidity Index score			0.199
0	611,433 (95.8)	631,656 (99.0)	
1	8822 (1.4)	2336 (0.4)	
≥2	18,092 (2.8)	4355 (0.7)	

Body mass index			0.023
Underweight (<18.5 kg/m ²)	340,250 (53.3)	346,047 (54.2)	
Normal (18.5-22.9 kg/m ²)	122,190 (19.1)	122,020 (19.1)	
Overweight (23.0-24.9 kg/m ²)	137,212 (21.5)	134,235 (21.0)	
Obese (≥25.0 kg/m ²)	37,728 (5.9)	35,140 (5.5)	
Unknown	967 (0.2)	905 (0.1)	
Blood pressure			0.041
SBP <140 mmHg and DBP <90 mmHg	551,836 (86.5)	544,450 (85.3)	
SBP ≥140mmHg or DBP ≥90 mmHg	38,184 (6.0)	42,382 (6.6)	
Unknown	48,327 (7.6)	51,515 (8.1)	
Fasting blood glucose			0.028
<100 mg/dL	408,131 (63.9)	404,033 (63.3)	
≥100 mg/dL	131,693 (20.6)	133,988 (21.0)	
Unknown	98,523 (15.4)	100,326 (15.7)	
Serum total cholesterol			0.042
<200 mg/dL	295,814 (46.3)	282,972 (44.3)	
200 to 239 mg/dL	230,730 (36.1)	234,567 (36.8)	
≥240 mg/dL	100,841 (15.8)	106,454 (16.7)	
Unknown	10,962 (1.7)	14,354 (2.3)	
Glomerular filtration rate			0.038
<60 mL/min/1.73 m ²	3577 (0.6)	2301 (0.4)	
60 to 89 mL/min/1.73 m ²	60,619 (9.5)	61,394 (9.6)	
≥90 mL/min/1.73 m ²	347,601 (54.5)	340,887 (53.4)	
Unknown	226,550 (35.5)	233,765 (36.6)	
Smoking status			0.062
Non-smoker	145,103 (22.7)	152,628 (23.9)	

Smoker	470,406 (73.7)	462,568 (72.5)	
Unknown	22,838 (3.6)	23,151 (3.6)	
Alcohol consumption			0.027
Everyday	131,967 (20.7)	130,385 (20.4)	
Sometimes	210,682 (33.0)	208,601 (32.7)	
Rarely	249,294 (39.1)	253,317 (39.7)	
Unknown	46,404 (7.3)	46,044 (7.2)	
Aerobic physical activity			0.027
Insufficient	130,291 (20.4)	138,213 (21.7)	
Sufficient	450,219 (70.5)	441,711 (69.2)	
Unknown	57,837 (9.1)	58,423 (9.2)	
Strain of SARS-CoV-2			0.007
Original	268,315 (42.0)	270,612 (42.4)	
Delta	370,032 (58.0)	367,735 (57.6)	

DBP, diastolic blood pressure; SARS-CoV-2, severe acute respiratory syndrome coronavirus 2; SBP, systolic blood pressure; SD, standard deviation; SMD, standardized mean difference.

* An SMD <0.1 indicates no significant imbalance. All SMDs were <0.100 in the propensity score-matched cohorts.

† Unmatched covariates were included as adjustment factors in statistical analyses.

Table S7. Overlap-weighted HR (95% CI) for the **long-term post-acute respiratory sequelae** or **short-term acute respiratory complication** subtypes following COVID-19 diagnosis of patients in binational cohorts (COVID-19 vs. general population) in South Korea (main) and Japan (replication)

	Main cohort (n=754,742)			Replication cohort (n=1,276,694)		
	Events, n (%)	HR (95% CI)		Events, n (%)	HR (95% CI)	
		Model 1*	Model 2†		Model 3*	Model 4 [‡]
Long-term post-acute respiratory sequelae						
Comparators	3008 (0.80)	1.0 (reference)	1.0 (reference)	10,645 (1.67)	1.0 (reference)	1.0 (reference)
Patients with COVID-19	4735 (1.25)	1.62 (1.55-1.70)	1.68 (1.60-1.77)	35,451 (5.55)	3.45 (3.38-3.53)	3.26 (3.19-3.33)
Chronic respiratory failure						
Comparators	3 (0.00079)	1.0 (reference)	1.0 (reference)	40 (0.0063)	1.0 (reference)	1.0 (reference)
Patients with COVID-19	30 (0.0079)	10.30 (3.18-33.35)	10.84 (3.21-36.64)	565 (0.089)	14.00 (10.17-19.27)	8.66 (6.27-11.97)
Pulmonary hypertension						
Comparators	2 (0.00053)	1.0 (reference)	1.0 (reference)	43 (0.0067)	1.0 (reference)	1.0 (reference)
Patients with COVID-19	4 (0.0011)	2.43 (0.39-15.22)	0.47 (0.01-28.71)	175 (0.027)	4.10 (2.94-5.72)	3.05 (2.17-4.28)
Sleep apnea						
Comparators	209 (0.065)	1.0 (reference)	1.0 (reference)	1883 (0.30)	1.0 (reference)	1.0 (reference)
Patients with COVID-19	230 (0.061)	1.12 (0.93-1.35)	1.17 (0.95-1.45)	4210 (0.66)	2.26 (2.14-2.39)	2.16 (2.05-2.29)
COPD						
Comparators	2059 (0.55)	1.0 (reference)	1.0 (reference)	3046 (0.48)	1.0 (reference)	1.0 (reference)
Patients with COVID-19	3052 (0.81)	1.52 (1.44-1.61)	1.54 (1.45-1.64)	12,484 (1.96)	4.17 (4.01-4.34)	3.91 (3.76-4.07)
Emphysema						
Comparators	72 (0.019)	1.0 (reference)	1.0 (reference)	431 (0.068)	1.0 (reference)	1.0 (reference)
Patients with COVID-19	103 (0.027)	1.50 (1.11-2.02)	1.52 (1.09-2.10)	1719 (0.27)	4.00 (3.60-4.44)	3.44 (3.10-3.83)

Asthma						
Comparators	706 (0.19)	1.0 (reference)	1.0 (reference)	5877 (0.92)	1.0 (reference)	1.0 (reference)
Patients with COVID-19	1189 (0.32)	1.72 (1.57-1.89)	1.88 (1.70-2.08)	20,135 (3.15)	3.50 (3.40-3.60)	3.41 (3.31-3.51)
Pulmonary sarcoidosis						
Comparators	4 (0.0011)	1.0 (reference)	1.0 (reference)	267 (0.042)	1.0 (reference)	1.0 (reference)
Patients with COVID-19	4 (0.0011)	0.95 (0.24-3.85)	1.53 (0.37-6.30)	1021 (0.16)	3.85 (3.36-4.40)	3.49 (3.04-3.99)
Interstitial lung disease						
Comparators	35 (0.0093)	1.0 (reference)	1.0 (reference)	805 (0.13)	1.0 (reference)	1.0 (reference)
Patients with COVID-19	333 (0.088)	9.85 (6.96-13.94)	10.57 (7.42-15.06)	4913 (0.77)	6.16 (5.72-6.64)	4.79 (4.44-5.17)
Acute respiratory complication						
Comparators	88 (0.023)	1.0 (reference)	1.0 (reference)	685 (0.11)	1.0 (reference)	1.0 (reference)
Patients with COVID-19	812 (0.22)	9.37 (7.52-11.68)	8.76 (6.98-10.98)	7273 (1.14)	10.64 (9.84-11.51)	9.09 (8.40-9.84)
Pneumocystis pneumonia						
Comparators	1 (0.00026)	1.0 (reference)	1.0 (reference)	112 (0.018)	1.0 (reference)	1.0 (reference)
Patients with COVID-19	0 (0.00)	NA	NA	512 (0.080)	4.55 (3.71-5.57)	3.34 (2.72-4.12)
Aspergillosis pneumonia						
Comparators	2 (0.00053)	1.0 (reference)	1.0 (reference)	15 (0.0023)	1.0 (reference)	1.0 (reference)
Patients with COVID-19	31 (0.01)	18.64 (3.91-88.82)	17.12 (3.50-83.88)	171 (0.027)	11.27 (6.68-19.02)	8.18 (4.82-13.89)
Pleural empyema						
Comparators	3 (0.00079)	1.0 (reference)	1.0 (reference)	3 (0.00047)	1.0 (reference)	1.0 (reference)
Patients with COVID-19	8 (0.0021)	2.56 (0.68-9.58)	1.99 (0.44-8.96)	120 (0.018)	35.48 (12.07-104.32)	29.66 (10.06-87.44)
Lung abscess						
Comparators	5 (0.0013)	1.0 (reference)	1.0 (reference)	3 (0.00047)	1.0 (reference)	1.0 (reference)

Patients with COVID-19	8 (0.0021)	1.56 (0.51-4.82)	1.37 (0.36-5.26)	153 (0.024)	48.16 (15.89-145.97)	44.01 (14.49-133.64)
Pneumothorax						
Comparators	17 (0.0045)	1.0 (reference)	1.0 (reference)	121 (0.019)	1.0 (reference)	1.0 (reference)
Patients with COVID-19	88 (0.023)	5.29 (3.15-8.90)	5.16 (2.99-8.89)	751 (0.12)	6.19 (5.11-7.50)	5.85 (4.82-7.09)
Acute respiratory failure						
Comparators	4 (0.0011)	1.0 (reference)	1.0 (reference)	319 (0.050)	1.0 (reference)	1.0 (reference)
Patients with COVID-19	487 (0.13)	120.86 (45.63-320.13)	106.37 (40.03-282.65)	4741 (0.74)	14.86 (13.27-16.64)	12.6 (11.25-14.12)
Pulmonary embolism						
Comparators	56 (0.015)	1.0 (reference)	1.0 (reference)	130 (0.020)	1.0 (reference)	1.0 (reference)
Patients with COVID-19	195 (0.052)	3.48 (2.59-4.68)	3.64 (2.66-4.98)	1290 (0.20)	9.88 (8.25-11.82)	8.21 (6.85-9.84)

CCI, Charlson comorbidity index; CI, confidence interval; COPD, chronic obstructive pulmonary disease; NA, not available; OR, odds ratio; SARS-CoV-2, severe acute respiratory syndrome coronavirus 2.

The data in bold indicate significant differences ($P < 0.05$).

***Model 1 and 3:** Adjusted for age (20–39, 40–59, and ≥ 60 years) and sex.

†**Model 2:** Adjusted for age (20–39, 40–59, and ≥ 60 years); sex, household income (low income, middle income, and high income); region of residence (urban and rural); CCI score (0, 1, and ≥ 2); obesity (underweight [$< 18.5 \text{ kg/m}^2$], normal [$18.5\text{--}22.9 \text{ kg/m}^2$], overweight [$23.0\text{--}24.9 \text{ kg/m}^2$], obese [$\geq 25.0 \text{ kg/m}^2$], and unknown); blood pressure (systolic blood pressure $< 140 \text{ mmHg}$ and diastolic blood pressure $< 90 \text{ mmHg}$, systolic blood pressure $\geq 140 \text{ mmHg}$ or diastolic blood pressure $\geq 90 \text{ mmHg}$, and unknown); fasting blood glucose (< 100 , $\geq 100 \text{ mg/dL}$, and unknown); serum total cholesterol (< 200 , $200\text{--}239$, $\geq 240 \text{ mg/dL}$, and unknown); glomerular filtration rate (< 60 , $60\text{--}89$, $\geq 90 \text{ mL/min/1.73 m}^2$, and unknown); smoking status (never, former, current smoker, and unknown); alcoholic drinks (< 1 , $1\text{--}2$, $3\text{--}4$, ≥ 5 days per week, and unknown); aerobic physical activity (sufficient, insufficient, and unknown); previous history of cardiovascular disease, and chronic kidney disease; history of medication use for diabetes mellitus, dyslipidemia, and hypertension; and strain of SARS-CoV-2 (original and delta).

‡ **Model 4 (replication):** Adjusted for age (20–39, 40–59, and ≥ 60 years); sex; insurance status (insured and dependent); CCI score (0, 1, and ≥ 2); body mass index (underweight [$< 18.5 \text{ kg/m}^2$], normal [$18.5\text{--}22.9 \text{ kg/m}^2$], overweight [$23.0\text{--}25.0 \text{ kg/m}^2$], obese [$\geq 25.0 \text{ kg/m}^2$], and unknown); blood pressure (systolic blood pressure $< 140 \text{ mmHg}$ and diastolic blood pressure $< 90 \text{ mmHg}$, systolic blood pressure $\geq 140 \text{ mmHg}$ or diastolic blood pressure $\geq 90 \text{ mmHg}$, and unknown); fasting blood glucose (< 100 , $\geq 100 \text{ mg/dL}$, and unknown); serum total cholesterol (< 200 ,

200–239, ≥ 240 mg/dL, and unknown); glomerular filtration rate (<60 , 60–89, ≥ 90 mL/min/1.73 m², and unknown); smoking status (non- and current smoker, and unknown); alcoholic drinks (rarely, sometimes, everyday, and unknown); aerobic physical activity (sufficient, insufficient, and unknown); previous history of cardiovascular disease, and chronic kidney disease; history of medication use for diabetes mellitus, dyslipidemia, and hypertension; and strain of SARS-CoV-2 (original and delta).

Comment 6.

Table 4 attempts to evaluate the risk of acute and long-term sequelae by comparing vaccination status and COVID severity. The reference group seems to be the non-infected control. Similar to previous comments, it is unclear how the non-infected control could have acute and long-term sequelae respiratory complications and at what time point the authors are referring to. Additionally, vaccination is initially protective against COVID-19 infection, and those who had COVID-19 post-vaccination represent breakthrough infections. The authors seem to conflate multiple groups (i.e., vaccine status among people without COVID-19). To properly assess the protective effect of vaccination against long COVID, the authors might consider focusing on people with breakthrough infections, comparing their vaccination status and acute/long-term COVID outcomes.

Response:

We sincerely appreciate your invaluable comments and suggestions. Our initial analysis was designed to investigate the increased risk of post-acute respiratory sequelae following SARS-CoV-2 infection or influenza, compared to the general natural risk of respiratory disease within a non-infected control group during the follow-up period. Therefore, it is more accurate to describe the non-infected control group in terms of the risk of respiratory diseases rather than post-acute respiratory sequelae. Accordingly, we have clarified this distinction by adding statements to the table caption, indicating that "HR of the non-infected control represents the risk of respiratory diseases, and HRs of patients with COVID-19 indicate the risk of post-acute respiratory complications following SARS-CoV-2 infection."

We also acknowledge that our initial methods, particularly descriptions regarding the index date and follow-up period, were insufficient; thus, we have revised and elaborated the Methods. In addition, we fully agree that selecting the patient group with COVID-19 but without SARS-CoV-2 vaccination (equivalent to 'people with breakthrough infections' from your comments) as the reference group provides a clearer perspective on the association between SARS-CoV-2 infection and both long-term post-acute respiratory sequelae and acute respiratory complications, with a focus on breakthrough infections. Therefore, we have re-analyzed and revised Table 4, designating patients with COVID-19 without SARS-CoV-2 vaccination as the reference group. This would be a better method to examine the association of the number and types of SARS-CoV-2 vaccination for respiratory symptoms, especially in breakthrough infections. Thank you for your invaluable input.

Changes in text:

Methods/Statistical analysis

- To estimate the hazard ratios (HRs) with 95% confidence intervals (CIs), we applied Cox proportional hazards model. Therefore, we assigned the 'individual index date' by following the criterion. For the exposed group, it is the date of the first diagnosis of SARS-CoV-2. For the individuals in the non-exposed group, the index date was allocated to match the index date of the corresponding exposed case. This approach was implemented to mitigate immortal time bias, ensuring an equitable comparison between groups.

Results

- The risk of acute respiratory complication showed decreasing trends according to the number of SARS-CoV-2 vaccinations from individuals after once receiving vaccination (HR, 0.51 [95% CI, 0.38-0.68]) to those with two or more vaccinations (HR, 0.24 [95% CI, 0.19-0.30]). Interestingly, mixed types of vaccination showed the lowest risk of developing long-term respiratory sequelae of all SARS-CoV-2 vaccination methods (HR, 0.18 [95% CI, 0.08-0.38]).

Discussion/Findings of this study

- Third, people who were vaccinated, especially multiple vaccinations and mixed vaccination, had a lower risk of developing long-term post-acute respiratory sequelae than infected patients of SARS-CoV-2 without vaccination.

Figure S2. Study flow of cohorts

Table 4. Subgroup analysis (COVID-19 vs. general population) of HR (95% CI) of the **long-term post-acute respiratory sequelae** or **short-term acute respiratory complication** after SARS-CoV-2 infection stratified by vaccination, COVID-19 severity, and SARS-CoV-2 strain in the cohort of South Korea (main)

Variable	Events/total, n/N (%)	HR (95% CI)	
		Model 1*	Model 2†
Long-term post-acute respiratory sequelae			
Number of SARS-CoV-2 vaccinations			
Non-infected control	16,122/1,918,150 (0.84)	0.62 (0.60-0.65)	0.60 (0.58-0.62)
COVID-19 without SARS-CoV-2 vaccination	4331/200,539 (2.16)	1.0 (reference)	1.0 (reference)
COVID-19 after SARS-CoV-2 vaccination received once	493/38,852 (1.27)	0.90 (0.82-0.99)	0.85 (0.77-0.94)
COVID-19 after SARS-CoV-2 vaccination received twice or more	468/155,207 (0.30)	0.69 (0.62-0.76)	0.64 (0.57-0.71)
Type of SARS-CoV-2 vaccinations			
Non-infected control	16,122/1,918,150 (0.84)	0.62 (0.60-0.65)	0.60 (0.58-0.62)
COVID-19 without SARS-CoV-2 vaccination	4331/200,539 (2.16)	1.0 (reference)	1.0 (reference)
COVID-19 with viral vector SARS-CoV-2 vaccination	465/109,066 (0.43)	1.12 (1.01-1.23)	0.99 (0.90-1.10)
COVID-19 with mRNA SARS-CoV-2 vaccination	477/66,891 (0.71)	1.20 (1.09-1.33)	1.11 (0.99-1.22)
COVID-19 with both types of SARS-CoV-2 vaccination	19/18,102 (0.10)	0.74 (0.47-1.16)	0.66 (0.42-1.03)
COVID-19 severity			
Non-infected control	16,122/1,918,150 (0.84)	1.0 (reference)	1.0 (reference)
Mild COVID-19	3492/340,813 (1.02)	1.28 (1.24-1.33)	1.37 (1.32-1.43)
Moderate to severe COVID-19	1800/53785 (3.35)	3.60 (3.43-3.78)	3.20 (3.03-3.38)
Original strain of SARS-CoV-2 (overall population)			
Non-infected control before the delta-dominant phase [§]	11,667/594,134 (1.96)	1.0 (reference)	1.0 (reference)

Infection with original strain	3649/121,521 (3.00)	1.58 (1.52-1.64)	1.59 (1.52-1.66)
Delta variant of SARS-CoV-2 (overall population)			
Non-infected control during the delta-dominant phase [§]	4455/1,324,016 (0.34)	1.0 (reference)	1.0 (reference)
Infection with Delta variant	1643/273,077 (0.60)	1.81 (1.71-1.92)	1.94 (1.81-2.08)
Acute respiratory complication			
Number of SARS-CoV-2 vaccinations			
Non-infected control	311/1,918,150 (0.016)	0.07 (0.06-0.08)	0.08 (0.06-0.09)
COVID-19 without SARS-CoV-2 vaccination	415/200,539 (0.21)	1.0 (reference)	1.0 (reference)
COVID-19 after SARS-CoV-2 vaccination received once	56/38,852 (0.14)	0.62 (0.47-0.83)	0.51 (0.38-0.68)
COVID-19 after SARS-CoV-2 vaccination received twice or more	147/155,207 (0.09)	0.32 (0.26-0.39)	0.24 (0.19-0.30)
Type of SARS-CoV-2 vaccinations			
Non-infected control	311/1,918,150 (0.016)	0.07 (0.06-0.08)	0.08 (0.06-0.09)
COVID-19 without SARS-CoV-2 vaccination	415/200,539 (0.21)	1.0 (reference)	1.0 (reference)
COVID-19 with viral vector SARS-CoV-2 vaccination	79/109,066 (0.072)	0.44 (0.34-0.56)	0.36 (0.28-0.47)
COVID-19 with mRNA SARS-CoV-2 vaccination	117/66,891 (0.17)	0.40 (0.32-0.50)	0.42 (0.33-0.53)
COVID-19 with both types of SARS-CoV-2 vaccination	7/18,102 (0.039)	0.19 (0.09-0.41)	0.18 (0.08-0.38)
COVID-19 severity			
Non-infected control	311/1,918,150 (0.016)	1.0 (reference)	1.0 (reference)
Mild COVID-19	50/340,813 (0.015)	0.95 (0.71-1.28)	0.99 (0.73-1.34)
Moderate to severe COVID-19	568/53,783 (1.06)	51.18 (44.38-59.02)	39.54 (33.54-46.62)
Original strain of SARS-CoV-2 (overall population)			
Non-infected control before the delta-dominant phase [§]	110/594,134 (0.019)	1.0 (reference)	1.0 (reference)
Infection with original strain	230/121,521 (0.19)	10.27 (8.18-12.88)	9.21 (7.19-11.80)

Delta variant of SARS-CoV-2 (overall population)

Non-infected control during the delta-dominant phase §	201/1,324,016 (0.015)	1.0 (reference)	1.0 (reference)
Infection with Delta variant	388/273,077 (0.14)	9.39 (7.92-11.14)	7.44 (6.13-9.03)

CCI, Charlson comorbidity index; CI, confidence interval; HR, hazard ratio; SARS-CoV-2, severe acute respiratory syndrome coronavirus 2.

The data in bold indicate significant differences ($P < 0.05$).

|| HR of the non-infected control represents the risk of respiratory diseases, and HRs of patients with COVID-19 indicate the risk of post-acute respiratory complications following SARS-CoV-2 infection.

§ Only 1:5-matched comparators in each patient group at the same index date were included to reduce immortal time bias.

***Model 1:** Adjusted for age (20–39, 40–59, and ≥ 60 years) and sex.

†**Model 2:** Adjusted for age (20–39, 40–59, and ≥ 60 years); sex, household income (low income, middle income, and high income); region of residence (urban and rural); CCI score (0, 1, and ≥ 2); obesity (underweight [$< 18.5 \text{ kg/m}^2$], normal [$18.5\text{--}22.9 \text{ kg/m}^2$], overweight [$23.0\text{--}24.9 \text{ kg/m}^2$], obese [$\geq 25.0 \text{ kg/m}^2$], and unknown); blood pressure (systolic blood pressure $< 140 \text{ mmHg}$ and diastolic blood pressure $< 90 \text{ mmHg}$, systolic blood pressure $\geq 140 \text{ mmHg}$ or diastolic blood pressure $\geq 90 \text{ mmHg}$, and unknown); fasting blood glucose (< 100 , $\geq 100 \text{ mg/dL}$, and unknown); serum total cholesterol (< 200 , $200\text{--}239$, $\geq 240 \text{ mg/dL}$, and unknown); glomerular filtration rate (< 60 , $60\text{--}89$, $\geq 90 \text{ mL/min/1.73 m}^2$, and unknown); smoking status (never, former, current smoker, and unknown); alcoholic drinks (< 1 , $1\text{--}2$, $3\text{--}4$, ≥ 5 days per week, and unknown); aerobic physical activity (sufficient, insufficient, and unknown); previous history of cardiovascular disease, and chronic kidney disease; history of medication use for diabetes mellitus, dyslipidemia, and hypertension; and strain of SARS-CoV-2 (original and delta).

Table S15. Propensity-score-matched subgroup analysis (COVID-19 vs. general population) of HR (95% CI) of **long-term post-acute respiratory sequelae** or **short-term acute respiratory complication** following COVID-19 diagnosis stratified by COVID-19 vaccination in South Korea

Variable	Events/total, n/N (%)	HR (95% CI)	
		Model 1*	Model 2†
Long-term post-acute respiratory sequelae			
Number of SARS-CoV-2 vaccinations			
Non-infected control	16,122/1,918,150 (0.84)	1.0 (reference)	1.0 (reference)
COVID-19 without SARS-CoV-2 vaccination	4331/200,539 (2.16)	1.60 (1.55-1.66)	1.67 (1.61-1.74)
COVID-19 after SARS-CoV-2 vaccination received once	493/38,852 (1.27)	1.86 (1.70-2.04)	1.78 (1.62-1.96)
COVID-19 after SARS-CoV-2 vaccination received twice or more	468/155,207 (0.30)	1.81 (1.65-1.20)	1.68 (1.52-1.85)
Type of SARS-CoV-2 vaccinations			
Non-infected control	16,122/1,918,150 (0.84)	1.0 (reference)	1.0 (reference)
COVID-19 without SARS-CoV-2 vaccination	4331/200,539 (2.16)	1.60 (1.55-1.66)	1.67 (1.61-1.74)
COVID-19 with viral vector SARS-CoV-2 vaccination	465/109,066 (0.43)	1.79 (1.63-1.97)	1.66 (1.51-1.83)
COVID-19 with mRNA SARS-CoV-2 vaccination	477/66,891 (0.71)	1.93 (1.76-2.12)	1.85 (1.68-2.03)
COVID-19 with both types of SARS-CoV-2 vaccination	19/18,102 (0.10)	1.19 (0.76-1.87)	1.10 (0.70-1.73)
Acute respiratory complication			
Number of SARS-CoV-2 vaccinations			
Non-infected control	311/1,918,150 (0.016)	1.0 (reference)	1.0 (reference)
COVID-19 without SARS-CoV-2 vaccination	415/200,539 (0.21)	14.51 (12.52-16.83)	13.19 (11.07-15.71)
COVID-19 after SARS-CoV-2 vaccination received once	56/38,852 (0.14)	9.17 (6.90-12.19)	7.97 (5.96-10.67)
COVID-19 after SARS-CoV-2 vaccination received twice or more	147/155,207 (0.09)	5.06 (4.16-6.16)	4.23 (3.44-5.21)

Type of SARS-CoV-2 vaccinations

Non-infected control	311/1,918,150 (0.016)	1.0 (reference)	1.0 (reference)
COVID-19 without SARS-CoV-2 vaccination	415/200,539 (0.21)	14.51 (12.52-16.83)	13.19 (11.07-15.71)
COVID-19 with viral vector SARS-CoV-2 vaccination	79/109,066 (0.072)	6.35 (4.95-8.14)	4.69 (3.63-6.07)
COVID-19 with mRNA SARS-CoV-2 vaccination	117/66,891 (0.17)	5.80 (4.67-7.20)	5.42 (4.33-6.78)
COVID-19 with both types of SARS-CoV-2 vaccination	7/18,102 (0.039)	2.81 (1.33-5.94)	2.33 (1.10-4.94)

CCI, Charlson comorbidity index; CI, confidence interval; HR, hazard ratio; SARS-CoV-2, severe acute respiratory syndrome coronavirus 2.

The data in bold indicate significant differences ($P < 0.05$).

§ Only 1:5-matched comparators in each patient group at the same index date were included to reduce immortal time bias.

***Model 1:** Adjusted for age (20–39, 40–59, and ≥ 60 years) and sex.

†**Model 2:** Adjusted for age (20–39, 40–59, and ≥ 60 years); sex, household income (low income, middle income, and high income); region of residence (urban and rural); CCI score (0, 1, and ≥ 2); obesity (underweight [$< 18.5 \text{ kg/m}^2$], normal [$18.5\text{--}22.9 \text{ kg/m}^2$], overweight [$23.0\text{--}24.9 \text{ kg/m}^2$], obese [$\geq 25.0 \text{ kg/m}^2$], and unknown); blood pressure (systolic blood pressure $< 140 \text{ mmHg}$ and diastolic blood pressure $< 90 \text{ mmHg}$, systolic blood pressure $\geq 140 \text{ mmHg}$ or diastolic blood pressure $\geq 90 \text{ mmHg}$, and unknown); fasting blood glucose (< 100 , $\geq 100 \text{ mg/dL}$, and unknown); serum total cholesterol (< 200 , $200\text{--}239$, $\geq 240 \text{ mg/dL}$, and unknown); glomerular filtration rate (< 60 , $60\text{--}89$, $\geq 90 \text{ mL/min/1.73 m}^2$, and unknown); smoking status (never, former, current smoker, and unknown); alcoholic drinks (< 1 , $1\text{--}2$, $3\text{--}4$, ≥ 5 days per week, and unknown); aerobic physical activity (sufficient, insufficient, and unknown); previous history of cardiovascular disease, and chronic kidney disease; history of medication use for diabetes mellitus, dyslipidemia, and hypertension; and strain of SARS-CoV-2 (original and delta).

Comment 7.

The authors report a large number of non-COVID or non-flu comparison groups in their analyses. Given that this is claims data, can the author address how they dealt with the issue of underdiagnosis of COVID-19 and flu in their control (non-infected) group? Misclassification of COVID/flu cases to non-infected cases is highly likely given the nature of claims data.

Response:

Thank you for your invaluable input. We deeply understand your concerns regarding the potential underdiagnosis issues of COVID-19 and influenza. First and foremost, we want to explain the status of the response to the outbreak in South Korea and Japan. In South Korea, recognized for its effective control over the COVID-19 pandemic, the government rapidly implemented social distancing measures and adopted an aggressive "test, trace, isolate" strategy (Dighe et al., 2020, BMC Medicine). This comprehensive tracing approach included the use of credit card records, handwritten visitor logs, KI-pass, and the safe call system (Gong et al., 2022, Epidemiol Health). Once there was a patient tested positive for SARS-CoV-2, contacts in high-risk groups, such as household contacts of the patients and the healthcare personnel, were routinely tested (Park et al., 2020, Emerg Infect Dis). By April 2020, these efforts led to a significant reduction in case numbers to single digits, demonstrating the effectiveness of these measures in disease control and minimizing underdiagnoses by testing a large portion of the population at risk of SARS-CoV-2 infection. Similarly, the Japanese government employed similar strategies to ensure a high diagnostic rate during the pandemic. The postponement of the Tokyo Summer Olympics from 2020 to 2021, coupled with implementing over one million tests from June 29 to September 8, 2021, and introducing multiple restrictions to prevent major outbreaks (Yashio et al., 2021, Travel Med Infect Dis). These measures ensured that most of those at risk of SARS-CoV-2 infection in Japan were diagnosed with COVID-19. Both South Korea and Japan's strategies to maximize diagnostic rates during the pandemic resulted in more reliable diagnosis patterns for SARS-CoV-2 and influenza infections.

Despite the initiative of RT-PCR assay or antigen tests in both Korea and Japan contributing to a lower possibility of underdiagnoses, we have also taken your comments and conducted a sensitivity analysis. This analysis was restricted to non-infected individuals who tested negative in the RT-PCR assay, enabling a more detailed examination by excluding participants potentially having SARS-CoV-2 or influenza infections but not seeking

appropriate medical testing. The results from the sensitivity analysis are presented in Tables S4 and S25. We have also added a description of this sensitivity analysis in the Methods section, 'Sensitivity Analysis.' In addition, we acknowledge the limitations related to our dataset and address these issues in the Discussion section, 'Limitations and Strengths.' Overall, thank you for providing such an important comment, which has significantly contributed to improving our manuscript.

<References>

1. Dighe A, Cattarino L, Cuomo-Dannenburg G, et al. Response to COVID-19 in South Korea and implications for lifting stringent interventions. *BMC Med* 2020; 18(1): 321.
2. Gong S, Moon JY, Jung J. Perceived usefulness of COVID-19 tools for contact tracing among contact tracers in Korea. *Epidemiol Health* 2022; 44: e2022106.
3. Park YJ, Choe YJ, Park O, et al. Contact Tracing during Coronavirus Disease Outbreak, South Korea, 2020. *Emerg Infect Dis* 2020; 26(10): 2465-8.
4. Yashio T, Murayama A, Kami M, Ozaki A, Tanimoto T, Rodriguez-Morales AJ. COVID-19 infection during the Olympic and Paralympic Games Tokyo 2020. *Travel Med Infect Dis* 2021; 44: 102205.

Changes in text:

Methods/*Sensitivity analysis*

- Second, we conducted an exposure-driven propensity score matching analysis based on the claim record of individuals who tested positive for SARS-CoV-2 within two weeks following an RT-PCR test and those who did not, for a more sophisticated analysis of the association between SARS-CoV-2 infection and respiratory symptoms (**Table S4**).

Discussion/*Limitations and strengths*

- Fourth, the dataset we utilized has a risk of underdiagnosis of SARS-CoV-2 and influenza infection. There is a possibility of overlooking patients who were infected with SARS-CoV-2 or influenza but did not take the PCR test or visit a hospital to receive treatment. However, to assess the potential underdiagnosis, we analyzed the HR of long-term post-acute respiratory sequelae and short-term acute respiratory complications with participants after PCR tests.

Table S4. Baseline characteristics for 1:5 propensity score–matched cohort (positive for COVID-19 vs. negative for COVID-19 after the PCR test) in South Korea

Characteristic	COVID-19 vs. general population (n=492,774)		SMD*
	COVID-19 (n=82,938)	General population (n=409,836)	
Mean age (SD), y	50.3 (18.0)	48.8 (15.1)	0.094
Age, n (%)			<0.001
20–39 y	27150 (32.7)	135507 (33.1)	
40–59 y	28084 (33.9)	139600 (34.1)	
≥60 y	27704 (33.4)	134729 (32.9)	
Sex, n (%)			0.007
Male	41715 (50.3)	205410 (50.1)	
Female	41223 (49.7)	204426 (49.9)	
Region of residence, n (%)			<0.001
Urban	42950 (51.8)	212152 (51.8)	
Rural	39988 (48.2)	197684 (48.2)	
Medical history, n (%)			
Cardiovascular disease	6367 (7.7)	30945 (7.6)	0.005
Chronic kidney disease	3127 (3.8)	14807 (3.6)	0.008
Medication use for diabetes	19346 (23.3)	94165 (23.0)	0.008
Medication use for hyperlipidemia	15948 (19.2)	78045 (19.0)	0.005
Medication use for hypertension	9556 (11.5)	45993 (11.2)	0.009
Unmatching covariates, n (%)[†]			
Charlson Comorbidity Index score			0.172
0	65267 (78.7)	337680 (82.4)	

1	10720 (12.9)	33672 (8.2)	
≥2	6951 (8.4)	38484 (9.4)	
Household income			<0.001
Low (0th–39th percentile)	36117 (43.6)	178323 (43.5)	
Middle (40th –79th percentile)	30486 (36.8)	150810 (36.8)	
High (80th–100th percentile)	16335 (19.7)	80703 (19.7)	
Body mass index			0.965
Underweight (<18.5 kg/m ²)	1478 (1.8)	13305 (3.3)	
Normal (18.5-22.9 kg/m ²)	16334 (19.7)	137555 (33.6)	
Overweight (23.0-24.9 kg/m ²)	11688 (14.1)	90952 (22.2)	
Obese (≥25.0 kg/m ²)	21462 (25.9)	157427 (38.4)	
Unknown	31976 (38.6)	10597 (2.6)	
Blood pressure			0.986
SBP <140 mmHg and DBP <90 mmHg	42980 (51.8)	345942 (84.4)	
SBP ≥140mmHg or DBP ≥90 mmHg	7576 (9.1)	51749 (12.6)	
Unknown	32382 (39.0)	12145 (3.0)	
Fasting blood glucose			0.986
<100 mg/dL	29647 (35.8)	240935 (58.8)	
≥100 mg/dL	20904 (25.2)	156723 (38.2)	
Unknown	32387 (39.1)	12178 (3.0)	
Serum total cholesterol			0.371
<200 mg/dL	15373 (18.5)	117059 (28.6)	
200 to 239 mg/dL	8155 (9.8)	64787 (15.8)	
≥240 mg/dL	3458 (4.2)	26592 (6.5)	
Unknown	55952 (67.5)	201398 (49.1)	
Glomerular filtration rate			0.989
<60 mL/min/1.73 m ²	2538 (3.1)	15911 (3.9)	

60 to 89 mL/min/1.73 m ²	22089 (26.6)	165622 (40.4)	
≥90 mL/min/1.73 m ²	25887 (31.2)	215705 (52.6)	
Unknown	32424 (39.1)	12598 (3.1)	
Smoking status			1.040
Never	34178 (41.2)	256807 (62.7)	
Former	9441 (11.4)	64616 (15.8)	
Current	7351 (8.9)	77779 (19.0)	
Unknown	31968 (38.5)	10634 (2.6)	
Alcohol consumption			0.964
<1 day/week	31148 (37.6)	247938 (60.5)	
1 to 2 days/week	13581 (16.4)	106578 (26)	
3 to 4 days/week	4614 (5.6)	33186 (8.1)	
≥5 days/week	1624 (2.0)	11508 (2.8)	
Unknown	31971 (38.6)	10626 (2.6)	
Aerobic physical activity			0.962
Insufficient	26430 (31.9)	202726 (49.5)	
Sufficient	24521 (29.6)	196414 (47.9)	
Unknown	31987 (38.6)	10696 (2.6)	
Strain of SARS-CoV-2			0.004
Original	32636 (39.4)	162008 (39.5)	
Delta	50302 (60.7)	247828 (60.5)	

DBP, diastolic blood pressure; PCR, polymerase chain reaction; SARS-CoV-2, severe acute respiratory syndrome coronavirus 2. SBP, systolic blood pressure; SD, standard deviation; SMD, standardized mean difference.

* An SMD <0.1 indicates no significant imbalance. All SMDs were <0.100 in the propensity score-matched cohorts.

† Unmatched covariates were included as adjustment factors in statistical analyses.

Table S25. HR (95% CI) for the long-term post-acute respiratory sequelae or short-term acute respiratory complication subtypes after the PCR test in the propensity score-matched cohorts (COVID-19 vs. general population) in South Korea

	COVID-19 vs. general population (n=492,774)		
	Events, n (%)	HR (95% CI)	
		Model 1*	Model 2†
Long-term post-acute respiratory sequelae			
Comparators	1745 (0.43)	1.0 (reference)	1.0 (reference)
Patients with COVID-19	1675 (2.02)	7.95 (7.37-8.58)	8.10 (7.45-8.80)
Chronic respiratory failure			
Comparators	9 (0.0022)	1.0 (reference)	1.0 (reference)
Patients with COVID-19	33 (0.039)	21.16 (9.77-45.81)	19.64 (8.66-44.54)
Pulmonary hypertension			
Comparators	5 (0.0012)	1.0 (reference)	1.0 (reference)
Patients with COVID-19	1 (0.0012)	1.68 (0.17-16.10)	0.33 (0.02-4.72)
Sleep apnea			
Comparators	118 (0.029)	1.0 (reference)	1.0 (reference)
Patients with COVID-19	78 (0.094)	5.65 (4.09-7.80)	6.47 (4.58-9.14)
COPD			
Comparators	1154 (0.28)	1.0 (reference)	1.0 (reference)
Patients with COVID-19	1047 (1.26)	7.32 (6.66-8.04)	7.37 (6.65-8.17)
Emphysema			
Comparators	70 (0.0017)	1.0 (reference)	1.0 (reference)
Patients with COVID-19	56 (0.068)	7.18 (4.79-10.77)	7.42 (4.81-11.45)
Asthma			
Comparators	435 (0.11)	1.0 (reference)	1.0 (reference)
Patients with COVID-19	443 (0.53)	8.28 (7.12-9.63)	8.46 (7.19-9.96)
Pulmonary sarcoidosis			

Comparators	8 (0.0020)	1.0 (reference)	1.0 (reference)
Patients with COVID-19	2 (0.0024)	2.52 (0.46-13.74)	4.20 (0.76-23.20)
Interstitial lung disease			
Comparators	96 (0.023)	1.0 (reference)	1.0 (reference)
Patients with COVID-19	177 (0.21)	19.32 (14.01-26.65)	21.12 (15.15-29.45)
Acute respiratory complication			
Comparators	135 (0.033)	1.0 (reference)	1.0 (reference)
Patients with COVID-19	174 (0.21)	6.51 (5.20-8.16)	5.26 (4.07-6.80)
Pneumocystis pneumonia			
Comparators	0 (0.00)	1.0 (reference)	1.0 (reference)
Patients with COVID-19	0 (0.00)	NA	NA
Aspergillosis pneumonia			
Comparators	9 (0.0022)	1.0 (reference)	1.0 (reference)
Patients with COVID-19	10 (0.012)	5.68 (2.31-13.97)	5.54 (2.08-14.74)
Pleural empyema			
Comparators	6 (0.0015)	1.0 (reference)	1.0 (reference)
Patients with COVID-19	3 (0.0036)	2.53 (0.63-10.10)	0.71 (0.13-4.00)
Lung abscess			
Comparators	1 (0.00024)	1.0 (reference)	1.0 (reference)
Patients with COVID-19	1 (0.0012)	4.92 (0.31-78.60)	15.36 (0.74-319.84)
Pneumothorax			
Comparators	16 (0.0039)	1.0 (reference)	1.0 (reference)
Patients with COVID-19	13 (0.016)	4.09 (1.97-8.50)	3.80 (1.65-8.73)
Acute respiratory failure			
Comparators	48 (0.012)	1.0 (reference)	1.0 (reference)
Patients with COVID-19	102 (0.12)	10.70 (7.59-15.08)	7.78 (5.29-11.46)
Pulmonary embolism			

Comparators	56 (0.014)	1.0 (reference)	1.0 (reference)
Patients with COVID-19	45 (0.054)	4.07 (2.75-6.02)	3.72 (2.38-5.82)

CCI, Charlson comorbidity index; CI, confidence interval; COPD, chronic obstructive pulmonary disease; HR, hazard ratio; NA, not available; PCR, polymerase chain reaction; SARS-CoV-2, severe acute respiratory syndrome coronavirus 2.

The data in bold indicate significant differences ($P < 0.05$).

***Model 1:** Adjusted for age (20–39, 40–59, and ≥ 60 years) and sex.

†**Model 2:** Adjusted for age (20–39, 40–59, and ≥ 60 years); sex, household income (low income, middle income, and high income); region of residence (urban and rural); CCI score (0, 1, and ≥ 2); obesity (underweight [$< 18.5 \text{ kg/m}^2$], normal [$18.5\text{--}22.9 \text{ kg/m}^2$], overweight [$23.0\text{--}24.9 \text{ kg/m}^2$], obese [$\geq 25.0 \text{ kg/m}^2$], and unknown); blood pressure (systolic blood pressure $< 140 \text{ mmHg}$ and diastolic blood pressure $< 90 \text{ mmHg}$, systolic blood pressure $\geq 140 \text{ mmHg}$ or diastolic blood pressure $\geq 90 \text{ mmHg}$, and unknown); fasting blood glucose (< 100 , $\geq 100 \text{ mg/dL}$, and unknown); serum total cholesterol (< 200 , $200\text{--}239$, $\geq 240 \text{ mg/dL}$, and unknown); glomerular filtration rate (< 60 , $60\text{--}89$, $\geq 90 \text{ mL/min/1.73 m}^2$, and unknown); smoking status (never, former, current smoker, and unknown); alcoholic drinks (< 1 , $1\text{--}2$, $3\text{--}4$, ≥ 5 days per week, and unknown); aerobic physical activity (sufficient, insufficient, and unknown); previous history of cardiovascular disease, and chronic kidney disease; history of medication use for diabetes mellitus, dyslipidemia, and hypertension; and strain of SARS-CoV-2 (original and delta).

Comment 8.

Another limitation is the timing of the data. The study period appears to be pre-Omicron, which is less relevant to current times. Updating the dataset to a more recent period would substantially strengthen the paper.

Response:

Thank you for your constructive comment. If we had access to the database of a more recent period that includes the Omicron period, we would have utilized such data in our new analysis. However, it would take about 6 months to a year for the database to be upgraded, and the database that includes the Omicron period is not available at the moment. We hope the reviewer understands our current situation. We would like to submit a follow-up research article next year in a similar context but include the database of recent periods if possible.

Comment 9.

There are minor editorial errors throughout the paper and in the tables. For example, Table 2 incorrectly reports “Ratio of HR,” which is redundant as hazard ratio already includes the term ratio. Thorough editing and proofreading of the manuscript may enhance its quality.

Response:

We apologize for any misunderstanding our terms may have caused. In Table 2, we first analyzed the Hazard Ratio (HR) to show the increased risk of respiratory symptoms associated with COVID-19 or influenza infection. ‘Ratio of HR’ is the ratio of HR of the patients with COVID-19 compared to the HR of the patients with influenza. Therefore, the ratio of HR shows the relative association between COVID-19 infection and respiratory symptoms compared to influenza infection. We understand that the term ‘Ratio of HR’ might have caused a misunderstanding. However, throughout incorporating your previous comment, we acknowledged that an indirect comparison of COVID-19 and influenza may not be the best way to compare COVID-19 to another viral respiratory disease. Therefore, we re-analyzed using a direct 1:1 propensity score matching between the group infected with SARS-CoV-2 and the group with influenza. The baseline characteristics for the direct matching are shown in Tables S11 and S12. The results of the analysis are also shown in Tables S13, S14, and S16. After this revision and re-analyses, the ‘Ratio of HR’ concept was no longer needed since the comparison of COVID-19 and influenza was made in a direct manner. The changes in context and the Tables are shown below. Thank you for giving us advice that greatly enhanced the quality of our manuscript.

Changes in text:**Methods/Exposure**

- To examine the relative severity of COVID-19 in comparison with another contagious viral respiratory disease, additional exposure to influenza infection was defined. It refers to cases diagnosed through an RT-PCR assay or antigen test on nasal and pharyngeal swabs during the observation period. For individuals infected with both SARS-CoV-2 and influenza, it includes instances of influenza infection developing after the SARS-CoV-2 infection.

Methods/Propensity score matching

- In addition, to investigate the relative severity of COVID-19 compared to other infectious viral respiratory diseases, an influenza group within the general population was utilized as another control group, directly matching SARS-CoV-2 infections at a 1:1 ratio.

Results

- When directly comparing the risk for acute respiratory complication between SARS-CoV-2 and influenza infections, SARS-CoV-2 infection was significantly associated with an increased risk (main: HR, 4.32 [95% CI, 2.73-6.83]; replication: HR, 6.51 [95% CI, 5.38-7.87]) in **Tables S11-S13**.

- This tendency of increased risk for several subtypes of respiratory diseases was also shown when compared to patients with influenza infection (**Table S14**).

- HR of time attenuation effect after SARS-CoV-2 infection showed significance compared to influenza infection likewise (**Table S16**).

Table S11. Baseline characteristics for 1:1 propensity score-matched cohort (COVID-19 vs. influenza) in South Korea (main)

Characteristic	COVID-19 vs. influenza (n=223,000)		SMD*
	COVID-19 (n=111,500)	Influenza (n=111,500)	
Mean age (SD), y	45.3 (15.3)	45.0 (12.7)	0.017
Age, n (%)			0.060
20–39 y	42,625 (38.2)	40,131 (36.0)	
40–59 y	50,713 (45.5)	53,295 (47.8)	
≥60 y	18,162 (16.3)	18,074 (16.2)	
Sex, n (%)			0.046
Male	46,400 (41.6)	49,587 (44.5)	
Female	65,100 (58.4)	61,913 (55.5)	
Region of residence, n (%)			0.005
Urban	49,431 (44.3)	49,160 (44.1)	
Rural	62,069 (55.7)	62,340 (55.9)	
Medical history, n (%)			
Cardiovascular disease	4166 (3.7)	4258 (3.8)	0.004
Chronic kidney disease	2011 (1.8)	1991 (1.8)	0.001
Medication use for diabetes	17,218 (15.4)	16,665 (15.0)	0.014
Medication use for hyperlipidemia	17,656 (15.8)	17,167 (15.4)	0.012
Medication use for hypertension	7408 (6.6)	6640 (6.0)	0.028
Unmatching covariates, n (%)[†]			
Charlson Comorbidity Index score			0.103
0	99,673 (89.4)	103,133 (92.5)	
1	7867 (7.1)	5296 (4.8)	
≥2	3960 (3.6)	3071 (2.8)	

Household income			0.078
Low (0th–39th percentile)	48,878 (43.8)	46,012 (41.3)	
Middle (40th –79th percentile)	45,339 (40.7)	45,119 (40.5)	
High (80th–100th percentile)	17,283 (15.5)	20,369 (18.3)	
Body mass index			1.180
Underweight (<18.5 kg/m ²)	2120 (1.9)	3997 (3.6)	
Normal (18.5-22.9 kg/m ²)	22,894 (20.5)	39,561 (35.5)	
Overweight (23.0-24.9 kg/m ²)	14,256 (12.8)	24,459 (21.9)	
Obese (≥25.0 kg/m ²)	25,159 (22.6)	43,474 (39.0)	
Unknown	47,071 (42.2)	9 (0.0081)	
Blood pressure			1.207
SBP <140 mmHg and DBP <90 mmHg	55,801 (50.1)	99,099 (88.9)	
SBP ≥140mmHg or DBP ≥90 mmHg	8400 (7.5)	12,237 (11.0)	
Unknown	47299 (42.4)	164 (0.15)	
Fasting blood glucose			1.229
<100 mg/dL	40,399 (36.2)	72,206 (64.8)	
≥100 mg/dL	23,797 (21.3)	39,131 (35.1)	
Unknown	47,304 (42.4)	163 (0.15)	
Serum total cholesterol			0.462
<200 mg/dL	17,459 (15.7)	29,991 (26.9)	
200 to 239 mg/dL	10485 (9.4)	18,760 (16.8)	
≥240 mg/dL	4859 (4.4)	7923 (7.1)	
Unknown	78,697 (70.6)	54,826 (49.2)	
Glomerular filtration rate			1.208
<60 mL/min/1.73 m ²	1689 (1.5)	2223 (2.0)	
60 to 89 mL/min/1.73 m ²	26,592 (23.9)	45,211 (40.6)	
≥90 mL/min/1.73 m ²	35,867 (32.2)	63,785 (57.2)	

Unknown	47,352 (42.5)	281 (0.25)	
Smoking status			1.155
Never	45,487 (40.8)	75,040 (67.3)	
Former	9450 (8.5)	15,936 (14.3)	
Current	9506 (8.5)	20,488 (18.4)	
Unknown	47,057 (42.2)	36 (0.032)	
Alcohol consumption			1.208
<1 day/week	38,511 (34.5)	68078 (61.1)	
1 to 2 days/week	17,887 (16.0)	31025 (27.8)	
3 to 4 days/week	6125 (5.5)	9620 (8.6)	
≥5 days/week	1922 (1.7)	2741 (2.5)	
Unknown	47,055 (42.2)	36 (0.032)	
Aerobic physical activity			1.228
Insufficient	33,898 (30.4)	58,183 (52.2)	
Sufficient	30,539 (27.4)	53,276 (47.8)	
Unknown	47,063 (42.2)	41 (0.037)	
Strain of SARS-CoV-2			<0.001
Original	34685 (31.1)	34,718 (31.1)	
Delta	76,815 (68.9)	76,782 (68.9)	

DBP, diastolic blood pressure; SARS-CoV-2, severe acute respiratory syndrome coronavirus 2; SBP, systolic blood pressure; SD, standard deviation; SMD, standardized mean difference.

* An SMD <0.1 indicates no significant imbalance. All SMDs were <0.100 in the propensity score-matched cohorts.

† Unmatched covariates were included as adjustment factors in statistical analyses.

Table S12. Baseline characteristics for 1:1 propensity score-matched cohort (COVID-19 vs. influenza) in Japan (replication)

Characteristic	COVID-19 vs. influenza (n=178,648)		SMD*
	COVID-19 (n=89,324)	Influenza (n=89,324)	
Mean age (SD), y	44 (11.76)	44 (11.57)	0.018
Age, n (%)			<0.001
20–39 y	31,256 (34.99)	31,210 (34.94)	
40–59 y	48,959 (54.81)	49,241 (55.13)	
≥60 y	9109 (10.20)	8873 (9.93)	
Sex, n (%)			0.005
Male	55,617 (62.26)	55,396 (62.02)	
Female	33,707 (37.74)	33,928 (37.98)	
Insurance status, n (%)			<0.001
Insured	80,141 (89.72)	80,141 (89.72)	
Dependent	9183 (10.28)	9183 (10.28)	
Medical history, n (%)			
Cardiovascular disease	6235 (6.98)	6214 (6.96)	0.001
Chronic kidney disease	3031 (3.39)	3252 (3.64)	0.013
Medication use for diabetes	2422 (2.71)	2200 (2.46)	0.016
Medication use for hyperlipidemia	6159 (6.90)	5877 (6.58)	0.013
Medication use for hypertension	8016 (8.97)	7716 (8.64)	0.012
Unmatching covariates, n (%)[†]			
Charlson Comorbidity Index score			0.153
0	85,599 (95.83)	87,875 (98.38)	
1	1150 (1.29)	501 (0.56)	
≥2	2575 (2.88)	948 (1.06)	

Body mass index			0.007
Underweight (<18.5 kg/m ²)	47,532 (53.21)	47,767 (53.48)	
Normal (18.5-22.9 kg/m ²)	17,018 (19.05)	16,935 (18.96)	
Overweight (23.0-24.9 kg/m ²)	19,260 (21.56)	19,131 (21.42)	
Obese (≥25.0 kg/m ²)	5373 (6.02)	5366 (6.01)	
Unknown	125 (0.14)	141 (0.16)	
Blood pressure			<0.001
SBP <140 mmHg and DBP <90 mmHg	77,118 (86.34)	77,019 (86.22)	
SBP ≥140mmHg or DBP ≥90 mmHg	5456 (6.11)	5383 (6.03)	
Unknown	6750 (7.56)	6922 (7.75)	
Fasting blood glucose			0.028
<100 mg/dL	57,035 (63.85)	57,644 (64.53)	
≥100 mg/dL	18,237 (20.42)	18,284 (20.47)	
Unknown	14,052 (15.73)	13,396 (15.00)	
Serum total cholesterol			0.022
<200 mg/dL	41,317 (46.26)	39,896 (44.66)	
200 to 239 mg/dL	32,270 (36.13)	33,092 (37.05)	
≥240 mg/dL	14,211 (15.91)	14,672 (16.43)	
Unknown	1526 (1.71)	1664 (1.86)	
Glomerular filtration rate			0.073
<60 mL/min/1.73 m ²	501 (0.56)	296 (0.33)	
60 to 89 mL/min/1.73 m ²	8344 (9.34)	7987 (8.94)	
≥90 mL/min/1.73 m ²	48,818 (54.65)	46,549 (52.11)	
Unknown	31,661 (35.45)	34,492 (38.61)	
Smoking status			0.062
Non-smoker	20,354 (22.79)	21,458 (24.02)	

Smoker	65,734 (73.59)	64,706 (72.44)	
Unknown	3236 (3.62)	3160 (3.54)	
Alcohol consumption			0.049
Everyday	18,486 (20.70)	18,284 (20.47)	
Sometimes	29,295 (32.80)	28,687 (32.12)	
Rarely	35,096 (39.29)	35,716 (39.98)	
Unknown	6447 (7.22)	6637 (7.43)	
Aerobic physical activity			0.035
Insufficient	18,249 (20.43)	18,303 (20.49)	
Sufficient	62,979 (70.51)	62,858 (70.37)	
Unknown	8096 (9.06)	8163 (9.14)	
Strain of SARS-CoV-2			1.652
Original	36,886 (41.29)	88,835 (99.45)	
Delta	52,438 (58.71)	489 (0.55)	

DBP, diastolic blood pressure; SARS-CoV-2, severe acute respiratory syndrome coronavirus 2; SBP, systolic blood pressure; SD, standard deviation; SMD, standardized mean difference.

* An SMD <0.1 indicates no significant imbalance. All SMDs were <0.100 in the propensity score–matched cohorts.

† Unmatched covariates were included as adjustment factors in statistical analyses.

Table S13. Hazard ratio (95% CI) for the **long-term post-acute respiratory sequelae** or **short-term acute respiratory complication** after SARS-CoV-2 infection in the propensity score-matched cohorts (COVID-19 vs. influenza) of South Korea (main) and Japan (replication)

Cohort	South Korea			Japan		
	COVID-19 vs. influenza (n=223,000)			COVID-19 vs. influenza (n=169,924)		
	Events, n (%)	HR (95% CI)		Events, n (%)	HR (95% CI)	
Model 1*		Model 2†	Model 3*		Model 4‡	
Long-term post-acute respiratory sequelae						
Comparators (general population or patients with influenza)	1081 (0.97)	1.0 (reference)	1.0 (reference)	905 (1.07)	1.0 (reference)	1.0 (reference)
Patients with COVID-19	1500 (1.35)	1.55 (1.43-1.67)	1.66 (1.52-1.82)	4757 (5.60)	5.44 (5.07-5.84)	5.17 (4.82-5.55)
Acute respiratory complication						
Comparators	45 (0.040)	1.0 (reference)	1.0 (reference)	122 (0.14)	1.0 (reference)	1.0 (reference)
Patients with COVID-19	115 (0.10)	4.75 (3.08-7.33)	4.32 (2.73-6.83)	924 (1.09)	7.57 (6.27-9.14)	6.51 (5.38-7.87)

CCI, Charlson comorbidity index; CI, confidence interval; HR, hazard ratio.

The data in bold indicate significant differences ($P < 0.05$).

***Model 1 and 3:** Adjusted for age (20–39, 40–59, and ≥ 60 years) and sex.

†**Model 2:** Adjusted for age (20–39, 40–59, and ≥ 60 years); sex, household income (low income, middle income, and high income); region of residence (urban and rural); CCI score (0, 1, and ≥ 2); obesity (underweight [$< 18.5 \text{ kg/m}^2$], normal [$18.5\text{--}22.9 \text{ kg/m}^2$]; overweight [$23.0\text{--}24.9 \text{ kg/m}^2$], obese [$\geq 25.0 \text{ kg/m}^2$], and unknown); blood pressure (systolic blood pressure $< 140 \text{ mmHg}$ and diastolic blood pressure $< 90 \text{ mmHg}$, systolic blood pressure $\geq 140 \text{ mmHg}$ or diastolic blood pressure $\geq 90 \text{ mmHg}$, and unknown); fasting blood glucose (< 100 , $\geq 100 \text{ mg/dL}$, and unknown); serum total cholesterol (< 200 , $200\text{--}239$, $\geq 240 \text{ mg/dL}$, and unknown); glomerular filtration rate (< 60 , $60\text{--}89$, $\geq 90 \text{ mL/min/1.73 m}^2$, and unknown); smoking status (never, former, current smoker, and unknown); alcoholic drinks (< 1 , $1\text{--}2$, $3\text{--}4$, ≥ 5 days per week, and unknown); aerobic physical activity (sufficient, insufficient, and unknown); previous history of cardiovascular disease, and chronic kidney disease; history of medication use for diabetes mellitus, dyslipidemia, and hypertension; and strain of SARS-CoV-2 (original and delta).

‡**Model 4:** Adjusted for age (20–39, 40–59, and ≥ 60 years); sex; insurance status (insured and dependent); CCI score (0, 1, and ≥ 2); body mass index (underweight [$< 18.5 \text{ kg/m}^2$], normal [$18.5\text{--}22.9 \text{ kg/m}^2$], overweight [$23.0\text{--}25.0 \text{ kg/m}^2$], obese [$\geq 25.0 \text{ kg/m}^2$], and unknown); blood

pressure (systolic blood pressure <140 mmHg and diastolic blood pressure <90 mmHg, systolic blood pressure \geq 140 mmHg or diastolic blood pressure \geq 90 mmHg, and unknown); fasting blood glucose (<100, \geq 100 mg/dL, and unknown); serum total cholesterol (<200, 200–239, \geq 240 mg/dL, and unknown); glomerular filtration rate (<60, 60–89, \geq 90 mL/min/1.73 m², and unknown); smoking status (non- and current smoker, and unknown); alcoholic drinks (rarely, sometimes, everyday, and unknown); aerobic physical activity (sufficient, insufficient, and unknown); previous history of cardiovascular disease, and chronic kidney disease; history of medication use for diabetes mellitus, dyslipidemia, and hypertension; and strain of SARS-CoV-2 (original and delta).

Table S14. HR (95% CI) for the long-term post-acute respiratory sequelae or short-term acute respiratory complication subtypes after SARS-CoV-2 infection in the propensity score-matched cohorts (COVID-19 vs. influenza) in South Korea (main) and Japan (replication)

Cohort	South Korea			Japan		
	COVID-19 vs. influenza (n=223,000)			COVID-19 vs. influenza (n=169,924)		
	Events, n (%)	HR (95% CI)		Events, n (%)	HR (95% CI)	
Model 1*		Model 2†	Model 3*		Model 4	
Long-term post-acute respiratory sequelae						
Chronic respiratory failure						
Comparators (general population or patients with influenza)	3 (0.0027)	1.0 (reference)	1.0 (reference)	3 (0.0035)	1.0 (reference)	1.0 (reference)
Patients with COVID-19	12 (0.011)	4.19 (1.18-14.86)	4.82 (1.28-18.12)	77 (0.091)	25.55 (8.06-80.95)	16.15 (5.04-51.80)
Pulmonary hypertension						
Comparators	0 (0.00)	1.0 (reference)	1.0 (reference)	4 (0.0047)	1.0 (reference)	1.0 (reference)
Patients with COVID-19	0 (0.00)	NA	NA	28 (0.033)	6.97 (2.44-19.87)	5.59 (1.93-16.14)
Sleep apnea						
Comparators	23 (0.021)	1.0 (reference)	1.0 (reference)	88 (0.10)	1.0 (reference)	1.0 (reference)
Patients with COVID-19	49 (0.044)	4.26 (2.26-8.01)	4.12 (2.09-8.12)	549 (0.65)	6.26 (5.00-7.84)	5.95 (4.74-7.46)
COPD						
Comparators	734 (0.66)	1.0 (reference)	1.0 (reference)	271 (0.32)	1.0 (reference)	1.0 (reference)
Patients with COVID-19	994 (0.89)	1.48 (1.34-1.63)	1.60 (1.44-1.79)	1703 (2.00)	0.99 (0.91-1.09)	1.01 (0.91-1.12)
Emphysema						
Comparators	19 (0.017)	1.0 (reference)	1.0 (reference)	31 (0.036)	1.0 (reference)	1.0 (reference)
Patients with COVID-19	33 (0.030)	1.96 (1.10-3.47)	1.99 (1.06-3.74)	232 (0.27)	7.47 (5.14-10.87)	6.46 (4.43-9.43)
Asthma						
Comparators	337 (0.30)	1.0 (reference)	1.0 (reference)	477 (0.56)	1.0 (reference)	1.0 (reference)
Patients with COVID-19	421 (0.38)	1.37 (1.18-1.58)	1.45 (1.23-1.71)	2672 (3.14)	5.72 (5.19-6.30)	5.59 (5.07-6.16)
Pulmonary sarcoidosis						

Comparators	2 (0.0018)	1.0 (reference)	1.0 (reference)	25 (0.029)	1.0 (reference)	1.0 (reference)
Patients with COVID-19	2 (0.0018)	2.07 (0.19-22.84)	3.24 (0.29-36.47)	127 (0.15)	5.08 (3.31-7.81)	4.52 (2.93-6.96)
Interstitial lung disease						
Comparators	32 (0.029)	1.0 (reference)	1.0 (reference)	67 (0.079)	1.0 (reference)	1.0 (reference)
Patients with COVID-19	94 (0.084)	3.25 (2.15-4.90)	3.37 (2.19-5.19)	679 (0.80)	10.19 (7.93-13.1)	8.26 (6.42-10.64)
Acute respiratory complication						
Pneumocystis pneumonia						
Comparators	0 (0.00)	1.0 (reference)	1.0 (reference)	18 (0.021)	1.0 (reference)	1.0 (reference)
Patients with COVID-19	0 (0.00)	NA	NA	66 (0.078)	3.65 (2.17-6.14)	2.85 (1.68-4.85)
Aspergillosis pneumonia						
Comparators	0 (0.00)	1.0 (reference)	1.0 (reference)	4 (0.0047)	1.0 (reference)	1.0 (reference)
Patients with COVID-19	9 (0.0081)	NA	NA	17 (0.020)	4.21 (1.42-12.52)	3.49 (1.16-10.54)
Pleural empyema						
Comparators	2 (0.0018)	1.0 (reference)	1.0 (reference)	0 (0.00)	1.0 (reference)	1.0 (reference)
Patients with COVID-19	0 (0.00)	NA	NA	18 (0.021)	NA	NA
Lung abscess						
Comparators	6 (0.0054)	1.0 (reference)	1.0 (reference)	6 (0.0071)	1.0 (reference)	1.0 (reference)
Patients with COVID-19	0 (0.00)	NA	NA	18 (0.021)	2.98 (1.18-7.50)	2.25 (0.88-5.80)
Pneumothorax						
Comparators	18 (0.016)	1.0 (reference)	1.0 (reference)	22 (0.026)	1.0 (reference)	1.0 (reference)
Patients with COVID-19	10 (0.0090)	1.05 (0.44-2.53)	1.02 (0.37-2.81)	114 (0.13)	5.17 (3.28-8.16)	4.92 (3.11-7.77)
Acute respiratory failure						
Comparators	2 (0.0018)	1.0 (reference)	1.0 (reference)	58 (0.068)	1.0 (reference)	1.0 (reference)
Patients with COVID-19	67 (0.060)	NA	NA	583 (0.69)	10.04 (7.67-13.15)	8.44 (6.43-11.07)
Pulmonary embolism						
Comparators	18 (0.06)	1.0 (reference)	1.0 (reference)	18 (0.021)	1.0 (reference)	1.0 (reference)
Patients with COVID-19	29 (0.026)	2.94 (1.43-6.03)	3.27 (1.53-7.02)	175 (0.21)	9.68 (5.96-15.73)	7.96 (4.88-12.98)

CCI, Charlson comorbidity index; CI, confidence interval; COPD, chronic obstructive pulmonary disease; HR, hazard ratio; NA, not available.

The data in bold indicate significant differences ($P < 0.05$).

***Model 1 and 3:** Adjusted for age (20–39, 40–59, and ≥ 60 years) and sex.

†**Model 2:** Adjusted for age (20–39, 40–59, and ≥ 60 years); sex, household income (low income, middle income, and high income); region of residence (urban and rural); CCI score (0, 1, and ≥ 2); obesity (underweight [$< 18.5 \text{ kg/m}^2$], normal [$18.5\text{--}22.9 \text{ kg/m}^2$]; overweight [$23.0\text{--}24.9 \text{ kg/m}^2$], obese [$\geq 25.0 \text{ kg/m}^2$], and unknown); blood pressure (systolic blood pressure $< 140 \text{ mmHg}$ and diastolic blood pressure $< 90 \text{ mmHg}$, systolic blood pressure $\geq 140 \text{ mmHg}$ or diastolic blood pressure $\geq 90 \text{ mmHg}$, and unknown); fasting blood glucose (< 100 , $\geq 100 \text{ mg/dL}$, and unknown); serum total cholesterol (< 200 , $200\text{--}239$, $\geq 240 \text{ mg/dL}$, and unknown); glomerular filtration rate (< 60 , $60\text{--}89$, $\geq 90 \text{ mL/min/1.73 m}^2$, and unknown); smoking status (never, former, current smoker, and unknown); alcoholic drinks (< 1 , $1\text{--}2$, $3\text{--}4$, ≥ 5 days per week, and unknown); aerobic physical activity (sufficient, insufficient, and unknown); previous history of cardiovascular disease, and chronic kidney disease; history of medication use for diabetes mellitus, dyslipidemia, and hypertension; and strain of SARS-CoV-2 (original and delta).

‡ **Model 4:** Adjusted for age (20–39, 40–59, and ≥ 60 years); sex; insurance status (insured and dependent); CCI score (0, 1, and ≥ 2); body mass index (underweight [$< 18.5 \text{ kg/m}^2$], normal [$18.5\text{--}22.9 \text{ kg/m}^2$], overweight [$23.0\text{--}25.0 \text{ kg/m}^2$], obese [$\geq 25.0 \text{ kg/m}^2$], and unknown); blood pressure (systolic blood pressure $< 140 \text{ mmHg}$ and diastolic blood pressure $< 90 \text{ mmHg}$, systolic blood pressure $\geq 140 \text{ mmHg}$ or diastolic blood pressure $\geq 90 \text{ mmHg}$, and unknown); fasting blood glucose (< 100 , $\geq 100 \text{ mg/dL}$, and unknown); serum total cholesterol (< 200 , $200\text{--}239$, $\geq 240 \text{ mg/dL}$, and unknown); glomerular filtration rate (< 60 , $60\text{--}89$, $\geq 90 \text{ mL/min/1.73 m}^2$, and unknown); smoking status (non- and current smoker, and unknown); alcoholic drinks (rarely, sometimes, everyday, and unknown); aerobic physical activity (sufficient, insufficient, and unknown); previous history of cardiovascular disease, and chronic kidney disease; history of medication use for diabetes mellitus, dyslipidemia, and hypertension; and strain of SARS-CoV-2 (original and delta).

Table S16. Time attenuation effect analysis (COVID-19 vs. influenza) of HR (95% CI) for the risk of the **long-term post-acute respiratory sequelae** after SARS-CoV-2 infection in South Korea (main cohort) and Japan (replication cohort)

Time	COVID-19 vs. influenza	
	Main cohort [†]	Replication cohort
long-term post-acute respiratory sequelae		
<3 months	2.28 (2.01-2.59)	8.88 (7.76-10.16)
3–6 months	1.24 (1.04-1.47)	3.81 (3.27-4.44)
≥6 months	1.01 (0.83-1.23)	3.65 (3.30-4.05)

CCI, Charlson comorbidity index; CI, confidence interval; HR, hazard ratio; SARS-CoV-2, severe acute respiratory syndrome coronavirus 2.

The data in bold indicate significant differences ($P < 0.05$).

† Adjusted HR (main): Adjusted for age (20–39, 40–59, and ≥60 years); sex, household income (low income, middle income, and high income); region of residence (urban and rural); CCI score (0, 1, and ≥2); obesity (underweight [$<18.5 \text{ kg/m}^2$], normal [$18.5\text{--}22.9 \text{ kg/m}^2$]; overweight [$23.0\text{--}24.9 \text{ kg/m}^2$], obese [$\geq 25.0 \text{ kg/m}^2$], and unknown); blood pressure (systolic blood pressure $<140 \text{ mmHg}$ and diastolic blood pressure $<90 \text{ mmHg}$, systolic blood pressure $\geq 140 \text{ mmHg}$ or diastolic blood pressure $\geq 90 \text{ mmHg}$, and unknown); fasting blood glucose (<100 , $\geq 100 \text{ mg/dL}$, and unknown); serum total cholesterol (<200 , $200\text{--}239$, $\geq 240 \text{ mg/dL}$, and unknown); glomerular filtration rate (<60 , $60\text{--}89$, $\geq 90 \text{ mL/min/1.73 m}^2$, and unknown); smoking status (never, former, current smoker, and unknown); alcoholic drinks (<1 , $1\text{--}2$, $3\text{--}4$, ≥ 5 days per week, and unknown); aerobic physical activity (sufficient, insufficient, and unknown); previous history of cardiovascular disease, and chronic kidney disease; history of medication use for diabetes mellitus, dyslipidemia, and hypertension; and strain of SARS-CoV-2 (original and delta).

|| Adjusted HR (replication): Adjusted for age (20–39, 40–59, and ≥60 years); sex; insurance status (insured and dependent); CCI score (0, 1, and ≥ 2); body mass index (underweight [$<18.5 \text{ kg/m}^2$], normal [$18.5\text{--}22.9 \text{ kg/m}^2$], overweight [$23.0\text{--}25.0 \text{ kg/m}^2$], obese [$\geq 25.0 \text{ kg/m}^2$], and unknown); blood pressure (systolic blood pressure $<140 \text{ mmHg}$ and diastolic blood pressure $<90 \text{ mmHg}$, systolic blood pressure $\geq 140 \text{ mmHg}$ or diastolic blood pressure $\geq 90 \text{ mmHg}$, and unknown); fasting blood glucose (<100 , $\geq 100 \text{ mg/dL}$, and unknown); serum total cholesterol (<200 , $200\text{--}239$, $\geq 240 \text{ mg/dL}$, and unknown); glomerular filtration rate (<60 , $60\text{--}89$, $\geq 90 \text{ mL/min/1.73 m}^2$, and unknown); smoking status (non- and current smoker, and unknown); alcoholic drinks (rarely, sometimes, everyday, and unknown); aerobic physical activity (sufficient, insufficient, and unknown); previous history of cardiovascular disease, and chronic kidney disease; history of medication use for diabetes mellitus, dyslipidemia, and hypertension; and strain of SARS-CoV-2 (original and delta).

Reviewer #2**Comment 1.**

I enjoyed reviewing this manuscript. It is well written and the analysis appears to be robust. I have some concerns that I would like to be addressed. Most important is the potential for sample bias.

Response:

Thank you for your positive remarks and constructive feedback on our manuscript. We sincerely appreciate your invaluable comments and the opportunity to refine our manuscript. Your feedback has been instrumental in guiding our revisions. We have carefully considered your suggestions and have enhanced our manuscript as below. We hope these refinements align with your expectations and look forward to a favorable response.

Comment 2.

Can you be sure that participants did not have the other infection (i.e. that people with SARS-CoV-2 infection never had influenza and vice versa)?

Response:

Thank you for your invaluable comments. We understand your concern about the potential co-infection among participants. However, in our study, participants with a history of influenza in the look-back period (2018-individual index date) were excluded from this study to ignore the possible respiratory symptoms in relation to previous infection of influenza. Also, patients with previous records of long-term post-acute respiratory sequelae (chronic respiratory failure, pulmonary hypertension, sleep apnea, chronic obstructive pulmonary disease, emphysema, asthma, pulmonary sarcoidosis, and interstitial lung disease) before the index date were excluded from the analysis. This was done to minimize the risk of including patients with respiratory conditions related to a previous infection.

In addition, we have clarified the separation between the groups of participants in our study. Individuals diagnosed with SARS-CoV-2 infection were specifically identified and included in the COVID-19 group, ensuring they had no recorded influenza diagnosis within the look-back period. Conversely, those identified with influenza were placed in a separate group, with a similar exclusion criterion applied for SARS-CoV-2 infection.

Furthermore, we have re-analyzed our results according to comments from Reviewer #1 and your other comments. We divided the re-analysis into two comparisons. First, we investigated the risk of respiratory complications following SARS-CoV-2 infection compared to the general population as a key outcome of this study, presented in the main table. Second, instead of an indirect comparison, we conducted a direct comparison through 1:1 exposure-driven propensity score matching to compare COVID-19 with influenza. To aid your understanding, we present our revised study design, which is shown in the Venn diagram below.

(only for reviewer)

PS matching cohort

- A: participants infected with only SARS-CoV-2
- B: participants only infected with influenza from January 2020 to December 2021
- C: participants infected with SARS-CoV-2 and were infected with influenza after being diagnosed with COVID-19
- D: participants without SARS-CoV-2 or influenza infection

COVID-19 vs. general population: A, C vs. B, D

COVID-19 vs. influenza: A, C vs. B

We hope our explanation can address your concerns and help you understand our study. We appreciate the opportunity to clarify this aspect.

Changes in text:

Abstract

- We aimed to identify the risk of acute respiratory complications or long-term post-acute respiratory sequelae in **long COVID**.
- After exposure-driven 1:5 propensity score matching, we found that the risk of acute respiratory complication or long-term respiratory sequelae is significantly increased in people with SARS-CoV-2 infection **compared to the general population (acute respiratory complication: HR, 8.06 [95% CI, 6.92-9.38]; long-term respiratory sequelae: HR, 1.68 [95% CI, 1.62-1.75])**, and the risk increased with increasing COVID-19 severity. The SARS-CoV-2 infection induced a significantly increased risk for several acute respiratory complications (**aspergillosis pneumonia, pneumothorax, acute respiratory failure, and pulmonary embolism**)

and long-term respiratory sequelae (chronic respiratory failure, COPD, emphysema, asthma, and interstitial lung disease).

- Through this large-scale, binational, and population-based cohort study, acute or long-term respiratory sequelae in long COVID were observed compared to the general population.

Methods/Exposures

- To examine the relative severity of COVID-19 in comparison with another contagious viral respiratory disease, additional exposure to influenza infection was defined. It refers to cases diagnosed through an RT-PCR assay or antigen test on nasal and pharyngeal swabs during the observation period. For individuals infected with both SARS-CoV-2 and influenza, it includes instances of influenza infection developing after the SARS-CoV-2 infection.

Methods/Propensity score matching

- To enhance the robustness and generalizability of our primary findings and balance baseline covariates, we employed exposure-driven propensity score matching. This approach compared individuals with SARS-CoV-2 infection to those without infection as a general population.²¹ The propensity score was calculated by using a logistic regression model, adjusted for age (20–39, 40–59, and ≥60 years), sex (male and female), region of residence (urban and rural), history of cardiovascular and chronic disease, and medication use for diabetes, hyperlipidemia, and hypertension. Individuals were paired in 1:5 ratio between the exposure group (SARS-CoV-2) and the non-exposure group. Through the prior procedures, we generated multi-to-one matched cohorts utilized a ‘greedy nearest-neighbor’ algorithm, maintaining a caliper width of 0.001 standard deviations. The quality of the match was evaluated through the standardized mean differences (SMD), with an SMD less than 0.1 signifying minimal imbalances between the groups.²¹ In addition, to investigate the relative severity of COVID-19 compared to other infectious viral respiratory diseases, an influenza group within the general population was utilized as another control group, directly matching SARS-CoV-2 infections at a 1:1 ratio.

Results

- **Table 1** shows the baseline characteristics of 1:5 propensity score matched cohort of South Korea. After 1:5 propensity score matching based on SARS-CoV-2 infection, we identified 82.9% (1,918,150/2,312,748) of participants without SARS-CoV-2 infection and 17.1% (394,598/2,312,748) of participants with SARS-CoV-2 infection, respectively.

- In the 1:3 propensity score matched replication cohort, 74.4% (2,318,505/3,115,606) of participants without SARS-CoV-2 infection and 25.6% (797,101/3,115,606) of participants with SARS-CoV-2 infection were included in our final analyses (Table S10).
- In the main and replication cohorts, individuals with SARS-CoV-2 infection had a higher adjusted HR for long-term post-acute respiratory sequelae compared to the general population (main: HR, 1.68 [95% CI, 1.62-1.75]; replication: HR, 3.32 [95% CI, 3.27-3.37]) in Table 2. Furthermore, patients with SARS-CoV-2 infection had an increased risk for acute respiratory complication compared to non-infected controls (main: HR, 8.06 [95% CI, 6.92-9.38]; replication: HR, 4.17 [95% CI, 3.90-4.45]). When directly comparing the risk for acute respiratory complication between SARS-CoV-2 and influenza infections, SARS-CoV-2 infection was significantly associated with an increased risk (main: HR, 4.32 [95% CI, 2.73-6.83]; replication: HR, 6.51 [95% CI, 5.38-7.87]) in Tables S11-S13.
- Relative to the general population, patients with SARS-CoV-2 infection had significantly increased risk for several subtypes of long-term respiratory sequelae, including chronic respiratory failure (main: HR, 8.92 [95% CI, 4.92-16.17]; replication: HR, 7.55 [95% CI, 6.35-8.97]), COPD, emphysema, asthma, pulmonary sarcoidosis, and interstitial lung disease (main: HR, 10.38 [95% CI, 8.75-12.31]; replication: HR, 4.75 [95% CI, 4.54-4.97]) in Table 3. Notably, the risk for acute respiratory complication, including aspergillosis pneumonia (main: HR, 6.85 [95% CI, 3.48-13.50]; replication: HR, 4.97 [95% CI, 4.26-5.79]), pneumothorax, acute respiratory failure (main: HR, 112.04 [95% CI, 64.00-196.16]; replication: HR, 6.49 [95% CI, 6.32-6.65]) showed an increase in patients with SARS-CoV-2 infection compared to the general population. This tendency of increased risk for several subtypes of respiratory diseases was also shown when compared to patients with influenza infection (Table S14).
- The risk of acute respiratory complication showed decreasing trends according to the number of SARS-CoV-2 vaccinations from individuals after once receiving vaccination (HR, 0.51 [95% CI, 0.38-0.68]) to those with two or more vaccinations (HR, 0.24 [95% CI, 0.19-0.30]). Interestingly, mixed types of vaccination showed the lowest risk of developing long-term respiratory sequelae of all SARS-CoV-2 vaccination methods (HR, 0.18 [95% CI, 0.08-0.38]). The risks of acute respiratory complication were higher in patients with moderate to severe COVID-19 symptoms (HR, 39.54 [95% CI, 33.54-46.62]). Both the original strain and the delta variant of SARS-CoV-2 were shown to have a higher risk of acute respiratory complications (original strain: HR, 9.21 [95% CI, 7.19-11.80]; delta strain: HR, 7.44 [95% CI, 6.13-9.03]).

In addition, the risk for long-term post-acute respiratory sequelae also exhibited a similar pattern (Tables 4 and S15).

- Table 5 shows the risk of developing acute respiratory complication or long-term post-acute respiratory sequelae based on how long it has been since the participant was infected with SARS-CoV-2 compared to the general population.

- The first three months after infection with SARS-CoV-2 had the highest risk of developing long-term respiratory sequelae (main: HR, 2.51 [95% CI, 2.38-2.64]; replication: HR, 4.40 [95% CI, 4.30-4.51]). With increasing duration post-SARS-CoV-2 infection, the risk of long-term post-acute respiratory sequelae significantly decreased, but the risk remained even after 6 months (main: HR, 1.10 [95% CI, 1.01-1.19]; replication: HR, 2.67 [95% CI, 2.61-2.73]). HR of time attenuation effect after SARS-CoV-2 infection showed significance compared to influenza infection likewise (Table S16).

Discussion/Findings of this study

- First, the risk of acute respiratory complication or long-term post-acute respiratory sequelae is significantly increased in participants with SARS-CoV-2 infection, compared to the general population. Second, SARS-CoV-2 infection induced a significantly increased risk for several specific long-term respiratory sequelae, including chronic respiratory failure, COPD, emphysema, asthma, and interstitial lung disease, compared to the general. In addition, several acute respiratory complications, including aspergillosis pneumonia, pneumothorax, acute respiratory failure, and pulmonary embolism, also depicted a notable increase in risk after SARS-CoV-2 infection, compared to the general population.

Table S2. Statistical analyses and justification

No. Statistical analysis (1 to 9)	Cohort	Justification
1. Incident respiratory diseases in COVID-19 versus general population	1. Main cohort after 1:5 propensity score matching 2. Replication cohort 1:3 propensity score matching	- Main results - To investigate the association of acute respiratory complication and long-term respiratory sequelae and long COVID-19. - Matching covariates: age (20–39, 40–59, and ≥60 years); sex; household income (low income, middle income, and high income); region of residence (urban and rural); previous history of cardiovascular disease, chronic kidney disease; and history of medication use for diabetes mellitus, dyslipidemia, and hypertension. - Model (main cohort): adjusting covariates for age (20–39, 40–59, and ≥60 years); sex, household income (low income, middle income, and high income); region of residence (urban and rural); CCI score (0, 1, and ≥2); obesity (underweight [<18.5 kg/m ²], normal [18.5 – 22.9 kg/m ²], overweight [23.0 – 24.9 kg/m ²], obese [≥ 25.0 kg/m ²], and unknown); blood pressure (systolic blood pressure <140 mmHg and diastolic blood pressure <90 mmHg, systolic blood pressure ≥ 140 mmHg or diastolic blood pressure ≥ 90 mmHg, and unknown); fasting blood glucose (<100 , ≥ 100 mg/dL, and unknown); serum total cholesterol (<200 , 200 – 239 , ≥ 240 mg/dL, and unknown); glomerular filtration rate (<60 , 60 – 89 , ≥ 90 mL/min/1.73 m ² , and unknown); smoking status (never, former, current smoker, and unknown); alcoholic drinks (<1 , 1 – 2 , 3 – 4 , ≥ 5 days per week, and unknown); aerobic physical activity (sufficient, insufficient, and unknown); previous history of cardiovascular disease, and chronic kidney disease; history of medication use for diabetes mellitus, dyslipidemia, and hypertension; and strain of SARS-CoV-2 (original and delta). - Model (replication cohort): adjusting covariates for age (20–39, 40–59, and ≥60 years); sex; insurance status (insured and dependent); CCI score (0, 1, and ≥ 2); body mass index (underweight [<18.5 kg/m ²], normal [18.5 – 22.9 kg/m ²], overweight [23.0 – 25.0 kg/m ²], obese [≥ 25.0 kg/m ²], and unknown); blood pressure (systolic blood pressure <140 mmHg and diastolic blood pressure <90 mmHg, systolic blood pressure

		≥140 mmHg or diastolic blood pressure ≥90 mmHg, and unknown); fasting blood glucose (<100, ≥100 mg/dL, and unknown); serum total cholesterol (<200, 200–239, ≥240 mg/dL, and unknown); glomerular filtration rate (<60, 60–89, ≥90 mL/min/1.73 m², and unknown); smoking status (non- and current smoker, and unknown); alcoholic drinks (rarely, sometimes, everyday, and unknown); aerobic physical activity (sufficient, insufficient, and unknown); previous history of cardiovascular disease, and chronic kidney disease; history of medication use for diabetes mellitus, dyslipidemia, and hypertension; and strain of SARS-CoV-2 (original and delta).
2. Subgroup analysis of the risk of several respiratory diseases after SARS-CoV-2 infection	1. Main cohort after 1:5 propensity score matching 2. Replication cohort 1:3 propensity score matching	- Main results - To investigate the likelihood of respiratory diseases (long-term post-acute respiratory sequelae and acute respiratory complication) after SARS-CoV-2 infection, compared with general population. - Long-term post-acute respiratory sequelae: chronic respiratory failure, pulmonary hypertension, sleep apnea, chronic obstructive pulmonary disease (COPD), emphysema, asthma, pulmonary sarcoidosis, and interstitial lung disease. - Acute respiratory complication: pneumocystis pneumonia, aspergillosis pneumonia, pleural empyema, lung abscess, pneumothorax, acute respiratory failure, and pulmonary embolism.
3. Subgroup analysis of the risk of incident respiratory diseases after SARS-CoV-2 according to the SARS-CoV-2 vaccinations, severity of COVID-19, and strain type	Main cohort after 1:5 propensity score matching	- Main results - To investigate the association of subsequent all-cause respiratory diseases following COVID-19, stratified by vaccination dose (once and twice or more), and type of vaccinations (vector, mRNA, and mix), severity of COVID-19 (mild and moderate-to-severe), and strain of SARS-CoV-2 (original and delta).
4. Attenuation time effect of long-term post-acute respiratory sequelae development after COVID-19	1. Main cohort after 1:5 propensity score matching 2. Replication cohort 1:3 propensity score matching	- Main results - To investigate the time attenuation effect of long-term post-acute respiratory sequelae development after SARS-CoV-2 infection (<3, 3 to 6, and ≥6 months).
5. Incident respiratory diseases in COVID-19 versus contemporary control (influenza)	1. Main cohort after 1:1 propensity score matching 2. Replication cohort 1:1 propensity score matching	- Contemporary control results, consistent with the methodology employed in statistical analysis no. 1, 2, and 4.

6. Negative control analysis of the risk of incident tympanic membrane perforation disease.	1. Main cohort after 1:5 propensity score matching 2. Replication cohort 1:3 propensity score matching	- To verify the validity of our findings and identify potential misclassification bias, we conducted a negative control analysis of tympanic membrane perforation disease following the COVID-19 diagnosis.
7. Incident respiratory diseases in positive for COVID-19 versus negative for COVID-19 after the PCR test	Positive for COVID-19 versus negative for COVID-19 after the PCR test after 1:5 propensity score matching in main cohort	- To conduct more sophisticated analysis of the association between SARS-CoV-2 infection and respiratory disease - The cohort was based on the claim record of individuals who tested positive for SARS-CoV-2 within two weeks following an RT-PCR test and those who did not.
8. Subgroup analysis of the risk of respiratory diseases after SARS-CoV-2 infection in overlap-weighted cohort	1. Main cohort after overlap-weighted cohort 2. Replication cohort after overlap-weighted cohort	- To assess the risks and prevalence of respiratory conditions following COVID-19, the average treatment effect in the overlap (ATO) method was used, with overlap-weighted hazard ratios calculated for precision. - Matching covariates: age (20–39, 40–59, and ≥60 years); sex; household income (low income, middle income, and high income); region of residence (urban and rural); previous history of cardiovascular disease, chronic kidney disease; and history of medication use for diabetes mellitus, dyslipidemia, and hypertension. - Model: adjusting covariates for age (20–39, 40–59, and ≥60 years); sex, household income (low income, middle income, and high income); region of residence (urban and rural); CCI score (0, 1, and ≥2); obesity (underweight [$<18.5 \text{ kg/m}^2$], normal [$18.5\text{--}22.9 \text{ kg/m}^2$]; overweight [$23.0\text{--}24.9 \text{ kg/m}^2$], obese [$\geq 25.0 \text{ kg/m}^2$], and unknown); blood pressure (systolic blood pressure $<140 \text{ mmHg}$ and diastolic blood pressure $<90 \text{ mmHg}$, systolic blood pressure $\geq 140 \text{ mmHg}$ or diastolic blood pressure $\geq 90 \text{ mmHg}$, and unknown); fasting blood glucose (<100 , $\geq 100 \text{ mg/dL}$, and unknown); serum total cholesterol (<200 , $200\text{--}239$, $\geq 240 \text{ mg/dL}$, and unknown); glomerular filtration rate (<60 , $60\text{--}89$, $\geq 90 \text{ mL/min/1.73 m}^2$, and unknown); smoking status (never, former, current smoker, and unknown); alcoholic drinks (<1 , $1\text{--}2$, $3\text{--}4$, ≥ 5 days per week, and unknown); aerobic physical activity (sufficient, insufficient, and unknown); previous history of cardiovascular disease, and chronic kidney disease; history of medication use for diabetes mellitus, dyslipidemia, and hypertension; and strain of SARS-CoV-2 (original and delta).

9. Stratification analysis of the risk of respiratory diseases and its subtypes development after COVID-19	1. Main cohort after 1:5 propensity score matching 2. Replication cohort 1:3 propensity score matching 3. Main cohort after 1:1 propensity score matching 4. Replication cohort 1:1 propensity score matching	- To investigate unexpected mediated effects by sex, age, region of residence, household income, body mass index, Charlson comorbidity index, smoking status, alcohol use, aerobic physical activity, and strain of SARS-CoV-2.
--	--	---

CCI, Charlson comorbidity index; PCR, polymerase chain reaction; SARS-CoV-2, severe acute respiratory syndrome coronavirus 2.

Table 1. Baseline characteristics for 1:5 propensity score–matched cohort (COVID-19 vs. general population) in South Korea (main)

Characteristic	COVID-19 vs. general population (n=2,312,748)		SMD*
	COVID-19 (n=394,598)	General population (n=1,918,150)	
Mean age (SD), y	47.5 (16.8)	46.8 (14.3)	0.046
Age, n (%)			0.026
20–39 y	143,273 (36.3)	702,322 (36.6)	
40–59 y	145,169 (36.8)	709,343 (37.0)	
≥60 y	106,156 (26.9)	506,485 (26.4)	
Sex, n (%)			0.006
Male	205,058 (52.0)	997,982 (52.0)	
Female	189,540 (48.0)	920,168 (48.0)	
Region of residence, n (%)			<0.001
Urban	213,052 (54.0)	1,035,261 (54.0)	
Rural	181,546 (46.0)	882,889 (46.0)	
Medical history, n (%)			
Cardiovascular disease	59,947 (15.2)	286,623 (14.9)	0.009
Chronic kidney disease	18,963 (4.8)	88,687 (4.6)	0.005
Medication use for diabetes	71,625 (18.2)	342,656 (17.9)	0.008
Medication use for hyperlipidemia	59,947 (15.2)	286,623 (14.9)	0.007
Medication use for hypertension	32,402 (8.2)	154,789 (8.1)	0.005
Unmatching covariates, n (%)[†]			
Charlson Comorbidity Index score			0.230
0	346,579 (87.8)	1,806,906 (94.2)	
1	30,584 (7.8)	60,556 (3.2)	
≥2	17,435 (4.4)	50,688 (2.6)	

Household income			<0.001
Low (0th–39th percentile)	182,632 (46.3)	887,593 (46.3)	
Middle (40th –79th percentile)	140,084 (35.5)	681,180 (35.5)	
High (80th–100th percentile)	71,882 (18.2)	349,377 (18.2)	
Body mass index			1.212
Underweight (<18.5 kg/m ²)	6875 (1.7)	67,970 (3.5)	
Normal (18.5-22.9 kg/m ²)	77,779 (19.7)	685,687 (35.8)	
Overweight (23.0-24.9 kg/m ²)	54,427 (13.8)	440,197 (23.0)	
Obese (≥25.0 kg/m ²)	95,082 (24.1)	724,074 (37.8)	
Unknown	160,435 (40.7)	222 (0.012)	
Blood pressure			1.157
SBP <140 mmHg and DBP <90 mmHg	199,342 (50.5)	1,673,777 (87.3)	
SBP ≥140mmHg or DBP ≥90 mmHg	33,711 (8.5)	240,991 (12.6)	
Unknown	161,545 (40.9)	3382 (0.2)	
Fasting blood glucose			1.179
<100 mg/dL	140,656 (35.7)	1,189,021 (62.0)	
≥100 mg/dL	92,374 (23.4)	725,682 (37.8)	
Unknown	161568 (40.9)	3447 (0.2)	
Serum total cholesterol			0.416
<200 mg/dL	67,886 (17.2)	525,684 (27.4)	
200 to 239 mg/dL	39,508 (10.0)	316,959 (16.5)	
≥240 mg/dL	16,860 (4.3)	134,405 (7.0)	
Unknown	270,344 (68.5)	941,102 (49.1)	
Glomerular filtration rate			1.180
<60 mL/min/1.73 m ²	8332 (2.1)	52,421 (2.7)	
60 to 89 mL/min/1.73 m ²	102,604 (26.0)	812,214 (42.3)	
≥90 mL/min/1.73 m ²	121,914 (30.9)	1,048,318 (54.7)	

Unknown	161,748 (41.0)	5197 (0.3)	
Smoking status			1.191
Never	154,105 (39.1)	1,214,221 (63.3)	
Former	42,814 (10.9)	299,027 (15.6)	
Current	37,263 (9.4)	404,323 (21.1)	
Unknown	160,416 (40.7)	579 (0.030)	
Alcohol consumption			1.156
<1 day/week	136,870 (34.7)	1,143,488 (59.6)	
1 to 2 days/week	66,311 (16.8)	545,090 (28.4)	
3 to 4 days/week	23,113 (5.9)	173,245 (9.0)	
≥5 days/week	7893 (2.0)	55,687 (2.9)	
Unknown	160,411 (40.7)	640 (0.034)	
Aerobic physical activity			1.179
Insufficient	118,792 (30.1)	959,088 (50.0)	
Sufficient	115,321 (29.2)	958,194 (50.0)	
Unknown	160,485 (40.7)	868 (0.1)	
Strain of SARS-CoV-2			0.004
Original	121,521 (30.8)	594,134 (31.0)	
Delta	273,077 (69.2)	1,324,016 (69.0)	

DBP, diastolic blood pressure; SARS-CoV-2, severe acute respiratory syndrome coronavirus 2; SBP, systolic blood pressure; SD, standard deviation; SMD, standardized mean difference.

* An SMD <0.1 indicates no significant imbalance. All SMDs were <0.100 in the propensity score–matched cohorts.

† Unmatched covariates were included as adjustment factors in statistical analyses.

Table 2. Hazard ratio (95% CI) for the **long-term post-acute respiratory sequelae** or **short-term acute respiratory complication** after SARS-CoV-2 infection in the propensity score-matched cohorts of South Korea (main) and Japan (replication)

Cohort	South Korea			Japan		
	COVID-19 vs. general population (n=2,312,748)			COVID-19 vs. general population (n=3,115,606)		
	Events, n (%)	HR (95% CI)		Events, n (%)	HR (95% CI)	
Model 1*		Model 2†	Model 3*		Model 4‡	
Long-term post-acute respiratory sequelae						
Comparators (general population or patients with influenza)	16,122 (0.84)	1.0 (reference)	1.0 (reference)	35300 (1.52)	1.0 (reference)	1.0 (reference)
Patients with COVID-19	5292 (1.34)	1.64 (1.59-1.69)	1.68 (1.62-1.75)	41074 (5.15)	3.50 (3.45-3.55)	3.32 (3.27-3.37)
Acute respiratory complication						
Comparators	331 (0.017)	1.0 (reference)	1.0 (reference)	1468 (0.06)	1.0 (reference)	1.0 (reference)
Patients with COVID-19	618 (0.16)	9.70 (8.46-11.11)	8.06 (6.92-9.38)	2304 (0.29)	4.60 (4.31-4.91)	4.17 (3.90-4.45)

CCI, Charlson comorbidity index; CI, confidence interval; HR, hazard ratio.

The data in bold indicate significant differences ($P < 0.05$).

***Model 1 and 3:** Adjusted for age (20–39, 40–59, and ≥ 60 years) and sex.

†**Model 2:** Adjusted for age (20–39, 40–59, and ≥ 60 years); sex, household income (low income, middle income, and high income); region of residence (urban and rural); CCI score (0, 1, and ≥ 2); obesity (underweight [$< 18.5 \text{ kg/m}^2$], normal [$18.5\text{--}22.9 \text{ kg/m}^2$]; overweight [$23.0\text{--}24.9 \text{ kg/m}^2$], obese [$\geq 25.0 \text{ kg/m}^2$], and unknown); blood pressure (systolic blood pressure $< 140 \text{ mmHg}$ and diastolic blood pressure $< 90 \text{ mmHg}$, systolic blood pressure $\geq 140 \text{ mmHg}$ or diastolic blood pressure $\geq 90 \text{ mmHg}$, and unknown); fasting blood glucose (< 100 , $\geq 100 \text{ mg/dL}$, and unknown); serum total cholesterol (< 200 , $200\text{--}239$, $\geq 240 \text{ mg/dL}$, and unknown); glomerular filtration rate (< 60 , $60\text{--}89$, $\geq 90 \text{ mL/min/1.73 m}^2$, and unknown); smoking status (never, former, current smoker, and unknown); alcoholic drinks (< 1 , $1\text{--}2$, $3\text{--}4$, ≥ 5 days per week, and unknown); aerobic physical activity (sufficient, insufficient, and unknown); previous history of cardiovascular disease, and chronic kidney disease; history of medication use for diabetes mellitus, dyslipidemia, and hypertension; and strain of SARS-CoV-2 (original and delta).

‡**Model 4:** Adjusted for age (20–39, 40–59, and ≥ 60 years); sex; insurance status (insured and dependent); CCI score (0, 1, and ≥ 2); body mass index (underweight [$< 18.5 \text{ kg/m}^2$], normal [$18.5\text{--}22.9 \text{ kg/m}^2$], overweight [$23.0\text{--}25.0 \text{ kg/m}^2$], obese [$\geq 25.0 \text{ kg/m}^2$], and unknown); blood pressure (systolic blood pressure $< 140 \text{ mmHg}$ and diastolic blood pressure $< 90 \text{ mmHg}$, systolic blood pressure $\geq 140 \text{ mmHg}$ or diastolic blood

pressure ≥ 90 mmHg, and unknown); fasting blood glucose (< 100 , ≥ 100 mg/dL, and unknown); serum total cholesterol (< 200 , 200–239, ≥ 240 mg/dL, and unknown); glomerular filtration rate (< 60 , 60–89, ≥ 90 mL/min/1.73 m², and unknown); smoking status (non- and current smoker, and unknown); alcoholic drinks (rarely, sometimes, everyday, and unknown); aerobic physical activity (sufficient, insufficient, and unknown); previous history of cardiovascular disease, and chronic kidney disease; history of medication use for diabetes mellitus, dyslipidemia, and hypertension; and strain of SARS-CoV-2 (original and delta).

Table 3. HR (95% CI) for the long-term post-acute respiratory sequelae or short-term acute respiratory complication subtypes after SARS-CoV-2 infection in the propensity score-matched cohorts in South Korea (main) and Japan (replication)

Cohort	South Korea			Japan		
	COVID-19 vs. general population (n=2,312,748)			COVID-19 vs. general population (n=3,115,606)		
	Events, n (%)	HR (95% CI)		Events, n (%)	HR (95% CI)	
Model 1*		Model 2†	Model 3*		Model 4	
Long-term post-acute respiratory sequelae						
Chronic respiratory failure						
Comparators (general population or patients with influenza)	18 (0.00094)	1.0 (reference)	1.0 (reference)	170 (0.0073)	1.0 (reference)	1.0 (reference)
Patients with COVID-19	46 (0.012)	12.80 (7.42-22.07)	8.92 (4.92-16.17)	688 (0.086)	11.85 (10.02-14.02)	7.55 (6.35-8.97)
Pulmonary hypertension						
Comparators	15 (0.00078)	1.0 (reference)	1.0 (reference)	156 (0.0067)	1.0 (reference)	1.0 (reference)
Patients with COVID-19	3 (0.00076)	1.00 (0.29-3.45)	0.60 (0.11-3.39)	217 (0.027)	4.07 (3.31-5.00)	3.11 (2.51-3.85)
Sleep apnea						
Comparators	1143 (0.060)	1.0 (reference)	1.0 (reference)	6643 (0.29)	1.0 (reference)	1.0 (reference)
Patients with COVID-19	235 (0.060)	1.02 (0.89-1.17)	1.13 (0.95-1.33)	5198 (0.65)	2.30 (2.22-2.39)	2.21 (2.13-2.29)
COPD						
Comparators	10846 (0.57)	1.0 (reference)	1.0 (reference)	11003 (0.47)	1.0 (reference)	1.0 (reference)
Patients with COVID-19	3359 (0.85)	1.54 (1.49-1.61)	1.57 (1.50-1.65)	15520 (1.95)	4.17 (4.07-4.27)	3.93 (3.83-4.03)
Emphysema						
Comparators	386 (0.020)	1.0 (reference)	1.0 (reference)	1550 (0.067)	1.0 (reference)	1.0 (reference)
Patients with COVID-19	133 (0.034)	1.73 (1.42-2.10)	1.60 (1.27-2.01)	2085 (0.26)	3.95 (3.70-4.22)	3.44 (3.22-3.68)
Asthma						
Comparators	4197 (0.22)	1.0 (reference)	1.0 (reference)	21314 (0.92)	1.0 (reference)	1.0 (reference)
Patients with COVID-19	1431 (0.36)	1.70 (1.60-1.80)	1.74 (1.62-1.87)	25311 (3.18)	3.53 (3.46-3.59)	3.44 (3.38-3.50)

Pulmonary sarcoidosis

Comparators	29 (0.0015)	1.0 (reference)	1.0 (reference)	972 (0.042)	1.0 (reference)	1.0 (reference)
Patients with COVID-19	4 (0.0010)	0.69 (0.24-1.95)	0.96 (0.34-2.75)	1255 (0.16)	3.78 (3.48-4.11)	3.44 (3.16-3.75)

Interstitial lung disease

Comparators	223 (0.012)	1.0 (reference)	1.0 (reference)	2942 (0.13)	1.0 (reference)	1.0 (reference)
Patients with COVID-19	453 (0.11)	10.13 (8.63-11.90)	10.38 (8.75-12.31)	5996 (0.75)	6.00 (5.74-6.27)	4.75 (4.54-4.97)

Acute respiratory complication**Pneumocystis pneumonia**

Comparators	5 (0.00026)	1.0 (reference)	1.0 (reference)	934 (0.040)	1.0 (reference)	1.0 (reference)
Patients with COVID-19	2 (0.00051)	1.96 (0.38-10.10)	0.03 (0.00-8550.49)	1426 (0.18)	4.46 (4.10-4.84)	3.28 (3.01-3.58)

Aspergillosis pneumonia

Comparators	16 (0.00083)	1.0 (reference)	1.0 (reference)	249 (0.011)	1.0 (reference)	1.0 (reference)
Patients with COVID-19	32 (0.0081)	9.73 (5.34-17.72)	6.85 (3.48-13.50)	601 (0.075)	7.05 (6.08-8.17)	4.97 (4.26-5.79)

Pleural empyema

Comparators	10 (0.00052)	1.0 (reference)	1.0 (reference)	24 (0.0010)	1.0 (reference)	1.0 (reference)
Patients with COVID-19	6 (0.0015)	2.93 (1.06-8.05)	1.45 (0.32-6.63)	226 (0.028)	27.44 (18.02-41.80)	22.00 (14.38-33.65)

Lung abscess

Comparators	17 (0.00089)	1.0 (reference)	1.0 (reference)	57 (0.0025)	1.0 (reference)	1.0 (reference)
Patients with COVID-19	7 (0.0018)	2.01 (0.83-4.85)	2.20 (0.81-6.00)	301 (0.038)	15.39 (11.60-20.43)	13.57 (10.19-18.07)

Pneumothorax

Comparators	43 (0.0022)	1.0 (reference)	1.0 (reference)	3818 (0.16)	1.0 (reference)	1.0 (reference)
Patients with COVID-19	50 (0.013)	5.69 (3.78-8.55)	5.29 (3.32-8.42)	3234 (0.41)	2.49 (2.37-2.60)	2.41 (2.30-2.53)

Acute respiratory failure

Comparators	13 (0.00068)	1.0 (reference)	1.0 (reference)	8767 (0.38)	1.0 (reference)	1.0 (reference)
Patients with COVID-19	363 (0.092)	135.7 (78.05-235.91)	112.04 (64.00-196.16)	20983 (2.63)	7.10 (6.92-7.28)	6.49 (6.32-6.65)

Pulmonary embolism

Comparators	209 (0.011)	1.0 (reference)	1.0 (reference)	2212 (0.10)	1.0 (reference)	1.0 (reference)
Patients with COVID-19	162 (0.041)	3.79 (3.08-4.65)	2.98 (2.32-3.82)	3972 (0.50)	5.26 (5.00-5.55)	4.58 (4.34-4.83)

CCI, Charlson comorbidity index; CI, confidence interval; COPD, chronic obstructive pulmonary disease; HR, hazard ratio; NA, not available.

The data in bold indicate significant differences ($P < 0.05$).

***Model 1 and 3:** Adjusted for age (20–39, 40–59, and ≥ 60 years) and sex.

†**Model 2:** Adjusted for age (20–39, 40–59, and ≥ 60 years); sex, household income (low income, middle income, and high income); region of residence (urban and rural); CCI score (0, 1, and ≥ 2); obesity (underweight [$< 18.5 \text{ kg/m}^2$], normal [$18.5\text{--}22.9 \text{ kg/m}^2$]; overweight [$23.0\text{--}24.9 \text{ kg/m}^2$], obese [$\geq 25.0 \text{ kg/m}^2$], and unknown); blood pressure (systolic blood pressure $< 140 \text{ mmHg}$ and diastolic blood pressure $< 90 \text{ mmHg}$, systolic blood pressure $\geq 140 \text{ mmHg}$ or diastolic blood pressure $\geq 90 \text{ mmHg}$, and unknown); fasting blood glucose (< 100 , $\geq 100 \text{ mg/dL}$, and unknown); serum total cholesterol (< 200 , $200\text{--}239$, $\geq 240 \text{ mg/dL}$, and unknown); glomerular filtration rate (< 60 , $60\text{--}89$, $\geq 90 \text{ mL/min/1.73 m}^2$, and unknown); smoking status (never, former, current smoker, and unknown); alcoholic drinks (< 1 , $1\text{--}2$, $3\text{--}4$, ≥ 5 days per week, and unknown); aerobic physical activity (sufficient, insufficient, and unknown); previous history of cardiovascular disease, and chronic kidney disease; history of medication use for diabetes mellitus, dyslipidemia, and hypertension; and strain of SARS-CoV-2 (original and delta).

‡**Model 4:** Adjusted for age (20–39, 40–59, and ≥ 60 years); sex; insurance status (insured and dependent); CCI score (0, 1, and ≥ 2); body mass index (underweight [$< 18.5 \text{ kg/m}^2$], normal [$18.5\text{--}22.9 \text{ kg/m}^2$], overweight [$23.0\text{--}25.0 \text{ kg/m}^2$], obese [$\geq 25.0 \text{ kg/m}^2$], and unknown); blood pressure (systolic blood pressure $< 140 \text{ mmHg}$ and diastolic blood pressure $< 90 \text{ mmHg}$, systolic blood pressure $\geq 140 \text{ mmHg}$ or diastolic blood pressure $\geq 90 \text{ mmHg}$, and unknown); fasting blood glucose (< 100 , $\geq 100 \text{ mg/dL}$, and unknown); serum total cholesterol (< 200 , $200\text{--}239$, $\geq 240 \text{ mg/dL}$, and unknown); glomerular filtration rate (< 60 , $60\text{--}89$, $\geq 90 \text{ mL/min/1.73 m}^2$, and unknown); smoking status (non- and current smoker, and unknown); alcoholic drinks (rarely, sometimes, everyday, and unknown); aerobic physical activity (sufficient, insufficient, and unknown); previous history of cardiovascular disease, and chronic kidney disease; history of medication use for diabetes mellitus, dyslipidemia, and hypertension; and strain of SARS-CoV-2 (original and delta).

Table 4. Subgroup analysis (COVID-19 vs. general population) of HR (95% CI) of the **long-term post-acute respiratory sequelae** or **short-term acute respiratory complication** after SARS-CoV-2 infection stratified by vaccination, COVID-19 severity, and SARS-CoV-2 strain in the cohort of South Korea (main)

Variable	Events/total, n/N (%)	HR (95% CI)	
		Model 1*	Model 2†
Long-term post-acute respiratory sequelae			
Number of SARS-CoV-2 vaccinations			
Non-infected control	16,122/1,918,150 (0.84)	0.62 (0.60-0.65)	0.60 (0.58-0.62)
COVID-19 without SARS-CoV-2 vaccination	4331/200,539 (2.16)	1.0 (reference)	1.0 (reference)
COVID-19 after SARS-CoV-2 vaccination received once	493/38,852 (1.27)	0.90 (0.82-0.99)	0.85 (0.77-0.94)
COVID-19 after SARS-CoV-2 vaccination received twice or more	468/155,207 (0.30)	0.69 (0.62-0.76)	0.64 (0.57-0.71)
Type of SARS-CoV-2 vaccinations			
Non-infected control	16,122/1,918,150 (0.84)	0.62 (0.60-0.65)	0.60 (0.58-0.62)
COVID-19 without SARS-CoV-2 vaccination	4331/200,539 (2.16)	1.0 (reference)	1.0 (reference)
COVID-19 with viral vector SARS-CoV-2 vaccination	465/109,066 (0.43)	1.12 (1.01-1.23)	0.99 (0.90-1.10)
COVID-19 with mRNA SARS-CoV-2 vaccination	477/66,891 (0.71)	1.20 (1.09-1.33)	1.11 (0.99-1.22)
COVID-19 with both types of SARS-CoV-2 vaccination	19/18,102 (0.10)	0.74 (0.47-1.16)	0.66 (0.42-1.03)
COVID-19 severity			
Non-infected control	16,122/1,918,150 (0.84)	1.0 (reference)	1.0 (reference)
Mild COVID-19	3492/340,813 (1.02)	1.28 (1.24-1.33)	1.37 (1.32-1.43)
Moderate to severe COVID-19	1800/53785 (3.35)	3.60 (3.43-3.78)	3.20 (3.03-3.38)
Original strain of SARS-CoV-2 (overall population)			
Non-infected control before the delta-dominant phase [§]	11,667/594,134 (1.96)	1.0 (reference)	1.0 (reference)

Infection with original strain	3649/121,521 (3.00)	1.58 (1.52-1.64)	1.59 (1.52-1.66)
Delta variant of SARS-CoV-2 (overall population)			
Non-infected control during the delta-dominant phase [§]	4455/1,324,016 (0.34)	1.0 (reference)	1.0 (reference)
Infection with Delta variant	1643/273,077 (0.60)	1.81 (1.71-1.92)	1.94 (1.81-2.08)
Acute respiratory complication			
Number of SARS-CoV-2 vaccinations			
Non-infected control	311/1,918,150 (0.016)	0.07 (0.06-0.08)	0.08 (0.06-0.09)
COVID-19 without SARS-CoV-2 vaccination	415/200,539 (0.21)	1.0 (reference)	1.0 (reference)
COVID-19 after SARS-CoV-2 vaccination received once	56/38,852 (0.14)	0.62 (0.47-0.83)	0.51 (0.38-0.68)
COVID-19 after SARS-CoV-2 vaccination received twice or more	147/155,207 (0.09)	0.32 (0.26-0.39)	0.24 (0.19-0.30)
Type of SARS-CoV-2 vaccinations			
Non-infected control	311/1,918,150 (0.016)	0.07 (0.06-0.08)	0.08 (0.06-0.09)
COVID-19 without SARS-CoV-2 vaccination	415/200,539 (0.21)	1.0 (reference)	1.0 (reference)
COVID-19 with viral vector SARS-CoV-2 vaccination	79/109,066 (0.072)	0.44 (0.34-0.56)	0.36 (0.28-0.47)
COVID-19 with mRNA SARS-CoV-2 vaccination	117/66,891 (0.17)	0.40 (0.32-0.50)	0.42 (0.33-0.53)
COVID-19 with both types of SARS-CoV-2 vaccination	7/18,102 (0.039)	0.19 (0.09-0.41)	0.18 (0.08-0.38)
COVID-19 severity			
Non-infected control	311/1,918,150 (0.016)	1.0 (reference)	1.0 (reference)
Mild COVID-19	50/340,813 (0.015)	0.95 (0.71-1.28)	0.99 (0.73-1.34)
Moderate to severe COVID-19	568/53,783 (1.06)	51.18 (44.38-59.02)	39.54 (33.54-46.62)
Original strain of SARS-CoV-2 (overall population)			
Non-infected control before the delta-dominant phase [§]	110/594,134 (0.019)	1.0 (reference)	1.0 (reference)
Infection with original strain	230/121,521 (0.19)	10.27 (8.18-12.88)	9.21 (7.19-11.80)

Delta variant of SARS-CoV-2 (overall population)

Non-infected control during the delta-dominant phase §	201/1,324,016 (0.015)	1.0 (reference)	1.0 (reference)
Infection with Delta variant	388/273,077 (0.14)	9.39 (7.92-11.14)	7.44 (6.13-9.03)

CCI, Charlson comorbidity index; CI, confidence interval; HR, hazard ratio; SARS-CoV-2, severe acute respiratory syndrome coronavirus 2.

The data in bold indicate significant differences ($P < 0.05$).

|| HR of the non-infected control represents the risk of respiratory diseases, and HRs of patients with COVID-19 indicate the risk of post-acute respiratory complications following SARS-CoV-2 infection.

§ Only 1:5-matched comparators in each patient group at the same index date were included to reduce immortal time bias.

***Model 1:** Adjusted for age (20–39, 40–59, and ≥ 60 years) and sex.

†**Model 2:** Adjusted for age (20–39, 40–59, and ≥ 60 years); sex, household income (low income, middle income, and high income); region of residence (urban and rural); CCI score (0, 1, and ≥ 2); obesity (underweight [$< 18.5 \text{ kg/m}^2$], normal [$18.5\text{--}22.9 \text{ kg/m}^2$]; overweight [$23.0\text{--}24.9 \text{ kg/m}^2$], obese [$\geq 25.0 \text{ kg/m}^2$], and unknown); blood pressure (systolic blood pressure $< 140 \text{ mmHg}$ and diastolic blood pressure $< 90 \text{ mmHg}$, systolic blood pressure $\geq 140 \text{ mmHg}$ or diastolic blood pressure $\geq 90 \text{ mmHg}$, and unknown); fasting blood glucose (< 100 , $\geq 100 \text{ mg/dL}$, and unknown); serum total cholesterol (< 200 , $200\text{--}239$, $\geq 240 \text{ mg/dL}$, and unknown); glomerular filtration rate (< 60 , $60\text{--}89$, $\geq 90 \text{ mL/min/1.73 m}^2$, and unknown); smoking status (never, former, current smoker, and unknown); alcoholic drinks (< 1 , $1\text{--}2$, $3\text{--}4$, ≥ 5 days per week, and unknown); aerobic physical activity (sufficient, insufficient, and unknown); previous history of cardiovascular disease, and chronic kidney disease; history of medication use for diabetes mellitus, dyslipidemia, and hypertension; and strain of SARS-CoV-2 (original and delta).

Table 5. Time attenuation effect analysis of HR (95% CI) for the risk of **long-term post-acute respiratory sequelae** after SARS-CoV-2 infection in South Korea (main cohort) and Japan (replication cohort)

Time	COVID-19 vs. general population	
	Main cohort [†]	Replication cohort
long-term post-acute respiratory sequelae		
<3 months	2.51 (2.38-2.64)	4.40 (4.30-4.51)
3–6 months	1.24 (1.15-1.33)	2.66 (2.57-2.75)
≥6 months	1.10 (1.01-1.19)	2.67 (2.61-2.73)

CCI, Charlson comorbidity index; CI, confidence interval; HR, hazard ratio; SARS-CoV-2, severe acute respiratory syndrome coronavirus 2. The data in bold indicate significant differences ($P < 0.05$).

[†] **Adjusted HR (main):** Adjusted for age (20–39, 40–59, and ≥60 years); sex, household income (low income, middle income, and high income); region of residence (urban and rural); CCI score (0, 1, and ≥2); obesity (underweight [$<18.5 \text{ kg/m}^2$], normal [$18.5\text{--}22.9 \text{ kg/m}^2$]; overweight [$23.0\text{--}24.9 \text{ kg/m}^2$], obese [$\geq 25.0 \text{ kg/m}^2$], and unknown); blood pressure (systolic blood pressure $<140 \text{ mmHg}$ and diastolic blood pressure $<90 \text{ mmHg}$, systolic blood pressure $\geq 140 \text{ mmHg}$ or diastolic blood pressure $\geq 90 \text{ mmHg}$, and unknown); fasting blood glucose (<100 , $\geq 100 \text{ mg/dL}$, and unknown); serum total cholesterol (<200 , $200\text{--}239$, $\geq 240 \text{ mg/dL}$, and unknown); glomerular filtration rate (<60 , $60\text{--}89$, $\geq 90 \text{ mL/min/1.73 m}^2$, and unknown); smoking status (never, former, current smoker, and unknown); alcoholic drinks (<1 , $1\text{--}2$, $3\text{--}4$, ≥ 5 days per week, and unknown); aerobic physical activity (sufficient, insufficient, and unknown); previous history of cardiovascular disease, and chronic kidney disease; history of medication use for diabetes mellitus, dyslipidemia, and hypertension; and strain of SARS-CoV-2 (original and delta).

^{||} **Adjusted HR (replication):** Adjusted for age (20–39, 40–59, and ≥60 years); sex; insurance status (insured and dependent); CCI score (0, 1, and ≥ 2); body mass index (underweight [$<18.5 \text{ kg/m}^2$], normal [$18.5\text{--}22.9 \text{ kg/m}^2$], overweight [$23.0\text{--}25.0 \text{ kg/m}^2$], obese [$\geq 25.0 \text{ kg/m}^2$], and unknown); blood pressure (systolic blood pressure $<140 \text{ mmHg}$ and diastolic blood pressure $<90 \text{ mmHg}$, systolic blood pressure $\geq 140 \text{ mmHg}$ or diastolic blood pressure $\geq 90 \text{ mmHg}$, and unknown); fasting blood glucose (<100 , $\geq 100 \text{ mg/dL}$, and unknown); serum total cholesterol (<200 , $200\text{--}239$, $\geq 240 \text{ mg/dL}$, and unknown); glomerular filtration rate (<60 , $60\text{--}89$, $\geq 90 \text{ mL/min/1.73 m}^2$, and unknown); smoking status (non- and current smoker, and unknown); alcoholic drinks (rarely, sometimes, everyday, and unknown); aerobic physical activity (sufficient, insufficient, and unknown); previous history of cardiovascular disease, and chronic kidney disease; history of medication use for diabetes mellitus, dyslipidemia, and hypertension; and strain of SARS-CoV-2 (original and delta).

Table S10. Baseline characteristics for 1:3 propensity score–matched cohort (COVID-19 vs. general population) in Japan (replication)

Characteristic	COVID-19 vs. general population (n=3,115,606)		SMD*
	COVID-19 (n=797,101)	General population (n=2,318,505)	
Mean age (SD), y	44 (11.88)	44 (12.03)	0.034
Age, n (%)			<0.001
20–39 y	302,404 (37.94)	878,119 (37.87)	
40–59 y	411,351 (51.61)	1,199,717 (51.75)	
≥60 y	83,346 (10.46)	240,669 (10.38)	
Sex, n (%)			0.005
Male	495,460 (62.16)	1,447,190 (62.42)	
Female	301,641 (37.84)	871,315 (37.58)	
Insurance status, n (%)			0.001
Insured	718,811 (90.18)	2,091,308 (90.20)	
Dependent	78,290 (9.82)	227,197 (9.80)	
Medical history, n (%)			
Cardiovascular disease	60,450 (7.58)	170,601 (7.36)	0.009
Chronic kidney disease	32,292 (4.05)	90,680 (3.91)	0.007
Medication use for diabetes	19,777 (2.48)	56,919 (2.45)	0.002
Medication use for hyperlipidemia	49,255 (6.18)	141,631 (6.11)	0.003
Medication use for hypertension	67,167 (8.43)	191,951 (8.28)	0.005
Unmatching covariates, n (%)[†]			
Charlson Comorbidity Index score			0.203
0	764,052 (95.85)	2,293,774 (98.93)	
1	10,954 (1.37)	8,640 (0.37)	
≥2	22,095 (2.77)	16,091 (0.69)	

Body mass index			<0.001
Underweight (<18.5 kg/m ²)	427,244 (53.60)	1,263,598 (54.50)	
Normal (18.5-22.9 kg/m ²)	151,460 (19.00)	439,941 (18.98)	
Overweight (23.0-24.9 kg/m ²)	169,962 (21.32)	484,142 (20.88)	
Obese (≥25.0 kg/m ²)	47,210 (5.92)	127,534 (5.50)	
Unknown	1225 (0.15)	3290 (0.14)	
Blood pressure			0.038
SBP <140 mmHg and DBP <90 mmHg	691,676 (86.77)	1,986,000 (85.66)	
SBP ≥140mmHg or DBP ≥90 mmHg	46,474 (5.83)	149,954 (6.47)	
Unknown	58,951 (7.40)	182,551 (7.87)	
Fasting blood glucose			<0.001
<100 mg/dL	511,681 (64.19)	1,474,020 (63.58)	
≥100 mg/dL	160,434 (20.13)	474,726 (20.48)	
Unknown	124,986 (15.68)	369,759 (15.95)	
Serum total cholesterol			0.054
<200 mg/dL	373,377 (46.84)	1,040,983 (44.90)	
200 to 239 mg/dL	285,475 (35.81)	843,032 (36.36)	
≥240 mg/dL	123,823 (15.53)	379,819 (16.38)	
Unknown	14,426 (1.81)	54,671 (2.36)	
Glomerular filtration rate			0.037
<60 mL/min/1.73 m ²	4383 (0.55)	8174 (0.35)	
60 to 89 mL/min/1.73 m ²	72,932 (9.15)	215,112 (9.28)	
≥90 mL/min/1.73 m ²	433,528 (54.39)	1,237,792 (53.39)	
Unknown	286,258 (35.91)	857,427 (36.98)	
Smoking status			0.062
Non-smoker	181,033 (22.71)	554,190 (23.90)	
Smoker	586,552 (73.59)	1,677,800 (72.37)	

Unknown	29,516 (3.70)	86,515 (3.73)	
Alcohol consumption			0.040
Everyday	161,809 (20.30)	466,183 (20.11)	
Sometimes	264,583 (33.19)	762,501 (32.89)	
Rarely	311,632 (39.10)	919,772 (39.67)	
Unknown	59,077 (7.41)	170,049 (7.33)	
Aerobic physical activity			0.046
Insufficient	161,205 (20.22)	497,575 (21.46)	
Sufficient	562,300 (70.54)	1,605,294 (69.24)	
Unknown	73,596 (9.23)	215,636 (9.30)	
Strain of SARS-CoV-2			0.005
Original	335,571 (42.10)	981,871 (42.35)	
Delta	461,530 (57.90)	1,336,634 (57.65)	

DBP, diastolic blood pressure; SARS-CoV-2, severe acute respiratory syndrome coronavirus 2; SBP, systolic blood pressure; SD, standard deviation; SMD, standardized mean difference.

* An SMD <0.1 indicates no significant imbalance. All SMDs were <0.100 in the propensity score-matched cohorts.

† Unmatched covariates were included as adjustment factors in statistical analyses.

Table S11. Baseline characteristics for 1:1 propensity score-matched cohort (COVID-19 vs. influenza) in South Korea (main)

Characteristic	COVID-19 vs. influenza (n=223,000)		SMD*
	COVID-19 (n=111,500)	Influenza (n=111,500)	
Mean age (SD), y	45.3 (15.3)	45.0 (12.7)	0.017
Age, n (%)			0.060
20–39 y	42,625 (38.2)	40,131 (36.0)	
40–59 y	50,713 (45.5)	53,295 (47.8)	
≥60 y	18,162 (16.3)	18,074 (16.2)	
Sex, n (%)			0.046
Male	46,400 (41.6)	49,587 (44.5)	
Female	65,100 (58.4)	61,913 (55.5)	
Region of residence, n (%)			0.005
Urban	49,431 (44.3)	49,160 (44.1)	
Rural	62,069 (55.7)	62,340 (55.9)	
Medical history, n (%)			
Cardiovascular disease	4166 (3.7)	4258 (3.8)	0.004
Chronic kidney disease	2011 (1.8)	1991 (1.8)	0.001
Medication use for diabetes	17,218 (15.4)	16,665 (15.0)	0.014
Medication use for hyperlipidemia	17,656 (15.8)	17,167 (15.4)	0.012
Medication use for hypertension	7408 (6.6)	6640 (6.0)	0.028
Unmatching covariates, n (%)[†]			
Charlson Comorbidity Index score			0.103
0	99,673 (89.4)	103,133 (92.5)	
1	7867 (7.1)	5296 (4.8)	
≥2	3960 (3.6)	3071 (2.8)	

Household income			0.078
Low (0th–39th percentile)	48,878 (43.8)	46,012 (41.3)	
Middle (40th –79th percentile)	45,339 (40.7)	45,119 (40.5)	
High (80th–100th percentile)	17,283 (15.5)	20,369 (18.3)	
Body mass index			1.180
Underweight (<18.5 kg/m ²)	2120 (1.9)	3997 (3.6)	
Normal (18.5-22.9 kg/m ²)	22,894 (20.5)	39,561 (35.5)	
Overweight (23.0-24.9 kg/m ²)	14,256 (12.8)	24,459 (21.9)	
Obese (≥25.0 kg/m ²)	25,159 (22.6)	43,474 (39.0)	
Unknown	47,071 (42.2)	9 (0.0081)	
Blood pressure			1.207
SBP <140 mmHg and DBP <90 mmHg	55,801 (50.1)	99,099 (88.9)	
SBP ≥140mmHg or DBP ≥90 mmHg	8400 (7.5)	12,237 (11.0)	
Unknown	47299 (42.4)	164 (0.15)	
Fasting blood glucose			1.229
<100 mg/dL	40,399 (36.2)	72,206 (64.8)	
≥100 mg/dL	23,797 (21.3)	39,131 (35.1)	
Unknown	47,304 (42.4)	163 (0.15)	
Serum total cholesterol			0.462
<200 mg/dL	17,459 (15.7)	29,991 (26.9)	
200 to 239 mg/dL	10485 (9.4)	18,760 (16.8)	
≥240 mg/dL	4859 (4.4)	7923 (7.1)	
Unknown	78,697 (70.6)	54,826 (49.2)	
Glomerular filtration rate			1.208
<60 mL/min/1.73 m ²	1689 (1.5)	2223 (2.0)	
60 to 89 mL/min/1.73 m ²	26,592 (23.9)	45,211 (40.6)	
≥90 mL/min/1.73 m ²	35,867 (32.2)	63,785 (57.2)	

Unknown	47,352 (42.5)	281 (0.25)	
Smoking status			1.155
Never	45,487 (40.8)	75,040 (67.3)	
Former	9450 (8.5)	15,936 (14.3)	
Current	9506 (8.5)	20,488 (18.4)	
Unknown	47,057 (42.2)	36 (0.032)	
Alcohol consumption			1.208
<1 day/week	38,511 (34.5)	68078 (61.1)	
1 to 2 days/week	17,887 (16.0)	31025 (27.8)	
3 to 4 days/week	6125 (5.5)	9620 (8.6)	
≥5 days/week	1922 (1.7)	2741 (2.5)	
Unknown	47,055 (42.2)	36 (0.032)	
Aerobic physical activity			1.228
Insufficient	33,898 (30.4)	58,183 (52.2)	
Sufficient	30,539 (27.4)	53,276 (47.8)	
Unknown	47,063 (42.2)	41 (0.037)	
Strain of SARS-CoV-2			<0.001
Original	34685 (31.1)	34,718 (31.1)	
Delta	76,815 (68.9)	76,782 (68.9)	

DBP, diastolic blood pressure; SARS-CoV-2, severe acute respiratory syndrome coronavirus 2; SBP, systolic blood pressure; SD, standard deviation; SMD, standardized mean difference.

* An SMD <0.1 indicates no significant imbalance. All SMDs were <0.100 in the propensity score-matched cohorts.

† Unmatched covariates were included as adjustment factors in statistical analyses.

Table S12. Baseline characteristics for 1:1 propensity score-matched cohort (COVID-19 vs. influenza) in Japan (replication)

Characteristic	COVID-19 vs. influenza (n=178,648)		SMD*
	COVID-19 (n=89,324)	Influenza (n=89,324)	
Mean age (SD), y	44 (11.76)	44 (11.57)	0.018
Age, n (%)			<0.001
20–39 y	31,256 (34.99)	31,210 (34.94)	
40–59 y	48,959 (54.81)	49,241 (55.13)	
≥60 y	9109 (10.20)	8873 (9.93)	
Sex, n (%)			0.005
Male	55,617 (62.26)	55,396 (62.02)	
Female	33,707 (37.74)	33,928 (37.98)	
Insurance status, n (%)			<0.001
Insured	80,141 (89.72)	80,141 (89.72)	
Dependent	9183 (10.28)	9183 (10.28)	
Medical history, n (%)			
Cardiovascular disease	6235 (6.98)	6214 (6.96)	0.001
Chronic kidney disease	3031 (3.39)	3252 (3.64)	0.013
Medication use for diabetes	2422 (2.71)	2200 (2.46)	0.016
Medication use for hyperlipidemia	6159 (6.90)	5877 (6.58)	0.013
Medication use for hypertension	8016 (8.97)	7716 (8.64)	0.012
Unmatching covariates, n (%)[†]			
Charlson Comorbidity Index score			0.153
0	85,599 (95.83)	87,875 (98.38)	
1	1150 (1.29)	501 (0.56)	
≥2	2575 (2.88)	948 (1.06)	

Body mass index			0.007
Underweight (<18.5 kg/m ²)	47,532 (53.21)	47,767 (53.48)	
Normal (18.5-22.9 kg/m ²)	17,018 (19.05)	16,935 (18.96)	
Overweight (23.0-24.9 kg/m ²)	19,260 (21.56)	19,131 (21.42)	
Obese (≥25.0 kg/m ²)	5373 (6.02)	5366 (6.01)	
Unknown	125 (0.14)	141 (0.16)	
Blood pressure			<0.001
SBP <140 mmHg and DBP <90 mmHg	77,118 (86.34)	77,019 (86.22)	
SBP ≥140mmHg or DBP ≥90 mmHg	5456 (6.11)	5383 (6.03)	
Unknown	6750 (7.56)	6922 (7.75)	
Fasting blood glucose			0.028
<100 mg/dL	57,035 (63.85)	57,644 (64.53)	
≥100 mg/dL	18,237 (20.42)	18,284 (20.47)	
Unknown	14,052 (15.73)	13,396 (15.00)	
Serum total cholesterol			0.022
<200 mg/dL	41,317 (46.26)	39,896 (44.66)	
200 to 239 mg/dL	32,270 (36.13)	33,092 (37.05)	
≥240 mg/dL	14,211 (15.91)	14,672 (16.43)	
Unknown	1526 (1.71)	1664 (1.86)	
Glomerular filtration rate			0.073
<60 mL/min/1.73 m ²	501 (0.56)	296 (0.33)	
60 to 89 mL/min/1.73 m ²	8344 (9.34)	7987 (8.94)	
≥90 mL/min/1.73 m ²	48,818 (54.65)	46,549 (52.11)	
Unknown	31,661 (35.45)	34,492 (38.61)	
Smoking status			0.062
Non-smoker	20,354 (22.79)	21,458 (24.02)	

Smoker	65,734 (73.59)	64,706 (72.44)	
Unknown	3236 (3.62)	3160 (3.54)	
Alcohol consumption			0.049
Everyday	18,486 (20.70)	18,284 (20.47)	
Sometimes	29,295 (32.80)	28,687 (32.12)	
Rarely	35,096 (39.29)	35,716 (39.98)	
Unknown	6447 (7.22)	6637 (7.43)	
Aerobic physical activity			0.035
Insufficient	18,249 (20.43)	18,303 (20.49)	
Sufficient	62,979 (70.51)	62,858 (70.37)	
Unknown	8096 (9.06)	8163 (9.14)	
Strain of SARS-CoV-2			1.652
Original	36,886 (41.29)	88,835 (99.45)	
Delta	52,438 (58.71)	489 (0.55)	

DBP, diastolic blood pressure; SARS-CoV-2, severe acute respiratory syndrome coronavirus 2; SBP, systolic blood pressure; SD, standard deviation; SMD, standardized mean difference.

* An SMD <0.1 indicates no significant imbalance. All SMDs were <0.100 in the propensity score–matched cohorts.

† Unmatched covariates were included as adjustment factors in statistical analyses.

Table S13. Hazard ratio (95% CI) for the **long-term post-acute respiratory sequelae** or **short-term acute respiratory complication** after SARS-CoV-2 infection in the propensity score-matched cohorts (COVID-19 vs. influenza) of South Korea (main) and Japan (replication)

Cohort	South Korea			Japan		
	COVID-19 vs. influenza (n=223,000)			COVID-19 vs. influenza (n=169,924)		
	Events, n (%)	HR (95% CI)		Events, n (%)	HR (95% CI)	
Model 1*		Model 2†	Model 3*		Model 4‡	
Long-term post-acute respiratory sequelae						
Comparators (general population or patients with influenza)	1081 (0.97)	1.0 (reference)	1.0 (reference)	905 (1.07)	1.0 (reference)	1.0 (reference)
Patients with COVID-19	1500 (1.35)	1.55 (1.43-1.67)	1.66 (1.52-1.82)	4757 (5.60)	5.44 (5.07-5.84)	5.17 (4.82-5.55)
Acute respiratory complication						
Comparators	45 (0.040)	1.0 (reference)	1.0 (reference)	122 (0.14)	1.0 (reference)	1.0 (reference)
Patients with COVID-19	115 (0.10)	4.75 (3.08-7.33)	4.32 (2.73-6.83)	924 (1.09)	7.57 (6.27-9.14)	6.51 (5.38-7.87)

CCI, Charlson comorbidity index; CI, confidence interval; HR, hazard ratio.

The data in bold indicate significant differences ($P < 0.05$).

***Model 1 and 3:** Adjusted for age (20–39, 40–59, and ≥ 60 years) and sex.

†**Model 2:** Adjusted for age (20–39, 40–59, and ≥ 60 years); sex, household income (low income, middle income, and high income); region of residence (urban and rural); CCI score (0, 1, and ≥ 2); obesity (underweight [$< 18.5 \text{ kg/m}^2$], normal [$18.5\text{--}22.9 \text{ kg/m}^2$]; overweight [$23.0\text{--}24.9 \text{ kg/m}^2$], obese [$\geq 25.0 \text{ kg/m}^2$], and unknown); blood pressure (systolic blood pressure $< 140 \text{ mmHg}$ and diastolic blood pressure $< 90 \text{ mmHg}$, systolic blood pressure $\geq 140 \text{ mmHg}$ or diastolic blood pressure $\geq 90 \text{ mmHg}$, and unknown); fasting blood glucose (< 100 , $\geq 100 \text{ mg/dL}$, and unknown); serum total cholesterol (< 200 , $200\text{--}239$, $\geq 240 \text{ mg/dL}$, and unknown); glomerular filtration rate (< 60 , $60\text{--}89$, $\geq 90 \text{ mL/min/1.73 m}^2$, and unknown); smoking status (never, former, current smoker, and unknown); alcoholic drinks (< 1 , $1\text{--}2$, $3\text{--}4$, ≥ 5 days per week, and unknown); aerobic physical activity (sufficient, insufficient, and unknown); previous history of cardiovascular disease, and chronic kidney disease; history of medication use for diabetes mellitus, dyslipidemia, and hypertension; and strain of SARS-CoV-2 (original and delta).

‡**Model 4:** Adjusted for age (20–39, 40–59, and ≥ 60 years); sex; insurance status (insured and dependent); CCI score (0, 1, and ≥ 2); body mass index (underweight [$< 18.5 \text{ kg/m}^2$], normal [$18.5\text{--}22.9 \text{ kg/m}^2$], overweight [$23.0\text{--}25.0 \text{ kg/m}^2$], obese [$\geq 25.0 \text{ kg/m}^2$], and unknown); blood

pressure (systolic blood pressure <140 mmHg and diastolic blood pressure <90 mmHg, systolic blood pressure \geq 140 mmHg or diastolic blood pressure \geq 90 mmHg, and unknown); fasting blood glucose (<100, \geq 100 mg/dL, and unknown); serum total cholesterol (<200, 200–239, \geq 240 mg/dL, and unknown); glomerular filtration rate (<60, 60–89, \geq 90 mL/min/1.73 m², and unknown); smoking status (non- and current smoker, and unknown); alcoholic drinks (rarely, sometimes, everyday, and unknown); aerobic physical activity (sufficient, insufficient, and unknown); previous history of cardiovascular disease, and chronic kidney disease; history of medication use for diabetes mellitus, dyslipidemia, and hypertension; and strain of SARS-CoV-2 (original and delta).

Table S14. HR (95% CI) for the long-term post-acute respiratory sequelae or short-term acute respiratory complication subtypes after SARS-CoV-2 infection in the propensity score-matched cohorts (COVID-19 vs. influenza) in South Korea (main) and Japan (replication)

Cohort	South Korea			Japan		
	COVID-19 vs. influenza (n=223,000)			COVID-19 vs. influenza (n=169,924)		
	Events, n (%)	HR (95% CI)		Events, n (%)	HR (95% CI)	
Model 1*		Model 2†	Model 3*		Model 4	
Long-term post-acute respiratory sequelae						
Chronic respiratory failure						
Comparators (general population or patients with influenza)	3 (0.0027)	1.0 (reference)	1.0 (reference)	3 (0.0035)	1.0 (reference)	1.0 (reference)
Patients with COVID-19	12 (0.011)	4.19 (1.18-14.86)	4.82 (1.28-18.12)	77 (0.091)	25.55 (8.06-80.95)	16.15 (5.04-51.80)
Pulmonary hypertension						
Comparators	0 (0.00)	1.0 (reference)	1.0 (reference)	4 (0.0047)	1.0 (reference)	1.0 (reference)
Patients with COVID-19	0 (0.00)	NA	NA	28 (0.033)	6.97 (2.44-19.87)	5.59 (1.93-16.14)
Sleep apnea						
Comparators	23 (0.021)	1.0 (reference)	1.0 (reference)	88 (0.10)	1.0 (reference)	1.0 (reference)
Patients with COVID-19	49 (0.044)	4.26 (2.26-8.01)	4.12 (2.09-8.12)	549 (0.65)	6.26 (5.00-7.84)	5.95 (4.74-7.46)
COPD						
Comparators	734 (0.66)	1.0 (reference)	1.0 (reference)	271 (0.32)	1.0 (reference)	1.0 (reference)
Patients with COVID-19	994 (0.89)	1.48 (1.34-1.63)	1.60 (1.44-1.79)	1703 (2.00)	0.99 (0.91-1.09)	1.01 (0.91-1.12)
Emphysema						
Comparators	19 (0.017)	1.0 (reference)	1.0 (reference)	31 (0.036)	1.0 (reference)	1.0 (reference)
Patients with COVID-19	33 (0.030)	1.96 (1.10-3.47)	1.99 (1.06-3.74)	232 (0.27)	7.47 (5.14-10.87)	6.46 (4.43-9.43)
Asthma						
Comparators	337 (0.30)	1.0 (reference)	1.0 (reference)	477 (0.56)	1.0 (reference)	1.0 (reference)
Patients with COVID-19	421 (0.38)	1.37 (1.18-1.58)	1.45 (1.23-1.71)	2672 (3.14)	5.72 (5.19-6.30)	5.59 (5.07-6.16)
Pulmonary sarcoidosis						

Comparators	2 (0.0018)	1.0 (reference)	1.0 (reference)	25 (0.029)	1.0 (reference)	1.0 (reference)
Patients with COVID-19	2 (0.0018)	2.07 (0.19-22.84)	3.24 (0.29-36.47)	127 (0.15)	5.08 (3.31-7.81)	4.52 (2.93-6.96)
Interstitial lung disease						
Comparators	32 (0.029)	1.0 (reference)	1.0 (reference)	67 (0.079)	1.0 (reference)	1.0 (reference)
Patients with COVID-19	94 (0.084)	3.25 (2.15-4.90)	3.37 (2.19-5.19)	679 (0.80)	10.19 (7.93-13.1)	8.26 (6.42-10.64)
Acute respiratory complication						
Pneumocystis pneumonia						
Comparators	0 (0.00)	1.0 (reference)	1.0 (reference)	18 (0.021)	1.0 (reference)	1.0 (reference)
Patients with COVID-19	0 (0.00)	NA	NA	66 (0.078)	3.65 (2.17-6.14)	2.85 (1.68-4.85)
Aspergillosis pneumonia						
Comparators	0 (0.00)	1.0 (reference)	1.0 (reference)	4 (0.0047)	1.0 (reference)	1.0 (reference)
Patients with COVID-19	9 (0.0081)	NA	NA	17 (0.020)	4.21 (1.42-12.52)	3.49 (1.16-10.54)
Pleural empyema						
Comparators	2 (0.0018)	1.0 (reference)	1.0 (reference)	0 (0.00)	1.0 (reference)	1.0 (reference)
Patients with COVID-19	0 (0.00)	NA	NA	18 (0.021)	NA	NA
Lung abscess						
Comparators	6 (0.0054)	1.0 (reference)	1.0 (reference)	6 (0.0071)	1.0 (reference)	1.0 (reference)
Patients with COVID-19	0 (0.00)	NA	NA	18 (0.021)	2.98 (1.18-7.50)	2.25 (0.88-5.80)
Pneumothorax						
Comparators	18 (0.016)	1.0 (reference)	1.0 (reference)	22 (0.026)	1.0 (reference)	1.0 (reference)
Patients with COVID-19	10 (0.0090)	1.05 (0.44-2.53)	1.02 (0.37-2.81)	114 (0.13)	5.17 (3.28-8.16)	4.92 (3.11-7.77)
Acute respiratory failure						
Comparators	2 (0.0018)	1.0 (reference)	1.0 (reference)	58 (0.068)	1.0 (reference)	1.0 (reference)
Patients with COVID-19	67 (0.060)	NA	NA	583 (0.69)	10.04 (7.67-13.15)	8.44 (6.43-11.07)
Pulmonary embolism						
Comparators	18 (0.06)	1.0 (reference)	1.0 (reference)	18 (0.021)	1.0 (reference)	1.0 (reference)
Patients with COVID-19	29 (0.026)	2.94 (1.43-6.03)	3.27 (1.53-7.02)	175 (0.21)	9.68 (5.96-15.73)	7.96 (4.88-12.98)

CCI, Charlson comorbidity index; CI, confidence interval; COPD, chronic obstructive pulmonary disease; HR, hazard ratio; NA, not available.

The data in bold indicate significant differences ($P < 0.05$).

***Model 1 and 3:** Adjusted for age (20–39, 40–59, and ≥ 60 years) and sex.

†**Model 2:** Adjusted for age (20–39, 40–59, and ≥ 60 years); sex, household income (low income, middle income, and high income); region of residence (urban and rural); CCI score (0, 1, and ≥ 2); obesity (underweight [$< 18.5 \text{ kg/m}^2$], normal [$18.5\text{--}22.9 \text{ kg/m}^2$]; overweight [$23.0\text{--}24.9 \text{ kg/m}^2$], obese [$\geq 25.0 \text{ kg/m}^2$], and unknown); blood pressure (systolic blood pressure $< 140 \text{ mmHg}$ and diastolic blood pressure $< 90 \text{ mmHg}$, systolic blood pressure $\geq 140 \text{ mmHg}$ or diastolic blood pressure $\geq 90 \text{ mmHg}$, and unknown); fasting blood glucose (< 100 , $\geq 100 \text{ mg/dL}$, and unknown); serum total cholesterol (< 200 , $200\text{--}239$, $\geq 240 \text{ mg/dL}$, and unknown); glomerular filtration rate (< 60 , $60\text{--}89$, $\geq 90 \text{ mL/min/1.73 m}^2$, and unknown); smoking status (never, former, current smoker, and unknown); alcoholic drinks (< 1 , $1\text{--}2$, $3\text{--}4$, ≥ 5 days per week, and unknown); aerobic physical activity (sufficient, insufficient, and unknown); previous history of cardiovascular disease, and chronic kidney disease; history of medication use for diabetes mellitus, dyslipidemia, and hypertension; and strain of SARS-CoV-2 (original and delta).

‖ **Model 4:** Adjusted for age (20–39, 40–59, and ≥ 60 years); sex; insurance status (insured and dependent); CCI score (0, 1, and ≥ 2); body mass index (underweight [$< 18.5 \text{ kg/m}^2$], normal [$18.5\text{--}22.9 \text{ kg/m}^2$], overweight [$23.0\text{--}25.0 \text{ kg/m}^2$], obese [$\geq 25.0 \text{ kg/m}^2$], and unknown); blood pressure (systolic blood pressure $< 140 \text{ mmHg}$ and diastolic blood pressure $< 90 \text{ mmHg}$, systolic blood pressure $\geq 140 \text{ mmHg}$ or diastolic blood pressure $\geq 90 \text{ mmHg}$, and unknown); fasting blood glucose (< 100 , $\geq 100 \text{ mg/dL}$, and unknown); serum total cholesterol (< 200 , $200\text{--}239$, $\geq 240 \text{ mg/dL}$, and unknown); glomerular filtration rate (< 60 , $60\text{--}89$, $\geq 90 \text{ mL/min/1.73 m}^2$, and unknown); smoking status (non- and current smoker, and unknown); alcoholic drinks (rarely, sometimes, everyday, and unknown); aerobic physical activity (sufficient, insufficient, and unknown); previous history of cardiovascular disease, and chronic kidney disease; history of medication use for diabetes mellitus, dyslipidemia, and hypertension; and strain of SARS-CoV-2 (original and delta).

Table S16. Time attenuation effect analysis (COVID-19 vs. influenza) of HR (95% CI) for the risk of the **long-term post-acute respiratory sequelae** after SARS-CoV-2 infection in South Korea (main cohort) and Japan (replication cohort)

Time	COVID-19 vs. influenza	
	Main cohort [†]	Replication cohort
long-term post-acute respiratory sequelae		
<3 months	2.28 (2.01-2.59)	8.88 (7.76-10.16)
3–6 months	1.24 (1.04-1.47)	3.81 (3.27-4.44)
≥6 months	1.01 (0.83-1.23)	3.65 (3.30-4.05)

CCI, Charlson comorbidity index; CI, confidence interval; HR, hazard ratio; SARS-CoV-2, severe acute respiratory syndrome coronavirus 2.

The data in bold indicate significant differences ($P < 0.05$).

† Adjusted HR (main): Adjusted for age (20–39, 40–59, and ≥60 years); sex, household income (low income, middle income, and high income); region of residence (urban and rural); CCI score (0, 1, and ≥2); obesity (underweight [$<18.5 \text{ kg/m}^2$], normal [$18.5\text{--}22.9 \text{ kg/m}^2$]; overweight [$23.0\text{--}24.9 \text{ kg/m}^2$], obese [$\geq 25.0 \text{ kg/m}^2$], and unknown); blood pressure (systolic blood pressure $<140 \text{ mmHg}$ and diastolic blood pressure $<90 \text{ mmHg}$, systolic blood pressure $\geq 140 \text{ mmHg}$ or diastolic blood pressure $\geq 90 \text{ mmHg}$, and unknown); fasting blood glucose (<100 , $\geq 100 \text{ mg/dL}$, and unknown); serum total cholesterol (<200 , $200\text{--}239$, $\geq 240 \text{ mg/dL}$, and unknown); glomerular filtration rate (<60 , $60\text{--}89$, $\geq 90 \text{ mL/min/1.73 m}^2$, and unknown); smoking status (never, former, current smoker, and unknown); alcoholic drinks (<1 , $1\text{--}2$, $3\text{--}4$, ≥ 5 days per week, and unknown); aerobic physical activity (sufficient, insufficient, and unknown); previous history of cardiovascular disease, and chronic kidney disease; history of medication use for diabetes mellitus, dyslipidemia, and hypertension; and strain of SARS-CoV-2 (original and delta).

|| Adjusted HR (replication): Adjusted for age (20–39, 40–59, and ≥60 years); sex; insurance status (insured and dependent); CCI score (0, 1, and ≥ 2); body mass index (underweight [$<18.5 \text{ kg/m}^2$], normal [$18.5\text{--}22.9 \text{ kg/m}^2$], overweight [$23.0\text{--}25.0 \text{ kg/m}^2$], obese [$\geq 25.0 \text{ kg/m}^2$], and unknown); blood pressure (systolic blood pressure $<140 \text{ mmHg}$ and diastolic blood pressure $<90 \text{ mmHg}$, systolic blood pressure $\geq 140 \text{ mmHg}$ or diastolic blood pressure $\geq 90 \text{ mmHg}$, and unknown); fasting blood glucose (<100 , $\geq 100 \text{ mg/dL}$, and unknown); serum total cholesterol (<200 , $200\text{--}239$, $\geq 240 \text{ mg/dL}$, and unknown); glomerular filtration rate (<60 , $60\text{--}89$, $\geq 90 \text{ mL/min/1.73 m}^2$, and unknown); smoking status (non- and current smoker, and unknown); alcoholic drinks (rarely, sometimes, everyday, and unknown); aerobic physical activity (sufficient, insufficient, and unknown); previous history of cardiovascular disease, and chronic kidney disease; history of medication use for diabetes mellitus, dyslipidemia, and hypertension; and strain of SARS-CoV-2 (original and delta).

Table S17. Stratification analysis for the **long-term** risk of **post-acute respiratory sequelae** following COVID-19 in the propensity score matching cohorts (COVID-19 vs. general population) of South Korea (main)

	Events, n (%)	COVID-19 vs. general population (n =2,312,748)	
		HR (95% CI)	
		Model 1*	Model 2†
Sex			
Male			
Comparators (general population or patients with influenza)	7849 (0.79)	1.0 (reference)	1.0 (reference)
Patients with COVID-19	75 (1.28)	1.67 (1.60-1.75)	1.64 (1.55-1.73)
Female			
Comparators	8273 (0.90)	1.0 (reference)	1.0 (reference)
Patients with COVID-19	2666 (1.41)	1.61 (1.54-1.68)	1.67 (1.59-1.76)
Age			
20–39 y			
Comparators	5809 (0.83)	1.0 (reference)	1.0 (reference)
Patients with COVID-19	1501 (1.05)	1.27 (1.20-1.35)	1.33 (1.23-1.44)
40–59 years			
Comparators	5243 (0.74)	1.0 (reference)	1.0 (reference)
Patients with COVID-19	1861 (1.28)	1.76 (1.66-1.85)	1.79 (1.68-1.89)
≥60 y			
Comparators	5070 (1.0010)	1.0 (reference)	1.0 (reference)
Patients with COVID-19	1930 (1.82)	1.96 (1.86-2.07)	1.85 (1.74-1.96)
Region of residence			
Rural			
Comparators	8382 (0.81)	1.0 (reference)	1.0 (reference)
Patients with COVID-19	2821 (1.32)	1.68 (1.61-1.75)	1.71 (1.63-1.80)
Urban			
Comparators	7740 (0.88)	1.0 (reference)	1.0 (reference)

Patients with COVID-19	2471 (1.36)	1.60 (1.53-1.67)	1.60 (1.51-1.68)
Income level			
Low (0th–39th percentile)			
Comparators	7444 (0.84)	1.0 (reference)	1.0 (reference)
Patients with COVID-19	2370 (1.30)	1.59 (1.52-1.67)	1.60 (1.51-1.69)
Middle (40th –79th percentile)			
Comparators	5786 (0.85)	1.0 (reference)	1.0 (reference)
Patients with COVID-19	1861 (1.33)	1.60 (1.52-1.69)	1.61 (1.51-1.71)
High (80th–100th percentile)			
Comparators	2892 (0.83)	1.0 (reference)	1.0 (reference)
Patients with COVID-19	1061 (1.48)	1.86 (1.73-1.99)	1.86 (1.72-2.02)
CCI score			
0 score			
Comparators	14,263 (0.79)	1.0 (reference)	1.0 (reference)
Patients with COVID-19	4171 (1.20)	1.59 (1.54-1.65)	1.72 (1.66-1.80)
≥1 score			
Comparators	1859 (1.67)	1.0 (reference)	1.0 (reference)
Patients with COVID-19	1121 (2.33)	1.31 (1.22-1.42)	1.39 (1.27-1.51)
BMI			
<18.5 kg/m²			
Comparators	549 (0.81)	1.0 (reference)	1.0 (reference)
Patients with COVID-19	89 (1.29)	1.66 (1.32-2.07)	1.59 (1.27-2.00)
18.5-23.0 kg/m²			
Comparators	5575 (0.81)	1.0 (reference)	1.0 (reference)
Patients with COVID-19	989 (1.27)	1.58 (1.48-1.69)	1.49 (1.39-1.60)
23.0-25.0 kg/m²			
Comparators	6362 (0.88)	1.0 (reference)	1.0 (reference)
Patients with COVID-19	1548 (1.63)	1.83 (1.74-1.94)	1.72 (1.63-1.82)
≥25.0 kg/m²			
Comparators	2 (0.90)	1.0 (reference)	1.0 (reference)

Patients with COVID-19	1822 (1.14)	1.10 (0.27-4.40)	5.28 (0.41-67.90)
Unknown			
Comparators	3634 (0.83)	1.0 (reference)	1.0 (reference)
Patients with COVID-19	844 (1.55)	1.87 (1.73-2.01)	1.75 (1.62-1.89)
Smoking status			
Non-smoker			
Comparators	10,215 (0.84)	1.0 (reference)	1.0 (reference)
Patients with COVID-19	2333 (1.51)	1.79 (1.71-1.88)	1.70 (1.62-1.78)
Smoker			
Comparators	5905 (0.84)	1.0 (reference)	1.0 (reference)
Patients with COVID-19	1138 (1.42)	1.71 (1.60-1.82)	1.58 (1.48-1.69)
Unknown			
Comparators	2 (0.35)	1.0 (reference)	1.0 (reference)
Patients with COVID-19	1821 (1.14)	3.88 (0.97-15.49)	0.23 (0.00-60.08)
Alcohol consumption			
Non-drinker			
Comparators	10,042 (0.88)	1.0 (reference)	1.0 (reference)
Patients with COVID-19	2140 (1.56)	1.79 (1.70-1.87)	1.67 (1.60-1.75)
Drinker			
Comparators	6077 (0.79)	1.0 (reference)	1.0 (reference)
Patients with COVID-19	1331 (1.37)	1.73 (1.63-1.83)	1.63 (1.53-1.73)
Unknown			
Comparators	3 (0.47)	1.0 (reference)	1.0 (reference)
Patients with COVID-19	1821 (1.14)	2.62 (0.84-8.12)	0.18 (0.00-24.24)
Physical activity			
Insufficient physical activity			
Comparators	8479 (0.88)	1.0 (reference)	1.0 (reference)
Patients with COVID-19	1773 (1.49)	1.69 (1.60-1.77)	1.57 (1.49-1.65)
Sufficient physical activity			
Comparators	7635 (0.80)	1.0 (reference)	1.0 (reference)
Patients with COVID-19	1698 (1.47)	1.85 (1.75-1.95)	1.76 (1.67-1.85)

Unknown

Comparators	8 (0.92)	1.0 (reference)	1.0 (reference)
Patients with COVID-19	1821 (1.13)	1.48 (0.74-2.96)	0.09 (0.00-3.16)

BMI, body mass index; CCI, Charlson comorbidity index; CI, confidence interval; HR, hazard ratio; NA, not available; SARS-CoV-2, severe acute respiratory syndrome coronavirus 2.

The data in bold indicate significant differences ($P < 0.05$).

***Model 1:** Adjusted for age (20–39, 40–59, and ≥ 60 years) and sex.

†Model 2 (main): Adjusted for age (20–39, 40–59, and ≥ 60 years); sex, household income (low income, middle income, and high income); region of residence (urban and rural); CCI score (0, 1, and ≥ 2); obesity (underweight [$< 18.5 \text{ kg/m}^2$], normal [$18.5\text{--}22.9 \text{ kg/m}^2$]; overweight [$23.0\text{--}24.9 \text{ kg/m}^2$], obese [$\geq 25.0 \text{ kg/m}^2$], and unknown); blood pressure (systolic blood pressure $< 140 \text{ mmHg}$ and diastolic blood pressure $< 90 \text{ mmHg}$, systolic blood pressure $\geq 140 \text{ mmHg}$ or diastolic blood pressure $\geq 90 \text{ mmHg}$, and unknown); fasting blood glucose (< 100 , $\geq 100 \text{ mg/dL}$, and unknown); serum total cholesterol (< 200 , $200\text{--}239$, $\geq 240 \text{ mg/dL}$, and unknown); glomerular filtration rate (< 60 , $60\text{--}89$, $\geq 90 \text{ mL/min/1.73 m}^2$, and unknown); smoking status (never, former, current smoker, and unknown); alcoholic drinks (< 1 , $1\text{--}2$, $3\text{--}4$, ≥ 5 days per week, and unknown); aerobic physical activity (sufficient, insufficient, and unknown); previous history of cardiovascular disease, and chronic kidney disease; history of medication use for diabetes mellitus, dyslipidemia, and hypertension; and strain of SARS-CoV-2 (original and delta).

Table S18. Stratification analysis for the **long-term** risk of **post-acute respiratory sequelae** following COVID-19 in the propensity score matching cohorts of (COVID-19 vs. influenza) South Korea (main)

	Events, n (%)	COVID-19 vs. influenza (n =223,000)	
		Model 1*	Model 2†
Sex			
Male			
Comparators (general population or patients with influenza)	515 (1.04)	1.0 (reference)	1.0 (reference)
Patients with COVID-19	581 (1.25)	1.34 (1.19-1.52)	1.40 (1.21-1.61)
Female			
Comparators	566 (0.91)	1.0 (reference)	1.0 (reference)
Patients with COVID-19	919 (1.41)	1.69 (1.52-1.88)	1.82 (1.62-2.05)
Age			
20–39 y			
Comparators	481 (1.20)	1.0 (reference)	1.0 (reference)
Patients with COVID-19	465 (1.09)	0.97 (0.85-1.11)	1.05 (0.89-1.23)
40–59 years			
Comparators	436 (0.82)	1.0 (reference)	1.0 (reference)
Patients with COVID-19	651 (1.28)	1.68 (1.49-1.90)	1.75 (1.53-2.00)
≥60 y			
Comparators	164 (0.91)	1.0 (reference)	1.0 (reference)
Patients with COVID-19	384 (2.11)	2.91 (2.40-3.53)	3.15 (2.56-3.88)
Region of residence			
Rural			
Comparators	448 (0.91)	1.0 (reference)	1.0 (reference)
Patients with COVID-19	624 (1.26)	1.56 (1.38-1.77)	1.73 (1.51-1.99)
Urban			
Comparators	633 (1.02)	1.0 (reference)	1.0 (reference)

Patients with COVID-19	876 (1.41)	1.51 (1.36-1.68)	1.57 (1.39-1.76)
Income level			
Low (0th–39th percentile)			
Comparators	454 (0.99)	1.0 (reference)	1.0 (reference)
Patients with COVID-19	671 (1.37)	1.54 (1.36-1.74)	1.69 (1.47-1.94)
Middle (40th –79th percentile)			
Comparators	432 (0.96)	1.0 (reference)	1.0 (reference)
Patients with COVID-19	596 (1.31)	1.50 (1.32-1.70)	1.59 (1.38-1.83)
High (80th–100th percentile)			
Comparators	195 (0.96)	1.0 (reference)	1.0 (reference)
Patients with COVID-19	233 (1.35)	1.59 (1.31-1.94)	1.62 (1.30-2.02)
CCI score			
0 score			
Comparators	988 (0.96)	1.0 (reference)	1.0 (reference)
Patients with COVID-19	1195 (1.20)	1.39 (1.27-1.51)	1.52 (1.38-1.67)
≥1 score			
Comparators	93 (1.11)	1.0 (reference)	1.0 (reference)
Patients with COVID-19	305 (2.58)	2.34 (1.84-2.98)	2.75 (2.12-3.56)
BMI			
<18.5 kg/m²			
Comparators	35 (0.88)	1.0 (reference)	1.0 (reference)
Patients with COVID-19	26 (1.23)	1.62 (0.96-2.72)	1.72 (1.01-2.92)
18.5-23.0 kg/m²			
Comparators	357 (0.90)	1.0 (reference)	1.0 (reference)
Patients with COVID-19	311 (1.36)	1.67 (1.43-1.96)	1.68 (1.44-1.96)
23.0-25.0 kg/m²			
Comparators	235 (0.96)	1.0 (reference)	1.0 (reference)
Patients with COVID-19	233 (1.63)	1.78 (1.48-2.13)	1.80 (1.50-2.17)
≥25.0 kg/m²			
Comparators	454 (1.04)	1.0 (reference)	1.0 (reference)

Patients with COVID-19	384 (1.53)	1.56 (1.35-1.79)	1.52 (1.32-1.75)
Unknown			
Comparators	0 (0.00)	1.0 (reference)	1.0 (reference)
Patients with COVID-19	546 (1.16)	N/A	N/A
Smoking status			
Non-smoker			
Comparators	717 (0.96)	1.0 (reference)	1.0 (reference)
Patients with COVID-19	681 (1.50)	1.67 (1.50-1.86)	1.66 (1.49-1.85)
Smoker			
Comparators	364 (1.00)	1.0 (reference)	1.0 (reference)
Patients with COVID-19	274 (1.45)	1.59 (1.35-1.86)	1.59 (1.35-1.87)
Unknown			
Comparators	0 (0.00)	1.0 (reference)	1.0 (reference)
Patients with COVID-19	545 (1.16)	N/A	N/A
Alcohol consumption			
Non-drinker			
Comparators	631 (0.93)	1.0 (reference)	1.0 (reference)
Patients with COVID-19	609 (1.58)	1.87 (1.66-2.09)	1.84 (1.64-2.07)
Drinker			
Comparators	450 (1.04)	1.0 (reference)	1.0 (reference)
Patients with COVID-19	346 (1.33)	1.36 (1.18-1.57)	1.37 (1.19-1.58)
Unknown			
Comparators	0 (0.00)	1.0 (reference)	1.0 (reference)
Patients with COVID-19	545 (1.16)	N/A	N/A
Physical activity			
Insufficient physical activity			
Comparators	554 (0.95)	1.0 (reference)	1.0 (reference)
Patients with COVID-19	507 (1.50)	1.69 (1.49-1.91)	1.67 (1.48-1.89)
Sufficient physical activity			
Comparators	526 (0.99)	1.0 (reference)	1.0 (reference)
Patients with COVID-19	448 (1.47)	1.61 (1.41-1.83)	1.60 (1.41-1.82)

Unknown

Comparators	1 (2.44)	1.0 (reference)	1.0 (reference)
Patients with COVID-19	545 (1.16)	0.47 (0.07-3.37)	N/A

BMI, body mass index; CCI, Charlson comorbidity index; CI, confidence interval; HR, hazard ratio; NA, not available; SARS-CoV-2, severe acute respiratory syndrome coronavirus 2.

The data in bold indicate significant differences ($P < 0.05$).

***Model 1:** Adjusted for age (20–39, 40–59, and ≥ 60 years) and sex.

†Model 2 (main): Adjusted for age (20–39, 40–59, and ≥ 60 years); sex, household income (low income, middle income, and high income); region of residence (urban and rural); CCI score (0, 1, and ≥ 2); obesity (underweight [$< 18.5 \text{ kg/m}^2$], normal [$18.5\text{--}22.9 \text{ kg/m}^2$]; overweight [$23.0\text{--}24.9 \text{ kg/m}^2$], obese [$\geq 25.0 \text{ kg/m}^2$], and unknown); blood pressure (systolic blood pressure $< 140 \text{ mmHg}$ and diastolic blood pressure $< 90 \text{ mmHg}$, systolic blood pressure $\geq 140 \text{ mmHg}$ or diastolic blood pressure $\geq 90 \text{ mmHg}$, and unknown); fasting blood glucose (< 100 , $\geq 100 \text{ mg/dL}$, and unknown); serum total cholesterol (< 200 , $200\text{--}239$, $\geq 240 \text{ mg/dL}$, and unknown); glomerular filtration rate (< 60 , $60\text{--}89$, $\geq 90 \text{ mL/min/1.73 m}^2$, and unknown); smoking status (never, former, current smoker, and unknown); alcoholic drinks (< 1 , $1\text{--}2$, $3\text{--}4$, ≥ 5 days per week, and unknown); aerobic physical activity (sufficient, insufficient, and unknown); previous history of cardiovascular disease, and chronic kidney disease; history of medication use for diabetes mellitus, dyslipidemia, and hypertension; and strain of SARS-CoV-2 (original and delta).

Table S19. Stratification analysis for the **short-term risk of acute respiratory complication** following COVID-19 in the propensity score matching cohorts (COVID-19 vs. general population) of South Korea (main)

	Events, n (%)	COVID-19 vs. general population (n =2,312,748)	
		HR (95% CI)	
		Model 1*	Model 2†
Sex			
Male			
Comparators (general population or patients with influenza)	167 (0.017)	1.0 (reference)	1.0 (reference)
Patients with COVID-19	345 (0.17)	10.08 (8.38-12.13)	8.72 (7.11-10.68)
Female			
Comparators	144 (0.016)	1.0 (reference)	1.0 (reference)
Patients with COVID-19	273 (0.14)	9.26 (7.56-11.33)	6.67 (5.28-8.43)
Age			
20–39 y			
Comparators	61 (0.0087)	1.0 (reference)	1.0 (reference)
Patients with COVID-19	65 (0.045)	5.23 (3.69-7.41)	5.85 (3.85-8.91)
40–59 years			
Comparators	83 (0.012)	1.0 (reference)	1.0 (reference)
Patients with COVID-19	157 (0.11)	9.26 (7.10-12.09)	7.14 (5.31-9.60)
≥60 y			
Comparators	167 (0.033)	1.0 (reference)	1.0 (reference)
Patients with COVID-19	396 (0.37)	11.55 (9.64-13.84)	8.69 (7.10-10.62)
Region of residence			
Rural			
Comparators	185 (0.018)	1.0 (reference)	1.0 (reference)
Patients with COVID-19	271 (0.13)	7.15 (5.93-8.62)	5.61 (4.52-6.96)
Urban			
Comparators	126 (0.014)	1.0 (reference)	1.0 (reference)

Patients with COVID-19	347 (0.19)	13.46 (10.97-16.50)	10.81 (8.64-13.51)
Income level			
Low (0th–39th percentile)			
Comparators	142 (0.016)	1.0 (reference)	1.0 (reference)
Patients with COVID-19	241 (0.13)	8.28 (6.73-10.19)	7.06 (5.58-8.94)
Middle (40th –79th percentile)			
Comparators	99 (0.015)	1.0 (reference)	1.0 (reference)
Patients with COVID-19	229 (0.16)	11.26 (8.90-14.26)	8.64 (6.65-11.22)
High (80th–100th percentile)			
Comparators	70 (0.020)	1.0 (reference)	1.0 (reference)
Patients with COVID-19	148 (0.21)	10.36 (7.80-13.77)	7.72 (5.61-10.61)
CCI score			
0 score			
Comparators	197 (0.011)	1.0 (reference)	1.0 (reference)
Patients with COVID-19	338 (0.10)	9.36 (7.85-11.16)	8.83 (7.25-10.76)
≥1 scores			
Comparators	114 (0.10)	1.0 (reference)	1.0 (reference)
Patients with COVID-19	280 (0.58)	5.60 (4.50-6.96)	5.85 (4.60-7.44)
BMI			
<18.5 kg/m²			
Comparators	13 (0.019)	1.0 (reference)	1.0 (reference)
Patients with COVID-19	11 (0.16)	8.37 (3.73-18.78)	8.08 (3.57-18.32)
18.5-23.0 kg/m²			
Comparators	81 (0.012)	1.0 (reference)	1.0 (reference)
Patients with COVID-19	70 (0.090)	7.22 (5.24-9.95)	6.15 (4.43-8.53)
23.0-25.0 kg/m²			
Comparators	71 (0.016)	1.0 (reference)	1.0 (reference)
Patients with COVID-19	85 (0.16)	8.94 (6.52-12.25)	7.92 (5.75-10.91)
≥25.0 kg/m²			
Comparators	146 (0.020)	1.0 (reference)	1.0 (reference)
Patients with COVID-19	201 (0.21)	9.51 (7.68-11.78)	8.25 (6.64-10.26)

Unknown			
Comparators	0 (0.00)	1.0 (reference)	1.0 (reference)
Patients with COVID-19	251 (0.16)	N/A	N/A
Smoking status			
Non-smoker			
Comparators	189 (0.016)	1.0 (reference)	1.0 (reference)
Patients with COVID-19	239 (0.16)	8.95 (7.40-10.84)	7.72 (6.35-9.38)
Smoker			
Comparators	122 (0.017)	1.0 (reference)	1.0 (reference)
Patients with COVID-19	128 (0.16)	8.50 (6.63-10.90)	7.56 (5.88-9.73)
Unknown			
Comparators	0 (0.00)	1.0 (reference)	1.0 (reference)
Patients with COVID-19	251 (0.16)	N/A	N/A
Alcohol consumption			
Non-drinker			
Comparators	208 (0.018)	1.0 (reference)	1.0 (reference)
Patients with COVID-19	243 (0.18)	8.88 (7.38-10.69)	7.75 (6.41-9.36)
Drinker			
Comparators	103 (0.013)	1.0 (reference)	1.0 (reference)
Patients with COVID-19	124 (0.13)	8.90 (6.85-11.56)	7.53 (5.77-9.83)
Unknown			
Comparators	0 (0.00)	1.0 (reference)	1.0 (reference)
Patients with COVID-19	251 (0.16)	N/A	N/A
Physical activity			
Insufficient physical activity			
Comparators	154 (0.016)	1.0 (reference)	1.0 (reference)
Patients with COVID-19	212 (0.18)	9.84 (7.99-12.12)	8.46 (6.84-10.46)
Sufficient physical activity			
Comparators	157 (0.016)	1.0 (reference)	1.0 (reference)
Patients with COVID-19	155 (0.13)	7.68 (6.15-9.59)	6.82 (5.45-8.55)
Unknown			
Comparators	0 (0.00)	1.0 (reference)	1.0 (reference)

Patients with COVID-19

251 (0.16)

N/A

N/A

BMI, body mass index; CCI, Charlson comorbidity index; CI, confidence interval; HR, hazard ratio; NA, not available; SARS-CoV-2, severe acute respiratory syndrome coronavirus 2.

The data in bold indicate significant differences ($P < 0.05$).

***Model 1:** Adjusted for age (20–39, 40–59, and ≥ 60 years) and sex.

†Model 2 (main): Adjusted for age (20–39, 40–59, and ≥ 60 years); sex, household income (low income, middle income, and high income); region of residence (urban and rural); CCI score (0, 1, and ≥ 2); obesity (underweight [$< 18.5 \text{ kg/m}^2$], normal [$18.5\text{--}22.9 \text{ kg/m}^2$]; overweight [$23.0\text{--}24.9 \text{ kg/m}^2$], obese [$\geq 25.0 \text{ kg/m}^2$], and unknown); blood pressure (systolic blood pressure $< 140 \text{ mmHg}$ and diastolic blood pressure $< 90 \text{ mmHg}$, systolic blood pressure $\geq 140 \text{ mmHg}$ or diastolic blood pressure $\geq 90 \text{ mmHg}$, and unknown); fasting blood glucose (< 100 , $\geq 100 \text{ mg/dL}$, and unknown); serum total cholesterol (< 200 , $200\text{--}239$, $\geq 240 \text{ mg/dL}$, and unknown); glomerular filtration rate (< 60 , $60\text{--}89$, $\geq 90 \text{ mL/min/1.73 m}^2$, and unknown); smoking status (never, former, current smoker, and unknown); alcoholic drinks (< 1 , $1\text{--}2$, $3\text{--}4$, ≥ 5 days per week, and unknown); aerobic physical activity (sufficient, insufficient, and unknown); previous history of cardiovascular disease, and chronic kidney disease; history of medication use for diabetes mellitus, dyslipidemia, and hypertension; and strain of SARS-CoV-2 (original and delta).

Table S20. Stratification analysis for the **short-term risk of acute respiratory complication** following COVID-19 in the propensity score matching cohorts (COVID-19 vs. influenza) of South Korea (main)

	Events, n (%)	COVID-19 vs. influenza (n =223,000)	
		Model 1*	HR (95% CI) Model 2†
Sex			
Male			
Comparators (general population or patients with influenza)	23 (0.046)	1.0 (reference)	1.0 (reference)
Patients with COVID-19	51 (0.11)	27.75 (6.76-114.00)	25.32 (6.04-106.13)
Female			
Comparators	22 (0.036)	1.0 (reference)	1.0 (reference)
Patients with COVID-19	64 (0.098)	10.38 (4.50-23.97)	8.23 (3.46-19.58)
Age			
20–39 y			
Comparators	18 (0.045)	1.0 (reference)	1.0 (reference)
Patients with COVID-19	15 (0.035)	6.94 (1.59-30.34)	10.78 (2.35-49.52)
40–59 years			
Comparators	12 (0.023)	1.0 (reference)	1.0 (reference)
Patients with COVID-19	43 (0.085)	46.16 (6.36-335.06)	42.17 (5.74-309.82)
≥60 y			
Comparators	15 (0.083)	1.0 (reference)	1.0 (reference)
Patients with COVID-19	57 (0.31)	11.53 (4.62-28.76)	7.80 (3.00-20.28)
Region of residence			
Rural			
Comparators	29 (0.059)	1.0 (reference)	1.0 (reference)
Patients with COVID-19	39 (0.079)	5.57 (2.49-12.46)	5.05 (2.18-11.71)
Urban			
Comparators	16 (0.026)	1.0 (reference)	1.0 (reference)

Patients with COVID-19	76 (0.12)	79.28 (11.03-570.15)	65.32 (8.99-474.56)
Income level			
Low (0th–39th percentile)			
Comparators	16 (0.035)	1.0 (reference)	1.0 (reference)
Patients with COVID-19	36 (0.074)	17.62 (4.25-73.15)	17.31 (4.07-73.64)
Middle (40th –79th percentile)			
Comparators	21 (0.047)	1.0 (reference)	1.0 (reference)
Patients with COVID-19	55 (0.12)	9.33 (4.02-21.67)	8.26 (3.46-19.72)
High (80th–100th percentile)			
Comparators	8 (0.039)	1.0 (reference)	1.0 (reference)
Patients with COVID-19	24 (0.14)	N/A	N/A
CCI score			
0 score			
Comparators	37 (0.036)	1.0 (reference)	1.0 (reference)
Patients with COVID-19	61 (0.061)	8.25 (3.95-17.25)	6.79 (3.12-14.75)
≥1 scores			
Comparators	8 (0.10)	1.0 (reference)	1.0 (reference)
Patients with COVID-19	54 (0.46)	N/A	N/A
BMI			
<18.5 kg/m²			
Comparators	6 (0.15)	1.0 (reference)	1.0 (reference)
Patients with COVID-19	3 (0.14)	2.84 (0.47-17.04)	2.70 (0.38-19.31)
18.5-23.0 kg/m²			
Comparators	15 (0.038)	1.0 (reference)	1.0 (reference)
Patients with COVID-19	16 (0.070)	26.54 (3.52-200.29)	21.48 (2.83-163.25)
23.0-25.0 kg/m²			
Comparators	12 (0.049)	1.0 (reference)	1.0 (reference)
Patients with COVID-19	13 (0.091)	11.02 (2.48-48.95)	11.51 (2.57-51.54)
≥25.0 kg/m²			
Comparators	12 (0.028)	1.0 (reference)	1.0 (reference)

Patients with COVID-19	37 (0.147)	19.63 (6.05-63.68)	16.15 (4.95-52.73)
Unknown			
Comparators	0 (0.00)	1.0 (reference)	1.0 (reference)
Patients with COVID-19	46 (0.10)	N/A	N/A
			
Smoking status			
Non-smoker			
Comparators	25 (0.033)	1.0 (reference)	1.0 (reference)
Patients with COVID-19	50 (0.11)	13.10 (5.61-30.56)	11.09 (4.73-25.99)
Smoker			
Comparators	20 (0.055)	1.0 (reference)	1.0 (reference)
Patients with COVID-19	19 (0.10)	16.91 (3.94-72.67)	16.11 (3.73-69.58)
Unknown			
Comparators	0 (0.00)	1.0 (reference)	1.0 (reference)
Patients with COVID-19	46 (0.10)	N/A	N/A
			
Alcohol consumption			
Non-drinker			
Comparators	28 (0.041)	1.0 (reference)	1.0 (reference)
Patients with COVID-19	46 (0.12)	15.83 (6.28-39.88)	13.73 (5.43-34.71)
Drinker			
Comparators	17 (0.039)	1.0 (reference)	1.0 (reference)
Patients with COVID-19	23 (0.089)	11.81 (3.54-39.37)	10.20 (3.03-34.35)
Unknown			
Comparators	0 (0.00)	1.0 (reference)	1.0 (reference)
Patients with COVID-19	46 (0.10)	N/A	N/A
			
Physical activity			
Insufficient physical activity			
Comparators	23 (0.040)	1.0 (reference)	1.0 (reference)
Patients with COVID-19	36 (0.11)	18.51 (5.70-60.16)	15.87 (4.87-51.79)
Sufficient physical activity			
Comparators	22 (0.041)	1.0 (reference)	1.0 (reference)
Patients with COVID-19	33 (0.11)	11.29 (4.41-28.94)	9.81 (3.80-25.28)

Unknown

Comparators	0 (0.00)	1.0 (reference)	1.0 (reference)
Patients with COVID-19	46 (0.10)	N/A	N/A

BMI, body mass index; CCI, Charlson comorbidity index; CI, confidence interval; HR, hazard ratio; NA, not available; SARS-CoV-2, severe acute respiratory syndrome coronavirus 2.

The data in bold indicate significant differences ($P < 0.05$).

***Model 1:** Adjusted for age (20–39, 40–59, and ≥ 60 years) and sex.

†Model 2 (main): Adjusted for age (20–39, 40–59, and ≥ 60 years); sex, household income (low income, middle income, and high income); region of residence (urban and rural); CCI score (0, 1, and ≥ 2); obesity (underweight [$< 18.5 \text{ kg/m}^2$], normal [$18.5\text{--}22.9 \text{ kg/m}^2$]; overweight [$23.0\text{--}24.9 \text{ kg/m}^2$], obese [$\geq 25.0 \text{ kg/m}^2$], and unknown); blood pressure (systolic blood pressure $< 140 \text{ mmHg}$ and diastolic blood pressure $< 90 \text{ mmHg}$, systolic blood pressure $\geq 140 \text{ mmHg}$ or diastolic blood pressure $\geq 90 \text{ mmHg}$, and unknown); fasting blood glucose (< 100 , $\geq 100 \text{ mg/dL}$, and unknown); serum total cholesterol (< 200 , $200\text{--}239$, $\geq 240 \text{ mg/dL}$, and unknown); glomerular filtration rate (< 60 , $60\text{--}89$, $\geq 90 \text{ mL/min/1.73 m}^2$, and unknown); smoking status (never, former, current smoker, and unknown); alcoholic drinks (< 1 , $1\text{--}2$, $3\text{--}4$, ≥ 5 days per week, and unknown); aerobic physical activity (sufficient, insufficient, and unknown); previous history of cardiovascular disease, and chronic kidney disease; history of medication use for diabetes mellitus, dyslipidemia, and hypertension; and strain of SARS-CoV-2 (original and delta).

Table S21. Stratification analysis for the **long-term** risk of **post-acute respiratory sequelae** following COVID-19 in the propensity score matching cohorts (COVID-19 vs. general population) of Japan (replication)

	COVID-19 vs. general population (n=3,115,606)		
	Events, n (%)	HR (95% CI)	
		Model 3*	Model 4
Sex			
Male			
Comparators (general population or patients with influenza)	21,197 (1.46)	1.0 (reference)	1.0 (reference)
Patients with COVID-19	24,947 (5.035)	3.56 (3.49-3.62)	3.34 (3.28-3.40)
Female			
Comparators	14,103 (1.62)	1.0 (reference)	1.0 (reference)
Patients with COVID-19	16,127 (5.35)	3.41 (3.34-3.49)	3.29 (3.21-3.36)
Age			
20–39 y			
Comparators	12,546 (1.43)	1.0 (reference)	1.0 (reference)
Patients with COVID-19	14,595 (4.83)	3.47 (3.39-3.56)	3.38 (3.30-3.46)
40–59 years			
Comparators	18,329 (1.53)	1.0 (reference)	1.0 (reference)
Patients with COVID-19	20,739 (5.042)	3.41 (3.35-3.48)	3.22 (3.15-3.28)
≥60 y			
Comparators	4425 (1.84)	1.0 (reference)	1.0 (reference)
Patients with COVID-19	5740 (6.89)	3.93 (3.77-4.08)	3.56 (3.42-3.70)
CCI score			
0 score			
Comparators	34,452 (1.50)	1.0 (reference)	1.0 (reference)

Patients with COVID-19	36,256 (4.75)	3.28 (3.23-3.33)	3.28 (3.23-3.33)
≥1 scores			
Comparators	848 (3.43)	1.0 (reference)	1.0 (reference)
Patients with COVID-19	4818 (14.58)	4.15 (3.86-4.47)	4.09 (3.80-4.40)
BMI			
<18.5 kg/m²			
Comparators	24,324 (1.43)	1.0 (reference)	1.0 (reference)
Patients with COVID-19	28,166 (4.87)	3.53 (3.47-3.59)	3.36 (3.31-3.42)
18.5-23.0 kg/m²			
Comparators	8232 (1.70)	1.0 (reference)	1.0 (reference)
Patients with COVID-19	9483 (5.58)	3.39 (3.29-3.49)	3.19 (3.10-3.29)
23.0-25.0 kg/m²			
Comparators	2681 (2.10)	1.0 (reference)	1.0 (reference)
Patients with COVID-19	3368 (7.13)	3.45 (3.28-3.63)	3.21 (3.05-3.38)
≥25.0 kg/m²			
Comparators	63 (1.91)	1.0 (reference)	1.0 (reference)
Patients with COVID-19	57 (4.65)	2.57 (1.80-3.68)	2.50 (1.73-3.59)
Smoking status			
Non-smoker			
Comparators	34,154 (1.53)	1.0 (reference)	1.0 (reference)
Patients with COVID-19	39,793 (5.18)	3.50 (3.45-3.55)	3.31 (3.26-3.36)
Smoker			
Comparators	1146 (1.32)	1.0 (reference)	1.0 (reference)
Patients with COVID-19	1281 (4.34)	3.50 (3.23-3.79)	3.37 (3.11-3.66)
Alcohol consumption			
Non-drinker			
Comparators	18,088 (1.47)	1.0 (reference)	1.0 (reference)

Patients with COVID-19	20,797 (4.88)	3.44 (3.37-3.51)	3.27 (3.20-3.34)
Drinker			
Comparators	14,746 (1.60)	1.0 (reference)	1.0 (reference)
Patients with COVID-19	17,402 (5.58)	3.58 (3.50-3.66)	3.38 (3.31-3.46)
Unknown			
Comparators	2466 (1.45)	1.0 (reference)	1.0 (reference)
Patients with COVID-19	2875 (4.87)	3.49 (3.31-3.69)	3.36 (3.18-3.55)
Physical activity			
Insufficient physical activity			
Comparators	7205 (1.45)	1.0 (reference)	1.0 (reference)
Patients with COVID-19	7543 (4.68)	3.36 (3.25-3.47)	3.20 (3.09-3.30)
Sufficient physical activity			
Comparators	25,121 (1.56)	1.0 (reference)	1.0 (reference)
Patients with COVID-19	30,100 (5.35)	3.52 (3.47-3.58)	3.34 (3.28-3.40)
Unknown			
Comparators	2974 (1.38)	1.0 (reference)	1.0 (reference)
Patients with COVID-19	3431 (4.66)	3.55 (3.38-3.73)	3.39 (3.23-3.56)
Strain of SARS-CoV-2			
Original			
Comparators	27,023 (2.75)	1.0 (reference)	1.0 (reference)
Patients with COVID-19	29,447 (8.78)	3.31 (3.26-3.36)	3.13 (3.08-3.18)
Delta			
Comparators	8277 (0.62)	1.0 (reference)	1.0 (reference)
Patients with COVID-19	11,627 (2.52)	4.13 (4.01-4.24)	3.94 (3.83-4.05)

BMI, body mass index; CCI, Charlson comorbidity index; CI, confidence interval; HR, hazard ratio; NA, not available; SARS-CoV-2, severe acute respiratory syndrome coronavirus 2.

The data in bold indicate significant differences ($P < 0.05$).

***Model 3:** Adjusted for age (20–39, 40–59, and ≥ 60 years) and sex.

|| **Model 4 (replication):** Adjusted for age (20–39, 40–59, and ≥ 60 years); sex; insurance status (insured and dependent); CCI score (0, 1, and ≥ 2); body mass index (underweight [< 18.5 kg/m²], normal [18.5–22.9 kg/m²], overweight [23.0–25.0 kg/m²], obese [≥ 25.0 kg/m²], and unknown); blood pressure (systolic blood pressure < 140 mmHg and diastolic blood pressure < 90 mmHg, systolic blood pressure ≥ 140 mmHg or diastolic blood pressure ≥ 90 mmHg, and unknown); fasting blood glucose (< 100 , ≥ 100 mg/dL, and unknown); serum total cholesterol (< 200 , 200–239, ≥ 240 mg/dL, and unknown); glomerular filtration rate (< 60 , 60–89, ≥ 90 mL/min/1.73 m², and unknown); smoking status (non- and current smoker, and unknown); alcoholic drinks (rarely, sometimes, everyday, and unknown); aerobic physical activity (sufficient, insufficient, and unknown); previous history of cardiovascular disease, and chronic kidney disease; history of medication use for diabetes mellitus, dyslipidemia, and hypertension; and strain of SARS-CoV-2 (original and delta).

Table S22. Stratification analysis for the **long-term** risk of **post-acute respiratory sequelae** following COVID-19 in the propensity score matching cohorts (COVID-19 vs. influenza) of Japan (replication)

	Events, n (%)	COVID-19 vs. influenza (n =169,924)	
		Model 3*	Model 4 ^{II}
Sex			
Male			
Comparators (general population or patients with influenza)	526 (1.00)	1.0 (reference)	1.0 (reference)
Patients with COVID-19	2938 (5.54)	5.76 (5.25-6.32)	5.41 (4.93-5.94)
Female			
Comparators	379 (1.18)	1.0 (reference)	1.0 (reference)
Patients with COVID-19	1819 (5.69)	5.00 (4.48-5.59)	4.83 (4.33-5.40)
Age			
20–39 y			
Comparators	373 (1.25)	1.0 (reference)	1.0 (reference)
Patients with COVID-19	1556 (5.19)	4.29 (3.83-4.80)	4.19 (3.75-4.70)
40–59 years			
Comparators	429 (0.92)	1.0 (reference)	1.0 (reference)
Patients with COVID-19	2567 (5.51)	6.23 (5.63-6.90)	5.92 (5.34-6.56)
≥60 y			
Comparators	103 (1.25)	1.0 (reference)	1.0 (reference)
Patients with COVID-19	634 (7.51)	6.30 (5.11-7.75)	5.64 (4.57-6.96)
CCI score			
0 score			
Comparators	877 (1.05)	1.0 (reference)	1.0 (reference)
Patients with COVID-19	4196 (5.15)	5.11 (4.75-5.49)	5.11 (4.75-5.50)
≥1 scores			

Comparators	28 (2.43)	1.0 (reference)	1.0 (reference)
Patients with COVID-19	561 (16.01)	6.61 (4.52-9.66)	6.63 (4.53-9.71)
BMI			
<18.5 kg/m²			
Comparators	636 (1.03)	1.0 (reference)	1.0 (reference)
Patients with COVID-19	3290 (5.35)	5.39 (4.95-5.86)	5.16 (4.74-5.62)
18.5-23.0 kg/m²			
Comparators	195 (1.08)	1.0 (reference)	1.0 (reference)
Patients with COVID-19	1075 (5.88)	5.67 (4.86-6.60)	5.36 (4.59-6.24)
23.0-25.0 kg/m²			
Comparators	73 (1.47)	1.0 (reference)	1.0 (reference)
Patients with COVID-19	387 (7.58)	5.26 (4.10-6.76)	4.84 (3.76-6.23)
≥25.0 kg/m²			
Comparators	1 (0.85)	1.0 (reference)	1.0 (reference)
Patients with COVID-19	5 (3.88)	4.87 (0.56-42.31)	3.58 (0.32-40.57)
Smoking status			
Non-smoker			
Comparators	868 (1.06)	1.0 (reference)	1.0 (reference)
Patients with COVID-19	4604 (5.62)	5.49 (5.11-5.90)	5.20 (4.83-5.59)
Smoker			
Comparators	37 (1.23)	1.0 (reference)	1.0 (reference)
Patients with COVID-19	153 (4.96)	4.36 (3.04-6.25)	4.29 (2.98-6.16)
Alcohol consumption			
Non-drinker			
Comparators	443 (0.99)	1.0 (reference)	1.0 (reference)
Patients with COVID-19	2411 (5.30)	5.58 (5.04-6.17)	5.28 (4.77-5.84)
Drinker			
Comparators	397 (1.18)	1.0 (reference)	1.0 (reference)
Patients with COVID-19	2012 (6.03)	5.28 (4.74-5.88)	5.01 (4.50-5.59)
Unknown			

Comparators	65 (1.03)	1.0 (reference)	1.0 (reference)
Patients with COVID-19	334 (5.44)	5.54 (4.25-7.23)	5.53 (4.23-7.22)
Physical activity			
Insufficient physical activity			
Comparators	169 (0.97)	1.0 (reference)	1.0 (reference)
Patients with COVID-19	883 (5.10)	5.50 (4.67-6.49)	5.22 (4.43-6.16)
Sufficient physical activity			
Comparators	651 (1.09)	1.0 (reference)	1.0 (reference)
Patients with COVID-19	3477 (5.80)	5.48 (5.04-5.96)	5.20 (4.78-5.65)
Unknown			
Comparators	85 (1.10)	1.0 (reference)	1.0 (reference)
Patients with COVID-19	397 (5.15)	5.01 (3.96-6.33)	4.88 (3.86-6.18)
Strain of SARS-CoV-2			
Original			
Comparators	641 (1.80)	1.0 (reference)	1.0 (reference)
Patients with COVID-19	3269 (9.19)	5.33 (4.89-5.80)	5.05 (4.64-5.50)
Delta			
Comparators	264 (0.53)	1.0 (reference)	1.0 (reference)
Patients with COVID-19	1488 (3.01)	5.73 (5.02-6.53)	5.47 (4.79-6.23)

BMI, body mass index; CCI, Charlson comorbidity index; CI, confidence interval; HR, hazard ratio; NA, not available; SARS-CoV-2, severe acute respiratory syndrome coronavirus 2.

The data in bold indicate significant differences ($P < 0.05$).

***Model 3:** Adjusted for age (20–39, 40–59, and ≥ 60 years) and sex.

|| **Model 4 (replication):** Adjusted for age (20–39, 40–59, and ≥ 60 years); sex; insurance status (insured and dependent); CCI score (0, 1, and ≥ 2); body mass index (underweight [$< 18.5 \text{ kg/m}^2$], normal [$18.5\text{--}22.9 \text{ kg/m}^2$], overweight [$23.0\text{--}25.0 \text{ kg/m}^2$], obese [$\geq 25.0 \text{ kg/m}^2$], and unknown); blood pressure (systolic blood pressure $< 140 \text{ mmHg}$ and diastolic blood pressure $< 90 \text{ mmHg}$, systolic blood pressure $\geq 140 \text{ mmHg}$ or diastolic blood pressure $\geq 90 \text{ mmHg}$, and unknown); fasting blood glucose (< 100 , $\geq 100 \text{ mg/dL}$, and unknown); serum total cholesterol (< 200 , $200\text{--}239$, $\geq 240 \text{ mg/dL}$, and unknown); glomerular filtration rate (< 60 , $60\text{--}89$, $\geq 90 \text{ mL/min/1.73 m}^2$, and unknown); smoking status (non- and current smoker, and unknown); alcoholic drinks (rarely, sometimes, everyday, and unknown); aerobic physical activity (sufficient, insufficient, and unknown); previous history of cardiovascular disease, and chronic kidney disease; history of medication use for diabetes mellitus, dyslipidemia, and hypertension; and strain of SARS-CoV-2 (original and delta).

Table S23. Stratification analysis for the **short-term risk of acute respiratory complication** following COVID-19 in the propensity score matching cohorts (COVID-19 vs. general population) of Japan (replication)

	Events, n (%)	COVID-19 vs. general population (n =3,115,606)	
		HR (95% CI)	
		Model 3*	Model 4 ^{II}
Sex			
Male			
Comparators (general population or patients with influenza)	954 (0.066)	1.0 (reference)	1.0 (reference)
Patients with COVID-19	1569 (0.32)	4.84 (4.46-5.24)	4.36 (4.01-4.73)
Female			
Comparators	514 (0.059)	1.0 (reference)	1.0 (reference)
Patients with COVID-19	735 (0.24)	4.16 (3.72-4.66)	3.82 (3.40-4.28)
Age			
20–39 y			
Comparators	466 (0.053)	1.0 (reference)	1.0 (reference)
Patients with COVID-19	721 (0.24)	4.51 (4.01-5.07)	4.39 (3.91-4.94)
40–59 years			
Comparators	817 (0.068)	1.0 (reference)	1.0 (reference)
Patients with COVID-19	1176 (0.29)	4.23 (3.87-4.63)	3.77 (3.44-4.13)
≥60 y			
Comparators	185 (0.077)	1.0 (reference)	1.0 (reference)
Patients with COVID-19	407 (0.49)	6.43 (5.41-7.65)	5.33 (4.45-6.37)
CCI score			
0 score			
Comparators	1409 (0.061)	1.0 (reference)	1.0 (reference)
Patients with COVID-19	1923 (0.25)	4.16 (3.88-4.46)	4.14 (3.86-4.43)

≥1 scores			
Comparators	59 (0.24)	1.0 (reference)	1.0 (reference)
Patients with COVID-19	381 (1.15)	4.32 (3.28-5.68)	4.40 (3.34-5.79)
BMI			
<18.5 kg/m²			
Comparators	1097 (0.064)	1.0 (reference)	1.0 (reference)
Patients with COVID-19	1636 (0.28)	4.44 (4.12-4.80)	4.05 (3.75-4.38)
18.5-23.0 kg/m²			
Comparators	288 (0.059)	1.0 (reference)	1.0 (reference)
Patients with COVID-19	508 (0.30)	5.04 (4.37-5.83)	4.51 (3.89-5.22)
23.0-25.0 kg/m²			
Comparators	82 (0.064)	1.0 (reference)	1.0 (reference)
Patients with COVID-19	154 (0.33)	4.90 (3.75-6.41)	4.37 (3.32-5.74)
≥25.0 kg/m²			
Comparators	1 (0.030)	1.0 (reference)	1.0 (reference)
Patients with COVID-19	6 (0.49)	16.98 (2.04-141.11)	18.83 (2.12-167.43)
Smoking status			
Non-smoker			
Comparators	1396 (0.063)	1.0 (reference)	1.0 (reference)
Patients with COVID-19	2211 (0.29)	4.63 (4.33-4.95)	4.19 (3.91-4.49)
Smoker			
Comparators	72 (0.083)	1.0 (reference)	1.0 (reference)
Patients with COVID-19	93 (0.32)	4.05 (2.97-5.51)	3.84 (2.82-5.24)
Alcohol consumption			
Non-drinker			
Comparators	752 (0.061)	1.0 (reference)	1.0 (reference)
Patients with COVID-19	1193 (0.28)	4.64 (4.24-5.09)	4.23 (3.86-4.65)
Drinker			
Comparators	590 (0.064)	1.0 (reference)	1.0 (reference)

Patients with COVID-19	935 (0.30)	4.65 (4.19-5.15)	4.16 (3.75-4.62)
Unknown			
Comparators	126 (0.074)	1.0 (reference)	1.0 (reference)
Patients with COVID-19	176 (0.30)	4.11 (3.27-5.17)	3.91 (3.10-4.92)
Physical activity			
Insufficient physical activity			
Comparators	339 (0.068)	1.0 (reference)	1.0 (reference)
Patients with COVID-19	471 (0.29)	4.36 (3.79-5.02)	3.91 (3.39-4.50)
Sufficient physical activity			
Comparators	967 (0.060)	1.0 (reference)	1.0 (reference)
Patients with COVID-19	1610 (0.29)	4.76 (4.39-5.15)	4.30 (3.97-4.66)
Unknown			
Comparators	162 (0.075)	1.0 (reference)	1.0 (reference)
Patients with COVID-19	223 (0.30)	4.18 (3.42-5.12)	3.97 (3.23-4.87)
Strain of SARS-CoV-2			
Original			
Comparators	612 (0.062)	1.0 (reference)	1.0 (reference)
Patients with COVID-19	951 (0.28)	4.58 (4.14-5.07)	4.01 (3.61-4.45)
Delta			
Comparators	856 (0.064)	1.0 (reference)	1.0 (reference)
Patients with COVID-19	1353 (0.29)	4.59 (4.21-5.00)	4.28 (3.92-4.66)

BMI, body mass index; CCI, Charlson comorbidity index; CI, confidence interval; HR, hazard ratio; NA, not available; SARS-CoV-2, severe acute respiratory syndrome coronavirus 2.

The data in bold indicate significant differences ($P < 0.05$).

***Model 3:** Adjusted for age (20–39, 40–59, and ≥ 60 years) and sex.

|| **Model 4 (replication):** Adjusted for age (20–39, 40–59, and ≥ 60 years); sex; insurance status (insured and dependent); CCI score (0, 1, and ≥ 2); body mass index (underweight [$< 18.5 \text{ kg/m}^2$], normal [$18.5\text{--}22.9 \text{ kg/m}^2$], overweight [$23.0\text{--}25.0 \text{ kg/m}^2$], obese [$\geq 25.0 \text{ kg/m}^2$], and unknown); blood pressure (systolic blood pressure $< 140 \text{ mmHg}$ and diastolic blood pressure $< 90 \text{ mmHg}$, systolic blood pressure $\geq 140 \text{ mmHg}$ or diastolic blood pressure $\geq 90 \text{ mmHg}$, and unknown); fasting blood glucose (< 100 , $\geq 100 \text{ mg/dL}$, and unknown); serum total cholesterol (< 200 , $200\text{--}239$, $\geq 240 \text{ mg/dL}$, and unknown); glomerular filtration rate (< 60 , $60\text{--}89$, $\geq 90 \text{ mL/min/1.73 m}^2$, and unknown); smoking status

(non- and current smoker, and unknown); alcoholic drinks (rarely, sometimes, everyday, and unknown); aerobic physical activity (sufficient, insufficient, and unknown); previous history of cardiovascular disease, and chronic kidney disease; history of medication use for diabetes mellitus, dyslipidemia, and hypertension; and strain of SARS-CoV-2 (original and delta).

Table S24. Stratification analysis for the **short-term** risk of **acute respiratory complication** following COVID-19 in the propensity score matching cohorts (COVID-19 vs. influenza) of Japan (replication)

	COVID-19 vs. influenza (n=169,924)		
	Events, n (%)	HR (95% CI)	
		Model 3*	Model 4 ^{II}
Sex			
Male			
Comparators (general population or patients with influenza)	80 (0.15)	1.0 (reference)	1.0 (reference)
Patients with COVID-19	641 (1.21)	7.97 (6.32-10.05)	6.76 (5.35-8.54)
Female			
Comparators	42 (0.13)	1.0 (reference)	1.0 (reference)
Patients with COVID-19	283 (0.89)	6.80 (4.92-9.40)	6.04 (4.36-8.37)
Age			
20–39 y			
Comparators	31 (0.10)	1.0 (reference)	1.0 (reference)
Patients with COVID-19	205 (0.68)	6.61 (4.53-9.64)	6.21 (4.25-9.06)
40–59 years			
Comparators	64 (0.14)	1.0 (reference)	1.0 (reference)
Patients with COVID-19	535 (1.15)	8.42 (6.50-10.91)	7.05 (5.43-9.16)
≥60 y			
Comparators	27 (0.33)	1.0 (reference)	1.0 (reference)
Patients with COVID-19	184 (2.18)	6.67 (4.45-9.99)	5.61 (3.73-8.44)
CCI score			
0 score			
Comparators	107 (0.13)	1.0 (reference)	1.0 (reference)

Patients with COVID-19	694 (0.85)	6.71 (5.47-8.22)	6.71 (5.47-8.22)
≥1 scores			
Comparators	15 (1.30)	1.0 (reference)	1.0 (reference)
Patients with COVID-19	230 (6.56)	4.88 (2.89-8.23)	4.82 (2.85-8.16)
BMI			
<18.5 kg/m²			
Comparators	81 (0.13)	1.0 (reference)	1.0 (reference)
Patients with COVID-19	629 (1.02)	7.84 (6.22-9.88)	6.94 (5.50-8.76)
18.5-23.0 kg/m²			
Comparators	29 (0.16)	1.0 (reference)	1.0 (reference)
Patients with COVID-19	214 (1.17)	7.24 (4.91-10.67)	5.90 (3.99-8.74)
23.0-25.0 kg/m²			
Comparators	12 (0.24)	1.0 (reference)	1.0 (reference)
Patients with COVID-19	81 (1.59)	6.52 (3.55-11.95)	5.29 (2.86-9.79)
≥25.0 kg/m²			
Comparators	0 (0.00)	1.0 (reference)	1.0 (reference)
Patients with COVID-19	0 (0.00)	1.56 (1.54 to 1.59)	1.54 (1.51 to 1.57)
Smoking status			
Non-smoker			
Comparators	118 (0.14)	1.0 (reference)	1.0 (reference)
Patients with COVID-19	892 (1.09)	7.57 (6.25-9.17)	6.50 (5.36-7.89)
Smoker			
Comparators	4 (0.13)	1.0 (reference)	1.0 (reference)
Patients with COVID-19	32 (1.04)	8.00 (2.83-22.64)	6.43 (2.24-18.45)
Alcohol consumption			
Non-drinker			
Comparators	54 (0.12)	1.0 (reference)	1.0 (reference)

Patients with COVID-19	471 (1.04)	8.64 (6.52-11.45)	7.28 (5.48-9.66)
Drinker			
Comparators	60 (0.18)	1.0 (reference)	1.0 (reference)
Patients with COVID-19	398 (1.19)	6.72 (5.12-8.81)	5.87 (4.47-7.72)
Unknown			
Comparators	8 (0.13)	1.0 (reference)	1.0 (reference)
Patients with COVID-19	55 (0.90)	6.92 (3.30-14.53)	6.22 (2.95-13.14)
			
Physical activity			
Insufficient physical activity			
Comparators	24 (0.14)	1.0 (reference)	1.0 (reference)
Patients with COVID-19	184 (1.06)	7.76 (5.07-11.87)	6.78 (4.42-10.41)
Sufficient physical activity			
Comparators	89 (0.15)	1.0 (reference)	1.0 (reference)
Patients with COVID-19	662 (1.10)	7.42 (5.95-9.26)	6.32 (5.06-7.90)
Unknown			
Comparators	9 (0.12)	1.0 (reference)	1.0 (reference)
Patients with COVID-19	78 (1.01)	8.56 (4.30-17.07)	7.55 (3.77-15.11)
			
Strain of SARS-CoV-2			
Original			
Comparators	59 (0.17)	1.0 (reference)	1.0 (reference)
Patients with COVID-19	496 (1.39)	8.41 (6.42-11.02)	6.81 (5.18-8.94)
Delta			
Comparators	63 (0.13)	1.0 (reference)	1.0 (reference)
Patients with COVID-19	428 (0.87)	6.80 (5.22-8.85)	6.21 (4.76-8.10)

BMI, body mass index; CCI, Charlson comorbidity index; CI, confidence interval; HR, hazard ratio; NA, not available; SARS-CoV-2, severe acute respiratory syndrome coronavirus 2.

The data in bold indicate significant differences ($P < 0.05$).

***Model 3:** Adjusted for age (20–39, 40–59, and ≥ 60 years) and sex.

|| **Model 4 (replication):** Adjusted for age (20–39, 40–59, and ≥ 60 years); sex; insurance status (insured and dependent); CCI score (0, 1, and ≥ 2); body mass index (underweight [$< 18.5 \text{ kg/m}^2$], normal [$18.5\text{--}22.9 \text{ kg/m}^2$], overweight [$23.0\text{--}25.0 \text{ kg/m}^2$], obese [$\geq 25.0 \text{ kg/m}^2$], and unknown); blood pressure (systolic blood pressure $< 140 \text{ mmHg}$ and diastolic blood pressure $< 90 \text{ mmHg}$, systolic blood pressure $\geq 140 \text{ mmHg}$ or diastolic blood pressure $\geq 90 \text{ mmHg}$, and unknown); fasting blood glucose (< 100 , $\geq 100 \text{ mg/dL}$, and unknown); serum total cholesterol (< 200 , $200\text{--}239$, $\geq 240 \text{ mg/dL}$, and unknown); glomerular filtration rate (< 60 , $60\text{--}89$, $\geq 90 \text{ mL/min/1.73 m}^2$, and unknown); smoking status (non- and current smoker, and unknown); alcoholic drinks (rarely, sometimes, everyday, and unknown); aerobic physical activity (sufficient, insufficient, and unknown); previous history of cardiovascular disease, and chronic kidney disease; history of medication use for diabetes mellitus, dyslipidemia, and hypertension; and strain of SARS-CoV-2 (original and delta).

Comment 3.

The statement “the majority of patients with SARS-CoV-2 infection report persistent, long-lasting comorbidities after infection termed long COVID” is untrue. Please correct it. The balance of evidence suggests that long-COVID prevalence following symptomatic SARS-CoV-2 infection is between 5 and 10%.

Response:

Thank you for your valuable feedback. We agree with your suggestion that describing long COVID as affecting the “majority” of patients was an overstatement. To incorporate your opinion, we have revised the sentence as below.

Changes in text:**Introduction, Paragraph #1**

Despite its relatively low fatality rate of approximately 1.3%,³ approximately 10 % of patients with SARS-CoV-2 infection report persistent, long-lasting comorbidities after infection termed long COVID.⁴

Comment 4.

Please elaborate on the meaning of claim-based cohort, and any potential bias that may be introduced. How representative are these cohorts of the populations of South Korea and Japan?

Response:

Thank you for your valuable comment regarding the potential bias and representativeness of our dataset. A claim-based cohort is derived from health insurance claim data, which includes electronic medical records claimed for medical services per individual. The National Health Insurance Service (NHIS) organized a single-payer system for healthcare in South Korea, establishing a comprehensive and representative database of the country's healthcare. The NHIS database we utilized for our main cohort includes over 50 million people, which covers about 98% of the Korean population. The K-CoV-N cohort is a reliable cohort generated by NHIS that covers over 10 million people who are 20 years old or older and have a medical examination record during the pre-observation and follow-up period. However, the NHIS database has its limitations since undocumented residents in South Korea are excluded from the database, as mentioned in the Discussion section '*Limitations and strengths.*'

Similarly, in Japan, we employed data from the Japan Medical Data Center (JMDC) claims database. The JMDC claims database is a database with accumulated receipts and medical examination data. The database includes data from over 17 million individuals, including healthy individuals, allowing researchers to analyze the prevalence rate and incidence of the general population. The strength of the JMDC database is that since the database is an epidemiological receipt database, the data can be tracked even when the participant uses more than one medical facility. JMDC database is, therefore, a credible database that more than 50 universities in Japan utilize and is used in a great number of research papers. Despite the broad data JMDC provides, the JMDC database has its limitations in that it does not include the entire population of Japan. We also added such context in the Discussion section '*Limitations and strengths.*' Additionally, we supplemented the content by adding a more detailed description of each cohort to the Method and Supplementary material section. We sincerely appreciate your suggestions for improving our findings.

Changes in text:

Methods/K-CoV-N cohort for main cohort

- We utilized the NHIS database, which is a large-scale, nationwide, general, population-based cohort in South Korea, covering 98% of the population for the main cohort.¹⁰
- The constructed K-CoV-N database embodies the following characteristics, thereby affirming its significance: (1) the Korean government has established an extensive healthcare system to provide coverage for individuals infected with SARS-CoV-2; (2) all patient-related data was anonymized by the Korean government;¹⁰ and (3) according to prior study, the diagnostic records from the NHIS had a predictive accuracy of 82%.¹¹

Supplement material: JMDC cohort/Data source and study design

- Japan has health insurance provided by the universal insurance system.¹ The JMDC has agreements with over 60 insurance providers and includes health insurance claims records of insured individuals, mostly employees of relatively large companies in Japan.² JMDC has generated a database, using data collected from medical institutions in Japan, consisting of patient-level data (unique identifier, family identifiers, relationship to the insured individual, age, sex) and claims for inpatient and outpatient treatment (disease class according to International Classification of Diseases [ICD]-10 code, prescribed drugs based on Anatomical Therapeutic Chemical class, and drug dosage form), diagnosis or therapeutic procedure, institutional information (hospital character), and health checkup (i.e., body mass index, blood pressure, clinical laboratory test, medication status, and self-administered questionnaire such as smoking status, alcohol consumption, and physical activity).³ For more information, please see this webpage (<https://www.jmdc.co.jp/en/jmdc-claims-database/>). Similar to the South Korea cohort, we collected a total of 4,909,861 subjects aged ≥ 20 years who had information on medical examinations in the JMDC data from January 1, 2018, to December 31, 2021. We excluded participants from the study who did not meet the following criteria: We excluded participants from the study with following criteria: (1) insufficient demographic information and those who died before (excluded n=916,070); and (2) previous history of chronic respiratory disease in the pre-observation period (excluded n=386,937).

Supplement material: JMDC cohort/Covariates

- Participant demographic data was sourced from the JMDC database, which included age (20–39, 40–59, and ≥60 years), sex, insurance status (insured and dependent), CCI score (0, 1, and ≥ 2); body mass index (underweight [$<18.5 \text{ kg/m}^2$], normal [$18.5\text{--}22.9 \text{ kg/m}^2$], overweight [$23.0\text{--}25.0 \text{ kg/m}^2$], obese [$\geq 25.0 \text{ kg/m}^2$], and unknown), blood pressure (systolic blood

pressure <140 mmHg and diastolic blood pressure <90 mmHg, systolic blood pressure ≥140 mmHg or diastolic blood pressure ≥90 mmHg, and unknown), fasting blood glucose (<100, ≥100 mg/dL, and unknown), serum total cholesterol (<200, 200–239, ≥240 mg/dL, and unknown), glomerular filtration rate (<60, 60–89, ≥90 mL/min/1.73 m², and unknown), smoking status (non- and current smoker, and unknown), alcoholic drinks (rarely, sometimes, everyday, and unknown), aerobic physical activity (sufficient, insufficient, and unknown), previous history of cardiovascular disease and chronic kidney disease, history of medication use for diabetes mellitus, dyslipidemia, and hypertension, and strain of SARS-CoV-2 (original and delta).

References (Supplement material)

- 1 Reich, M. R., Ikegami, N., Shibuya, K. & Takemi, K. 50 years of pursuing a healthy society in Japan. *Lancet* **378**, 1051-1053 (2011). [https://doi.org/10.1016/s0140-6736\(11\)60274-2](https://doi.org/10.1016/s0140-6736(11)60274-2)
- 2 Setogawa, N., Ohbe, H., Isogai, T., Matsui, H. & Yasunaga, H. Characteristics and short-term outcomes of outpatient and inpatient cardiac catheterizations: A descriptive study using a nationwide claim database in Japan. *J Cardiol* **82**, 201-206 (2023). <https://doi.org/10.1016/j.jjcc.2023.05.010>
- 3 Kaneko, H. *et al.* Medication-Naïve Blood Pressure and Incident Cancers: Analysis of 2 Nationwide Population-Based Databases. *Am J Hypertens* **35**, 731-739 (2022). <https://doi.org/10.1093/ajh/hpac054>

Discussion/Limitations and strengths

- First, although the database used is a highly credible database that covers 98% of the Korean population and 40% of the Japanese population, individuals who could be vulnerable to influenza and COVID-19, such as immigrants and undocumented immigrants, are left out of the database.^{10, 49}

References (Manuscript)

- 10 Shin, Y. H. *et al.* Autoimmune inflammatory rheumatic diseases and COVID-19 outcomes in South Korea: a nationwide cohort study. *The Lancet Rheumatology* **3**, e698-e706 (2021).

11 Park, B., Sung, J., Park, K., Seo, S. & Kim, S. Report of the evaluation for validity of discharged diagnoses in Korean Health Insurance database. Seoul: Seoul National University, 19-52 (2003).

49 Sato, M. *et al.* Effect of periodontal therapy on glycaemic control in type 2 diabetes. *J Clin Periodontol* (2024). <https://doi.org/10.1111/jcpe.13939>

Comment 5.

Be more specific about the time period of infection and follow-up. Did you consider including dominant variant as a covariate in analysis?

Response:

Thank you for addressing an important aspect that would help us improve our manuscript. Upon reviewing your comment, we realized the lack of details in explaining the index dates of both infected and uninfected groups. Therefore, we included specified details of the index date in our manuscript and Figure S2. In addition, your feedback prompted us to enhance our research by including the dominant variant as a covariate in the analysis. In the K-CoV-N cohort, the SARS-CoV-2 infection from January 2020 to July 31, 2021, was defined as the original strain, and the delta variant was defined from August 1, 2021 to December 31, 2021.¹ In the JMDC cohort, diagnosis of SARS-CoV-2 infection was categorized as infection with original strain until May 31, 2021 and delta variants from June 1, 2021, to December 31, 2021.² Based on these distinctions, we performed additional subgroup analyses, assessing the hazard ratio of the original and delta strains separately compared to a control group of uninfected individuals with matching index dates. In addition, based on your suggestions, we have added the dominant variant as a covariate in the revised analysis. The re-analyzed results, including HR and 95% confidence intervals (CIs) for both long-term post-acute respiratory sequelae and short-term acute respiratory complications, are now presented in Table 4. Furthermore, we have elaborated on the index date and the timeframe for the original and delta variants in the Methods section. Once again, we sincerely appreciate your suggestions for improving our findings.

<Reference>

1. Ryu, B. H. et al. Clinical Features of Adult COVID-19 Patients without Risk Factors before and after the Nationwide SARS-CoV-2 B.1.617.2 (Delta)-variant Outbreak in Korea: Experience from Gyeongsangnam-do. *Journal of Korean medical science* 36, e341 (2021). <https://doi.org/10.3346/jkms.2021.36.e341>
2. Ito K, Piantham C, Nishiura H. Predicted dominance of variant Delta of SARS-CoV-2 before Tokyo Olympic Games, Japan, July 2021. *Euro Surveill* 2021; 26(27).

Changes in text:

Methods/Statistical analysis

- Therefore, we assigned the 'individual index date' by following the criterion. For the exposed group, it is the date of the first diagnosis of SARS-CoV-2. For the individuals in the non-exposed group, the index date was allocated to match the index date of the corresponding exposed case. This approach was implemented to mitigate immortal time bias, ensuring an equitable comparison between groups.

Methods/Exposures

- In South Korea, SARS-CoV-2 infection from January 2020 to July 31, 2021, was defined as the original strain, and the SARS-CoV-2 infection from August 1, 2021, to December 31, 2021, was defined as the delta.¹⁵ In Japan, diagnosis of SARS-CoV-2 infection was categorized as infection with original strain until May 31, 2021, and delta variants from June 1, 2021, to December 31, 2021.¹⁶

Methods/Covariates

- ...aerobic physical activity (sufficient [≥ 150 min/week of moderate-intensity activity or ≥ 75 min/week of vigorous-intensity activity or greater than an equivalent combination], insufficient, and unknown), and type of SARS-CoV-2 (original and delta) were obtained.¹²

Methods/Statistical analysis

- These analyses involved respiratory disease (long-term post-acute respiratory sequelae and acute respiratory complication) and its subtypes, number of vaccinations (without, 1 time, and ≥ 2 times), type of previous vaccination (mRNA, viral vector, and both), and strain of SARS-CoV-2 (original and delta).

- To minimize the impact of potential confounders, the following variables were used for the adjusted model: age, sex, household income, region of residence, CCI score, obesity, blood pressure, fasting blood glucose, serum total cholesterol, glomerular filtration rate, smoking status, alcoholic drinks, aerobic physical activity, previous history of cardiovascular and chronic kidney diseases, history of medication use for diabetes mellitus, dyslipidemia, and hypertension, and strain of SARS-CoV-2 (original and delta).

Results

- Both the original strain and the delta variant of SARS-CoV-2 were shown to have a higher risk of acute respiratory complications (original strain: HR, 9.21 [95% CI, 7.19-11.80]; delta strain: HR, 7.44 [95% CI, 6.13-9.03]).

Discussion/Findings of this study

Fifth, infection of SARS-CoV-2 was associated with increase of long-term post-acute respiratory sequelae and acute respiratory complications regardless of the strain type.

Supplementary material: JMDC cohort/Covariates

- ...previous history of cardiovascular disease and chronic kidney disease, history of medication use for diabetes mellitus, dyslipidemia, and hypertension, and strain of SARS-CoV-2 (original and delta).

References (Manuscript)

15 Ryu, B. H. *et al.* Clinical Features of Adult COVID-19 Patients without Risk Factors before and after the Nationwide SARS-CoV-2 B.1.617.2 (Delta)-variant Outbreak in Korea: Experience from Gyeongsangnam-do. *Journal of Korean medical science* **36**, e341 (2021). <https://doi.org/10.3346/jkms.2021.36.e341>

16 Ito K, Piantham C, Nishiura H. Predicted dominance of variant Delta of SARS-CoV-2 before Tokyo Olympic Games, Japan, July 2021. *Euro Surveill* 2021; 26(27).

Figure S2. Study flow of cohorts

Table 4. Subgroup analysis (COVID-19 vs. general population) of HR (95% CI) of the **long-term post-acute respiratory sequelae** or **short-term acute respiratory complication** after SARS-CoV-2 infection stratified by vaccination, COVID-19 severity, and SARS-CoV-2 strain in the cohort of South Korea (main)

Variable	Events/total, n/N (%)	HR (95% CI)	
		Model 1*	Model 2†
Long-term post-acute respiratory sequelae			
Number of SARS-CoV-2 vaccinations			
Non-infected control	16,122/1,918,150 (0.84)	0.62 (0.60-0.65)	0.60 (0.58-0.62)
COVID-19 without SARS-CoV-2 vaccination	4331/200,539 (2.16)	1.0 (reference)	1.0 (reference)
COVID-19 after SARS-CoV-2 vaccination received once	493/38,852 (1.27)	0.90 (0.82-0.99)	0.85 (0.77-0.94)
COVID-19 after SARS-CoV-2 vaccination received twice or more	468/155,207 (0.30)	0.69 (0.62-0.76)	0.64 (0.57-0.71)
Type of SARS-CoV-2 vaccinations			
Non-infected control	16,122/1,918,150 (0.84)	0.62 (0.60-0.65)	0.60 (0.58-0.62)
COVID-19 without SARS-CoV-2 vaccination	4331/200,539 (2.16)	1.0 (reference)	1.0 (reference)
COVID-19 with viral vector SARS-CoV-2 vaccination	465/109,066 (0.43)	1.12 (1.01-1.23)	0.99 (0.90-1.10)
COVID-19 with mRNA SARS-CoV-2 vaccination	477/66,891 (0.71)	1.20 (1.09-1.33)	1.11 (0.99-1.22)
COVID-19 with both types of SARS-CoV-2 vaccination	19/18,102 (0.10)	0.74 (0.47-1.16)	0.66 (0.42-1.03)
COVID-19 severity			
Non-infected control	16,122/1,918,150 (0.84)	1.0 (reference)	1.0 (reference)
Mild COVID-19	3492/340,813 (1.02)	1.28 (1.24-1.33)	1.37 (1.32-1.43)
Moderate to severe COVID-19	1800/53785 (3.35)	3.60 (3.43-3.78)	3.20 (3.03-3.38)
Original strain of SARS-CoV-2 (overall population)			
Non-infected control before the delta-dominant phase [§]	11,667/594,134 (1.96)	1.0 (reference)	1.0 (reference)

Infection with original strain	3649/121,521 (3.00)	1.58 (1.52-1.64)	1.59 (1.52-1.66)
Delta variant of SARS-CoV-2 (overall population)			
Non-infected control during the delta-dominant phase [§]	4455/1,324,016 (0.34)	1.0 (reference)	1.0 (reference)
Infection with Delta variant	1643/273,077 (0.60)	1.81 (1.71-1.92)	1.94 (1.81-2.08)
Acute respiratory complication			
Number of SARS-CoV-2 vaccinations			
Non-infected control	311/1,918,150 (0.016)	0.07 (0.06-0.08)	0.08 (0.06-0.09)
COVID-19 without SARS-CoV-2 vaccination	415/200,539 (0.21)	1.0 (reference)	1.0 (reference)
COVID-19 after SARS-CoV-2 vaccination received once	56/38,852 (0.14)	0.62 (0.47-0.83)	0.51 (0.38-0.68)
COVID-19 after SARS-CoV-2 vaccination received twice or more	147/155,207 (0.09)	0.32 (0.26-0.39)	0.24 (0.19-0.30)
Type of SARS-CoV-2 vaccinations			
Non-infected control	311/1,918,150 (0.016)	0.07 (0.06-0.08)	0.08 (0.06-0.09)
COVID-19 without SARS-CoV-2 vaccination	415/200,539 (0.21)	1.0 (reference)	1.0 (reference)
COVID-19 with viral vector SARS-CoV-2 vaccination	79/109,066 (0.072)	0.44 (0.34-0.56)	0.36 (0.28-0.47)
COVID-19 with mRNA SARS-CoV-2 vaccination	117/66,891 (0.17)	0.40 (0.32-0.50)	0.42 (0.33-0.53)
COVID-19 with both types of SARS-CoV-2 vaccination	7/18,102 (0.039)	0.19 (0.09-0.41)	0.18 (0.08-0.38)
COVID-19 severity			
Non-infected control	311/1,918,150 (0.016)	1.0 (reference)	1.0 (reference)
Mild COVID-19	50/340,813 (0.015)	0.95 (0.71-1.28)	0.99 (0.73-1.34)
Moderate to severe COVID-19	568/53,783 (1.06)	51.18 (44.38-59.02)	39.54 (33.54-46.62)
Original strain of SARS-CoV-2 (overall population)			
Non-infected control before the delta-dominant phase [§]	110/594,134 (0.019)	1.0 (reference)	1.0 (reference)
Infection with original strain	230/121,521 (0.19)	10.27 (8.18-12.88)	9.21 (7.19-11.80)

Delta variant of SARS-CoV-2 (overall population)

Non-infected control during the delta-dominant phase §	201/1,324,016 (0.015)	1.0 (reference)	1.0 (reference)
Infection with Delta variant	388/273,077 (0.14)	9.39 (7.92-11.14)	7.44 (6.13-9.03)

CCI, Charlson comorbidity index; CI, confidence interval; HR, hazard ratio; SARS-CoV-2, severe acute respiratory syndrome coronavirus 2.

The data in bold indicate significant differences ($P < 0.05$).

|| HR of the non-infected control represents the risk of respiratory diseases, and HRs of patients with COVID-19 indicate the risk of post-acute respiratory complications following SARS-CoV-2 infection.

§ Only 1:5-matched comparators in each patient group at the same index date were included to reduce immortal time bias.

***Model 1:** Adjusted for age (20–39, 40–59, and ≥ 60 years) and sex.

†**Model 2:** Adjusted for age (20–39, 40–59, and ≥ 60 years); sex, household income (low income, middle income, and high income); region of residence (urban and rural); CCI score (0, 1, and ≥ 2); obesity (underweight [$< 18.5 \text{ kg/m}^2$], normal [$18.5\text{--}22.9 \text{ kg/m}^2$]; overweight [$23.0\text{--}24.9 \text{ kg/m}^2$], obese [$\geq 25.0 \text{ kg/m}^2$], and unknown); blood pressure (systolic blood pressure $< 140 \text{ mmHg}$ and diastolic blood pressure $< 90 \text{ mmHg}$, systolic blood pressure $\geq 140 \text{ mmHg}$ or diastolic blood pressure $\geq 90 \text{ mmHg}$, and unknown); fasting blood glucose (< 100 , $\geq 100 \text{ mg/dL}$, and unknown); serum total cholesterol (< 200 , $200\text{--}239$, $\geq 240 \text{ mg/dL}$, and unknown); glomerular filtration rate (< 60 , $60\text{--}89$, $\geq 90 \text{ mL/min/1.73 m}^2$, and unknown); smoking status (never, former, current smoker, and unknown); alcoholic drinks (< 1 , $1\text{--}2$, $3\text{--}4$, ≥ 5 days per week, and unknown); aerobic physical activity (sufficient, insufficient, and unknown); previous history of cardiovascular disease, and chronic kidney disease; history of medication use for diabetes mellitus, dyslipidemia, and hypertension; and strain of SARS-CoV-2 (original and delta).

Comment 6.

Figure S1 requires clarification. Are the number of exclusions the total number of participants excluded in each group (A-D) for all reasons (insufficient information, death, previous history of respiratory disease)? Please provide numbers excluded for each reason. The proportions excluded from the Japanese cohort are far higher than those excluded from the South Korean cohort. Please comment on possible reasons for this. Could there be bias due to under or over-recording of disease?

Response:

Thank you for the comment that would aid us in increasing the credibility of our study. We deeply agree with your opinion regarding the need to clarify the number of exclusions. As you mentioned, Figure S1 shows the exclusions of the study population due to insufficient information, death, and history of chronic respiratory diseases. Therefore, we have added and revised the reasons for exclusion along with the number of participants excluded for each reason. Furthermore, we deeply apologize for the discrepancy between the recorded number of participants and the actual number constructed in our database for the Japanese cohort. After reviewing our datasets, we confirm that the initial Japanese cohorts number approximately 4.9 million. We apologize for any confusion caused. Your feedback has led to a more detailed Figure S1 and methods section, and we are truly grateful for this opportunity to improve our manuscript.

Changes in text:**Methods/K-CoV-N cohort for main cohort**

- We excluded participants with the following criteria: (1) insufficient demographic information and those who died before (excluded n=3,967,482); and (2) previous history of chronic respiratory disease in the pre-observation period (excluded n=710,468).

Supplement material: JMDC cohort/Data source and study design

- Similar to the South Korea cohort, we collected a total of 4,909,861 subjects aged ≥ 20 years who had information on medical examinations in the JMDC data from January 1, 2018, to December 31, 2021. We excluded participants with the following criteria: (1) insufficient demographic information and those who died before (excluded n=916,070); and (2) previous history of chronic respiratory disease in the pre-observation period (excluded n=386,937).

Figure S1. Study population in the main cohort (South Korea) and replication cohort (Japan)

Comment 7.

Table S1 is unnecessary and out of place. The content should be moved to the text in methods, results and discussion where relevant..

Response:

We appreciate your invaluable suggestions. Table S1 describes the disease exposure and outcome, baseline characteristics of the study population, other considerations, and overall representatives of the study. We thought that Table S1 would be helpful for comprehensively showing the characteristics of our study. However, following your comment, we have realized that most of the context of Table S1 is included in our manuscript already, and it would be unnecessary for us to include Table S1 in the Supplementary Material. Therefore, we deleted Table S1 from our Supplementary Material and added the necessary information in the manuscript as below. Thank you for helping us improve our article.

Changes in text:**Results**

- In the main cohort, there were a total of 10,027,506 participants with a mean age of 48.4 (standard deviation [SD], 13.4) years, of which 49.9% (5,000,621/10,027,506) were female (**Table S8**). The replication cohort includes 4,909,861 participants with a mean age of 46.8 (SD, 11.9) years and 38.3% (1,882,174/4,909,861) females (**Table S9**).

Comment 8.

Table S3 is referred to out of context at the bottom of page 10.

Response:

Thank you for the detailed comment indicating the mistakes we would have missed otherwise. Considering your comment, we have removed Table S3, which is mentioned in the Method section, ‘*Outcome.*’ In addition, since we deleted Table S1 according to your previous comment, the original Table S3 is now Table S2. Table S2 is mentioned in the first sentence of the Method section, ‘*Statistical analysis.*’

Changes in text:

Methods/ *Statistical analysis*

In the matched COVID-19 cohort, we conducted various statistical analyses, detailed in **Table S2.**

Comment 9.

Table S4 is missing urban/rural residence for validation cohort.

Response:

Thank you for your feedback. South Korean and Japanese data were collected by different institutions. Therefore, the two databases have different components and data structures. While the Korean database includes data on the participant's region of residence, the Japanese database does not include such characteristics, hence the missing information on urban and rural residence. To address this issue, we marked such instances as NA (not available) in the initial analysis. Consequently, in this revision, we divided Table S4 into two tables to demonstrate the data more clearly for each cohort: Table S8 and Table S9, which are shown in the Supplementary Materials. We also addressed this issue, about differences in data structures between datasets from Korea and Japan, in the Discussion section, 'Limitations and strengths.

Changes in text:**Discussion/*Limitations and Strengths***

- Third, the K-CoV-N and JMDC datasets are heterogeneous. Therefore, we opted against merging the datasets, using the K-CoV-N data for the main cohort and the JMDC data for the replication cohort. In addition, we used different lists of the covariates for each main and replication cohort due to the difference in data structure.

Table S8. Baseline characteristics for the full unmatched cohorts of South Korea (main cohort)

Characteristics	Main cohort
Total, n	10,027,506
Age, years, mean (SD)	48.4 (13.4)
Age, years, n (%)	
20 to 39	2,756,102 (27.5)
40 to 59	4,799,784 (47.9)
≥ 60	2,471,620 (24.7)
Sex, n (%)	
Male	5,026,885 (50.1)
Female	5,000,621 (49.9)
Region of residence, n (%)	
Rural	4,460,562 (44.5)
Urban	5,566,944 (55.5)

SD, standard deviation.

Table S9. Baseline characteristics for the full unmatched cohorts of Japan (replication cohort)

Characteristics	Replication cohort
Total, n	4,909,861
Age, years, mean (SD)	46.8 (11.9)
Age, years, n (%)	
20 to 39	1,362,496 (27.8)
40 to 59	2,801,078 (57.1)
≥ 60	746,287 (15.2)
Sex, n (%)	
Male	3,027,687 (61.7)
Female	1,882,174 (38.3)
Insurance status, n (%)	
Insured	4,291,062 (87.4)
Dependent	618,799 (12.6)

SD, standard deviation.

Comment 10.

Table 1 reports substantial missingness. How was this dealt with?

Response:

Thank you for your valuable feedback. Table 1 depicts the baseline characteristics for the propensity score-matched main cohort. Missingness in our database may be due to several procedures that were excluded. Our cohort only includes participants with medical examination records during the examination period. This is because of the confounders we may not be able to take into consideration. We attempted to control and correct any possible confounding variables by using the indicators investigated during the health examination. Therefore, we composed the cohort with only participants with health examination records, which is the main reason for the exclusion of most excluded participants.

Due to the uniqueness of our study, we excluded the patients with insufficient demographic information, those who died, and those with a history of chronic respiratory disease in the pre-observation period or influenza before the index date. We then conducted 1:5 propensity score matching to minimize the possible bias. Each patient infected with SARS-CoV-2 was matched with around 5 people from the general population according to propensity score. During the multi-to-one propensity score matching, there were excluded participants, resulting in 2,312,748 participants in the final matched cohort. We hope the reviewer understands our effort to minimize the sample bias in conducting the research. Furthermore, we have enhanced the content by detailing the number of individuals excluded based on specific criteria prior to matching for each cohort in the Methods section. We sincerely appreciate the valuable feedback provided by the reviewer.

Changes in text:**Methods/K-CoV-N cohort for main cohort**

- We excluded participants with the following criteria: (1) insufficient demographic information and those who died before (excluded n=3,967,482); and (2) previous history of chronic respiratory disease in the pre-observation period (excluded n=710,468).

Supplement material/JMDC cohort

- Similar to the South Korea cohort, we collected a total of 4,909,861 subjects aged ≥ 20 years who had information on medical examinations in the JMDC data from January 1, 2018, to

December 31, 2021. We excluded participants with the following criteria: (1) insufficient demographic information and those who died before (excluded n=916,070); and (2) previous history of chronic respiratory disease in the pre-observation period (excluded n=386,937).

Table 1. Baseline characteristics for 1:5 propensity score–matched cohort (COVID-19 vs. general population) in South Korea (main)

Characteristic	COVID-19 vs. general population (n=2,312,748)		SMD*
	COVID-19 (n=394,598)	General population (n=1,918,150)	
Mean age (SD), y	47.5 (16.8)	46.8 (14.3)	0.046
Age, n (%)			0.026
20–39 y	143,273 (36.3)	702,322 (36.6)	
40–59 y	145,169 (36.8)	709,343 (37.0)	
≥60 y	106,156 (26.9)	506,485 (26.4)	
Sex, n (%)			0.006
Male	205,058 (52.0)	997,982 (52.0)	
Female	189,540 (48.0)	920,168 (48.0)	
Region of residence, n (%)			<0.001
Urban	213,052 (54.0)	1,035,261 (54.0)	
Rural	181,546 (46.0)	882,889 (46.0)	
Medical history, n (%)			
Cardiovascular disease	59,947 (15.2)	286,623 (14.9)	0.009
Chronic kidney disease	18,963 (4.8)	88,687 (4.6)	0.005
Medication use for diabetes	71,625 (18.2)	342,656 (17.9)	0.008
Medication use for hyperlipidemia	59,947 (15.2)	286,623 (14.9)	0.007
Medication use for hypertension	32,402 (8.2)	154,789 (8.1)	0.005
Unmatching covariates, n (%)[†]			
Charlson Comorbidity Index score			0.230
0	346,579 (87.8)	1,806,906 (94.2)	
1	30,584 (7.8)	60,556 (3.2)	

≥2	17,435 (4.4)	50,688 (2.6)	
Household income			<0.001
Low (0th–39th percentile)	182,632 (46.3)	887,593 (46.3)	
Middle (40th –79th percentile)	140,084 (35.5)	681,180 (35.5)	
High (80th–100th percentile)	71,882 (18.2)	349,377 (18.2)	
Body mass index			1.212
Underweight (<18.5 kg/m ²)	6875 (1.7)	67,970 (3.5)	
Normal (18.5-22.9 kg/m ²)	77,779 (19.7)	685,687 (35.8)	
Overweight (23.0-24.9 kg/m ²)	54,427 (13.8)	440,197 (23.0)	
Obese (≥25.0 kg/m ²)	95,082 (24.1)	724,074 (37.8)	
Unknown	160,435 (40.7)	222 (0.012)	
Blood pressure			1.157
SBP <140 mmHg and DBP <90 mmHg	199,342 (50.5)	1,673,777 (87.3)	
SBP ≥140mmHg or DBP ≥90 mmHg	33,711 (8.5)	240,991 (12.6)	
Unknown	161,545 (40.9)	3382 (0.2)	
Fasting blood glucose			1.179
<100 mg/dL	140,656 (35.7)	1,189,021 (62.0)	
≥100 mg/dL	92,374 (23.4)	725,682 (37.8)	
Unknown	161568 (40.9)	3447 (0.2)	
Serum total cholesterol			0.416
<200 mg/dL	67,886 (17.2)	525,684 (27.4)	
200 to 239 mg/dL	39,508 (10.0)	316,959 (16.5)	
≥240 mg/dL	16,860 (4.3)	134,405 (7.0)	
Unknown	270,344 (68.5)	941,102 (49.1)	
Glomerular filtration rate			1.180
<60 mL/min/1.73 m ²	8332 (2.1)	52,421 (2.7)	
60 to 89 mL/min/1.73 m ²	102,604 (26.0)	812,214 (42.3)	

≥90 mL/min/1.73 m ²	121,914 (30.9)	1,048,318 (54.7)	
Unknown	161,748 (41.0)	5197 (0.3)	
Smoking status			1.191
Never	154,105 (39.1)	1,214,221 (63.3)	
Former	42,814 (10.9)	299,027 (15.6)	
Current	37,263 (9.4)	404,323 (21.1)	
Unknown	160,416 (40.7)	579 (0.030)	
Alcohol consumption			1.156
<1 day/week	136,870 (34.7)	1,143,488 (59.6)	
1 to 2 days/week	66,311 (16.8)	545,090 (28.4)	
3 to 4 days/week	23,113 (5.9)	173,245 (9.0)	
≥5 days/week	7893 (2.0)	55,687 (2.9)	
Unknown	160,411 (40.7)	640 (0.034)	
Aerobic physical activity			1.179
Insufficient	118,792 (30.1)	959,088 (50.0)	
Sufficient	115,321 (29.2)	958,194 (50.0)	
Unknown	160,485 (40.7)	868 (0.1)	
Strain of SARS-CoV-2			0.004
Original	121,521 (30.8)	594,134 (31.0)	
Delta	273,077 (69.2)	1,324,016 (69.0)	

DBP, diastolic blood pressure; SARS-CoV-2, severe acute respiratory syndrome coronavirus 2; SBP, systolic blood pressure; SD, standard deviation; SMD, standardized mean difference.

* An SMD <0.1 indicates no significant imbalance. All SMDs were <0.100 in the propensity score-matched cohorts.

† Unmatched covariates were included as adjustment factors in statistical analyses.

Table S10. Baseline characteristics for 1:3 propensity score–matched cohort (COVID-19 vs. general population) in Japan (replication)

Characteristic	COVID-19 vs. general population (n=3,115,606)		SMD*
	COVID-19 (n=797,101)	General population (n=2,318,505)	
Mean age (SD), y	44 (11.88)	44 (12.03)	0.034
Age, n (%)			<0.001
20–39 y	302,404 (37.94)	878,119 (37.87)	
40–59 y	411,351 (51.61)	1,199,717 (51.75)	
≥60 y	83,346 (10.46)	240,669 (10.38)	
Sex, n (%)			0.005
Male	495,460 (62.16)	1,447,190 (62.42)	
Female	301,641 (37.84)	871,315 (37.58)	
Insurance status, n (%)			0.001
Insured	718,811 (90.18)	2,091,308 (90.20)	
Dependent	78,290 (9.82)	227,197 (9.80)	
Medical history, n (%)			
Cardiovascular disease	60,450 (7.58)	170,601 (7.36)	0.009
Chronic kidney disease	32,292 (4.05)	90,680 (3.91)	0.007
Medication use for diabetes	19,777 (2.48)	56,919 (2.45)	0.002
Medication use for hyperlipidemia	49,255 (6.18)	141,631 (6.11)	0.003
Medication use for hypertension	67,167 (8.43)	191,951 (8.28)	0.005
Unmatching covariates, n (%)[†]			
Charlson Comorbidity Index score			0.203
0	764,052 (95.85)	2,293,774 (98.93)	
1	10,954 (1.37)	8,640 (0.37)	
≥2	22,095 (2.77)	16,091 (0.69)	

Body mass index			<0.001
Underweight (<18.5 kg/m ²)	427,244 (53.60)	1,263,598 (54.50)	
Normal (18.5-22.9 kg/m ²)	151,460 (19.00)	439,941 (18.98)	
Overweight (23.0-24.9 kg/m ²)	169,962 (21.32)	484,142 (20.88)	
Obese (≥25.0 kg/m ²)	47,210 (5.92)	127,534 (5.50)	
Unknown	1225 (0.15)	3290 (0.14)	
Blood pressure			0.038
SBP <140 mmHg and DBP <90 mmHg	691,676 (86.77)	1,986,000 (85.66)	
SBP ≥140mmHg or DBP ≥90 mmHg	46,474 (5.83)	149,954 (6.47)	
Unknown	58,951 (7.40)	182,551 (7.87)	
Fasting blood glucose			<0.001
<100 mg/dL	511,681 (64.19)	1,474,020 (63.58)	
≥100 mg/dL	160,434 (20.13)	474,726 (20.48)	
Unknown	124,986 (15.68)	369,759 (15.95)	
Serum total cholesterol			0.054
<200 mg/dL	373,377 (46.84)	1,040,983 (44.90)	
200 to 239 mg/dL	285,475 (35.81)	843,032 (36.36)	
≥240 mg/dL	123,823 (15.53)	379,819 (16.38)	
Unknown	14,426 (1.81)	54,671 (2.36)	
Glomerular filtration rate			0.037
<60 mL/min/1.73 m ²	4383 (0.55)	8174 (0.35)	
60 to 89 mL/min/1.73 m ²	72,932 (9.15)	215,112 (9.28)	
≥90 mL/min/1.73 m ²	433,528 (54.39)	1,237,792 (53.39)	
Unknown	286,258 (35.91)	857,427 (36.98)	
Smoking status			0.062
Non-smoker	181,033 (22.71)	554,190 (23.90)	
Smoker	586,552 (73.59)	1,677,800 (72.37)	

Unknown	29,516 (3.70)	86,515 (3.73)	
Alcohol consumption			0.040
Everyday	161,809 (20.30)	466,183 (20.11)	
Sometimes	264,583 (33.19)	762,501 (32.89)	
Rarely	311,632 (39.10)	919,772 (39.67)	
Unknown	59,077 (7.41)	170,049 (7.33)	
Aerobic physical activity			0.046
Insufficient	161,205 (20.22)	497,575 (21.46)	
Sufficient	562,300 (70.54)	1,605,294 (69.24)	
Unknown	73,596 (9.23)	215,636 (9.30)	
Strain of SARS-CoV-2			0.005
Original	335,571 (42.10)	981,871 (42.35)	
Delta	461,530 (57.90)	1,336,634 (57.65)	

DBP, diastolic blood pressure; SARS-CoV-2, severe acute respiratory syndrome coronavirus 2; SBP, systolic blood pressure; SD, standard deviation; SMD, standardized mean difference.

* An SMD <0.1 indicates no significant imbalance. All SMDs were <0.100 in the propensity score-matched cohorts.

† Unmatched covariates were included as adjustment factors in statistical analyses.

Comment 11.

Tables 2 and 3 report very large differences in effect size between the discovery and validation cohorts. Please comment on this and its implications.

Response:

Thank you for addressing an important difference of effect size in the two cohorts. Difference of effect size may be shown in the analysis because of two main reasons.

Firstly, the HR of the two cohorts may depict a size difference due to the different therapeutic regimens applied in the two different countries. South Korea and Japan show different patterns of changes in the COVID-19 situation and strategies for dealing with the pandemic. Due to the limitation of not being able to secure enough vaccines, vaccination in Japan was slower than that of South Korea.¹ The South Korean government achieved high control over the COVID-19 pandemic through aggressive case identification and the “test, track, isolate” method. This resulted in a high possibility of diagnosing people with SARS-CoV-2 infection and confirmed patients were quarantined for two weeks or more. However, the Japanese government relied more on voluntary behaviors. The Japanese government recommended quarantine rather than compelling confirmed patients, especially those with mild symptoms, and focused on treating serious cases.² Due to the difference in medical policies and responses to the pandemic, the effect size difference between the two cohorts is inevitable.

We also figured that the difference may be due to the difference in the method of collecting the data and the data structure. Therefore, we did additional research on the collection of data and the construction of the database. NHIS runs a comprehensive health insurance system that covers data from both hospitals and primary care settings. The database NHIS formed is a credible database that includes mild cases diagnosed in primary care, increasing the possibility of covering a wide range of health cases.³ Likewise, the JMDC billing database of Japan is an epidemiological database that collects receipts and health examination data from several medical associations. Therefore, even in the cases where the patient visits more than one medical institution, all the medical data can be included in the JMDC database, including patients with mild symptoms. (<https://www.jmdc.co.jp/en/jmdc-claims-database/>)

Editorial Note: Image below redacted where no permission to publish could be obtained.

[REDACTED]

However, although the datasets being collected under similar conditions regarding the timing of medical visits, differences in data construction and participant characteristics have been identified. For JMDC, the dataset is constructed primarily around claim data received from over 60 distinct insurance providers.⁴ In addition, it mainly targets employed individuals and their families within Japan, indicating a potential limitation in demographic scope.⁵ On the other hand, the K-CoV-2 dataset is focused on the entire South Korean population (98% coverage).⁶ Moreover, there can be inherent differences in the coding of diseases across different countries. Consequently, the difference in the method of collecting the data and the data structure inevitably imply the possibility of leading to variations in the outcomes.

Therefore, acknowledging the possibility of bias that we were not able to take into consideration, we conducted an additional negative control analysis. Negative control analysis is an analysis method of that can detect and verify the biases or confounder factor that may form during the research construction or data analysis. Negative control analysis enables the researchers to verify if the database is widely affected by other factors and is not limited to the desired outcome. In the negative control analysis, we therefore conducted an analysis with tympanic membrane perforation as the outcome. According to the analysis, it was confirmed that the values in both cohorts are statistically insignificant. We added explanation about the new analysis regarding the association of tympanic membrane perforation and long COVID by making a new section in the **Methods** called '**Sensitivity analysis**'.

These results show that the research construction and analysis methods we used had little potential bias or influence from confounding variables. However, our previous manuscript did not mention enough about the differences in the two datasets and did not explain the potential limitation of a bi-national study. We mentioned this in the Limitations and Strengths section and made further refinements to our methodology for each cohort, as detailed in the

Methods section, to enhance the completeness of our study. Thank you for mentioning such an important factor and help us improve our study.

<References>

1. Ma, M. et al. Comparison of COVID-19 vaccine policies and their effectiveness in Korea, Japan, and Singapore. *Int J Equity Health* 22, 224 (2023). <https://doi.org/10.1186/s12939-023-02034-x>
2. Wang, X. et al. A Comparative Retrospective Study of COVID-19 Responses in Four Representative Asian Countries. *Risk Manag Healthc Policy* 15, 13-25 (2022). <https://doi.org/10.2147/rmhp.S334326>
3. Cheol Seong S, Kim YY, Khang YH, Heon Park J, Kang HJ, Lee H, Do CH, Song JS, Hyon Bang J, Ha S, Lee EJ, Ae Shin S. Data Resource Profile: The National Health Information Database of the National Health Insurance Service in South Korea. *Int J Epidemiol.* 2017 Jun 1;46(3):799-800. doi: 10.1093/ije/dyw253. PMID: 27794523; PMCID: PMC5837262.
4. Reich MR, Ikegami N, Shibuya K, Takemi K. 50 years of pursuing a healthy society in Japan. *Lancet.* 2011 Sep 17;378(9796):1051-3. doi: 10.1016/S0140-6736(11)60274-2IF: 168.9 Q1 . Epub 2011 Aug 30. PMID: 21885101.
5. Setogawa N, Ohbe H, Isogai T, Matsui H, Yasunaga H. Characteristics and short-term outcomes of outpatient and inpatient cardiac catheterizations: A descriptive study using a nationwide claim database in Japan. *J Cardiol.* 2023 Sep;82(3):201-206. doi: 10.1016/j.jjcc.2023.05.010. Epub 2023 May 28. PMID: 37247658.
6. Cheol Seong S, Kim YY, Khang YH, Heon Park J, Kang HJ, Lee H, Do CH, Song JS, Hyon Bang J, Ha S, Lee EJ, Ae Shin S. Data Resource Profile: The National Health Information Database of the National Health Insurance Service in South Korea. *Int J Epidemiol.* 2017 Jun 1;46(3):799-800. doi: 10.1093/ije/dyw253. PMID: 27794523; PMCID: PMC5837262.

Changes in text:

Methods/*Sensitivity analysis*

- Several sensitivity analyses were conducted to enhance the credibility of the manuscript and our primary analyses. First, to identify detection bias and validate the results of our cohort, we

performed a negative control analysis by exploring the association between tympanic membrane perforation disease and SARS-CoV-2 infection (**Table S3**).

Methods/K-CoV-N cohort for main cohort

- We utilized the NHIS database, which is a large-scale, nationwide, general, population-based cohort in South Korea, covering 98% of the population for the main cohort.¹⁰
- The constructed K-CoV-N database embodies the following characteristics, thereby affirming its significance: (1) the Korean government has established an extensive healthcare system to provide coverage for individuals infected with SARS-CoV-2; (2) all patient-related data was anonymized by the Korean government;¹⁰ and (3) according to prior study, the diagnostic records from the NHIS had a predictive accuracy of 82%.¹¹

Discussion/Limitations and strengths

- Third, the K-CoV-N and JMDC datasets are heterogeneous. Therefore, we opted against merging the datasets, using the K-CoV-N data for the main cohort and the JMDC data for the replication cohort. In addition, we used different lists of the covariates for each main and replication cohort due to the difference in data structure.

Supplementary material: JMDC cohort/Data source and study design

- Japan has health insurance provided by the universal insurance system.¹ The JMDC has agreements with over 60 insurance providers and includes health insurance claims records of insured individuals, mostly employees of relatively large companies in Japan.² JMDC has generated a database, using data collected from medical institutions in Japan, consisting of patient-level data (unique identifier, family identifiers, relationship to the insured individual, age, sex) and claims for inpatient and outpatient treatment (disease class according to International Classification of Diseases [ICD]-10 code, prescribed drugs based on Anatomical Therapeutic Chemical class, and drug dosage form), diagnosis or therapeutic procedure, institutional information (hospital character), and health checkup (i.e., body mass index, blood pressure, clinical laboratory test, medication status, and self-administered questionnaire such as smoking status, alcohol consumption, and physical activity).³ For more information, please see this webpage (<https://www.jmdc.co.jp/en/jmdc-claims-database/>).

References (Manuscript)

10 Shin, Y. H. et al. Autoimmune inflammatory rheumatic diseases and COVID-19 outcomes in South Korea: a nationwide cohort study. *The Lancet Rheumatology* 3, e698-e706 (2021).

11 Park, B., Sung, J., Park, K., Seo, S. & Kim, S. Report of the evaluation for validity of discharged diagnoses in Korean Health Insurance database. Seoul: Seoul National University, 19-52 (2003).

47 Sato, M. *et al.* Effect of periodontal therapy on glycaemic control in type 2 diabetes. *J Clin Periodontol* (2024). <https://doi.org/10.1111/jcpe.13939>

References (Supplementary material)

1 Reich, M. R., Ikegami, N., Shibuya, K. & Takemi, K. 50 years of pursuing a healthy society in Japan. *Lancet* 378, 1051-1053 (2011). [https://doi.org/10.1016/s0140-6736\(11\)60274-2](https://doi.org/10.1016/s0140-6736(11)60274-2)

2 Setogawa, N., Ohbe, H., Isogai, T., Matsui, H. & Yasunaga, H. Characteristics and short-term outcomes of outpatient and inpatient cardiac catheterizations: A descriptive study using a nationwide claim database in Japan. *J Cardiol* 82, 201-206 (2023). <https://doi.org/10.1016/j.jjcc.2023.05.010>

3 Kaneko, H. *et al.* Medication-Naïve Blood Pressure and Incident Cancers: Analysis of 2 Nationwide Population-Based Databases. *Am J Hypertens* 35, 731-739 (2022). <https://doi.org/10.1093/ajh/hpac054>

Table S3. HR (95% CI) for the **long-term post-acute respiratory sequelae** or **short-term acute respiratory complication** in **negative control analysis** using non-COVID-19 disease (tympanic membrane perforation) in the propensity score-matched main cohort (South Korea) and replication cohort (Japan)

Cohort	South Korea			Japan		
	Events, n (%)	HR (95% CI)		Events, n (%)	HR (95% CI)	
Model 1*		Model 2†	Model 3*		Model 4‡	
Tympanic membrane perforation						
Comparators	77 (0.0040)	1.0 (reference)	1.0 (reference)	654 (0.028)	1.0 (reference)	1.0 (reference)
Patients with COVID-19	14 (0.0036)	0.91 (0.51-1.60)	0.72 (0.33-1.56)	294 (0.038)	1.08 (0.69-1.73)	0.94 (0.52-1.80)

CCI, Charlson comorbidity index; CI, confidence interval; HR, hazard ratio.

***Model 1 and 3:** Adjusted for age (20–39, 40–59, and ≥60 years) and sex.

†**Model 2:** Adjusted for age (20–39, 40–59, and ≥60 years); sex, household income (low income, middle income, and high income); region of residence (urban and rural); CCI score (0, 1, and ≥2); obesity (underweight [$<18.5 \text{ kg/m}^2$], normal [$18.5\text{--}22.9 \text{ kg/m}^2$]; overweight [$23.0\text{--}24.9 \text{ kg/m}^2$], obese [$\geq 25.0 \text{ kg/m}^2$], and unknown); blood pressure (systolic blood pressure $<140 \text{ mmHg}$ and diastolic blood pressure $<90 \text{ mmHg}$, systolic blood pressure $\geq 140 \text{ mmHg}$ or diastolic blood pressure $\geq 90 \text{ mmHg}$, and unknown); fasting blood glucose (<100 , $\geq 100 \text{ mg/dL}$, and unknown); serum total cholesterol (<200 , $200\text{--}239$, $\geq 240 \text{ mg/dL}$, and unknown); glomerular filtration rate (<60 , $60\text{--}89$, $\geq 90 \text{ mL/min/1.73 m}^2$, and unknown); smoking status (never, former, current smoker, and unknown); alcoholic drinks (<1 , $1\text{--}2$, $3\text{--}4$, ≥ 5 days per week, and unknown); aerobic physical activity (sufficient, insufficient, and unknown); previous history of cardiovascular disease, and chronic kidney disease; history of medication use for diabetes mellitus, dyslipidemia, and hypertension; and strain of SARS-CoV-2 (original and delta).

‡**Model 4:** Adjusted for age (20–39, 40–59, and ≥60 years); sex; insurance status (insured and dependent); CCI score (0, 1, and ≥ 2); body mass index (underweight [$<18.5 \text{ kg/m}^2$], normal [$18.5\text{--}22.9 \text{ kg/m}^2$], overweight [$23.0\text{--}25.0 \text{ kg/m}^2$], obese [$\geq 25.0 \text{ kg/m}^2$], and unknown); blood pressure (systolic blood pressure $<140 \text{ mmHg}$ and diastolic blood pressure $<90 \text{ mmHg}$, systolic blood pressure $\geq 140 \text{ mmHg}$ or diastolic blood pressure $\geq 90 \text{ mmHg}$, and unknown); fasting blood glucose (<100 , $\geq 100 \text{ mg/dL}$, and unknown); serum total cholesterol (<200 , $200\text{--}239$, $\geq 240 \text{ mg/dL}$, and unknown); glomerular filtration rate (<60 , $60\text{--}89$, $\geq 90 \text{ mL/min/1.73 m}^2$, and unknown); smoking status (non- and current smoker, and unknown); alcoholic drinks (rarely, sometimes, everyday, and unknown); aerobic physical activity (sufficient, insufficient, and unknown); previous history of cardiovascular disease, and chronic kidney disease; history of medication use for diabetes mellitus, dyslipidemia, and hypertension; and strain of SARS-CoV-2 (original and delta).

Comment 12.

How can the risk of acute respiratory complication remain after 6 months (page 16)? By definition that is not acute.

Response:

Thank you for your comment. As we classified the respiratory complications into long-term post-acute respiratory sequelae and acute respiratory complications, we attempted to examine the long-term influence of COVID-19 and influenza on both groups. However, as we read your comment, we thought that we should have made a more thorough decision on the definition of acute respiratory complication. The definition of acute may be vague and differ depending on the disease and symptoms. Through a systematic review, we were able to investigate how other research defined ‘acute’ in terms of various respiratory sequelae. We determined acute respiratory complications as pneumocystis pneumonia, aspergillosis pneumonia, pleural empyema, lung abscess, pneumothorax, acute respiratory failure, and pulmonary embolism shown a month after SARS-CoV-2 or influenza infection.

The table below shows the studies that we consulted on while deciding the time attenuation for examining acute respiratory complications after SARS-CoV-2 influenza infection. This table is for reviewers only.

Author, Year	Title	Country	Outcome	Observation period of acute respiratory complication
Vilar et al, 1999 ¹	The management of Pneumocystis carinii pneumonia	United Kingdom	pneumocystis pneumonia	21 days
Harman, 2021 ²	Aspergillosis Guidelines	-	aspergillosis pneumonia	6 to 12 weeks
Ampofo et al, 2008 ³	Management of Parapneumonic Empyema	USA	pleural empyema	2 to 6 weeks
Sabbulaet al, 2023 ⁴	Lung Abscess	USA	lung abscess	4 weeks
Toufen et al, 2011 ⁵	Follow-up after acute respiratory distress syndrome caused by	Brazil	acute respiratory failure	2 to 3 months

	influenza a (H1N1) virus infection			
Jill Jin, 2023 ⁶	Treatment Duration for Pulmonary Embolism		pulmonary embolism	3 to 6 months

As a result, we decided the observation period of acute respiratory complications as one month from the first diagnosis of SARS-CoV-2 or influenza infection and conducted a reanalysis. Table 5 and Table S21, therefore, do not include the time attenuation effect analysis of acute respiratory complications. The changes in the Tables and manuscript are depicted below.

<References>

1. Vilar FJ, Khoo SH, Walley T. The management of *Pneumocystis carinii* pneumonia. *Br J Clin Pharmacol* 1999; 47(6): 605-9.
2. <https://emedicine.medscape.com/article/296052-guidelines?form=fpf>
3. Ampofo K, Byington C. Management of parapneumonic empyema. *Pediatr Infect Dis J.* 2007 May;26(5):445-6. doi: 10.1097/01.inf.0000261011.23728.dd. PMID: 17468658; PMCID: PMC2330267.
4. Sabbula BR, Rammohan G, Athavale A, et al. Lung Abscess. [Updated 2023 Aug 15]. In: StatPearls [Internet]. Treasure Island (FL): StatPearls Publishing; 2024 Jan-. Available from: <https://www.ncbi.nlm.nih.gov/books/NBK555920/>
5. Toufen C, Costa ELV, Hirota AS, Li HY, Amato MBP, Carvalho CRR. Follow-up after acute respiratory distress syndrome caused by influenza a (H1N1) virus infection. *Clinics* 2011; 66(6): 933-7.
6. Jin J. Treatment Duration for Pulmonary Embolism. *JAMA.* 2015;314(1):98. doi:10.1001/jama.2015.7431

Changes in text:

Methods/Outcomes

Second, the ‘primary outcome’ for acute respiratory complication is the incidence of various respiratory diseases within one month following a diagnosis of SARS-CoV-2.¹⁷⁻²⁰

References (Manuscript)

- 17 Vilar, F., Khoo, S. & Walley, T. The management of *Pneumocystis carinii* pneumonia. *British journal of clinical pharmacology* **47**, 605 (1999).
- 18 Ampofo, K. & Byington, C. Management of parapneumonic empyema. *The Pediatric infectious disease journal* **26**, 445-446 (2007).
<https://doi.org/10.1097/01.inf.0000261011.23728.dd>
- 19 Toufen, C., Jr. *et al.* Follow-up after acute respiratory distress syndrome caused by influenza a (H1N1) virus infection. *Clinics (Sao Paulo, Brazil)* **66**, 933-937 (2011).
<https://doi.org/10.1590/s1807-59322011000600002>
- 20 Jin, J. JAMA PATIENT PAGE. Treatment Duration for Pulmonary Embolism. *Jama* **314**, 98 (2015). <https://doi.org/10.1001/jama.2015.7431>

Table 5. Time attenuation effect analysis of HR (95% CI) for the risk of **long-term post-acute respiratory sequelae** after SARS-CoV-2 infection in South Korea (main cohort) and Japan (replication cohort)

Time	COVID-19 vs. general population	
	Main cohort [†]	Replication cohort
long-term post-acute respiratory sequelae		
<3 months	2.51 (2.38-2.64)	4.40 (4.30-4.51)
3–6 months	1.24 (1.15-1.33)	2.66 (2.57-2.75)
≥6 months	1.10 (1.01-1.19)	2.67 (2.61-2.73)

CCI, Charlson comorbidity index; CI, confidence interval; HR, hazard ratio; SARS-CoV-2, severe acute respiratory syndrome coronavirus 2.

The data in bold indicate significant differences ($P < 0.05$).

[†] **Adjusted HR (main):** Adjusted for age (20–39, 40–59, and ≥60 years); sex, household income (low income, middle income, and high income); region of residence (urban and rural); CCI score (0, 1, and ≥2); obesity (underweight [$<18.5 \text{ kg/m}^2$], normal [$18.5\text{--}22.9 \text{ kg/m}^2$]; overweight [$23.0\text{--}24.9 \text{ kg/m}^2$], obese [$\geq 25.0 \text{ kg/m}^2$], and unknown); blood pressure (systolic blood pressure $<140 \text{ mmHg}$ and diastolic blood pressure $<90 \text{ mmHg}$, systolic blood pressure $\geq 140 \text{ mmHg}$ or diastolic blood pressure $\geq 90 \text{ mmHg}$, and unknown); fasting blood glucose (<100 , $\geq 100 \text{ mg/dL}$, and unknown); serum total cholesterol (<200 , $200\text{--}239$, $\geq 240 \text{ mg/dL}$, and unknown); glomerular filtration rate (<60 , $60\text{--}89$, $\geq 90 \text{ mL/min/1.73 m}^2$, and unknown); smoking status (never, former, current smoker, and unknown); alcoholic drinks (<1 , $1\text{--}2$, $3\text{--}4$, ≥ 5 days per week, and unknown); aerobic physical activity (sufficient, insufficient, and unknown); previous history of cardiovascular disease, and chronic kidney disease; history of medication use for diabetes mellitus, dyslipidemia, and hypertension; and strain of SARS-CoV-2 (original and delta).

^{||} **Adjusted HR (replication):** Adjusted for age (20–39, 40–59, and ≥60 years); sex; insurance status (insured and dependent); CCI score (0, 1, and ≥ 2); body mass index (underweight [$<18.5 \text{ kg/m}^2$], normal [$18.5\text{--}22.9 \text{ kg/m}^2$], overweight [$23.0\text{--}25.0 \text{ kg/m}^2$], obese [$\geq 25.0 \text{ kg/m}^2$], and unknown); blood pressure (systolic blood pressure $<140 \text{ mmHg}$ and diastolic blood pressure $<90 \text{ mmHg}$, systolic blood pressure $\geq 140 \text{ mmHg}$ or diastolic blood pressure $\geq 90 \text{ mmHg}$, and unknown); fasting blood glucose (<100 , $\geq 100 \text{ mg/dL}$, and unknown); serum total cholesterol (<200 , $200\text{--}239$, $\geq 240 \text{ mg/dL}$, and unknown); glomerular filtration rate (<60 , $60\text{--}89$, $\geq 90 \text{ mL/min/1.73 m}^2$, and unknown); smoking status (non- and current smoker, and unknown); alcoholic drinks (rarely, sometimes, everyday, and unknown); aerobic physical activity (sufficient, insufficient, and unknown); previous history of cardiovascular disease, and chronic kidney disease; history of medication use for diabetes mellitus, dyslipidemia, and hypertension; and strain of SARS-CoV-2 (original and delta).

Table S16. Time attenuation effect analysis (COVID-19 vs. influenza) of HR (95% CI) for the risk of the **long-term post-acute respiratory sequelae** after SARS-CoV-2 infection in South Korea (main cohort) and Japan (replication cohort)

Time	COVID-19 vs. influenza	
	Main cohort [†]	Replication cohort
long-term post-acute respiratory sequelae		
<3 months	2.28 (2.01-2.59)	8.88 (7.76-10.16)
3–6 months	1.24 (1.04-1.47)	3.81 (3.27-4.44)
≥6 months	1.01 (0.83-1.23)	3.65 (3.30-4.05)

CCI, Charlson comorbidity index; CI, confidence interval; HR, hazard ratio; SARS-CoV-2, severe acute respiratory syndrome coronavirus 2.

The data in bold indicate significant differences ($P < 0.05$).

[†] **Adjusted HR (main):** Adjusted for age (20–39, 40–59, and ≥60 years); sex, household income (low income, middle income, and high income); region of residence (urban and rural); CCI score (0, 1, and ≥2); obesity (underweight [$<18.5 \text{ kg/m}^2$], normal [$18.5\text{--}22.9 \text{ kg/m}^2$]; overweight [$23.0\text{--}24.9 \text{ kg/m}^2$], obese [$\geq 25.0 \text{ kg/m}^2$], and unknown); blood pressure (systolic blood pressure $<140 \text{ mmHg}$ and diastolic blood pressure $<90 \text{ mmHg}$, systolic blood pressure $\geq 140 \text{ mmHg}$ or diastolic blood pressure $\geq 90 \text{ mmHg}$, and unknown); fasting blood glucose (<100 , $\geq 100 \text{ mg/dL}$, and unknown); serum total cholesterol (<200 , $200\text{--}239$, $\geq 240 \text{ mg/dL}$, and unknown); glomerular filtration rate (<60 , $60\text{--}89$, $\geq 90 \text{ mL/min/1.73 m}^2$, and unknown); smoking status (never, former, current smoker, and unknown); alcoholic drinks (<1 , $1\text{--}2$, $3\text{--}4$, ≥ 5 days per week, and unknown); aerobic physical activity (sufficient, insufficient, and unknown); previous history of cardiovascular disease, and chronic kidney disease; history of medication use for diabetes mellitus, dyslipidemia, and hypertension; and strain of SARS-CoV-2 (original and delta).

^{||} **Adjusted HR (replication):** Adjusted for age (20–39, 40–59, and ≥60 years); sex; insurance status (insured and dependent); CCI score (0, 1, and ≥ 2); body mass index (underweight [$<18.5 \text{ kg/m}^2$], normal [$18.5\text{--}22.9 \text{ kg/m}^2$], overweight [$23.0\text{--}25.0 \text{ kg/m}^2$], obese [$\geq 25.0 \text{ kg/m}^2$], and unknown); blood pressure (systolic blood pressure $<140 \text{ mmHg}$ and diastolic blood pressure $<90 \text{ mmHg}$, systolic blood pressure $\geq 140 \text{ mmHg}$ or diastolic blood pressure $\geq 90 \text{ mmHg}$, and unknown); fasting blood glucose (<100 , $\geq 100 \text{ mg/dL}$, and unknown); serum total cholesterol (<200 , $200\text{--}239$, $\geq 240 \text{ mg/dL}$, and unknown); glomerular filtration rate (<60 , $60\text{--}89$, $\geq 90 \text{ mL/min/1.73 m}^2$, and unknown); smoking status (non- and current smoker, and unknown); alcoholic drinks (rarely, sometimes, everyday, and unknown); aerobic physical activity (sufficient, insufficient, and unknown); previous history of cardiovascular disease, and chronic kidney disease; history of medication use for diabetes mellitus, dyslipidemia, and hypertension; and strain of SARS-CoV-2 (original and delta).

Comment 13.

It needs to be made more clear that people with SARS-CoV-2 infection and influenza were not directly compared. Rather, each group was compared separately with uninfected individuals. The language used in interpretation should reflect this.

Response:

Thank you for your insightful feedback. We originally intended to make a comparison of the influence of COVID-19 and influenza by comparing each group separately with the uninfected group and making further comparisons between the two. However, based on your comments, we realized that our initial analysis provided a limited perspective by not directly comparing the risks of the post-acute respiratory sequelae between participants with SARS-CoV-2 infection and participants with influenza. Therefore, we have conducted a new matching of the entire dataset and re-analysis. To aid your understanding, we present our revised study design as shown in the Venn diagram below.

In response to your other comments and those from Reviewer #1 regarding control groups, we divided the re-analysis into two comparisons. First, we investigated the risk of respiratory complications following SARS-CoV-2 infection compared to the general population as a key outcome of this study, presented in the main table. We defined the general population as individuals not infected with SARS-CoV-2, which may include patients with

influenza. We conducted analyses of the hazard ratio (HR) of COVID-19, using the general population as a reference, as shown in Tables 1-5. Second, to incorporate your suggestion, instead of an indirect comparison, we conducted a direct comparison through 1:1 exposure-driven propensity score matching to compare COVID-19 with influenza. To examine the relative severity of COVID-19 in comparison with another contagious viral respiratory disease, the patient group with influenza infection excluded patients with both SARS-CoV-2 and influenza infections. We conducted analyses of the HR of COVID-19, using patients with influenza as a reference. The results of these analyses have been added to supplementary Tables S11-S14 and S16. Furthermore, we have also revised the Methods and Results sections accordingly. For your information, the overall findings of our study through re-analysis are consistent with the findings of the original version. Once again, we sincerely appreciate the opportunity to improve our manuscript.

Changes in text:

Abstract

- We aimed to identify the risk of acute respiratory complications or long-term post-acute respiratory sequelae in **long COVID**.
- After exposure-driven 1:5 propensity score matching, we found that the risk of acute respiratory complication or long-term respiratory sequelae is significantly increased in people with SARS-CoV-2 infection **compared to the general population (acute respiratory complication: HR, 8.06 [95% CI, 6.92-9.38]; long-term respiratory sequelae: HR, 1.68 [95% CI, 1.62-1.75])**, and the risk increased with increasing COVID-19 severity. The SARS-CoV-2 infection induced a significantly increased risk for several acute respiratory complications **(aspergillosis pneumonia, pneumothorax, acute respiratory failure, and pulmonary embolism)** and long-term respiratory sequelae **(chronic respiratory failure, COPD, emphysema, asthma, and interstitial lung disease)**.
- Through this large-scale, binational, and population-based cohort study, acute or long-term respiratory sequelae in long COVID were observed **compared to the general population**.

Methods/Exposures

- **To examine the relative severity of COVID-19 in comparison with another contagious viral respiratory disease, additional exposure to influenza infection was defined. It refers to cases diagnosed through an RT-PCR assay or antigen test on nasal and pharyngeal swabs during the**

observation period. For individuals infected with both SARS-CoV-2 and influenza, it includes instances of influenza infection developing after the SARS-CoV-2 infection.

Methods/Propensity score matching

- To enhance the robustness and generalizability of our primary findings and balance baseline covariates, we employed exposure-driven propensity score matching. This approach compared individuals with SARS-CoV-2 infection to those without infection as a general population.²² The propensity score was calculated by using a logistic regression model, adjusted for age (20–39, 40–59, and ≥60 years), sex (male and female), region of residence (urban and rural), history of cardiovascular and chronic disease, and medication use for diabetes, hyperlipidemia, and hypertension. Individuals were paired in 1:5 ratio between the exposure group (SARS-CoV-2) and the non-exposure group. Through the prior procedures, we generated multi-to-one matched cohorts utilized a ‘greedy nearest-neighbor’ algorithm, maintaining a caliper width of 0.001 standard deviations. The quality of the match was evaluated through the standardized mean differences (SMD), with an SMD less than 0.1 signifying minimal imbalances between the groups.²² In addition, to investigate the relative severity of COVID-19 compared to other infectious viral respiratory diseases, an influenza group within the general population was utilized as another control group, directly matching SARS-CoV-2 infections at a 1:1 ratio.

Results

- **Table 1** shows the baseline characteristics of 1:5 propensity score matched cohort of South Korea. After 1:5 propensity score matching based on SARS-CoV-2 infection, we identified 82.9% (1,918,150/2,312,748) of participants without SARS-CoV-2 infection and 17.1% (394,598/2,312,748) of participants with SARS-CoV-2 infection, respectively.

- In the 1:3 propensity score matched replication cohort, 74.4% (2,318,505/3,115,606) of participants without SARS-CoV-2 infection and 25.6% (797,101/3,115,606) of participants with SARS-CoV-2 infection were included in our final analyses (**Table S10**).

- In the main and replication cohorts, individuals with SARS-CoV-2 infection had a higher adjusted HR for long-term post-acute respiratory sequelae compared to the general population (main: HR, 1.68 [95% CI, 1.62-1.75]; replication: HR, 3.32 [95% CI, 3.27-3.37]) in **Table 2**. Furthermore, patients with SARS-CoV-2 infection had an increased risk for acute respiratory complication compared to non-infected controls (main: HR, 8.06 [95% CI, 6.92-9.38]; replication: HR, 4.17 [95% CI, 3.90-4.45]). When directly comparing the risk for acute

respiratory complication between SARS-CoV-2 and influenza infections, SARS-CoV-2 infection was significantly associated with an increased risk (main: HR, 4.32 [95% CI, 2.73-6.83]; replication: HR, 6.51 [95% CI, 5.38-7.87]) in **Tables S11-S13**.

- Relative to the general population, patients with SARS-CoV-2 infection had significantly increased risk for several subtypes of long-term respiratory sequelae, including chronic respiratory failure (main: HR, 8.92 [95% CI, 4.92-16.17]; replication: HR, 7.55 [95% CI, 6.35-8.97]), COPD, emphysema, asthma, pulmonary sarcoidosis, and interstitial lung disease (main: HR, 10.38 [95% CI, 8.75-12.31]; replication: HR, 4.75 [95% CI, 4.54-4.97]) in **Table 3**. Notably, the risk for acute respiratory complication, including aspergillosis pneumonia (main: HR, 6.85 [95% CI, 3.48-13.50]; replication: HR, 4.97 [95% CI, 4.26-5.79]), pneumothorax, acute respiratory failure (main: HR, 112.04 [95% CI, 64.00-196.16]; replication: HR, 6.49 [95% CI, 6.32-6.65]) showed an increase in patients with SARS-CoV-2 infection compared to the general population. This tendency of increased risk for several subtypes of respiratory diseases was also shown when compared to patients with influenza infection (**Table S14**).

- The risk of acute respiratory complication showed decreasing trends according to the number of SARS-CoV-2 vaccinations from individuals after once receiving vaccination (HR, 0.51 [95% CI, 0.38-0.68]) to those with two or more vaccinations (HR, 0.24 [95% CI, 0.19-0.30]). Interestingly, mixed types of vaccination showed the lowest risk of developing long-term respiratory sequelae of all SARS-CoV-2 vaccination methods (HR, 0.18 [95% CI, 0.08-0.38]). The risks of acute respiratory complication were higher in patients with moderate to severe COVID-19 symptoms (HR, 39.54 [95% CI, 33.54-46.62]). Both the original strain and the delta variant of SARS-CoV-2 were shown to have a higher risk of acute respiratory complications (original strain: HR, 9.21 [95% CI, 7.19-11.80]; delta strain: HR, 7.44 [95% CI, 6.13-9.03]). In addition, the risk for long-term post-acute respiratory sequelae also exhibited a similar pattern (**Tables 4 and S15**).

- **Table 5** shows the risk of developing acute respiratory complication or long-term post-acute respiratory sequelae based on how long it has been since the participant was infected with SARS-CoV-2 compared to the general population.

- The first three months after infection with SARS-CoV-2 had the highest risk of developing long-term respiratory sequelae (main: HR, 2.51 [95% CI, 2.38-2.64]; replication: HR, 4.40 [95% CI, 4.30-4.51]). With increasing duration post-SARS-CoV-2 infection, the risk of long-term post-acute respiratory sequelae significantly decreased, but the risk remained even after 6 months (main: HR, 1.10 [95% CI, 1.01-1.19]; replication: HR, 2.67 [95% CI, 2.61-2.73]). HR

of time attenuation effect after SARS-CoV-2 infection showed significance compared to influenza infection likewise (**Table S16**).

Discussion/Findings of this study

- First, the risk of acute respiratory complication or long-term post-acute respiratory sequelae is significantly increased in participants with SARS-CoV-2 infection, compared to the general population. Second, SARS-CoV-2 infection induced a significantly increased risk for several specific long-term respiratory sequelae, including chronic respiratory failure, COPD, emphysema, asthma, and interstitial lung disease, compared to the general. In addition, several acute respiratory complications, including aspergillosis pneumonia, pneumothorax, acute respiratory failure, and pulmonary embolism, also depicted a notable increase in risk after SARS-CoV-2 infection, compared to the general population.

Table 1. Baseline characteristics for 1:5 propensity score–matched cohort (COVID-19 vs. general population) in South Korea (main)

Characteristic	COVID-19 vs. general population (n=2,312,748)		SMD*
	COVID-19 (n=394,598)	General population (n=1,918,150)	
Mean age (SD), y	47.5 (16.8)	46.8 (14.3)	0.046
Age, n (%)			0.026
20–39 y	143,273 (36.3)	702,322 (36.6)	
40–59 y	145,169 (36.8)	709,343 (37.0)	
≥60 y	106,156 (26.9)	506,485 (26.4)	
Sex, n (%)			0.006
Male	205,058 (52.0)	997,982 (52.0)	
Female	189,540 (48.0)	920,168 (48.0)	
Region of residence, n (%)			<0.001
Urban	213,052 (54.0)	1,035,261 (54.0)	
Rural	181,546 (46.0)	882,889 (46.0)	
Medical history, n (%)			
Cardiovascular disease	59,947 (15.2)	286,623 (14.9)	0.009
Chronic kidney disease	18,963 (4.8)	88,687 (4.6)	0.005
Medication use for diabetes	71,625 (18.2)	342,656 (17.9)	0.008
Medication use for hyperlipidemia	59,947 (15.2)	286,623 (14.9)	0.007
Medication use for hypertension	32,402 (8.2)	154,789 (8.1)	0.005
Unmatching covariates, n (%)[†]			
Charlson Comorbidity Index score			0.230
0	346,579 (87.8)	1,806,906 (94.2)	
1	30,584 (7.8)	60,556 (3.2)	
≥2	17,435 (4.4)	50,688 (2.6)	

Household income			<0.001
Low (0th–39th percentile)	182,632 (46.3)	887,593 (46.3)	
Middle (40th –79th percentile)	140,084 (35.5)	681,180 (35.5)	
High (80th–100th percentile)	71,882 (18.2)	349,377 (18.2)	
Body mass index			1.212
Underweight (<18.5 kg/m ²)	6875 (1.7)	67,970 (3.5)	
Normal (18.5-22.9 kg/m ²)	77,779 (19.7)	685,687 (35.8)	
Overweight (23.0-24.9 kg/m ²)	54,427 (13.8)	440,197 (23.0)	
Obese (≥25.0 kg/m ²)	95,082 (24.1)	724,074 (37.8)	
Unknown	160,435 (40.7)	222 (0.012)	
Blood pressure			1.157
SBP <140 mmHg and DBP <90 mmHg	199,342 (50.5)	1,673,777 (87.3)	
SBP ≥140mmHg or DBP ≥90 mmHg	33,711 (8.5)	240,991 (12.6)	
Unknown	161,545 (40.9)	3382 (0.2)	
Fasting blood glucose			1.179
<100 mg/dL	140,656 (35.7)	1,189,021 (62.0)	
≥100 mg/dL	92,374 (23.4)	725,682 (37.8)	
Unknown	161568 (40.9)	3447 (0.2)	
Serum total cholesterol			0.416
<200 mg/dL	67,886 (17.2)	525,684 (27.4)	
200 to 239 mg/dL	39,508 (10.0)	316,959 (16.5)	
≥240 mg/dL	16,860 (4.3)	134,405 (7.0)	
Unknown	270,344 (68.5)	941,102 (49.1)	
Glomerular filtration rate			1.180
<60 mL/min/1.73 m ²	8332 (2.1)	52,421 (2.7)	
60 to 89 mL/min/1.73 m ²	102,604 (26.0)	812,214 (42.3)	
≥90 mL/min/1.73 m ²	121,914 (30.9)	1,048,318 (54.7)	

Unknown	161,748 (41.0)	5197 (0.3)	
Smoking status			1.191
Never	154,105 (39.1)	1,214,221 (63.3)	
Former	42,814 (10.9)	299,027 (15.6)	
Current	37,263 (9.4)	404,323 (21.1)	
Unknown	160,416 (40.7)	579 (0.030)	
Alcohol consumption			1.156
<1 day/week	136,870 (34.7)	1,143,488 (59.6)	
1 to 2 days/week	66,311 (16.8)	545,090 (28.4)	
3 to 4 days/week	23,113 (5.9)	173,245 (9.0)	
≥5 days/week	7893 (2.0)	55,687 (2.9)	
Unknown	160,411 (40.7)	640 (0.034)	
Aerobic physical activity			1.179
Insufficient	118,792 (30.1)	959,088 (50.0)	
Sufficient	115,321 (29.2)	958,194 (50.0)	
Unknown	160,485 (40.7)	868 (0.1)	
Strain of SARS-CoV-2			0.004
Original	121,521 (30.8)	594,134 (31.0)	
Delta	273,077 (69.2)	1,324,016 (69.0)	

DBP, diastolic blood pressure; SARS-CoV-2, severe acute respiratory syndrome coronavirus 2; SBP, systolic blood pressure; SD, standard deviation; SMD, standardized mean difference.

* An SMD <0.1 indicates no significant imbalance. All SMDs were <0.100 in the propensity score–matched cohorts.

† Unmatched covariates were included as adjustment factors in statistical analyses.

Table 2. Hazard ratio (95% CI) for the **long-term post-acute respiratory sequelae** or **short-term acute respiratory complication** after SARS-CoV-2 infection in the propensity score-matched cohorts of South Korea (main) and Japan (replication)

Cohort	South Korea			Japan		
	COVID-19 vs. general population (n=2,312,748)			COVID-19 vs. general population (n=3,115,606)		
	Events, n (%)	HR (95% CI)		Events, n (%)	HR (95% CI)	
Model 1*		Model 2†	Model 3*		Model 4‡	
Long-term post-acute respiratory sequelae						
Comparators (general population or patients with influenza)	16,122 (0.84)	1.0 (reference)	1.0 (reference)	35300 (1.52)	1.0 (reference)	1.0 (reference)
Patients with COVID-19	5292 (1.34)	1.64 (1.59-1.69)	1.68 (1.62-1.75)	41074 (5.15)	3.50 (3.45-3.55)	3.32 (3.27-3.37)
Acute respiratory complication						
Comparators	331 (0.017)	1.0 (reference)	1.0 (reference)	1468 (0.06)	1.0 (reference)	1.0 (reference)
Patients with COVID-19	618 (0.16)	9.70 (8.46-11.11)	8.06 (6.92-9.38)	2304 (0.29)	4.60 (4.31-4.91)	4.17 (3.90-4.45)

CCI, Charlson comorbidity index; CI, confidence interval; HR, hazard ratio.

The data in bold indicate significant differences ($P < 0.05$).

***Model 1 and 3:** Adjusted for age (20–39, 40–59, and ≥ 60 years) and sex.

†**Model 2:** Adjusted for age (20–39, 40–59, and ≥ 60 years); sex, household income (low income, middle income, and high income); region of residence (urban and rural); CCI score (0, 1, and ≥ 2); obesity (underweight [$< 18.5 \text{ kg/m}^2$], normal [$18.5\text{--}22.9 \text{ kg/m}^2$]; overweight [$23.0\text{--}24.9 \text{ kg/m}^2$], obese [$\geq 25.0 \text{ kg/m}^2$], and unknown); blood pressure (systolic blood pressure $< 140 \text{ mmHg}$ and diastolic blood pressure $< 90 \text{ mmHg}$, systolic blood pressure $\geq 140 \text{ mmHg}$ or diastolic blood pressure $\geq 90 \text{ mmHg}$, and unknown); fasting blood glucose (< 100 , $\geq 100 \text{ mg/dL}$, and unknown); serum total cholesterol (< 200 , $200\text{--}239$, $\geq 240 \text{ mg/dL}$, and unknown); glomerular filtration rate (< 60 , $60\text{--}89$, $\geq 90 \text{ mL/min/1.73 m}^2$, and unknown); smoking status (never, former, current smoker, and unknown); alcoholic drinks (< 1 , $1\text{--}2$, $3\text{--}4$, ≥ 5 days per week, and unknown); aerobic physical activity (sufficient, insufficient, and unknown); previous history of cardiovascular disease, and chronic kidney disease; history of medication use for diabetes mellitus, dyslipidemia, and hypertension; and strain of SARS-CoV-2 (original and delta).

‡**Model 4:** Adjusted for age (20–39, 40–59, and ≥ 60 years); sex; insurance status (insured and dependent); CCI score (0, 1, and ≥ 2); body mass index (underweight [$< 18.5 \text{ kg/m}^2$], normal [$18.5\text{--}22.9 \text{ kg/m}^2$], overweight [$23.0\text{--}25.0 \text{ kg/m}^2$], obese [$\geq 25.0 \text{ kg/m}^2$], and unknown); blood pressure (systolic blood pressure $< 140 \text{ mmHg}$ and diastolic blood pressure $< 90 \text{ mmHg}$, systolic blood pressure $\geq 140 \text{ mmHg}$ or diastolic blood

pressure ≥ 90 mmHg, and unknown); fasting blood glucose (< 100 , ≥ 100 mg/dL, and unknown); serum total cholesterol (< 200 , 200–239, ≥ 240 mg/dL, and unknown); glomerular filtration rate (< 60 , 60–89, ≥ 90 mL/min/1.73 m², and unknown); smoking status (non- and current smoker, and unknown); alcoholic drinks (rarely, sometimes, everyday, and unknown); aerobic physical activity (sufficient, insufficient, and unknown); previous history of cardiovascular disease, and chronic kidney disease; history of medication use for diabetes mellitus, dyslipidemia, and hypertension; and strain of SARS-CoV-2 (original and delta).

Table 3. HR (95% CI) for the long-term post-acute respiratory sequelae or short-term acute respiratory complication subtypes after SARS-CoV-2 infection in the propensity score-matched cohorts in South Korea (main) and Japan (replication)

Cohort	South Korea			Japan		
	COVID-19 vs. general population (n=2,312,748)			COVID-19 vs. general population (n=3,115,606)		
	Events, n (%)	HR (95% CI)		Events, n (%)	HR (95% CI)	
Model 1*		Model 2†	Model 3*		Model 4	
Long-term post-acute respiratory sequelae						
Chronic respiratory failure						
Comparators (general population or patients with influenza)	18 (0.00094)	1.0 (reference)	1.0 (reference)	170 (0.0073)	1.0 (reference)	1.0 (reference)
Patients with COVID-19	46 (0.012)	12.80 (7.42-22.07)	8.92 (4.92-16.17)	688 (0.086)	11.85 (10.02-14.02)	7.55 (6.35-8.97)
Pulmonary hypertension						
Comparators	15 (0.00078)	1.0 (reference)	1.0 (reference)	156 (0.0067)	1.0 (reference)	1.0 (reference)
Patients with COVID-19	3 (0.00076)	1.00 (0.29-3.45)	0.60 (0.11-3.39)	217 (0.027)	4.07 (3.31-5.00)	3.11 (2.51-3.85)
Sleep apnea						
Comparators	1143 (0.060)	1.0 (reference)	1.0 (reference)	6643 (0.29)	1.0 (reference)	1.0 (reference)
Patients with COVID-19	235 (0.060)	1.02 (0.89-1.17)	1.13 (0.95-1.33)	5198 (0.65)	2.30 (2.22-2.39)	2.21 (2.13-2.29)
COPD						
Comparators	10846 (0.57)	1.0 (reference)	1.0 (reference)	11003 (0.47)	1.0 (reference)	1.0 (reference)
Patients with COVID-19	3359 (0.85)	1.54 (1.49-1.61)	1.57 (1.50-1.65)	15520 (1.95)	4.17 (4.07-4.27)	3.93 (3.83-4.03)
Emphysema						
Comparators	386 (0.020)	1.0 (reference)	1.0 (reference)	1550 (0.067)	1.0 (reference)	1.0 (reference)
Patients with COVID-19	133 (0.034)	1.73 (1.42-2.10)	1.60 (1.27-2.01)	2085 (0.26)	3.95 (3.70-4.22)	3.44 (3.22-3.68)
Asthma						
Comparators	4197 (0.22)	1.0 (reference)	1.0 (reference)	21314 (0.92)	1.0 (reference)	1.0 (reference)
Patients with COVID-19	1431 (0.36)	1.70 (1.60-1.80)	1.74 (1.62-1.87)	25311 (3.18)	3.53 (3.46-3.59)	3.44 (3.38-3.50)

Pulmonary sarcoidosis

Comparators	29 (0.0015)	1.0 (reference)	1.0 (reference)	972 (0.042)	1.0 (reference)	1.0 (reference)
Patients with COVID-19	4 (0.0010)	0.69 (0.24-1.95)	0.96 (0.34-2.75)	1255 (0.16)	3.78 (3.48-4.11)	3.44 (3.16-3.75)

Interstitial lung disease

Comparators	223 (0.012)	1.0 (reference)	1.0 (reference)	2942 (0.13)	1.0 (reference)	1.0 (reference)
Patients with COVID-19	453 (0.11)	10.13 (8.63-11.90)	10.38 (8.75-12.31)	5996 (0.75)	6.00 (5.74-6.27)	4.75 (4.54-4.97)

Acute respiratory complication**Pneumocystis pneumonia**

Comparators	5 (0.00026)	1.0 (reference)	1.0 (reference)	934 (0.040)	1.0 (reference)	1.0 (reference)
Patients with COVID-19	2 (0.00051)	1.96 (0.38-10.10)	0.03 (0.00-8550.49)	1426 (0.18)	4.46 (4.10-4.84)	3.28 (3.01-3.58)

Aspergillosis pneumonia

Comparators	16 (0.00083)	1.0 (reference)	1.0 (reference)	249 (0.011)	1.0 (reference)	1.0 (reference)
Patients with COVID-19	32 (0.0081)	9.73 (5.34-17.72)	6.85 (3.48-13.50)	601 (0.075)	7.05 (6.08-8.17)	4.97 (4.26-5.79)

Pleural empyema

Comparators	10 (0.00052)	1.0 (reference)	1.0 (reference)	24 (0.0010)	1.0 (reference)	1.0 (reference)
Patients with COVID-19	6 (0.0015)	2.93 (1.06-8.05)	1.45 (0.32-6.63)	226 (0.028)	27.44 (18.02-41.80)	22.00 (14.38-33.65)

Lung abscess

Comparators	17 (0.00089)	1.0 (reference)	1.0 (reference)	57 (0.0025)	1.0 (reference)	1.0 (reference)
Patients with COVID-19	7 (0.0018)	2.01 (0.83-4.85)	2.20 (0.81-6.00)	301 (0.038)	15.39 (11.60-20.43)	13.57 (10.19-18.07)

Pneumothorax

Comparators	43 (0.0022)	1.0 (reference)	1.0 (reference)	3818 (0.16)	1.0 (reference)	1.0 (reference)
Patients with COVID-19	50 (0.013)	5.69 (3.78-8.55)	5.29 (3.32-8.42)	3234 (0.41)	2.49 (2.37-2.60)	2.41 (2.30-2.53)

Acute respiratory failure

Comparators	13 (0.00068)	1.0 (reference)	1.0 (reference)	8767 (0.38)	1.0 (reference)	1.0 (reference)
Patients with COVID-19	363 (0.092)	135.7 (78.05-235.91)	112.04 (64.00-196.16)	20983 (2.63)	7.10 (6.92-7.28)	6.49 (6.32-6.65)

Pulmonary embolism

Comparators	209 (0.011)	1.0 (reference)	1.0 (reference)	2212 (0.10)	1.0 (reference)	1.0 (reference)
Patients with COVID-19	162 (0.041)	3.79 (3.08-4.65)	2.98 (2.32-3.82)	3972 (0.50)	5.26 (5.00-5.55)	4.58 (4.34-4.83)

CCI, Charlson comorbidity index; CI, confidence interval; COPD, chronic obstructive pulmonary disease; HR, hazard ratio; NA, not available.

The data in bold indicate significant differences ($P < 0.05$).

***Model 1 and 3:** Adjusted for age (20–39, 40–59, and ≥ 60 years) and sex.

†**Model 2:** Adjusted for age (20–39, 40–59, and ≥ 60 years); sex, household income (low income, middle income, and high income); region of residence (urban and rural); CCI score (0, 1, and ≥ 2); obesity (underweight [$< 18.5 \text{ kg/m}^2$], normal [$18.5\text{--}22.9 \text{ kg/m}^2$]; overweight [$23.0\text{--}24.9 \text{ kg/m}^2$], obese [$\geq 25.0 \text{ kg/m}^2$], and unknown); blood pressure (systolic blood pressure $< 140 \text{ mmHg}$ and diastolic blood pressure $< 90 \text{ mmHg}$, systolic blood pressure $\geq 140 \text{ mmHg}$ or diastolic blood pressure $\geq 90 \text{ mmHg}$, and unknown); fasting blood glucose (< 100 , $\geq 100 \text{ mg/dL}$, and unknown); serum total cholesterol (< 200 , $200\text{--}239$, $\geq 240 \text{ mg/dL}$, and unknown); glomerular filtration rate (< 60 , $60\text{--}89$, $\geq 90 \text{ mL/min/1.73 m}^2$, and unknown); smoking status (never, former, current smoker, and unknown); alcoholic drinks (< 1 , $1\text{--}2$, $3\text{--}4$, ≥ 5 days per week, and unknown); aerobic physical activity (sufficient, insufficient, and unknown); previous history of cardiovascular disease, and chronic kidney disease; history of medication use for diabetes mellitus, dyslipidemia, and hypertension; and strain of SARS-CoV-2 (original and delta).

‡ **Model 4:** Adjusted for age (20–39, 40–59, and ≥ 60 years); sex; insurance status (insured and dependent); CCI score (0, 1, and ≥ 2); body mass index (underweight [$< 18.5 \text{ kg/m}^2$], normal [$18.5\text{--}22.9 \text{ kg/m}^2$], overweight [$23.0\text{--}25.0 \text{ kg/m}^2$], obese [$\geq 25.0 \text{ kg/m}^2$], and unknown); blood pressure (systolic blood pressure $< 140 \text{ mmHg}$ and diastolic blood pressure $< 90 \text{ mmHg}$, systolic blood pressure $\geq 140 \text{ mmHg}$ or diastolic blood pressure $\geq 90 \text{ mmHg}$, and unknown); fasting blood glucose (< 100 , $\geq 100 \text{ mg/dL}$, and unknown); serum total cholesterol (< 200 , $200\text{--}239$, $\geq 240 \text{ mg/dL}$, and unknown); glomerular filtration rate (< 60 , $60\text{--}89$, $\geq 90 \text{ mL/min/1.73 m}^2$, and unknown); smoking status (non- and current smoker, and unknown); alcoholic drinks (rarely, sometimes, everyday, and unknown); aerobic physical activity (sufficient, insufficient, and unknown); previous history of cardiovascular disease, and chronic kidney disease; history of medication use for diabetes mellitus, dyslipidemia, and hypertension; and strain of SARS-CoV-2 (original and delta).

Table 4. Subgroup analysis (COVID-19 vs. general population) of HR (95% CI) of the **long-term post-acute respiratory sequelae** or **short-term acute respiratory complication** after SARS-CoV-2 infection stratified by vaccination, COVID-19 severity, and SARS-CoV-2 strain in the cohort of South Korea (main)

Variable	Events/total, n/N (%)	HR (95% CI)	
		Model 1*	Model 2†
Long-term post-acute respiratory sequelae			
Number of SARS-CoV-2 vaccinations			
Non-infected control	16,122/1,918,150 (0.84)	0.62 (0.60-0.65)	0.60 (0.58-0.62)
COVID-19 without SARS-CoV-2 vaccination	4331/200,539 (2.16)	1.0 (reference)	1.0 (reference)
COVID-19 after SARS-CoV-2 vaccination received once	493/38,852 (1.27)	0.90 (0.82-0.99)	0.85 (0.77-0.94)
COVID-19 after SARS-CoV-2 vaccination received twice or more	468/155,207 (0.30)	0.69 (0.62-0.76)	0.64 (0.57-0.71)
Type of SARS-CoV-2 vaccinations			
Non-infected control	16,122/1,918,150 (0.84)	0.62 (0.60-0.65)	0.60 (0.58-0.62)
COVID-19 without SARS-CoV-2 vaccination	4331/200,539 (2.16)	1.0 (reference)	1.0 (reference)
COVID-19 with viral vector SARS-CoV-2 vaccination	465/109,066 (0.43)	1.12 (1.01-1.23)	0.99 (0.90-1.10)
COVID-19 with mRNA SARS-CoV-2 vaccination	477/66,891 (0.71)	1.20 (1.09-1.33)	1.11 (0.99-1.22)
COVID-19 with both types of SARS-CoV-2 vaccination	19/18,102 (0.10)	0.74 (0.47-1.16)	0.66 (0.42-1.03)
COVID-19 severity			
Non-infected control	16,122/1,918,150 (0.84)	1.0 (reference)	1.0 (reference)
Mild COVID-19	3492/340,813 (1.02)	1.28 (1.24-1.33)	1.37 (1.32-1.43)
Moderate to severe COVID-19	1800/53785 (3.35)	3.60 (3.43-3.78)	3.20 (3.03-3.38)
Original strain of SARS-CoV-2 (overall population)			
Non-infected control before the delta-dominant phase [§]	11,667/594,134 (1.96)	1.0 (reference)	1.0 (reference)

Infection with original strain	3649/121,521 (3.00)	1.58 (1.52-1.64)	1.59 (1.52-1.66)
Delta variant of SARS-CoV-2 (overall population)			
Non-infected control during the delta-dominant phase [§]	4455/1,324,016 (0.34)	1.0 (reference)	1.0 (reference)
Infection with Delta variant	1643/273,077 (0.60)	1.81 (1.71-1.92)	1.94 (1.81-2.08)
Acute respiratory complication			
Number of SARS-CoV-2 vaccinations			
Non-infected control	311/1,918,150 (0.016)	0.07 (0.06-0.08)	0.08 (0.06-0.09)
COVID-19 without SARS-CoV-2 vaccination	415/200,539 (0.21)	1.0 (reference)	1.0 (reference)
COVID-19 after SARS-CoV-2 vaccination received once	56/38,852 (0.14)	0.62 (0.47-0.83)	0.51 (0.38-0.68)
COVID-19 after SARS-CoV-2 vaccination received twice or more	147/155,207 (0.09)	0.32 (0.26-0.39)	0.24 (0.19-0.30)
Type of SARS-CoV-2 vaccinations			
Non-infected control	311/1,918,150 (0.016)	0.07 (0.06-0.08)	0.08 (0.06-0.09)
COVID-19 without SARS-CoV-2 vaccination	415/200,539 (0.21)	1.0 (reference)	1.0 (reference)
COVID-19 with viral vector SARS-CoV-2 vaccination	79/109,066 (0.072)	0.44 (0.34-0.56)	0.36 (0.28-0.47)
COVID-19 with mRNA SARS-CoV-2 vaccination	117/66,891 (0.17)	0.40 (0.32-0.50)	0.42 (0.33-0.53)
COVID-19 with both types of SARS-CoV-2 vaccination	7/18,102 (0.039)	0.19 (0.09-0.41)	0.18 (0.08-0.38)
COVID-19 severity			
Non-infected control	311/1,918,150 (0.016)	1.0 (reference)	1.0 (reference)
Mild COVID-19	50/340,813 (0.015)	0.95 (0.71-1.28)	0.99 (0.73-1.34)
Moderate to severe COVID-19	568/53,783 (1.06)	51.18 (44.38-59.02)	39.54 (33.54-46.62)
Original strain of SARS-CoV-2 (overall population)			
Non-infected control before the delta-dominant phase [§]	110/594,134 (0.019)	1.0 (reference)	1.0 (reference)
Infection with original strain	230/121,521 (0.19)	10.27 (8.18-12.88)	9.21 (7.19-11.80)

Delta variant of SARS-CoV-2 (overall population)

Non-infected control during the delta-dominant phase §	201/1,324,016 (0.015)	1.0 (reference)	1.0 (reference)
Infection with Delta variant	388/273,077 (0.14)	9.39 (7.92-11.14)	7.44 (6.13-9.03)

CCI, Charlson comorbidity index; CI, confidence interval; HR, hazard ratio; SARS-CoV-2, severe acute respiratory syndrome coronavirus 2.

The data in bold indicate significant differences (P < 0.05).

|| HR of the non-infected control represents the risk of respiratory diseases, and HRs of patients with COVID-19 indicate the risk of post-acute respiratory complications following SARS-CoV-2 infection.

§ Only 1:5-matched comparators in each patient group at the same index date were included to reduce immortal time bias.

***Model 1:** Adjusted for age (20–39, 40–59, and ≥60 years) and sex.

†**Model 2:** Adjusted for age (20–39, 40–59, and ≥60 years); sex, household income (low income, middle income, and high income); region of residence (urban and rural); CCI score (0, 1, and ≥2); obesity (underweight [$<18.5 \text{ kg/m}^2$], normal [$18.5\text{--}22.9 \text{ kg/m}^2$]; overweight [$23.0\text{--}24.9 \text{ kg/m}^2$], obese [$\geq 25.0 \text{ kg/m}^2$], and unknown); blood pressure (systolic blood pressure $<140 \text{ mmHg}$ and diastolic blood pressure $<90 \text{ mmHg}$, systolic blood pressure $\geq 140 \text{ mmHg}$ or diastolic blood pressure $\geq 90 \text{ mmHg}$, and unknown); fasting blood glucose (<100 , $\geq 100 \text{ mg/dL}$, and unknown); serum total cholesterol (<200 , $200\text{--}239$, $\geq 240 \text{ mg/dL}$, and unknown); glomerular filtration rate (<60 , $60\text{--}89$, $\geq 90 \text{ mL/min/1.73 m}^2$, and unknown); smoking status (never, former, current smoker, and unknown); alcoholic drinks (<1 , $1\text{--}2$, $3\text{--}4$, ≥ 5 days per week, and unknown); aerobic physical activity (sufficient, insufficient, and unknown); previous history of cardiovascular disease, and chronic kidney disease; history of medication use for diabetes mellitus, dyslipidemia, and hypertension; and strain of SARS-CoV-2 (original and delta).

Table 5. Time attenuation effect analysis of HR (95% CI) for the risk of **long-term post-acute respiratory sequelae** after SARS-CoV-2 infection in South Korea (main cohort) and Japan (replication cohort)

Time	COVID-19 vs. general population	
	Main cohort [†]	Replication cohort
long-term post-acute respiratory sequelae		
<3 months	2.51 (2.38-2.64)	4.40 (4.30-4.51)

3–6 months	1.24 (1.15-1.33)	2.66 (2.57-2.75)
≥6 months	1.10 (1.01-1.19)	2.67 (2.61-2.73)

CCI, Charlson comorbidity index; CI, confidence interval; HR, hazard ratio; SARS-CoV-2, severe acute respiratory syndrome coronavirus 2.

The data in bold indicate significant differences ($P < 0.05$).

† **Adjusted HR (main):** Adjusted for age (20–39, 40–59, and ≥60 years); sex, household income (low income, middle income, and high income); region of residence (urban and rural); CCI score (0, 1, and ≥2); obesity (underweight [$<18.5 \text{ kg/m}^2$], normal [$18.5\text{--}22.9 \text{ kg/m}^2$]; overweight [$23.0\text{--}24.9 \text{ kg/m}^2$], obese [$\geq 25.0 \text{ kg/m}^2$], and unknown); blood pressure (systolic blood pressure $<140 \text{ mmHg}$ and diastolic blood pressure $<90 \text{ mmHg}$, systolic blood pressure $\geq 140 \text{ mmHg}$ or diastolic blood pressure $\geq 90 \text{ mmHg}$, and unknown); fasting blood glucose (<100 , $\geq 100 \text{ mg/dL}$, and unknown); serum total cholesterol (<200 , $200\text{--}239$, $\geq 240 \text{ mg/dL}$, and unknown); glomerular filtration rate (<60 , $60\text{--}89$, $\geq 90 \text{ mL/min/1.73 m}^2$, and unknown); smoking status (never, former, current smoker, and unknown); alcoholic drinks (<1 , $1\text{--}2$, $3\text{--}4$, ≥ 5 days per week, and unknown); aerobic physical activity (sufficient, insufficient, and unknown); previous history of cardiovascular disease, and chronic kidney disease; history of medication use for diabetes mellitus, dyslipidemia, and hypertension; and strain of SARS-CoV-2 (original and delta).

‖ **Adjusted HR (replication):** Adjusted for age (20–39, 40–59, and ≥60 years); sex; insurance status (insured and dependent); CCI score (0, 1, and ≥ 2); body mass index (underweight [$<18.5 \text{ kg/m}^2$], normal [$18.5\text{--}22.9 \text{ kg/m}^2$], overweight [$23.0\text{--}25.0 \text{ kg/m}^2$], obese [$\geq 25.0 \text{ kg/m}^2$], and unknown); blood pressure (systolic blood pressure $<140 \text{ mmHg}$ and diastolic blood pressure $<90 \text{ mmHg}$, systolic blood pressure $\geq 140 \text{ mmHg}$ or diastolic blood pressure $\geq 90 \text{ mmHg}$, and unknown); fasting blood glucose (<100 , $\geq 100 \text{ mg/dL}$, and unknown); serum total cholesterol (<200 , $200\text{--}239$, $\geq 240 \text{ mg/dL}$, and unknown); glomerular filtration rate (<60 , $60\text{--}89$, $\geq 90 \text{ mL/min/1.73 m}^2$, and unknown); smoking status (non- and current smoker, and unknown); alcoholic drinks (rarely, sometimes, everyday, and unknown); aerobic physical activity (sufficient, insufficient, and unknown); previous history of cardiovascular disease, and chronic kidney disease; history of medication use for diabetes mellitus, dyslipidemia, and hypertension; and strain of SARS-CoV-2 (original and delta).

Table S10. Baseline characteristics for 1:3 propensity score–matched cohort (COVID-19 vs. general population) in Japan (replication)

Characteristic	COVID-19 vs. general population (n=3,115,606)		SMD*
	COVID-19 (n=797,101)	General population (n=2,318,505)	
Mean age (SD), y	44 (11.88)	44 (12.03)	0.034
Age, n (%)			<0.001
20–39 y	302,404 (37.94)	878,119 (37.87)	
40–59 y	411,351 (51.61)	1,199,717 (51.75)	
≥60 y	83,346 (10.46)	240,669 (10.38)	
Sex, n (%)			0.005
Male	495,460 (62.16)	1,447,190 (62.42)	
Female	301,641 (37.84)	871,315 (37.58)	
Insurance status, n (%)			0.001
Insured	718,811 (90.18)	2,091,308 (90.20)	
Dependent	78,290 (9.82)	227,197 (9.80)	
Medical history, n (%)			
Cardiovascular disease	60,450 (7.58)	170,601 (7.36)	0.009
Chronic kidney disease	32,292 (4.05)	90,680 (3.91)	0.007
Medication use for diabetes	19,777 (2.48)	56,919 (2.45)	0.002
Medication use for hyperlipidemia	49,255 (6.18)	141,631 (6.11)	0.003
Medication use for hypertension	67,167 (8.43)	191,951 (8.28)	0.005
Unmatching covariates, n (%)[†]			
Charlson Comorbidity Index score			0.203
0	764,052 (95.85)	2,293,774 (98.93)	
1	10,954 (1.37)	8,640 (0.37)	
≥2	22,095 (2.77)	16,091 (0.69)	

Body mass index			<0.001
Underweight (<18.5 kg/m ²)	427,244 (53.60)	1,263,598 (54.50)	
Normal (18.5-22.9 kg/m ²)	151,460 (19.00)	439,941 (18.98)	
Overweight (23.0-24.9 kg/m ²)	169,962 (21.32)	484,142 (20.88)	
Obese (≥25.0 kg/m ²)	47,210 (5.92)	127,534 (5.50)	
Unknown	1225 (0.15)	3290 (0.14)	
Blood pressure			0.038
SBP <140 mmHg and DBP <90 mmHg	691,676 (86.77)	1,986,000 (85.66)	
SBP ≥140mmHg or DBP ≥90 mmHg	46,474 (5.83)	149,954 (6.47)	
Unknown	58,951 (7.40)	182,551 (7.87)	
Fasting blood glucose			<0.001
<100 mg/dL	511,681 (64.19)	1,474,020 (63.58)	
≥100 mg/dL	160,434 (20.13)	474,726 (20.48)	
Unknown	124,986 (15.68)	369,759 (15.95)	
Serum total cholesterol			0.054
<200 mg/dL	373,377 (46.84)	1,040,983 (44.90)	
200 to 239 mg/dL	285,475 (35.81)	843,032 (36.36)	
≥240 mg/dL	123,823 (15.53)	379,819 (16.38)	
Unknown	14,426 (1.81)	54,671 (2.36)	
Glomerular filtration rate			0.037
<60 mL/min/1.73 m ²	4383 (0.55)	8174 (0.35)	
60 to 89 mL/min/1.73 m ²	72,932 (9.15)	215,112 (9.28)	
≥90 mL/min/1.73 m ²	433,528 (54.39)	1,237,792 (53.39)	
Unknown	286,258 (35.91)	857,427 (36.98)	
Smoking status			0.062
Non-smoker	181,033 (22.71)	554,190 (23.90)	
Smoker	586,552 (73.59)	1,677,800 (72.37)	

Unknown	29,516 (3.70)	86,515 (3.73)	
Alcohol consumption			0.040
Everyday	161,809 (20.30)	466,183 (20.11)	
Sometimes	264,583 (33.19)	762,501 (32.89)	
Rarely	311,632 (39.10)	919,772 (39.67)	
Unknown	59,077 (7.41)	170,049 (7.33)	
Aerobic physical activity			0.046
Insufficient	161,205 (20.22)	497,575 (21.46)	
Sufficient	562,300 (70.54)	1,605,294 (69.24)	
Unknown	73,596 (9.23)	215,636 (9.30)	
Strain of SARS-CoV-2			0.005
Original	335,571 (42.10)	981,871 (42.35)	
Delta	461,530 (57.90)	1,336,634 (57.65)	

DBP, diastolic blood pressure; SARS-CoV-2, severe acute respiratory syndrome coronavirus 2; SBP, systolic blood pressure; SD, standard deviation; SMD, standardized mean difference.

* An SMD <0.1 indicates no significant imbalance. All SMDs were <0.100 in the propensity score–matched cohorts.

† Unmatched covariates were included as adjustment factors in statistical analyses.

Table S11. Baseline characteristics for 1:1 propensity score-matched cohort (COVID-19 vs. influenza) in South Korea (main)

Characteristic	COVID-19 vs. influenza (n=223,000)		SMD*
	COVID-19 (n=111,500)	Influenza (n=111,500)	
Mean age (SD), y	45.3 (15.3)	45.0 (12.7)	0.017
Age, n (%)			0.060
20–39 y	42,625 (38.2)	40,131 (36.0)	
40–59 y	50,713 (45.5)	53,295 (47.8)	
≥60 y	18,162 (16.3)	18,074 (16.2)	
Sex, n (%)			0.046
Male	46,400 (41.6)	49,587 (44.5)	
Female	65,100 (58.4)	61,913 (55.5)	
Region of residence, n (%)			0.005
Urban	49,431 (44.3)	49,160 (44.1)	
Rural	62,069 (55.7)	62,340 (55.9)	
Medical history, n (%)			
Cardiovascular disease	4166 (3.7)	4258 (3.8)	0.004
Chronic kidney disease	2011 (1.8)	1991 (1.8)	0.001
Medication use for diabetes	17,218 (15.4)	16,665 (15.0)	0.014
Medication use for hyperlipidemia	17,656 (15.8)	17,167 (15.4)	0.012
Medication use for hypertension	7408 (6.6)	6640 (6.0)	0.028
Unmatching covariates, n (%)[†]			
Charlson Comorbidity Index score			0.103
0	99,673 (89.4)	103,133 (92.5)	
1	7867 (7.1)	5296 (4.8)	
≥2	3960 (3.6)	3071 (2.8)	

Household income			0.078
Low (0th–39th percentile)	48,878 (43.8)	46,012 (41.3)	
Middle (40th –79th percentile)	45,339 (40.7)	45,119 (40.5)	
High (80th–100th percentile)	17,283 (15.5)	20,369 (18.3)	
Body mass index			1.180
Underweight (<18.5 kg/m ²)	2120 (1.9)	3997 (3.6)	
Normal (18.5-22.9 kg/m ²)	22,894 (20.5)	39,561 (35.5)	
Overweight (23.0-24.9 kg/m ²)	14,256 (12.8)	24,459 (21.9)	
Obese (≥25.0 kg/m ²)	25,159 (22.6)	43,474 (39.0)	
Unknown	47,071 (42.2)	9 (0.0081)	
Blood pressure			1.207
SBP <140 mmHg and DBP <90 mmHg	55,801 (50.1)	99,099 (88.9)	
SBP ≥140mmHg or DBP ≥90 mmHg	8400 (7.5)	12,237 (11.0)	
Unknown	47299 (42.4)	164 (0.15)	
Fasting blood glucose			1.229
<100 mg/dL	40,399 (36.2)	72,206 (64.8)	
≥100 mg/dL	23,797 (21.3)	39,131 (35.1)	
Unknown	47,304 (42.4)	163 (0.15)	
Serum total cholesterol			0.462
<200 mg/dL	17,459 (15.7)	29,991 (26.9)	
200 to 239 mg/dL	10485 (9.4)	18,760 (16.8)	
≥240 mg/dL	4859 (4.4)	7923 (7.1)	
Unknown	78,697 (70.6)	54,826 (49.2)	
Glomerular filtration rate			1.208
<60 mL/min/1.73 m ²	1689 (1.5)	2223 (2.0)	
60 to 89 mL/min/1.73 m ²	26,592 (23.9)	45,211 (40.6)	
≥90 mL/min/1.73 m ²	35,867 (32.2)	63,785 (57.2)	

Unknown	47,352 (42.5)	281 (0.25)	
Smoking status			1.155
Never	45,487 (40.8)	75,040 (67.3)	
Former	9450 (8.5)	15,936 (14.3)	
Current	9506 (8.5)	20,488 (18.4)	
Unknown	47,057 (42.2)	36 (0.032)	
Alcohol consumption			1.208
<1 day/week	38,511 (34.5)	68078 (61.1)	
1 to 2 days/week	17,887 (16.0)	31025 (27.8)	
3 to 4 days/week	6125 (5.5)	9620 (8.6)	
≥5 days/week	1922 (1.7)	2741 (2.5)	
Unknown	47,055 (42.2)	36 (0.032)	
Aerobic physical activity			1.228
Insufficient	33,898 (30.4)	58,183 (52.2)	
Sufficient	30,539 (27.4)	53,276 (47.8)	
Unknown	47,063 (42.2)	41 (0.037)	
Strain of SARS-CoV-2			<0.001
Original	34685 (31.1)	34,718 (31.1)	
Delta	76,815 (68.9)	76,782 (68.9)	

DBP, diastolic blood pressure; SARS-CoV-2, severe acute respiratory syndrome coronavirus 2; SBP, systolic blood pressure; SD, standard deviation; SMD, standardized mean difference.

* An SMD <0.1 indicates no significant imbalance. All SMDs were <0.100 in the propensity score-matched cohorts.

† Unmatched covariates were included as adjustment factors in statistical analyses.

Table S12. Baseline characteristics for 1:1 propensity score-matched cohort (COVID-19 vs. influenza) in Japan (replication)

Characteristic	COVID-19 vs. influenza (n=178,648)		SMD*
	COVID-19 (n=89,324)	Influenza (n=89,324)	
Mean age (SD), y	44 (11.76)	44 (11.57)	0.018
Age, n (%)			<0.001
20–39 y	31,256 (34.99)	31,210 (34.94)	
40–59 y	48,959 (54.81)	49,241 (55.13)	
≥60 y	9109 (10.20)	8873 (9.93)	
Sex, n (%)			0.005
Male	55,617 (62.26)	55,396 (62.02)	
Female	33,707 (37.74)	33,928 (37.98)	
Insurance status, n (%)			<0.001
Insured	80,141 (89.72)	80,141 (89.72)	
Dependent	9183 (10.28)	9183 (10.28)	
Medical history, n (%)			
Cardiovascular disease	6235 (6.98)	6214 (6.96)	0.001
Chronic kidney disease	3031 (3.39)	3252 (3.64)	0.013
Medication use for diabetes	2422 (2.71)	2200 (2.46)	0.016
Medication use for hyperlipidemia	6159 (6.90)	5877 (6.58)	0.013
Medication use for hypertension	8016 (8.97)	7716 (8.64)	0.012
Unmatching covariates, n (%)[†]			
Charlson Comorbidity Index score			0.153
0	85,599 (95.83)	87,875 (98.38)	
1	1150 (1.29)	501 (0.56)	
≥2	2575 (2.88)	948 (1.06)	

Body mass index			0.007
Underweight (<18.5 kg/m ²)	47,532 (53.21)	47,767 (53.48)	
Normal (18.5-22.9 kg/m ²)	17,018 (19.05)	16,935 (18.96)	
Overweight (23.0-24.9 kg/m ²)	19,260 (21.56)	19,131 (21.42)	
Obese (≥25.0 kg/m ²)	5373 (6.02)	5366 (6.01)	
Unknown	125 (0.14)	141 (0.16)	
Blood pressure			<0.001
SBP <140 mmHg and DBP <90 mmHg	77,118 (86.34)	77,019 (86.22)	
SBP ≥140mmHg or DBP ≥90 mmHg	5456 (6.11)	5383 (6.03)	
Unknown	6750 (7.56)	6922 (7.75)	
Fasting blood glucose			0.028
<100 mg/dL	57,035 (63.85)	57,644 (64.53)	
≥100 mg/dL	18,237 (20.42)	18,284 (20.47)	
Unknown	14,052 (15.73)	13,396 (15.00)	
Serum total cholesterol			0.022
<200 mg/dL	41,317 (46.26)	39,896 (44.66)	
200 to 239 mg/dL	32,270 (36.13)	33,092 (37.05)	
≥240 mg/dL	14,211 (15.91)	14,672 (16.43)	
Unknown	1526 (1.71)	1664 (1.86)	
Glomerular filtration rate			0.073
<60 mL/min/1.73 m ²	501 (0.56)	296 (0.33)	
60 to 89 mL/min/1.73 m ²	8344 (9.34)	7987 (8.94)	
≥90 mL/min/1.73 m ²	48,818 (54.65)	46,549 (52.11)	
Unknown	31,661 (35.45)	34,492 (38.61)	
Smoking status			0.062
Non-smoker	20,354 (22.79)	21,458 (24.02)	

Smoker	65,734 (73.59)	64,706 (72.44)	
Unknown	3236 (3.62)	3160 (3.54)	
Alcohol consumption			0.049
Everyday	18,486 (20.70)	18,284 (20.47)	
Sometimes	29,295 (32.80)	28,687 (32.12)	
Rarely	35,096 (39.29)	35,716 (39.98)	
Unknown	6447 (7.22)	6637 (7.43)	
Aerobic physical activity			0.035
Insufficient	18,249 (20.43)	18,303 (20.49)	
Sufficient	62,979 (70.51)	62,858 (70.37)	
Unknown	8096 (9.06)	8163 (9.14)	
Strain of SARS-CoV-2			1.652
Original	36,886 (41.29)	88,835 (99.45)	
Delta	52,438 (58.71)	489 (0.55)	

DBP, diastolic blood pressure; SARS-CoV-2, severe acute respiratory syndrome coronavirus 2; SBP, systolic blood pressure; SD, standard deviation; SMD, standardized mean difference.

* An SMD <0.1 indicates no significant imbalance. All SMDs were <0.100 in the propensity score–matched cohorts.

† Unmatched covariates were included as adjustment factors in statistical analyses.

Table S13. Hazard ratio (95% CI) for the **long-term post-acute respiratory sequelae** or **short-term acute respiratory complication** after SARS-CoV-2 infection in the propensity score-matched cohorts (COVID-19 vs. influenza) of South Korea (main) and Japan (replication)

Cohort	South Korea			Japan		
	COVID-19 vs. influenza (n=223,000)			COVID-19 vs. influenza (n=169,924)		
	Events, n (%)	HR (95% CI)		Events, n (%)	HR (95% CI)	
Model 1*		Model 2†	Model 3*		Model 4‡	
Long-term post-acute respiratory sequelae						
Comparators (general population or patients with influenza)	1081 (0.97)	1.0 (reference)	1.0 (reference)	905 (1.07)	1.0 (reference)	1.0 (reference)
Patients with COVID-19	1500 (1.35)	1.55 (1.43-1.67)	1.66 (1.52-1.82)	4757 (5.60)	5.44 (5.07-5.84)	5.17 (4.82-5.55)
Acute respiratory complication						
Comparators	45 (0.040)	1.0 (reference)	1.0 (reference)	122 (0.14)	1.0 (reference)	1.0 (reference)
Patients with COVID-19	115 (0.10)	4.75 (3.08-7.33)	4.32 (2.73-6.83)	924 (1.09)	7.57 (6.27-9.14)	6.51 (5.38-7.87)

CCI, Charlson comorbidity index; CI, confidence interval; HR, hazard ratio.

The data in bold indicate significant differences ($P < 0.05$).

***Model 1 and 3:** Adjusted for age (20–39, 40–59, and ≥ 60 years) and sex.

†**Model 2:** Adjusted for age (20–39, 40–59, and ≥ 60 years); sex, household income (low income, middle income, and high income); region of residence (urban and rural); CCI score (0, 1, and ≥ 2); obesity (underweight [$< 18.5 \text{ kg/m}^2$], normal [$18.5\text{--}22.9 \text{ kg/m}^2$]; overweight [$23.0\text{--}24.9 \text{ kg/m}^2$], obese [$\geq 25.0 \text{ kg/m}^2$], and unknown); blood pressure (systolic blood pressure $< 140 \text{ mmHg}$ and diastolic blood pressure $< 90 \text{ mmHg}$, systolic blood pressure $\geq 140 \text{ mmHg}$ or diastolic blood pressure $\geq 90 \text{ mmHg}$, and unknown); fasting blood glucose (< 100 , $\geq 100 \text{ mg/dL}$, and unknown); serum total cholesterol (< 200 , $200\text{--}239$, $\geq 240 \text{ mg/dL}$, and unknown); glomerular filtration rate (< 60 , $60\text{--}89$, $\geq 90 \text{ mL/min/1.73 m}^2$, and unknown); smoking status (never, former, current smoker, and unknown); alcoholic drinks (< 1 , $1\text{--}2$, $3\text{--}4$, ≥ 5 days per week, and unknown); aerobic physical activity (sufficient, insufficient, and unknown); previous history of cardiovascular disease, and chronic kidney disease; history of medication use for diabetes mellitus, dyslipidemia, and hypertension; and strain of SARS-CoV-2 (original and delta).

‡**Model 4:** Adjusted for age (20–39, 40–59, and ≥ 60 years); sex; insurance status (insured and dependent); CCI score (0, 1, and ≥ 2); body mass index (underweight [$< 18.5 \text{ kg/m}^2$], normal [$18.5\text{--}22.9 \text{ kg/m}^2$], overweight [$23.0\text{--}25.0 \text{ kg/m}^2$], obese [$\geq 25.0 \text{ kg/m}^2$], and unknown); blood pressure (systolic blood pressure $< 140 \text{ mmHg}$ and diastolic blood pressure $< 90 \text{ mmHg}$, systolic blood pressure $\geq 140 \text{ mmHg}$ or diastolic blood

pressure ≥ 90 mmHg, and unknown); fasting blood glucose (< 100 , ≥ 100 mg/dL, and unknown); serum total cholesterol (< 200 , 200–239, ≥ 240 mg/dL, and unknown); glomerular filtration rate (< 60 , 60–89, ≥ 90 mL/min/1.73 m², and unknown); smoking status (non- and current smoker, and unknown); alcoholic drinks (rarely, sometimes, everyday, and unknown); aerobic physical activity (sufficient, insufficient, and unknown); previous history of cardiovascular disease, and chronic kidney disease; history of medication use for diabetes mellitus, dyslipidemia, and hypertension; and strain of SARS-CoV-2 (original and delta).

Table S14. HR (95% CI) for the long-term post-acute respiratory sequelae or short-term acute respiratory complication subtypes after SARS-CoV-2 infection in the propensity score-matched cohorts (COVID-19 vs. influenza) in South Korea (main) and Japan (replication)

Cohort	South Korea			Japan		
	COVID-19 vs. influenza (n=223,000)			COVID-19 vs. influenza (n=169,924)		
	Events, n (%)	HR (95% CI)		Events, n (%)	HR (95% CI)	
Model 1*		Model 2†	Model 3*		Model 4	
Long-term post-acute respiratory sequelae						
Chronic respiratory failure						
Comparators (general population or patients with influenza)	3 (0.0027)	1.0 (reference)	1.0 (reference)	3 (0.0035)	1.0 (reference)	1.0 (reference)
Patients with COVID-19	12 (0.011)	4.19 (1.18-14.86)	4.82 (1.28-18.12)	77 (0.091)	25.55 (8.06-80.95)	16.15 (5.04-51.80)
Pulmonary hypertension						
Comparators	0 (0.00)	1.0 (reference)	1.0 (reference)	4 (0.0047)	1.0 (reference)	1.0 (reference)
Patients with COVID-19	0 (0.00)	NA	NA	28 (0.033)	6.97 (2.44-19.87)	5.59 (1.93-16.14)
Sleep apnea						
Comparators	23 (0.021)	1.0 (reference)	1.0 (reference)	88 (0.10)	1.0 (reference)	1.0 (reference)
Patients with COVID-19	49 (0.044)	4.26 (2.26-8.01)	4.12 (2.09-8.12)	549 (0.65)	6.26 (5.00-7.84)	5.95 (4.74-7.46)
COPD						
Comparators	734 (0.66)	1.0 (reference)	1.0 (reference)	271 (0.32)	1.0 (reference)	1.0 (reference)
Patients with COVID-19	994 (0.89)	1.48 (1.34-1.63)	1.60 (1.44-1.79)	1703 (2.00)	0.99 (0.91-1.09)	1.01 (0.91-1.12)
Emphysema						
Comparators	19 (0.017)	1.0 (reference)	1.0 (reference)	31 (0.036)	1.0 (reference)	1.0 (reference)
Patients with COVID-19	33 (0.030)	1.96 (1.10-3.47)	1.99 (1.06-3.74)	232 (0.27)	7.47 (5.14-10.87)	6.46 (4.43-9.43)
Asthma						
Comparators	337 (0.30)	1.0 (reference)	1.0 (reference)	477 (0.56)	1.0 (reference)	1.0 (reference)
Patients with COVID-19	421 (0.38)	1.37 (1.18-1.58)	1.45 (1.23-1.71)	2672 (3.14)	5.72 (5.19-6.30)	5.59 (5.07-6.16)
Pulmonary sarcoidosis						

Comparators	2 (0.0018)	1.0 (reference)	1.0 (reference)	25 (0.029)	1.0 (reference)	1.0 (reference)
Patients with COVID-19	2 (0.0018)	2.07 (0.19-22.84)	3.24 (0.29-36.47)	127 (0.15)	5.08 (3.31-7.81)	4.52 (2.93-6.96)
Interstitial lung disease						
Comparators	32 (0.029)	1.0 (reference)	1.0 (reference)	67 (0.079)	1.0 (reference)	1.0 (reference)
Patients with COVID-19	94 (0.084)	3.25 (2.15-4.90)	3.37 (2.19-5.19)	679 (0.80)	10.19 (7.93-13.1)	8.26 (6.42-10.64)
Acute respiratory complication						
Pneumocystis pneumonia						
Comparators	0 (0.00)	1.0 (reference)	1.0 (reference)	18 (0.021)	1.0 (reference)	1.0 (reference)
Patients with COVID-19	0 (0.00)	NA	NA	66 (0.078)	3.65 (2.17-6.14)	2.85 (1.68-4.85)
Aspergillosis pneumonia						
Comparators	0 (0.00)	1.0 (reference)	1.0 (reference)	4 (0.0047)	1.0 (reference)	1.0 (reference)
Patients with COVID-19	9 (0.0081)	NA	NA	17 (0.020)	4.21 (1.42-12.52)	3.49 (1.16-10.54)
Pleural empyema						
Comparators	2 (0.0018)	1.0 (reference)	1.0 (reference)	0 (0.00)	1.0 (reference)	1.0 (reference)
Patients with COVID-19	0 (0.00)	NA	NA	18 (0.021)	NA	NA
Lung abscess						
Comparators	6 (0.0054)	1.0 (reference)	1.0 (reference)	6 (0.0071)	1.0 (reference)	1.0 (reference)
Patients with COVID-19	0 (0.00)	NA	NA	18 (0.021)	2.98 (1.18-7.50)	2.25 (0.88-5.80)
Pneumothorax						
Comparators	18 (0.016)	1.0 (reference)	1.0 (reference)	22 (0.026)	1.0 (reference)	1.0 (reference)
Patients with COVID-19	10 (0.0090)	1.05 (0.44-2.53)	1.02 (0.37-2.81)	114 (0.13)	5.17 (3.28-8.16)	4.92 (3.11-7.77)
Acute respiratory failure						
Comparators	2 (0.0018)	1.0 (reference)	1.0 (reference)	58 (0.068)	1.0 (reference)	1.0 (reference)
Patients with COVID-19	67 (0.060)	NA	NA	583 (0.69)	10.04 (7.67-13.15)	8.44 (6.43-11.07)
Pulmonary embolism						
Comparators	18 (0.06)	1.0 (reference)	1.0 (reference)	18 (0.021)	1.0 (reference)	1.0 (reference)
Patients with COVID-19	29 (0.026)	2.94 (1.43-6.03)	3.27 (1.53-7.02)	175 (0.21)	9.68 (5.96-15.73)	7.96 (4.88-12.98)

CCI, Charlson comorbidity index; CI, confidence interval; COPD, chronic obstructive pulmonary disease; HR, hazard ratio; NA, not available.

The data in bold indicate significant differences ($P < 0.05$).

***Model 1 and 3:** Adjusted for age (20–39, 40–59, and ≥ 60 years) and sex.

†Model 2: Adjusted for age (20–39, 40–59, and ≥ 60 years); sex, household income (low income, middle income, and high income); region of residence (urban and rural); CCI score (0, 1, and ≥ 2); obesity (underweight [$< 18.5 \text{ kg/m}^2$], normal [$18.5\text{--}22.9 \text{ kg/m}^2$]; overweight [$23.0\text{--}24.9 \text{ kg/m}^2$], obese [$\geq 25.0 \text{ kg/m}^2$], and unknown); blood pressure (systolic blood pressure $< 140 \text{ mmHg}$ and diastolic blood pressure $< 90 \text{ mmHg}$, systolic blood pressure $\geq 140 \text{ mmHg}$ or diastolic blood pressure $\geq 90 \text{ mmHg}$, and unknown); fasting blood glucose (< 100 , $\geq 100 \text{ mg/dL}$, and unknown); serum total cholesterol (< 200 , $200\text{--}239$, $\geq 240 \text{ mg/dL}$, and unknown); glomerular filtration rate (< 60 , $60\text{--}89$, $\geq 90 \text{ mL/min/1.73 m}^2$, and unknown); smoking status (never, former, current smoker, and unknown); alcoholic drinks (< 1 , $1\text{--}2$, $3\text{--}4$, ≥ 5 days per week, and unknown); aerobic physical activity (sufficient, insufficient, and unknown); previous history of cardiovascular disease, and chronic kidney disease; history of medication use for diabetes mellitus, dyslipidemia, and hypertension; and strain of SARS-CoV-2 (original and delta).

‡Model 4: Adjusted for age (20–39, 40–59, and ≥ 60 years); sex; insurance status (insured and dependent); CCI score (0, 1, and ≥ 2); body mass index (underweight [$< 18.5 \text{ kg/m}^2$], normal [$18.5\text{--}22.9 \text{ kg/m}^2$], overweight [$23.0\text{--}25.0 \text{ kg/m}^2$], obese [$\geq 25.0 \text{ kg/m}^2$], and unknown); blood pressure (systolic blood pressure $< 140 \text{ mmHg}$ and diastolic blood pressure $< 90 \text{ mmHg}$, systolic blood pressure $\geq 140 \text{ mmHg}$ or diastolic blood pressure $\geq 90 \text{ mmHg}$, and unknown); fasting blood glucose (< 100 , $\geq 100 \text{ mg/dL}$, and unknown); serum total cholesterol (< 200 , $200\text{--}239$, $\geq 240 \text{ mg/dL}$, and unknown); glomerular filtration rate (< 60 , $60\text{--}89$, $\geq 90 \text{ mL/min/1.73 m}^2$, and unknown); smoking status (non- and current smoker, and unknown); alcoholic drinks (rarely, sometimes, everyday, and unknown); aerobic physical activity (sufficient, insufficient, and unknown); previous history of cardiovascular disease, and chronic kidney disease; history of medication use for diabetes mellitus, dyslipidemia, and hypertension; and strain of SARS-CoV-2 (original and delta).

Table S16. Time attenuation effect analysis (COVID-19 vs. influenza) of HR (95% CI) for the risk of the **long-term post-acute respiratory sequelae** after SARS-CoV-2 infection in South Korea (main cohort) and Japan (replication cohort)

Time	COVID-19 vs. influenza	
	Main cohort [†]	Replication cohort
long-term post-acute respiratory sequelae		
<3 months	2.28 (2.01-2.59)	8.88 (7.76-10.16)
3–6 months	1.24 (1.04-1.47)	3.81 (3.27-4.44)
≥6 months	1.01 (0.83-1.23)	3.65 (3.30-4.05)

CCI, Charlson comorbidity index; CI, confidence interval; HR, hazard ratio; SARS-CoV-2, severe acute respiratory syndrome coronavirus 2.

The data in bold indicate significant differences ($P < 0.05$).

† Adjusted HR (main): Adjusted for age (20–39, 40–59, and ≥60 years); sex, household income (low income, middle income, and high income); region of residence (urban and rural); CCI score (0, 1, and ≥2); obesity (underweight [$<18.5 \text{ kg/m}^2$], normal [$18.5\text{--}22.9 \text{ kg/m}^2$]; overweight [$23.0\text{--}24.9 \text{ kg/m}^2$], obese [$\geq 25.0 \text{ kg/m}^2$], and unknown); blood pressure (systolic blood pressure $<140 \text{ mmHg}$ and diastolic blood pressure $<90 \text{ mmHg}$, systolic blood pressure $\geq 140 \text{ mmHg}$ or diastolic blood pressure $\geq 90 \text{ mmHg}$, and unknown); fasting blood glucose (<100 , $\geq 100 \text{ mg/dL}$, and unknown); serum total cholesterol (<200 , $200\text{--}239$, $\geq 240 \text{ mg/dL}$, and unknown); glomerular filtration rate (<60 , $60\text{--}89$, $\geq 90 \text{ mL/min/1.73 m}^2$, and unknown); smoking status (never, former, current smoker, and unknown); alcoholic drinks (<1 , $1\text{--}2$, $3\text{--}4$, ≥ 5 days per week, and unknown); aerobic physical activity (sufficient, insufficient, and unknown); previous history of cardiovascular disease, and chronic kidney disease; history of medication use for diabetes mellitus, dyslipidemia, and hypertension; and strain of SARS-CoV-2 (original and delta).

|| Adjusted HR (replication): Adjusted for age (20–39, 40–59, and ≥60 years); sex; insurance status (insured and dependent); CCI score (0, 1, and ≥ 2); body mass index (underweight [$<18.5 \text{ kg/m}^2$], normal [$18.5\text{--}22.9 \text{ kg/m}^2$], overweight [$23.0\text{--}25.0 \text{ kg/m}^2$], obese [$\geq 25.0 \text{ kg/m}^2$], and unknown); blood pressure (systolic blood pressure $<140 \text{ mmHg}$ and diastolic blood pressure $<90 \text{ mmHg}$, systolic blood pressure $\geq 140 \text{ mmHg}$ or diastolic blood pressure $\geq 90 \text{ mmHg}$, and unknown); fasting blood glucose (<100 , $\geq 100 \text{ mg/dL}$, and unknown); serum total cholesterol (<200 , $200\text{--}239$, $\geq 240 \text{ mg/dL}$, and unknown); glomerular filtration rate (<60 , $60\text{--}89$, $\geq 90 \text{ mL/min/1.73 m}^2$, and unknown); smoking status (non- and current smoker, and unknown); alcoholic drinks (rarely, sometimes, everyday, and unknown); aerobic physical activity (sufficient, insufficient,

and unknown); previous history of cardiovascular disease, and chronic kidney disease; history of medication use for diabetes mellitus, dyslipidemia, and hypertension; and strain of SARS-CoV-2 (original and delta).

Comment 14.

The population of South Korea is >50 million people. Therefore, how can you claim that the discovery cohort of ~10 million covers 98% of the population?

Response:

Thank you for your important comment, which would enhance the credibility of our manuscript. The NHIS data is national health insurance claims records data that includes over 50 million people and covers about 98% of the Korean population. Therefore, the NHIS database is credible in representing the Korean population. However, the database we utilized is not the full database generated by NHIS. The K-CoV-N cohort is a cohort generated by NHIS that covers people who are 20 years old or older and have a record of medical examination during the pre-observation (2018 to 2019) and the follow-up period (2020 to 2021). This cohort is specially generated by NHIS for research purposes and includes over 10 million participants. We understand your concern about the claim that our dataset covers 98% of the population and admit that the previous manuscript lacked an explanation of the main cohort. Therefore, we further explained the South Korean database in the Methods section.

Changes in text:**Methods/*K-CoV-N cohort for main cohort***

- We utilized the NHIS database, which is a large-scale, nationwide, general, population-based cohort in South Korea, covering 98% of the population for the main cohort.¹⁰
- The constructed K-CoV-N database embodies the following characteristics, thereby affirming its significance: (1) the Korean government has established an extensive healthcare system to provide coverage for individuals infected with SARS-CoV-2; (2) all patient-related data was anonymized by the Korean government;¹⁰ and (3) according to prior study, the diagnostic records from the NHIS had a predictive accuracy of 82%.¹¹

Discussion/*Limitations and strengths*

- First, although the database used is a highly credible database that covers 98% of the Korean population and 40% of the Japanese population, individuals who could be vulnerable to influenza and COVID-19, such as immigrants and undocumented immigrants, are left out of the database.^{10, 49}

References (Manuscript)

- 10 Shin, Y. H. et al. Autoimmune inflammatory rheumatic diseases and COVID-19 outcomes in South Korea: a nationwide cohort study. *The Lancet Rheumatology* 3, e698-e706 (2021).
- 11 Park, B., Sung, J., Park, K., Seo, S. & Kim, S. Report of the evaluation for validity of discharged diagnoses in Korean Health Insurance database. Seoul: Seoul National University, 19-52 (2003).
- 49 Sato, M. *et al.* Effect of periodontal therapy on glycaemic control in type 2 diabetes. *J Clin Periodontol* (2024). <https://doi.org/10.1111/jcpe.13939>

Comment 15.

The methods section implies that vaccination status including type of vaccine is recorded at an individual level. However, on page 20 is the sentence “Third, this study illustrated the method of mixed vaccination in general, not showing the specific vaccines used on each person.” Please clarify whether vaccination is recorded for each individual or not.

Response:

Thank you for bringing this to our attention, and we apologize for any confusion caused by our initial wording in the manuscript. Vaccination status, including the type of vaccine received, is recorded for each individual in our dataset. This information allowed us to analyze the impact of mixed vaccination types on developing acute respiratory complications and long-term post-acute respiratory sequelae among patients with COVID-19. However, we did not consider the order of the vaccination the individuals took when they received both types of SARS-CoV-2 vaccination. Instead, we grouped every individual who took both types of vaccinations in the ‘patients with COVID-19 with both types of SARS-CoV-2 vaccination’ group in Table 4. Also, we excluded the following limitation from the manuscript for a clearer description and removal of any ambiguity. Once again, we apologize for the ambiguous description in the original version and thank you for the opportunity to improve the clarity of our manuscript.

Changes in text:**Discussion/Limitations and strengths**

~~Third, this study illustrated the method of mixed vaccination in general, not showing the specific vaccines used on each person. Further studies should be conducted to identify the vaccine combination to minimize the side effects and COVID-19 complications.~~

Table 4. Subgroup analysis (COVID-19 vs. general population) of HR (95% CI) of the **long-term post-acute respiratory sequelae** or **short-term acute respiratory complication** after SARS-CoV-2 infection stratified by vaccination, COVID-19 severity, and SARS-CoV-2 strain in the cohort of South Korea (main)

Variable	Events/total, n/N (%)	HR (95% CI)	
		Model 1*	Model 2†
Long-term post-acute respiratory sequelae			
Number of SARS-CoV-2 vaccinations			
Non-infected control	16,122/1,918,150 (0.84)	0.62 (0.60-0.65)	0.60 (0.58-0.62)
COVID-19 without SARS-CoV-2 vaccination	4331/200,539 (2.16)	1.0 (reference)	1.0 (reference)
COVID-19 after SARS-CoV-2 vaccination received once	493/38,852 (1.27)	0.90 (0.82-0.99)	0.85 (0.77-0.94)
COVID-19 after SARS-CoV-2 vaccination received twice or more	468/155,207 (0.30)	0.69 (0.62-0.76)	0.64 (0.57-0.71)
Type of SARS-CoV-2 vaccinations			
Non-infected control	16,122/1,918,150 (0.84)	0.62 (0.60-0.65)	0.60 (0.58-0.62)
COVID-19 without SARS-CoV-2 vaccination	4331/200,539 (2.16)	1.0 (reference)	1.0 (reference)
COVID-19 with viral vector SARS-CoV-2 vaccination	465/109,066 (0.43)	1.12 (1.01-1.23)	0.99 (0.90-1.10)
COVID-19 with mRNA SARS-CoV-2 vaccination	477/66,891 (0.71)	1.20 (1.09-1.33)	1.11 (0.99-1.22)
COVID-19 with both types of SARS-CoV-2 vaccination	19/18,102 (0.10)	0.74 (0.47-1.16)	0.66 (0.42-1.03)
COVID-19 severity			
Non-infected control	16,122/1,918,150 (0.84)	1.0 (reference)	1.0 (reference)
Mild COVID-19	3492/340,813 (1.02)	1.28 (1.24-1.33)	1.37 (1.32-1.43)
Moderate to severe COVID-19	1800/53785 (3.35)	3.60 (3.43-3.78)	3.20 (3.03-3.38)
Original strain of SARS-CoV-2 (overall population)			

Non-infected control before the delta-dominant phase [§]	11,667/594,134 (1.96)	1.0 (reference)	1.0 (reference)
Infection with original strain	3649/121,521 (3.00)	1.58 (1.52-1.64)	1.59 (1.52-1.66)
Delta variant of SARS-CoV-2 (overall population)			
Non-infected control during the delta-dominant phase [§]	4455/1,324,016 (0.34)	1.0 (reference)	1.0 (reference)
Infection with Delta variant	1643/273,077 (0.60)	1.81 (1.71-1.92)	1.94 (1.81-2.08)
Acute respiratory complication			
Number of SARS-CoV-2 vaccinations			
Non-infected control	311/1,918,150 (0.016)	0.07 (0.06-0.08)	0.08 (0.06-0.09)
COVID-19 without SARS-CoV-2 vaccination	415/200,539 (0.21)	1.0 (reference)	1.0 (reference)
COVID-19 after SARS-CoV-2 vaccination received once	56/38,852 (0.14)	0.62 (0.47-0.83)	0.51 (0.38-0.68)
COVID-19 after SARS-CoV-2 vaccination received twice or more	147/155,207 (0.09)	0.32 (0.26-0.39)	0.24 (0.19-0.30)
Type of SARS-CoV-2 vaccinations			
Non-infected control	311/1,918,150 (0.016)	0.07 (0.06-0.08)	0.08 (0.06-0.09)
COVID-19 without SARS-CoV-2 vaccination	415/200,539 (0.21)	1.0 (reference)	1.0 (reference)
COVID-19 with viral vector SARS-CoV-2 vaccination	79/109,066 (0.072)	0.44 (0.34-0.56)	0.36 (0.28-0.47)
COVID-19 with mRNA SARS-CoV-2 vaccination	117/66,891 (0.17)	0.40 (0.32-0.50)	0.42 (0.33-0.53)
COVID-19 with both types of SARS-CoV-2 vaccination	7/18,102 (0.039)	0.19 (0.09-0.41)	0.18 (0.08-0.38)
COVID-19 severity			
Non-infected control	311/1,918,150 (0.016)	1.0 (reference)	1.0 (reference)
Mild COVID-19	50/340,813 (0.015)	0.95 (0.71-1.28)	0.99 (0.73-1.34)
Moderate to severe COVID-19	568/53,783 (1.06)	51.18 (44.38-59.02)	39.54 (33.54-46.62)
Original strain of SARS-CoV-2 (overall population)			
Non-infected control before the delta-dominant phase [§]	110/594,134 (0.019)	1.0 (reference)	1.0 (reference)

Infection with original strain	230/121,521 (0.19)	10.27 (8.18-12.88)	9.21 (7.19-11.80)
Delta variant of SARS-CoV-2 (overall population)			
Non-infected control during the delta-dominant phase §	201/1,324,016 (0.015)	1.0 (reference)	1.0 (reference)
Infection with Delta variant	388/273,077 (0.14)	9.39 (7.92-11.14)	7.44 (6.13-9.03)

CCI, Charlson comorbidity index; CI, confidence interval; HR, hazard ratio; SARS-CoV-2, severe acute respiratory syndrome coronavirus 2. The data in bold indicate significant differences ($P < 0.05$).

|| HR of the non-infected control represents the risk of respiratory diseases, and HRs of patients with COVID-19 indicate the risk of post-acute respiratory complications following SARS-CoV-2 infection.

§ Only 1:5-matched comparators in each patient group at the same index date were included to reduce immortal time bias.

***Model 1:** Adjusted for age (20–39, 40–59, and ≥ 60 years) and sex.

†**Model 2:** Adjusted for age (20–39, 40–59, and ≥ 60 years); sex, household income (low income, middle income, and high income); region of residence (urban and rural); CCI score (0, 1, and ≥ 2); obesity (underweight [$< 18.5 \text{ kg/m}^2$], normal [$18.5\text{--}22.9 \text{ kg/m}^2$]; overweight [$23.0\text{--}24.9 \text{ kg/m}^2$], obese [$\geq 25.0 \text{ kg/m}^2$], and unknown); blood pressure (systolic blood pressure $< 140 \text{ mmHg}$ and diastolic blood pressure $< 90 \text{ mmHg}$, systolic blood pressure $\geq 140 \text{ mmHg}$ or diastolic blood pressure $\geq 90 \text{ mmHg}$, and unknown); fasting blood glucose (< 100 , $\geq 100 \text{ mg/dL}$, and unknown); serum total cholesterol (< 200 , $200\text{--}239$, $\geq 240 \text{ mg/dL}$, and unknown); glomerular filtration rate (< 60 , $60\text{--}89$, $\geq 90 \text{ mL/min/1.73 m}^2$, and unknown); smoking status (never, former, current smoker, and unknown); alcoholic drinks (< 1 , $1\text{--}2$, $3\text{--}4$, ≥ 5 days per week, and unknown); aerobic physical activity (sufficient, insufficient, and unknown); previous history of cardiovascular disease, and chronic kidney disease; history of medication use for diabetes mellitus, dyslipidemia, and hypertension; and strain of SARS-CoV-2 (original and delta).

Comment 16.

I do not expect you to act on this advice posthoc. However, public and patient involvement is beneficial in the secondary analysis and interpretation of data. Their insights can be very useful and there are many people with long-COVID who are interested and engaged in research.

Response:

Thank you for your constructive comment. We thoroughly agree with your comment that we need an explanation of the public and patient involvement in the manuscript. Therefore, we added a patient and public involvement section to the last part of the method section. We hope all our comments and changes in the text met your expectations.

Changes in text:**Method/*Patient and public involvement***

- The Korean government and JMDC anonymized patient data by excluding patient-related data such as personal identification numbers or names for confidentiality. While direct identification of individuals was rendered impossible due to the removal of names, all other pertinent data remained intact and accessible for our analyses. Research questions and outcome measures were autonomously determined without the intervention of individuals. The research design and implementation proceeded without external consultation. However, upon request, the researchers intend to disseminate the results of this research to all research participants and relevant communities.

REVIEWERS' COMMENTS

Reviewer #1 (Remarks to the Author):

I appreciate the authors and their team for the throughout responses to my comments. I have one additional comment following #5, as I suggesting presenting the marginal predicted prevalence for each conditions. First, I think these information should be presented as one of the primary information--if the manuscript has maxed out in number of tables/figure, the predicted prevalence can be mentioned in the text. Secondly, I am having trouble finding where the predicted prevalence are. Supplementary Table S7 presented risk related to individual conditions, but they only presented the HR in each condition, which is different from what I was initially asking for. Would you please clarify?

I have no additional comments. Thank you for the time and efforts!

Reviewer #2 (Remarks to the Author):

Thank you for your response to reviewer comments and the substantial changes made to the manuscript. I believe it is much improved. However, I have some outstanding concerns that were not fully addressed:

1. Please include in the limitations section the differential missingness between cases and controls. For example >40% versus <1% missingness for BMI, blood pressure, glucose and lifestyle variables. The reason for this is not given.
2. Thank you for your explanation of the NHIS data subset you are using (blue text page 260 of rebuttal). Please add this explanation to the methods section.
3. To clarify, public and patient involvement does not necessarily mean involvement from the study participants themselves. This is not possible when performing secondary analysis of anonymised data. Rather, it can involve other relevant members of the public who are qualified to comment on the study design, analysis and interpretation. I feel there has been some misunderstanding here. My comment was a suggestion for good practice in future work.

Response Letter

Table of Contents

1. Reviewer #1	3
2. Reviewer #2	9

Reviewer #1

Comment 1.

I appreciate the authors and their team for the throughout responses to my comments. I have one additional comment following #5, as I suggesting presenting the marginal predicted prevalence for each conditions. First, I think these information should be presented as one of the primary information- if the manuscript has maxed out in number of tables/figure, the predicted prevalence can be mentioned in the text. Secondaryly, I am having trouble finding where the predicted prevalence are. Supplementary Table S7 presented risk related to individual conditions, but they only presented the HR in each condition, which is different from what I was initially asking for. Would you please clarify? I have no additional comments. Thank you for the time and efforts!

Response:

Thank you for your feedback and for recognizing our efforts to address your comments thoroughly. We have carefully considered your suggestion and have incorporated the results of predicted prevalence into the manuscript. In addition, we have added tables (Tables S11 and S12) that present the predicted prevalence values for each condition, ensuring that these important findings are clearly reflected and accessible. We hope these refinements will address your comments and enhance the detail of the manuscript. Thank you again for your thoughtful input.

Changes in text:

Result

This tendency of increased risk for several subtypes of respiratory diseases was also shown when compared to patients with influenza infection and the overlap-weighted cohort. (Tables S7-S10). Estimates of marginal prevalence showed that patients with COVID-19 had a higher prevalence compared to the general population (Tables S11 and S12).

Table S11. Marginal predicted prevalence (percent; 95% CI) of general population vs. patients with COVID-19 in South Korea

	South Korea, percent (95% CI)			
	Post-acute respiratory sequelae		Acute respiratory complication	
	Model 1*	Model 2†	Model 1*	Model 2†
Age group				
20-39 y				
Comparators (general population)	0.83 (0.83-0.83)	0.83 (0.83-0.83)	0.01 (0.01-0.01)	0.01 (0.01-0.01)
Patients with COVID-19	1.05 (1.05-1.05)	1.04 (1.04-1.05)	0.05 (0.05-0.05)	0.05 (0.04-0.05)
40-59 y				
Comparators	0.74 (0.74-0.74)	0.74 (0.74-0.74)	0.01 (0.01-0.01)	0.01 (0.01-0.01)
Patients with COVID-19	1.28 (1.28-1.28)	1.28 (1.27-1.28)	0.11 (0.11-0.11)	0.11 (0.11-0.11)
≥60 y				
Comparators	1.00 (1.00-1.00)	0.97 (0.97-0.97)	0.03 (0.03-0.03)	0.03 (0.03-0.03)
Patients with COVID-19	1.82 (1.82-1.82)	1.75 (1.74-1.75)	0.37 (0.37-0.37)	0.37 (0.37-0.38)
Sex				
Male				
Comparators	0.79 (0.79-0.79)	0.77 (0.77-0.77)	0.02 (0.02-0.02)	0.02 (0.02-0.02)
Patients with COVID-19	1.28 (1.28-1.28)	1.25 (1.25-1.25)	0.17 (0.17-0.17)	0.17 (0.17-0.17)
Female				
Comparators	0.90 (0.90-0.90)	0.89 (0.89-0.89)	0.02 (0.02-0.02)	0.02 (0.02-0.02)
Patients with COVID-19	1.41 (1.41-1.41)	1.47 (1.46-1.48)	0.14 (0.14-0.14)	0.14 (0.14-0.15)
Household income				
Low (0th–39th percentile)				
Comparators	0.84 (0.84-0.84)	0.83 (0.83-0.83)	0.02 (0.02-0.02)	0.02 (0.02-0.02)
Patients with COVID-19	1.30 (1.30-1.30)	1.27 (1.27-1.28)	0.13 (0.13-0.13)	0.13 (0.13-0.13)
Middle (40th –79th percentile)				
Comparators	0.85 (0.85-0.85)	0.84 (0.84-0.84)	0.01 (0.01-0.01)	0.01 (0.01-0.01)

Patients with COVID-19	1.33 (1.33-1.33)	1.31 (1.31-1.31)	0.16 (0.16-0.16)	0.16 (0.16-0.17)
High (80th–100th percentile)				
Comparators	0.83 (0.83-0.83)	0.81 (0.81-0.81)	0.02 (0.02-0.02)	0.02 (0.02-0.02)
Patients with COVID-19	1.48 (1.47-1.48)	1.45 (1.44-1.45)	0.21 (0.20-0.21)	0.21 (0.20-0.21)
Region of residence				
Urban				
Comparators	0.81 (0.81-0.81)	0.80 (0.80-0.80)	0.02 (0.02-0.02)	0.02 (0.02-0.02)
Patients with COVID-19	1.32 (1.32-1.33)	1.30 (1.30-1.30)	0.13 (0.13-0.13)	0.13 (0.13-0.13)
Rural				
Comparators	0.88 (0.88-0.88)	0.87 (0.87-0.87)	0.01 (0.01-0.01)	0.01 (0.01-0.01)
Patients with COVID-19	1.36 (1.36-1.36)	1.34 (1.34-1.34)	0.19 (0.19-0.19)	0.19 (0.19-0.19)
Charlson comorbidity index score				
0				
Comparators	0.79 (0.79-0.79)	0.78 (0.78-0.78)	0.01 (0.01-0.01)	0.01 (0.01-0.01)
Patients with COVID-19	1.20 (1.20-1.20)	1.19 (1.19-1.19)	0.10 (0.10-0.10)	0.10 (0.10-0.10)
≥1				
Comparators	1.67 (1.67-1.67)	1.61 (1.61-1.61)	0.10 (0.10-0.10)	0.10 (0.10-0.10)
Patients with COVID-19	2.33 (2.33-2.34)	2.38 (2.35-2.41)	0.58 (0.58-0.59)	0.58 (0.58-0.59)

CI, confidence interval.

***Model 1:** Adjusted for age (20–39, 40–59, and ≥60 years) and sex.

†**Model 2:** Adjusted for age (20–39, 40–59, and ≥60 years); sex, household income (low income, middle income, and high income); region of residence (urban and rural); CCI score (0, 1, and ≥2); obesity (underweight [$<18.5 \text{ kg/m}^2$], normal [$18.5\text{--}22.9 \text{ kg/m}^2$]; overweight [$23.0\text{--}24.9 \text{ kg/m}^2$], obese [$\geq 25.0 \text{ kg/m}^2$], and unknown); blood pressure (systolic blood pressure $<140 \text{ mmHg}$ and diastolic blood pressure $<90 \text{ mmHg}$, systolic blood pressure $\geq 140 \text{ mmHg}$ or diastolic blood pressure $\geq 90 \text{ mmHg}$, and unknown); fasting blood glucose (<100 , $\geq 100 \text{ mg/dL}$, and unknown); serum total cholesterol (<200 , $200\text{--}239$, $\geq 240 \text{ mg/dL}$, and unknown); glomerular filtration rate (<60 , $60\text{--}89$, $\geq 90 \text{ mL/min/1.73 m}^2$, and unknown); smoking status (never, former, current smoker, and unknown); alcoholic drinks (<1 , $1\text{--}2$, $3\text{--}4$, ≥ 5 days per week, and unknown); aerobic physical activity (sufficient, insufficient, and unknown); previous history of cardiovascular disease, and chronic kidney disease; history

of medication use for diabetes mellitus, dyslipidemia, and hypertension; and strain of SARS-CoV-2 (original and delta).

Table S12. Marginal predicted prevalence (percent; 95% CI) of general population vs. patients with COVID-19 in Japan

	Japan, percent (95% CI)			
	Post-acute respiratory sequelae		Acute respiratory complication	
	Model 3*	Model 4 ^{II}	Model 3*	Model 4 ^{II}
Age group				
20-39 y				
Comparators (general population)	1.43 (1.43-1.44)	1.43 (1.43-1.43)	0.05 (0.04-0.05)	0.05 (0.05-0.05)
Patients with COVID-19	4.83 (4.82-4.83)	4.83 (4.82-4.83)	0.24 (0.24-0.24)	0.24 (0.24-0.24)
40-59 y				
Comparators	1.53 (1.53-1.54)	1.53 (1.52-1.53)	0.07 (0.07-0.08)	0.07 (0.07-0.07)
Patients with COVID-19	5.04 (5.04-5.04)	5.04 (5.04-5.05)	0.29 (0.29-0.29)	0.29 (0.29-0.29)
≥60 y				
Comparators	1.84 (1.83-1.84)	1.84 (1.84-1.84)	0.08 (0.08-0.08)	0.08 (0.08-0.08)
Patients with COVID-19	6.89 (6.88-6.89)	6.89 (6.87-6.91)	0.49 (0.49-0.49)	0.49 (0.49-0.49)
Sex				
Male				
Comparators	1.46 (1.46-1.47)	1.46 (1.46-1.47)	0.07 (0.06-0.07)	0.07 (0.07-0.08)
Patients with COVID-19	5.04 (5.03-5.04)	5.04 (5.03-5.04)	0.32 (0.32-0.32)	0.32 (0.32-0.32)
Female				
Comparators	1.62 (1.61-1.63)	1.62 (1.62-1.62)	0.06 (0.06-0.06)	0.06 (0.06-0.06)
Patients with COVID-19	5.35 (5.35-5.35)	5.35 (5.34-5.35)	0.24 (0.24-0.24)	0.24 (0.24-0.24)
Charlson comorbidity index score				
0				
Comparators	1.51 (1.50-1.52)	1.50 (1.49-1.51)	0.06 (0.06-0.07)	0.06 (0.06-0.06)
Patients with COVID-19	4.75 (4.74-4.75)	4.75 (4.74-4.75)	0.25 (0.24-0.25)	0.25 (0.25-0.25)
≥1				

Comparators	3.43 (3.43-3.43)	3.43 (3.42-3.43)	0.23 (0.22-0.24)	0.24 (0.24-0.24)
Patients with COVID-19	14.58 (14.56-14.59)	14.58 (14.56-14.60)	1.15 (1.15-1.16)	1.15 (1.15-1.16)

CI, confidence interval.

***Model 3:** Adjusted for age (20–39, 40–59, and ≥60 years) and sex.

|| **Model 4:** Adjusted for age (20–39, 40–59, and ≥60 years); sex; insurance status (insured and dependent); CCI score (0, 1, and ≥ 2); body mass index (underweight [$<18.5 \text{ kg/m}^2$], normal [$18.5\text{--}22.9 \text{ kg/m}^2$], overweight [$23.0\text{--}25.0 \text{ kg/m}^2$], obese [$\geq 25.0 \text{ kg/m}^2$], and unknown); blood pressure (systolic blood pressure $<140 \text{ mmHg}$ and diastolic blood pressure $<90 \text{ mmHg}$, systolic blood pressure $\geq 140 \text{ mmHg}$ or diastolic blood pressure $\geq 90 \text{ mmHg}$, and unknown); fasting blood glucose (<100 , $\geq 100 \text{ mg/dL}$, and unknown); serum total cholesterol (<200 , $200\text{--}239$, $\geq 240 \text{ mg/dL}$, and unknown); glomerular filtration rate (<60 , $60\text{--}89$, $\geq 90 \text{ mL/min/1.73 m}^2$, and unknown); smoking status (non- and current smoker, and unknown); alcoholic drinks (rarely, sometimes, everyday, and unknown); aerobic physical activity (sufficient, insufficient, and unknown); previous history of cardiovascular disease, and chronic kidney disease; history of medication use for diabetes mellitus, dyslipidemia, and hypertension; and strain of SARS-CoV-2 (original and delta).

Reviewer #2

Comment 1.

Thank you for your response to reviewer comments and the substantial changes made to the manuscript. I believe it is much improved. However, I have some outstanding concerns that were not fully addressed:

- 1. Please include in the limitations section the differential missingness between cases and controls. For example >40% versus <1% missingness for BMI, blood pressure, glucose and lifestyle variables. The reason for this is not given.*

Response:

Thank you for your feedback and the opportunity to clarify the manuscript further. The differential missingness proportion (>40% in cases vs. <1% in controls), as you mentioned, was regarding the national health examination information variables. Since these variables were not included in the propensity score matching process, these differences inevitably occurred. Therefore, we have updated the Discussion ‘*Limitations and strengths*’ section to address your concern regarding the differential missingness of data. We appreciate your invaluable input in improving our manuscript.

Changes in text:

Discussion/*Limitations and strengths*

Sixth, the propensity score-matched cohort had differential missingness between those infected with SARS-CoV-2 and the general population for national health examination information variables (BMI, blood pressure, fasting blood glucose, glomerular filtration rate, smoking status, alcohol consumption, and aerobic physical activity; >40% versus <1%), due to their exclusion from matching criteria.

Comment 2.

2. Thank you for your explanation of the NHIS data subset you are using (blue text page 260 of rebuttal).

Please add this explanation to the methods section.

Response:

Thank you for your feedback. In response to your suggestion, we have added a detailed explanation of the NHIS datasets within the method sections of the main manuscript, as below.

Changes in text:

Methods/*K-CoV-N cohort for main cohort*

The NHIS and the KDCA provided data for the cohort constructed for study purposes, which includes participants ≥ 20 years old with a record of medical examination from January 1, 2018, to December 31, 2021 (total N=10,027,506).

Comment 3.

3. To clarify, public and patient involvement does not necessarily mean involvement from the study participants themselves. This is not possible when performing secondary analysis of anonymised data. Rather, it can involve other relevant members of the public who are qualified to comment on the study design, analysis and interpretation. I feel there has been some misunderstanding here. My comment was a suggestion for good practice in future work.

Response:

Thank you for your insightful comment regarding public and patient involvement. We acknowledge that conducting a secondary analysis may not be feasible due to the anonymization of the data. Therefore, we have carefully considered your suggestions and have enhanced our manuscript as below. Once again, we sincerely appreciate the feedback and hope this revision aligns with your expectations.

Changes in text:***Methods/Patient and public involvement***

The Korean government and JMDC anonymized patient data by excluding patient-related data such as personal identification numbers or names for confidentiality. While direct identification of individuals was rendered impossible due to the removal of names, all other pertinent data remained intact and accessible for our analyses. Research questions and outcome measures were autonomously determined without the intervention of individuals. The research design and implementation proceeded without external consultation. **However, it can be extended to include contributions from other qualified public participants who are able to offer valuable insights into the research design, analysis, and interpretation.** Upon request, the researchers intend to disseminate the results of this research to all research participants and relevant communities.